# A connectome of the *Drosophila* central complex reveals network motifs suitable for flexible navigation and context-dependent action selection

Brad K Hulse*†, Hannah Haberkern*†, Romain Franconville*†,
Daniel Turner-Evans*†, Shin-ya Takemura, Tanya Wolff, Marcella Noorman,
Marisa Dreher, Chuntao Dan, Ruchi Parekh, Ann M Hermundstad, Gerald M Rubin,
Vivek Jayaraman*

Janelia Research Campus, Howard Hughes Medical Institute, Ashburn, United States

**Abstract** Flexible behaviors over long timescales are thought to engage recurrent neural networks in deep brain regions, which are experimentally challenging to study. In insects, recurrent circuit dynamics in a brain region called the central complex (CX) enable directed locomotion, sleep, and context- and experience-dependent spatial navigation. We describe the first complete electron microscopy-based connectome of the *Drosophila* CX, including all its neurons and circuits at synaptic resolution. We identified new CX neuron types, novel sensory and motor pathways, and network motifs that likely enable the CX to extract the fly's head direction, maintain it with attractor dynamics, and combine it with other sensorimotor information to perform vector-based navigational computations. We also identified numerous pathways that may facilitate the selection of CX-driven behavioral patterns by context and internal state. The CX connectome provides a comprehensive blueprint necessary for a detailed understanding of network dynamics underlying sleep, flexible navigation, and state-dependent action selection.

**\*For correspondence:**
hulseb@janelia.hhmi.org (BKH);
haberkernh@janelia.hhmi.org
(HH);
franconviller@janelia.hhmi.org
(RF);
turnerevansd@janelia.hhmi.org
(DT-E);
vivek@janelia.hhmi.org (VJ)

†These authors contributed
equally to this work

**Competing interest:** The authors
declare that no competing
interests exist.

**Reviewing Editor:** Ronald L
Calabrese, Emory University,
United States

## Introduction

Flexible, goal-oriented behavior requires combining diverse streams of sensory and internal state information from the present with knowledge gathered from the past to determine context-appropriate patterns of actions into the future. This presents the nervous system with several challenges. For many animals, selecting actions based on sensory input requires the dynamical transformation of information from sensors on one body part into a reference frame suitable for the activation of muscles on another (*Buneo et al., 2002*; *Huston and Jayaraman, 2011*; *Huston and Krapp, 2008*; *Pouget et al., 2002*). When integrating information from different sensors, the brain must also resolve any conflicts that arise between different cues. Further, maintaining goal-oriented behavioral programs over long timescales requires the brain to ignore transient sensory distractions and to compensate for fluctuations in the quality of sensory information or perhaps even its temporary unavailability. Brains are thought to solve such complex computational challenges by relying not just on direct sensory to motor transformations, but also on abstract internal representations (*Moser et al., 2008*). Abstract representations are useful not just in animal brains, but also in artificial agents trained to solve challenging navigational tasks (*Banino et al., 2018*; *Cueva and Wei, 2018*). In the brain, representations that persist in the absence of direct sensory input are thought to rely on attractor dynamics (*Knierim and Zhang, 2012*), which are typically generated by recurrent neural circuits in deep brain regions rather than just those at the sensory and motor periphery. A major challenge in understanding the

dynamics and function of deep brain circuits is that—in contrast to early sensory circuits—their inputs and outputs are usually difficult to identify and characterize. Further, the attractor dynamics (*Knierim and Zhang, 2012*) and vector computations (*Bicanski and Burgess, 2020*) that characterize circuits involved in flexible navigation are thought to rely on structured connectivity between large populations of neurons. This connectivity is difficult to determine, at least in large-brained animals. Insects, with their identified neurons and smaller brains (*Haberkern and Jayaraman, 2016*), present an excellent opportunity to obtain a detailed understanding of how neural circuits generate behavior that unfolds flexibly and over longer timescales.

Insects maintain a specific pattern of action selection over many minutes and even hours during behaviors like foraging or migration, and maintain a prolonged state of inaction during quiet wakefulness or sleep (*Hendricks et al., 2000*; *Shaw et al., 2000*). Both types of behaviors are initiated and modulated based on environmental conditions (e.g., humidity, heat, and the availability of food) and an insect's internal needs (e.g., sleep drive and nutritive state) (*Griffith, 2013*). The context-dependent initiation and control of many such behaviors is thought to depend on a conserved insect brain region called the central complex (CX) (*Figure 1*, *Figure 1—figure supplement 1*; *Helfrich-Forster, 2018*; *Pfeiffer and Homberg, 2014*; *Strauss, 2002*; *Turner-Evans and Jayaraman, 2016*). In *Drosophila*, this highly recurrent central brain region, which is composed of ~3000 identified neurons, enables flies to modulate their locomotor activity by time of day (*Liang et al., 2019*), maintain an arbitrary heading when flying (*Giraldo et al., 2018*) and walking (*Green et al., 2019*; *Turner-Evans et al., 2020*), form short- and long-term visual memories that aid in spatial navigation (*Kuntz et al., 2017*; *Liu et al., 2006*; *Neuser et al., 2008*; *Ofstad et al., 2011*), use internal models of their body size when performing motor tasks (*Krause et al., 2019*), track sleep need and induce sleep (*Donlea et al., 2018*), and consolidate memories during sleep (*Dag et al., 2019*).

The precise role of CX circuits in generating these behaviors is an area of active investigation. Neural activity in the region has been linked to sensory maps and directed actions using electrophysiology in a variety of different insects (*el Jundi et al., 2015*; *Guo and Ritzmann, 2013*; *Heinze and Homberg, 2007*; *Heinze and Reppert, 2011*; *Stone et al., 2017*; *Varga and Ritzmann, 2016*). In the fly, CX neurons have been shown to track the insect's angular orientation during navigation in environments with directional sensory cues and also in their absence (*Fisher et al., 2019*; *Green et al., 2017*; *Kim et al., 2019*; *Okubo et al., 2020*; *Seelig and Jayaraman, 2015*; *Turner-Evans et al., 2017*). Many computational models have been proposed to explain how the CX generates such activity patterns during navigation (*Arena et al., 2013*; *Cope et al., 2017*; *Kakaria and de Bivort, 2017*; *Kim and Dickinson, 2017a*; *Kim et al., 2019*; *Stone et al., 2017*; *Su et al., 2017*; *Turner-Evans et al., 2017*). However, a key untested assumption in most computational and conceptual models of CX function is the connectivity of CX circuits. Connectivity and circuit structure, in turn, can inspire models of function.

Although the anatomy of the CX (*Figure 1A–E*) and the morphology of its neurons have been examined in a wide variety of insects using light-level microscopy (*el Jundi et al., 2018*; *Hanesch et al., 1989*; *Heinze et al., 2013*; *Heinze and Homberg, 2008*, *Heinze and Homberg, 2009*; *Homberg, 2008*; *Lin et al., 2013*; *Omoto et al., 2018*; *Pfeiffer and Homberg, 2014*; *Strausfeld, 1999*; *Williams, 1975*; *Wolff et al., 2015*; *Wolff and Rubin, 2018*; *Young and Armstrong, 2010b*), the synaptic connectivity of CX neurons has mainly been estimated indirectly from the light-level overlap of the bouton-like processes of one neuron and the spine-like processes of another. GFP-reconstitution-across-synaptic-partners (GRASP) (*Xie et al., 2017*) and trans-Tango (*Omoto et al., 2018*), methods that are limited in accuracy and reliability (*Lee et al., 2017*; *Talay et al., 2017*), have also been used to infer synaptic connectivity in the fly CX. Optogenetic stimulation of one candidate neural population and two-photon imaging of the calcium responses of another (*Franconville et al., 2018*) has allowed estimations of coarse functional connectivity within the CX, but this technique currently lacks the throughput to comprehensively determine connectivity at the single-neuron (rather than neuron-type) level, and cannot easily discriminate direct from indirect synaptic connectivity. There have also been efforts to characterize synaptic structure in the CX and associated regions with electron microscopy (EM) in the bee and locust (*Held et al., 2016*; *Homberg and Muller, 2016*), and a combination of coarse-scale and synaptic-resolution EM has been used to infer connectivity in the sweat bee (*Stone et al., 2017*). Most recently, the reconstruction of a small set of CX neurons within a whole-brain volume acquired using transmission electron microscopy (TEM) (*Zheng et al., 2018*) was

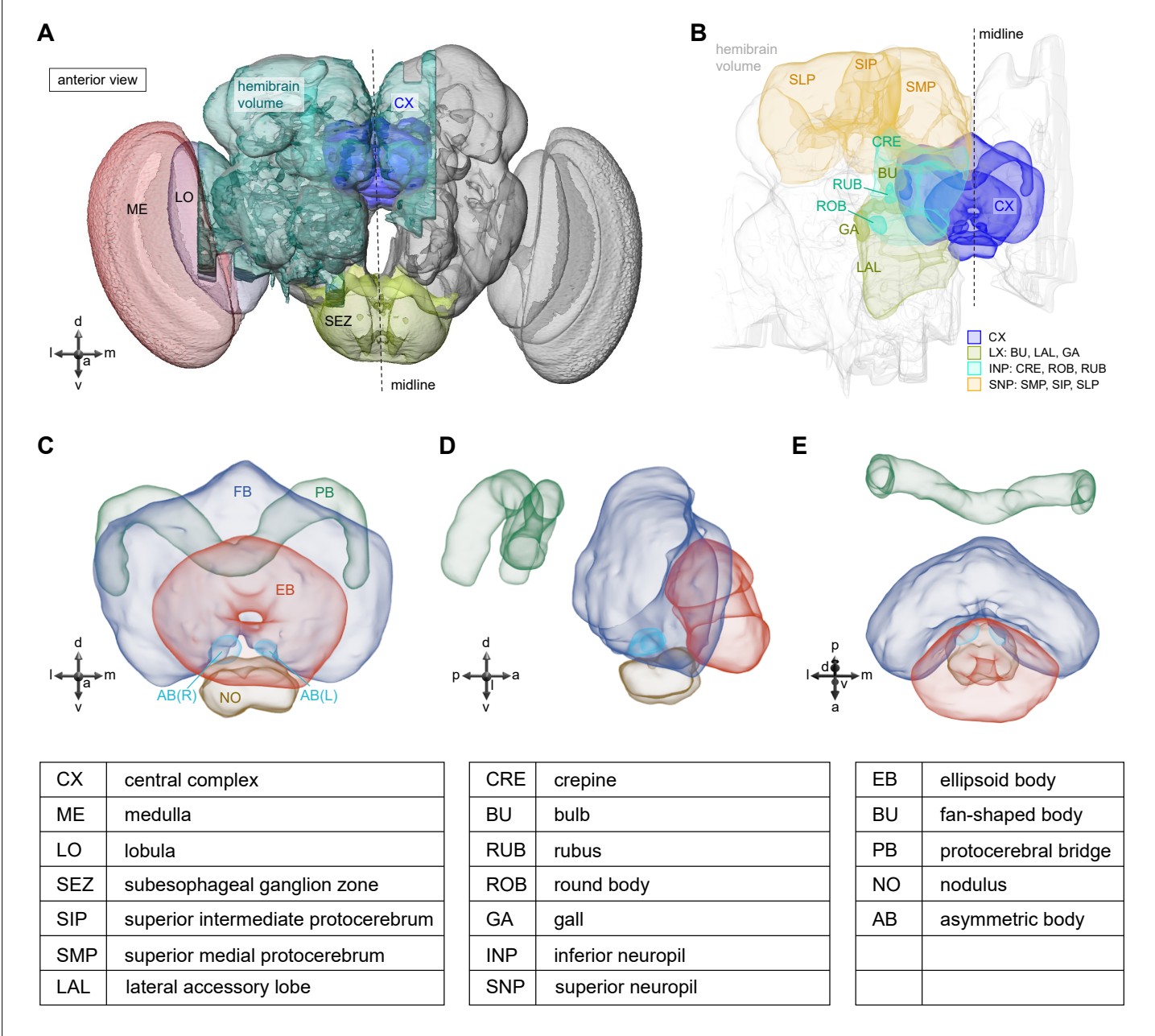

**Figure 1.** The central complex (CX) and accessory brain regions. (**A**) The portion of the central brain (aquamarine) that was imaged and reconstructed to generate the hemibrain volume (*Scheffer et al., 2020*) is superimposed on a frontal view of a grayscale representation of the entire *Drosophila melanogaster* brain (JRC 2018 unisex template [*Bogovic et al., 2020*]). The CX is shown in dark blue. The midline is indicated by the dotted black line. The brain areas LO, ME, and SEZ, which lie largely outside the hemibrain, are labeled. (**B**) A zoomed-in view of the hemibrain volume, highlighting the CX and accessory brain regions. (**C**) A zoomed-in view of the structures that make up the CX, the ellipsoid body (EB), protocerebral bridge (PB), fan-shaped body (FB), asymmetrical body (AB), and paired noduli (NO). (**D**) The same structures viewed from the lateral side of the brain. (**E**) The same structures viewed from the dorsal side of the brain. The table below shows the abbreviations and full names for most of the brain regions discussedin this paper. See (*Scheffer, 2020*) for details. Anatomical axis labels: d: dorsal; v: ventral; l: lateral; m: medial; p: posterior; a: anterior.

The online version of this article includes the following figure supplement(s) for figure 1:

**Figure supplement 1.** The central complex (CX) and additional accessory brain regions.

**Figure supplement 2.** Fan-shaped body (FB) neurons tracts.

**Figure supplement 3.** Main neurite diameter of central complex (CX) neurons.

used to examine the relationship between circuit structure and function in the head direction system (*Turner-Evans et al., 2020*).

Here we analyzed the arborizations and connectivity of the ~3000 CX neurons in version 1.1 of the 'hemibrain' connectome—a dataset with 25,000 semiautomatically reconstructed neurons and 20 million synapses from the central brain of a 5-day-old female fly (*Scheffer et al., 2020*)(see Materials and methods). For most of the analyses, interpretations, and hypotheses in this article, we built on a large and foundational body of anatomical and functional work on the CX in a wide variety of insects. We could link data from different experiment types and insects because many CX neuron types are identifiable across individuals, and sometimes even across species. Thus, it was possible to map results from a large number of CX physiology and behavioral genetics experiments to specific neuron types in the hemibrain connectome.

Not all parts of the dataset have been manually proofread to the same level of completeness (measured as the percentage of synapses associated with neural fragments that are connected to identified cell bodies). In the CX, some substructures were proofread more densely and to a higher level of completion than the others. Comparing the connectivity maps obtained after different extents of proofreading indicated that synaptic connectivity ratios between CX neuron types were largely unchanged by proofreading beyond the level applied to the full CX. This validation step reassured us that our analyses and conclusions were not significantly compromised by the incompleteness of the connectome.

We analyzed the connectome throughout the CX and its accessory regions. We also identified pathways external to the CX that bring sensory input to the region and others that likely carry motor signals out. Further, we discovered multiple levels of recurrence within and across CX structures through pathways both internal and external to the CX. Overall, we found that neural connectivity in the fly's central brain is highly structured. We were able to extract circuit motifs from these patterns of connectivity, which, in turn, allowed us to hypothesize links between circuit structure and function (*Figure 2*, *Figure 2—figure supplement 1*).

We began by identifying multiple, parallel sensory pathways from visual and mechanosensory areas into the ellipsoid body (EB), a toroidal structure within the CX (*Figure 1C*). Neurons within each pathway make all-to-all synaptic connections with other neurons of their type and contact 'compass neurons' known to represent the fly's head direction, creating a potential neural substrate within the EB for extracting orientation information from a variety of environmental cues.

The compass neurons are part of a recurrent subnetwork with a topological and dynamical resemblance to theorized network structures called ring attractors (*Ben-Yishai et al., 1995*; *Skaggs et al., 1995*) that have been hypothesized to compute head direction in the mammalian brain (*Hulse and Jayaraman, 2019*; *Kim et al., 2017c*; *Turner-Evans et al., 2017*; *Turner-Evans et al., 2020*; *Xie et al., 2002*; *Zhang, 1996*). A key connection in the subnetwork is from the compass neurons to a population of interneurons in a handlebar-shaped structure called the protocerebral bridge (PB; *Figure 1C*). The synaptic connectivity profile of the interneurons suggests that they ensure that the head direction representation is maintained in a sinusoidal 'bump' of population activity before it is transferred to multiple types of so-called 'columnar' neurons. In addition to receiving head direction information, many of these columnar neurons likely also receive input related to the fly's self-motion in paired structures known as the noduli (NO; *Figure 1C*; *Currier et al., 2020*; *Lu et al., 2020b*; *Lyu et al., 2020*; *Stone et al., 2017*). The NO input may independently tune the amplitude of sinusoidal activity bumps in the left and right halves of the PB. Some classes of columnar neurons project from the PB back to the EB, while others project to localized areas within a coarsely multilayered and multicolumnar structure called the fan-shaped body (FB; *Figure 1C*). The loosely defined layers and columns of the FB form a rough, two-dimensional grid. FB columnar neurons convey activity bumps from the left and right halves of the PB to the FB. The PB-FB projection patterns of these neurons suggest that their activity bumps in the FB have neuron type-specific phase shifts relative to each other, similar to those observed between EB columnar neurons that are thought to update the head direction representation in the EB (*Green et al., 2017*; *Turner-Evans et al., 2017*; *Turner-Evans et al., 2020*).

Each FB columnar neuron type contacts multiple FB interneuron types, each of whose individual neurons collectively tile the width of the FB. These interneurons have neuronal morphologies that are either 'vertical,' in that they connect different layers within an FB column, or 'horizontal,' in that they extend arbors across intervening columns. These columnar gaps between synaptic connections made

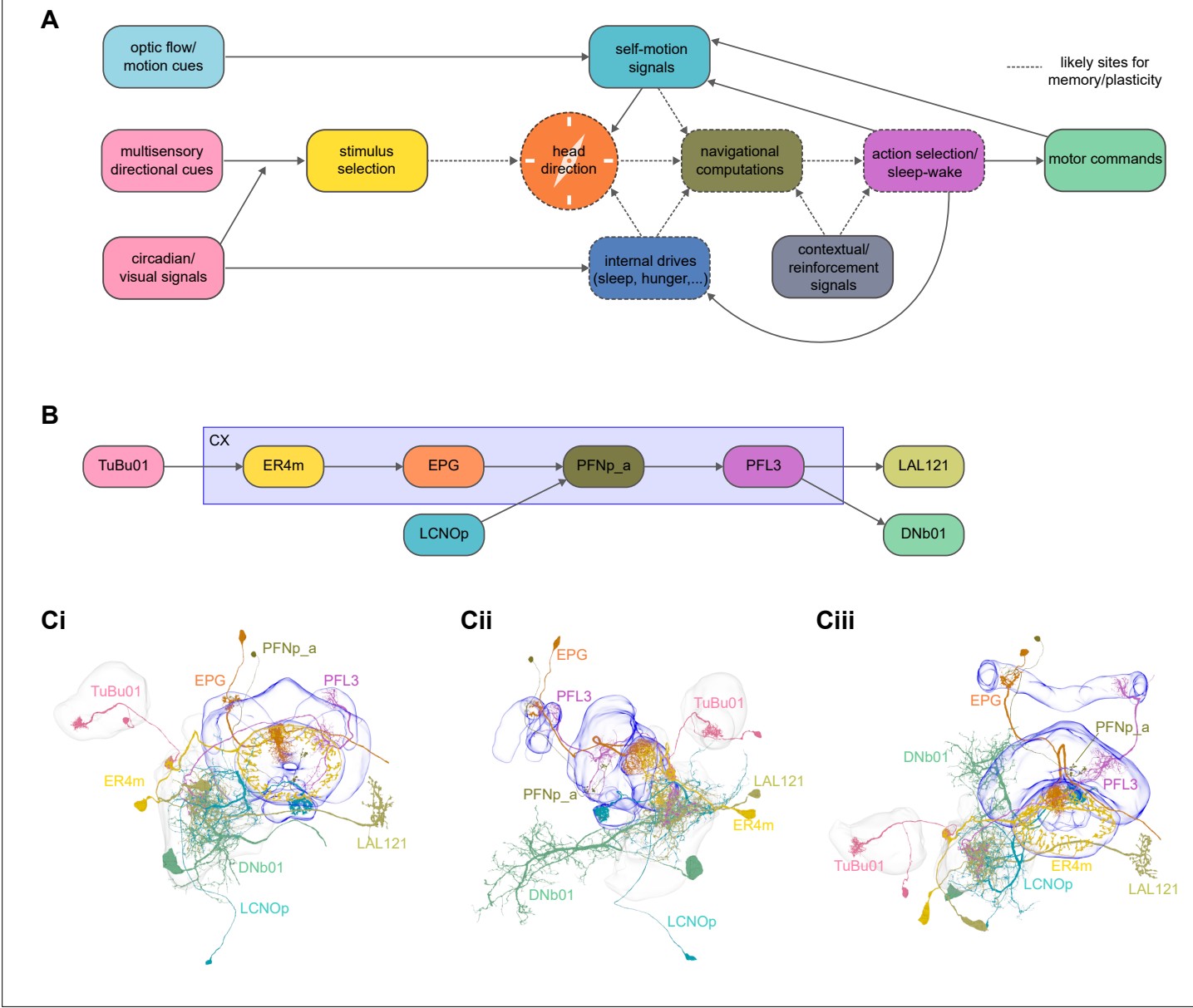

**Figure 2.** High-level schematic and an example sensorimotor pathway through the central complex (CX). (**A**) The CX integrates information from multiple sensory modalities to track the fly's internal drives and its orientation in its surroundings, enabling the fly to generate flexible, directed behavior, while also modulating its internal state. This high-level schematic provides an overview of computations that the CX has been associated with, loosely organized by known modules and interactions. (**B**) A sample neuron type-based pathway going from neuron types that provide information about sensory (here, visual) cues to neuron types within the core CX that generate head direction to self-motion-based modulation of the head direction input and ultimately to action selection through the activation of descending neurons (DNs). The neurons shown here will be fully introduced later in the article. Note that the schematic highlights a small subset of neurons that are connected to each other in a feedforward manner, but the pathway also features dense recurrence and feedback. (**C**) **Ci-iii** show three different views (anterior, lateral, dorsal, respectively) of individual, connected neurons of the types schematized in B.

The online version of this article includes the following figure supplement(s) for figure 2:

**Figure supplement 1.** Selected central complex (CX) input, intra, and output neurons.

by each horizontal interneuron can be interpreted as phase jumps in the context of activity bumps in the FB. The many FB interneuron types form a densely recurrent network with repeating connectivity motifs and phase shifts. Taken together with the self-motion and sinusoidal head direction input that many FB columnar neurons are known to receive in other structures, these motifs and phase shifts seem ideal to perform coordinate transformations necessary for a variety of vector-based navigational

computations. The FB interneurons also provide the major input to canonical CX output types. The phase shifts of the output columnar types between the PB and the FB are well suited to generate goal-directed motor commands based on the fly's current heading (*Stone et al., 2017*).

In addition to the columnar neurons and interneurons, the FB receives layer-specific synaptic inputs from multiple regions, including the superior medial protocerebrum (SMP). Some of these inputs link the FB to a brain region called the mushroom body (MB), which is involved in associative memory and is the subject of a companion manuscript (*Li et al., 2020*). These layer-specific inputs reinforce an existing view of the FB as a center for context-dependent navigational control, a structure that enables different behavioral modules to be switched on and off depending on internal state and external context. The architecture and connectivity of the FB suggest that the region may provide a sophisticated, genetically defined framework for flexible behavior in the fly.

Previous studies have established a role for both the dorsal FB (dFB) and the EB in tracking sleep need and controlling sleep-wake states (reviewed in *Donlea, 2019*). Our analysis of these circuits suggests that there are more putative sleep-promoting neuron types in the dFB than previously reported, many of which form reciprocal connections with wake-promoting dopaminergic neurons (DANs). Furthermore, these dFB neuron types have numerous inputs and outputs, and form bidirectional connections to sleep circuits in the EB. Our results highlight novel neuron types and pathways whose potential involvement in sleep-wake control requires functional investigation. Finally, we identified multiple novel output pathways from both the EB and ventral and dorsal layers of the FB to the lateral accessory lobe (LAL), SMP, crepine (CRE), and posterior slope (PS) (*Figure 1—figure supplement 1*). These regions themselves host networks that project to many brain areas and neuron types that ultimately feed descending neurons (DNs). DNs project to motor centers in the ventral nerve cord (VNC), allowing the CX to exert a wide-ranging influence on behavior, likely well beyond the navigational and orienting behaviors most often associated with the CX. Indeed, the regions that are targeted by the output pathways are associated not just with sensory-guided navigation, but with innate behaviors like feeding and oviposition and with associatively learned behaviors. Another remarkable feature of the CX's output pathways is the large number of collaterals that feed back into the CX at each stage. Such loops could implement various motor control functions, from simple gain adaptation to more complex forms of forward models.

In summary, our analysis revealed remarkable patterning in the connections made by the hundreds of neuron types that innervate different structures of the CX. In the 'Results' section, we describe this patterned connectivity in some detail. In the 'Discussion' section, we synthesize these findings and explore what this patterned circuit structure may imply about function, specifically in the context of vector-based navigation and action selection. Although many readers may prefer to jump directly to the 'Results' section that discusses neuron types, circuits, and brain regions related to their particular research focus, we recommend that the general reader identify the 'Results' section that most interest them by first reading the 'Discussion' section.

## Results

In contrast to many previous EM-based circuit reconstruction efforts, which have relied on sparse, manual tracing of neurons (*Eichler et al., 2017*; *Helmstaedter et al., 2013*; *Turner-Evans et al., 2020*; *Zheng et al., 2018*), the hemibrain connectome was generated using a combination of automatic, machine learning-based reconstruction techniques (*Januszewski et al., 2018*; *Li et al., 2019*; *Scheffer et al., 2020*) and manual proofreading (*Scheffer et al., 2020*). This semiautomatic process allowed us to reconstruct a large fraction of most neurons that project to the CX. This, as we explain below, aided our efforts to classify and name CX neurons. Of course, a complete CX connectome would contain the complete reconstruction of all neurons with processes in the CX, the detection of all their chemical and electrical synapses, and an identification of all pre- and postsynaptic partners at each of those synapses. The resolution of the techniques used to generate the FIBSEM connectome did not permit the detection of gap junctions. Nor does the connectome reveal neuromodulatory connections mediated by neuropeptides, which are known to be prevalent in the CX (*Kahsai and Winther, 2011*), although rapid progress in machine learning methods may soon make this possible (*Eckstein et al., 2020*). Glial cells, which perform important roles in neural circuit function (*Allen and Lyons, 2018*; *Bittern et al., 2021*; *De Pittà and Berry, 2019*; *Ma et al., 2016*; *Mu et al., 2019*), were not segmented. In addition, although the hemibrain volume contains the core structures of

**Table 1.** Brain regions of the central complex contained and defined in the hemibrain. The regions are hierarchical, with the more indented regions forming subsets of the less indented. Reproduced with permission from *Scheffer et al., 2020*.

| CX | Central complex |
|---|---|
| FB | Fan-shaped body |
| FBl1 | Fan-shaped body layer 1 |
| FBl2 | Fan-shaped body layer 2 |
| FBl3 | Fan-shaped body layer 4 |
| FBl4 | Fan-shaped body layer 4 |
| FBl5 | Fan-shaped body layer 5 |
| FBl6 | Fan-shaped body layer 6 |
| FBl7 | Fan-shaped body layer 7 |
| FBl8 | Fan-shaped body layer 8 |
| FBl9 | Fan-shaped body layer 9 |
| EB | Ellipsoid body |
| EBr1 | Ellipsoid body zone r1 |
| EBr2r4 | Ellipsoid body zone r2r4 |
| EBr3am | Ellipsoid body zone r3am |
| EBr3d | Ellipsoid body zone r3d |
| EBr3pw | Ellipsoid body zone r3pw |
| EBr5 | Ellipsoid body zone r5 |
| EBr6 | Ellipsoid body zone r6 |
| AB(R)/(L) | Asymmetrical body |
| PB | Protocerebral bridge |
| PB(R1) | PB glomerulus R1 |
| PB(R2) | PB glomerulus R2 |
| PB(R3) | PB glomerulus R3 |
| PB(R4) | PB glomerulus R4 |
| PB(R5) | PB glomerulus R5 |
| PB(R6) | PB glomerulus R6 |
| PB(R7) | PB glomerulus R7 |
| PB(R8) | PB glomerulus R8 |
| PB(R9) | PB glomerulus R9 |
| PB(L1) | PB glomerulus L1 |
| PB(L2) | PB glomerulus L2 |
| PB(L3) | PB glomerulus L3 |
| PB(L4) | PB glomerulus L4 |
| PB(L5) | PB glomerulus L5 |
| PB(L6) | PB glomerulus L6 |

*Table 1 continued on next page*

*Table 1 continued*

| CX | Central complex |
|---|---|
| PB(L7) | PB glomerulus L7 |
| PB(L8) | PB glomerulus L8 |
| PB(L9) | PB glomerulus L9 |
| NO | Noduli |
| NO1(R)/(L) | Nodulus 1 |
| NO2(R)/(L) | Nodulus 2 |
| NO3(R)/(L) | Nodulus 3 |

the CX—the entire PB, EB, and FB, and both the right and left NO and asymmetrical body (AB) (see *Table 1* for a hierarchy of the named CX brain regions in the volume)—it does not include all brain structures that are connected to the CX. Specifically, for many brain structures associated with the CX that are further from the midline, the hemibrain volume only contains complete structures within the right hemisphere (*Scheffer et al., 2020*). Thus, the hemibrain does not contain most of the LAL, CRE, wedge (WED), vest (VES), gall (GA), and bulb (BU) from the left hemisphere, and CX neurons whose arbors extend into these excluded brain structures are necessarily cut off at the borders of the hemibrain volume (their status is indicated in the connectome database, see Materials and methods). Even for structures within the right hemisphere, such as the LAL, CRE, and PS, some neural processes that connect these structures to each other were not captured in the volume, sometimes making it impossible to assign the orphaned arbors to known neurons. Nevertheless, we were able to identify the vast majority of CX neurons and many neurons in accessory regions as well.

## CX neuron classification and nomenclature

Historically, *Drosophila* CX neurons have been typed and named by using data from light microscopy (LM) (*Hanesch et al., 1989*; *Lin et al., 2013*; *Omoto et al., 2018*; *Wolff et al., 2015*; *Wolff and Rubin, 2018*; *Young and Armstrong, 2010b*). Light-level data, often acquired from GAL4 lines that genetically target small numbers of neuron types (*Jenett et al., 2012*; *Nern et al., 2015*; *Pfeiffer et al., 2010*; *Wolff et al., 2015*; *Wolff and Rubin, 2018*), reveal neuronal morphology in sufficient detail to classify neurons into types and subtypes. However, only neurons targeted by existing genetic lines can be identified through

light-level data, and morphologically similar neuron types can be hard to distinguish. For example, the PEN1 and PEN2 types (now called PEN_a and PEN_b; see Materials and methods), which seem identical at the light level (*Wolff et al., 2015*), were initially differentiated by their functional properties (*Green et al., 2017*). The hemibrain dataset provides much higher resolution maps of CX neurons than light-level data and enables a distinction to be drawn based on connectivity between neuron types that are morphologically identical in light-level samples. EM reconstructions confirm that PEN_a and PEN_b neurons are indeed strikingly different in their synaptic connectivity (*Scheffer et al., 2020*; *Turner-Evans et al., 2020*).

However, LM data offer far greater numbers of neurons per neuron type across different brains. To date, just this one fly CX has been densely reconstructed and analyzed at synaptic resolution with EM. Thus, for neuron types that are represented by just one neuron per hemibrain, at best two fully traced neurons are available for analysis (one per side), and only one if the arbor of the second extends outside the hemibrain volume. The inherent variability of individual neurons of a single type is not yet clear; for example, the LCNO neurons discussed in the NO section are known from LM to arborize in the CRE (*Wolff and Rubin, 2018*), but do not arborize in the region in this sample. It is possible that arbors in some neuropils exhibit a greater degree of variability than other neuron types; for example, the number and length of branches in a larger neuropil such as the LAL could be less tightly regulated than in a small neuropil, leading to greater variability in morphology and perhaps also in synaptic connections with upstream or downstream partners in that neuropil. Notably, the hemibrain dataset does not contain the previously identified 'canal' cell, an EB-PB columnar neuron type (*Wolff and Rubin, 2018*). The absence of this neuron type may be a developmental anomaly of the particular fly that was imaged for the hemibrain dataset or may hint at broader developmental differences across different wild-type and Gal4 lines.

The full spectrum of the morphology of a neuron type will likely not be verified until multiple single neurons are analyzed in a split-GAL4 line known to target just that one neuron type or until multiple brains are analyzed at EM resolution. In the meantime, EM and LM datasets contribute both partially overlapping as well as unique anatomical insights useful for classifying and naming neurons. Both datasets were therefore used to assign names to previously undescribed neurons of the CX, the overwhelming majority of which are FB neurons (see *Tables 2* and *3* for all new CX neuron types, with numbers for each type, and *Figure 1—figure supplement 2* for the positions of those new types in known FB fiber tracts; see *Table 4* for numbers of different EB, PB, and NO neuron types). All CX neurons have been given two names, a short name that is convenient for searching databases and that is used as a shorthand abbreviation throughout this article, and a longer name that provides sufficient anatomical insight to capture the overall morphology of the neuron. The long anatomical names have their roots in both the EM and LM datasets, emphasize overall morphology, and attempt to define neuron types based on features that we anticipate can be distinguished at the light level, ultimately in split-GAL4 lines (see details in Materials and methods). The short names are derived primarily from hemibrain connectivity information. Our overall method for connectivity-based neuron-type classification of CX neurons was described in *Scheffer et al., 2020*, but see Materials and methods for a short summary. Finally, to facilitate comparisons with neurons of other insect species, *Figure 1—figure supplement 3* provides the median diameter of the main neurite for each CX neuron type in the dataset.

## Validation of CX connectome

The manual proofreading procedure we used is labor-intensive and time-consuming. For this reason, it was not performed to the same extent on the entire connectome. For example, completeness within the core CX structures is generally higher than completeness within CX-associated regions (*Scheffer et al., 2020*), and completeness also differed for different CX regions. We therefore performed a series of validation analyses to examine how such differences in completeness might affect estimates of connectivity. In particular, we examined how connectivity estimates might be affected by the percentage of synapses that are assigned to known neuronal bodies rather than to unidentified neural fragments (partially reconstructed neural processes) within a given region. This analysis was performed both on the EB, comparing two different stages in the proofreading process, and in the PB and FB, comparing symmetric regions proofread to different levels of completion.

**Table 2.** Identified fan-shaped body (FB) tangential neuron types and the number of each type.

| Short | Long name | Right | Left | Short | Long | R | L | Short | Long | R | L |
|---|---|---|---|---|---|---|---|---|---|---|---|
| FB1A | SMPSIPFB1,3 | 2 | 2 | FB2A | NOaLALFB2 | 2 | 2 | FB3A | LALNO2FB3 | 2 | 2 |
| FB1B | SMPSLPFB1d | 2 | 2 | FB2B_a | LALCREFB2_1 | 2 | 2 | FB3B | EBCREFB3 | 1 | 1 |
| FB1C | LALNOmFB1 | 2 | 2 | FB2B_b | LALCREFB2_1 | 2 | 2 | FB3C | LALSMPFB3 | 4 | 5 |
| FB1D | SLPFB1d | 2 | 2 | FB2C | SMPCREFB2_1 | 3 | 3 | FB3D | LALCREFB3 | 1 | 1 |
| FB1E_a | SIPSMPFB1d | 2 | 2 | FB2D | LALCREFB2_2 | 3 | 3 | FB3E | SMPLALFB3 | 1 | 1 |
| FB1E_b | SLPSIPFB1d | 1 | 1 | FB2E | SCLSMPFB2 | 2 | 2 | | | | |
| FB1F | SMPSIPFB1d | 1 | 1 | FB2F_a | SIPSMPFB2 | 3 | 3 | | | | |
| FB1G | SMPSIPFB1d,3 | 1 | 1 | FB2F_b | SIPSMPFB2 | 3 | 3 | | | | |
| FB1H | CRENO2,3FB1-4 | 1 | 1 | FB2F_c | SIPSMPFB2 | 2 | 2 | | | | |
| FB1I | SMPSIPFB1d,7 | 1 | 1 | FB2G_a | SMPSIPFB2 | 1 | 1 | | | | |
| FB1J | SLPSIPFB1,7,8 | 1 | 1 | FB2G_b | SIPLALFB2 | 2 | 2 | | | | |
| | | | | FB2H_a | SIPSCLFB2 | 1 | 1 | | | | |
| | | | | FB2H_b | SIPSCLFB2 | 1 | 1 | | | | |
| | | | | FB2I_a | SMPATLFB2 | 5 | 4 | | | | |
| | | | | FB2I_b | SMPATLFB2 | 1 | 1 | | | | |
| | | | | FB2J | SMPPLPFB2 | 2 | 3 | | | | |
| | | | | FB2K | LALSMPFB2 | 3 | 3 | | | | |
| | | | | FB2L | SMPCREFB2_2 | 1 | 1 | | | | |
| | | | | FB2M | SIPCREFB2 | 3 | 3 | | | | |

| Short | Long name | Right | Left | Short | Long | R | L | Short | Long | R | L |
|---|---|---|---|---|---|---|---|---|---|---|---|
| FB4A | CRESMPFB4_1 | 4 | 4 | FB5A | LALCREFB5 | 2 | 2 | FB6A | SMPSIPFB6_1 | 3 | 3 |
| FB4B | NO2LALFB4 | 1 | 1 | FB5B | SMPSIPFB5d_1 | 3 | 3 | FB6B | SMPSIPFB6_2 | 1 | 2 |
| FB4C | CRENO2FB4_1 | 1 | 1 | FB5C | SMPCREFB5_1 | 1 | 1 | FB6C_a | SIPSMPFB6_1 | 1 | 1 |
| FB4D | CRESMPFB4_2 | 3 | 3 | FB5D | CRESMPFB5_1 | 1 | 1 | FB6C_b | SIPSMPFB6_1 | 2 | 3 |
| FB4E | CRELALFB4_1 | 5 | 6 | FB5E | CRESMPFB5_2 | 1 | 1 | FB6D | SMPFB6 | 1 | 1 |
| FB4F_a | CRELALFB4_2 | 5 | 4 | FB5F | SMPCREFB5_2 | 1 | 1 | FB6E | SIPSMPFB6_2 | 1 | 1 |
| FB4F_b | CRELALFB4_2 | 1 | 1 | FB5G | SMPSIPFB5,6 | 4 | 4 | FB6F | SMPSIPFB6_3 | 1 | 1 |
| FB4G | CRELALFB4_3 | 1 | 1 | FB5H | CRESMPFB5_3 | 1 | 1 | FB6G | SIPSMPFB6_3 | 1 | 1 |
| FB4H | CRELALFB4_4 | 1 | 1 | FB5I | SMPCREFB5_3 | 1 | 1 | FB6H | SMPSIPFB6_4 | 1 | 1 |
| FB4I | LALCREFB4 | 1 | 1 | FB5J | SMPFB5 | 1 | 1 | FB6I | SMPSIPFB6_5 | 1 | 1 |
| FB4J | CRELALFB4_5 | 1 | 1 | FB5K | CREFB5 | 1 | 1 | FB6J | FB6_1 | 4 | 4 |
| FB4K | CRESMPFB4_3 | 2 | 2 | FB5L | CRESMPFB5_4 | 1 | 1 | FB6K | SMPSIPFB6_6 | 2 | 2 |
| FB4L | LALSIPFB4 | 2 | 2 | FB5M | CRESMPFB5_5 | 1 | 1 | FB6L | FB6_2 | 3 | 3 |
| FB4M | CRENO2FB4_2 | 2 | 2 | FB5N | SMPCREFB5_4 | 1 | 1 | FB6M | WEDLALFB6 | 2 | 2 |
| FB4N | SMPCREFB4 | 1 | 1 | FB5O | SMPCREFB5_5 | 1 | 1 | FB6N | CRESMPFB6_1 | 1 | 1 |
| FB4O | CRESMPFB4d | 2 | 2 | FB5P | SMPCREFB5_6 | 2 | 2 | FB6O | SIPSMPFB6_4 | 1 | 1 |
| FB4P_a | CRESMPFB4_4 | 2 | 2 | FB5Q | SMPCREFB5d | 2 | 2 | FB6P | SMPCREFB6_1 | 1 | 1 |
| FB4P_b | CRESMPFB4_4 | 4 | 3 | FB5R | FB5 | 3 | 3 | FB6Q | SIPSMPFB6_5 | 1 | 1 |

*Table 2 continued on next page*

*Table 2 continued*

| Short | Long name | Right | Left | Short | Long | R | L | Short | Long | R | L |
|---|---|---|---|---|---|---|---|---|---|---|---|
| FB4Q_a | CRESMPFB4_5 | 1 | 1 | FB5S | FB5d,6v | 3 | 4 | FB6R | SMPSIPFB6_7 | 2 | 1 |
| FB4Q_b | CRESMPFB4_5 | 2 | 2 | FB5T | CRESMPFB5_6 | 1 | 1 | FB6S | SIPSMPFB6_6 | 3 | 3 |
| FB4R | CREFB4 | 1 | 3 | FB5U | FB5d | 2 | 1 | FB6T | SIPSMPFB6_7 | 2 | 2 |
| FB4X | CRESIPFB4,5 | 1 | 1 | FB5V | CRELALFB5 | 9 | 9 | FB6U | SMPCREFB6_2 | 2 | 2 |
| FB4Y | EBCREFB4,5 | 2 | 2 | FB5W | SMPCREFB5_7 | 4 | 4 | FB6V | SMPCREFB6_3 | 1 | 1 |
| FB4Z | FB4d5v | 8 | 8 | FB5X | SMPCREFB5_8 | 3 | 3 | FB6W | CRESMPFB6_2 | 1 | 1 |
| | | | | FB5Y | SMPSIPFB5d_2 | 2 | 2 | FB6X | SMPCREFB6_4 | 1 | 1 |
| | | | | FB5Z | SMPCREFB5_9 | 2 | 2 | FB6Y | SMPSIPFB6_8 | 1 | 1 |
| | | | | FB5AA | SMPCREFB5_10 | 1 | 1 | FB6Z | SMPSIPFB6_9 | 1 | 1 |
| | | | | FB5AB | SIPCREFB5d | 1 | 1 | | | | |

| Short | Long name | Right | Left | Short | Long | R | L | Short | Long | R | L |
|---|---|---|---|---|---|---|---|---|---|---|---|
| FB7A | SIPSLPFB7 | 3 | 3 | FB8A | SLPSMPFB8_1 | 3 | 2 | FB9A | SLPFB9_1 | 3 | 3 |
| FB7B | SMPSLPFB7 | 1 | 1 | FB8B | PLPSLPFB8 | 2 | 2 | FB9B_a | SLPFB9_2 | 2 | 2 |
| FB7C | SMPSIPFB7_1 | 1 | 2 | FB8C | SMPFB8 | 2 | 2 | FB9B_b | SLPFB9_2 | 1 | 2 |
| FB7D | FB7,6 | 2 | 2 | FB8D | SLPSMPFB8_2 | 1 | 1 | FB9B_c | SLPFB9_2 | 2 | 2 |
| FB7E | SMPSIPFB7_2 | 3 | 3 | FB8E | SMPSIPFB8_1 | 3 | 2 | FB9B_d | SLPFB9_2 | 2 | 2 |
| FB7F | SMPSIPFB7_3 | 1 | 1 | FB8F_a | SIPSLPFB8 | 4 | 4 | FB9B_e | SLPFB9_2 | 2 | 2 |
| FB7G | SMPFB7,8 | 2 | 2 | FB8F_b | SIPSLPFB8 | 4 | 4 | FB9C_a | SLPFB9_2 | 2 | 2 |
| FB7H | SMPFB7 | 1 | 1 | FB8G | SMPSIPFB8_2 | 3 | 3 | FB9C_b | SLPFB9_2 | 2 | 2 |
| FB7I | SMPSIPFB7,6 | 2 | 3 | FB8H | SMPSLPFB8 | 3 | 3 | | | | |
| FB7J | FB7,8 | 2 | 2 | FB8I | SMPSIPFB8_3 | 2 | 2 | | | | |
| FB7K | SLPSIPFB7 | 2 | 2 | | | | | | | | |
| FB7L | SMPSIPFB7_4 | 2 | 2 | | | | | | | | |
| FB7M | SMPSIPFB7_5 | 1 | 1 | | | | | | | | |

## Neuron-to-neuron connectivity before and after dense proofreading for the same neurons

To assess how sensitive connectivity estimates are to the completeness level of the tracing, we compared both the pre- and postsynaptic connectivity of a selection of neurons (see Materials and methods) arborizing in a specific brain structure, the EB, before and after it was subjected to focused proofreading (*Figure 3A*). Proofreading increased the number of pre- and postsynapses for which the synaptic partner could now be identified (*Figure 3B*), mainly by merging small dendritic fragments onto known neurons. However, the *relative* synaptic contributions that each neuron received from its various partners remained largely the same before and after dense proofreading (example fit for the EPG neuron of *Figure 3A* is shown in *Figure 3C*, the slopes from regression analyses of all neurons are shown in *Figure 3D*). Indeed, with the exception of some FB neurons with minor projections in the EB (green points in *Figure 3D*, right), the input and output connectivity of most neurons did not significantly change when expressed as relative weights (*Figure 3D*, see also *Figure 3—figure supplement 1*).

**Table 3.** Identified intrinsic columnar neuron types of the fan-shaped body (FB) and ellipsoid body (EB).

The types include hΔ and vΔ neuron types (total 598 neurons), and the columnar projection neurons, FR, FC, FS, and EL (total 585 cells) neuron types.

| Cell types | | # cells | Types | | # cells | Types | | # cells |
|---|---|---|---|---|---|---|---|---|
| hΔA | FB4D5FB4 | 12 | vΔA_a | AF | 54 | FR1 | FB2-5RUB | 18 |
| hΔB | FB3,4vD5FB3,4v | 19 | vΔA_b | FB1D0FB8 | 31 | FR2 | TBD | 18 |
| hΔC | FB2,6D7FB6 | 20 | vΔB | FB1D0FB7_1 | 32 | EL | EBGAs | 18 |
| hΔD | FB1,8D3FB8 | 8 | vΔC | FB1D0FB7_2 | 28 | FC1A | FB2CRE_1 | 16 |
| hΔE | FB1,7D3FB7 | 8 | vΔD | FB1D0FB6 | 18 | FC1B | FB2CRE_2 | 18 |
| hΔF | FB1,6d,7D2FB6,7 | 8 | vΔE | FB1,2,3D0FB6v | 23 | FC1C | FB2CRE_3 | 33 |
| hΔG | FB2,3,5d6vD3FB6v | 8 | vΔF | FB1,2,3D0FB5d | 12 | FC1D | FB2CRE_4 | 20 |
| hΔH | FB2d,4D3FB5 | 8 | vΔG | FB1,2D0FB5d | 15 | FC1E | FB2CRE_5 | 20 |
| hΔI | FB2,3,4,5D5FB4,5v | 18 | vΔH | FB1,2D0FB5 | 17 | FC1F | FB2CRE_6 | 17 |
| hΔJ | FB1,2,3,4D5FB4,5 | 29 | vΔI | FB1D0FB5 | 25 | FC2A | FB1-5CRE | 18 |
| hΔK | EBFB3,4D5FB6 | 31 | vΔJ | FB1D0FB5v | 11 | FC2B | FB1d,3,5,6CRE | 33 |
| hΔL | FB2,6D5FB6d | 12 | vΔK | FB1vD0FB4d5v | 46 | FC2C | FB1d,3,6,7CRE | 37 |
| hΔM | FB2,4D3FB5 | 9 | vΔL | FB1vD0FB4 | 38 | FC3 | FB2,3,5,6CRE | 42 |
| | | | vΔM | FB1vD0FB4 | 58 | FS1A | FB2-6SMPSMP | 44 |
| | | | | | | FS1B | FB2,5,SMPSMP | 24 |
| | | | | | | FS2 | FB3,6SMP | 32 |
| | | | | | | FS3 | FB1d,3,6,7SMP | 67 |
| | | | | | | FS4A | FB3,8ABSMP | 57 |
| | | | | | | FS4B | FB2,8ABSMP | 37 |
| | | | | | | FS4C | FB2,6,7SMP | 16 |

## Neuron-to-neuron connectivity at different completion percentages within the same brain region

In addition to comparing connectivity for the same neurons at different completion levels, we also examined connectivity in different subregions that were proofread to differing levels of completeness. Parts of the PB (glomeruli L4 and R3) and FB (the third column) were intentionally proofread to a denser level of completeness than other areas within those structures. In addition, although we did not perform an analysis of these differences, the right NO was more densely proofread than the left NO (*Scheffer et al., 2020*). These differences in level of completeness must be kept in mind when interpreting synapse counts in these regions.

In the PB, we compared connectivity within the densely proofread R3 and L4 glomeruli to their less densely proofread mirror-symmetric glomeruli, L3 and R4, respectively. For a meaningful comparison, we focused our connectivity analysis on neurons with arbors restricted to single PB glomeruli (*Figure 4Ai*, e.g., shows examples of PFNa neurons that innervate L3 and R3, respectively). The impact of dense proofreading in the PB was evident in the increased percentage of synapses from or to identified partners of neurons of the same type in the densely proofread glomeruli, L4 and R3 (*Figure 4Aii*, for presynaptic partners, left, and postsynaptic partners, right). However, as with the EB comparison before and after dense proofreading (*Figure 3*), we found that the relative contributions of different identified partners onto the selected types remained nearly unchanged across the two glomeruli (see *Figure 4Aiii* and Aiv for an example regression), both for inputs and outputs. As a control, we also performed the same analyses on two additional pairs of mirror-symmetric PB glomeruli, L5 and R5,

**Table 4.** Identified neuron types of the ellipsoid body (EB), protocerebral bridge (PB), and noduli (NO).

| Cell types | # cells | Cell types | # cells | Cell types | # cells |
|---|---|---|---|---|---|
| ER1_a | 16 | ExR1 | 4 | Delta7 | 42 |
| ER1_b | 14 | ExR2 | 4 | IbSpsP | 24 |
| ER2_a | 9 | ExR3 | 2 | LPsP | 2 |
| ER2_b | 6 | ExR4 | 2 | P1-9 | 2 |
| ER2_c | 21 | ExR5 | 4 | P6-8P9 | 4 |
| ER2_d | 6 | ExR6 | 2 | SpsP | 4 |
| ER3a_a | 12 | ExR7 | 4 | | |
| ER3a_b | 4 | ExR8 | 4 | Cell types | # cells |
| ER3a_c | 4 | | | PFGs | 18 |
| ER3a_d | 6 | Cell types | # cells | PFL1 | 14 |
| ER3d_a | 12 | | | PFL2 | 12 |
| ER3d_b | 10 | EL | 18 | PFL3 | 24 |
| ER3d_c | 12 | EPG | 46 | PFNa | 58 |
| ER3d_d | 10 | EPGt | 4 | PFNd | 40 |
| ER3m | 18 | PEG | 18 | PFNm_a | 26 |
| ER3p_a | 12 | PEN_a(PEN1) | 20 | PFNm_b | 18 |
| ER3p_b | 6 | PEN_b(PEN2) | 22 | PFNp_a | 60 |
| ER3w_a | 9 | | | PFNp_b | 115 |
| ER3w_b | 11 | Cell types | # cells | PFNp_c | 46 |
| ER4d | 25 | | | PFNp_d | 33 |
| ER4m | 10 | GLNO | 4 | PFNp_e | 21 |
| ER5 | 20 | LCNOp | 2 | PFNv | 20 |
| ER6 | 4 | LCNOpm | 2 | PFR_a | 29 |
| | | LNO1 | 4 | PFR_b | 16 |
| | | LNO2 | 2 | | |
| | | LNO3 | 1 | | |
| | | LNOa | 2 | | |

and L6 and R6 (*Figure 4—figure supplement 1*), finding no more differences in relative contributions than we found when comparing the L3-R3 and L4-R4 pairs.

Finally, we also selected a columnar region within the medial part of the FB ('C3') for focused proofreading (*Figure 4Bi*). As with the EB and PB, this process led to an increase in the percentage of synapses with identified partners (*Figure 4Bii*) without significant changes in relative connectivity when compared to other columns of the FB (*Figure 4Biv*). Note that correlations in connectivity in the FB are lower than those observed in the PB (compare *Figure 4Aiv* with *Figure 4Biv*). We believe this to be the result of both the lack of clear columnar definition in the FB and of true inhomogeneities across vertical sections of the FB, as we discuss further in the FB section.

## Assessing the relative importance of different synaptic inputs

Morphologically, many fly neurons feature a single process that emanates from the soma, which usually sits near the brain surface. This process then sends branches out into multiple brain structures

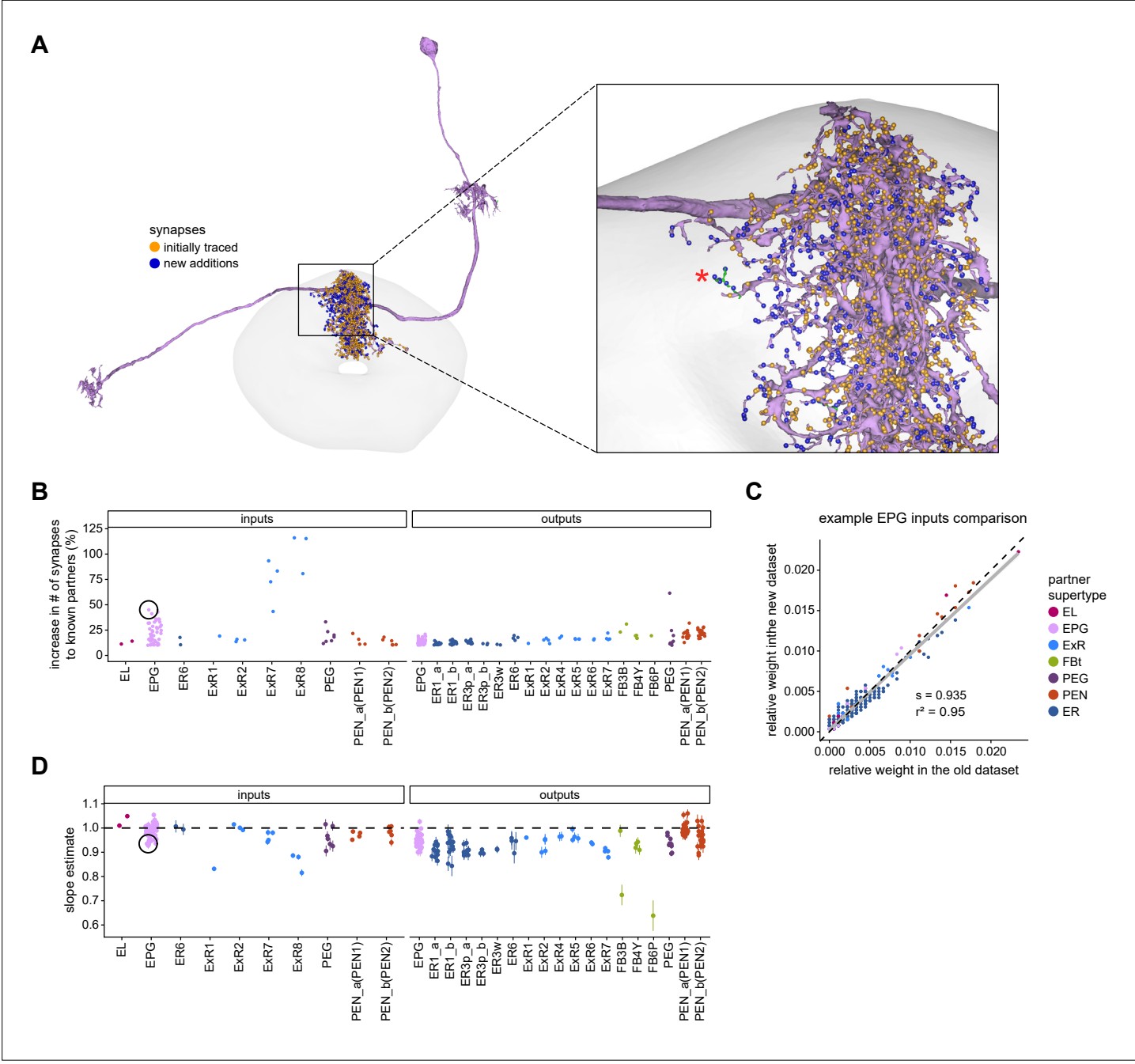

**Figure 3.** Quantitative impact of different levels of proofreading on neuronal connectivity in the ellipsoid body (EB). (**A**) Morphological rendering of an example EPG neuron before and after dense tracing in the EB. Inset, zoomed-in view of part of the EPG arbors highlighting changes resulting from dense reconstruction. The neuron segmentation is in pink. One newly added fragment is colored in green and marked with a red star. Synapses to neurons that were initially identified are in orange. Synapses to neurons that were identified after dense tracing are in blue. These new additions often resulted from joining previously unidentified fragments to their parent neurons, which partner with the example EPG neuron. (**B**) Change in the number of input synapses from known neurons (left panel) and output synapses to known neurons made with selected EB neurons after dense tracing. Each neuron in this subset had at least 200 presynaptic sites in the EB for the left panel, 200 postsynaptic sites in the EB for the right panel, and at least a 10% change in known synapse numbers after dense tracing. The EB neurons are ordered by type and colored by supertype (see Materials and methods). Each colored dot represents a single neuron of the type indicated. Throughout, we analyze input and output connectivity separately. The example neuron shown in (**A**) is circled in black. (**C**) Comparison of the input connectivity of the neuron shown in (**A**) before and after dense tracing. Each point is the relative weight of a connection between that EPG and a single other neuron. Relative weight refers to the fraction of the inputs that comes from the given partner (see Materials and methods). The color denotes the type of the partner neuron. The gray line is a linear fit with 95% confidence

*Figure 3 continued on next page*

*Figure 3 continued*

intervals (the confidence interval is too small to be seen). The dashed line is the identity line. (**D**) Slope of the linear fits (similar to the one in **C**) with 95% confidence intervals for all neurons considered. Many confidence intervals are too small to be seen. The example shown in (**A**) is circled in black.

The online version of this article includes the following figure supplement(s) for figure 3:

**Figure supplement 1.** Influence of the amount of change from tracing on fit results.

or substructures. Although there is sometimes one compartment with mainly postsynaptic specializations, the other compartments are typically 'mixed,' featuring both pre- and postsynaptic specializations. This heterogeneity and compartmentalization make it challenging to compare the relative weight of different synaptic inputs to the neuron's synaptic output. Even for fly neurons that spike, action potential initiation sites are largely unknown (although see *Gouwens and Wilson, 2009*; *Ravenscroft et al., 2020*). Furthermore, spiking neurons may perform local circuit computations involving synaptic transmission without action potentials.

In this study, we will, as a default, analyze the relative contributions of different presynaptic neurons separately for different neuropils. In some cases, we will assume a polarity for neurons based on compartments in which they are mainly postsynaptic. For spiking neurons, these 'dendritic' areas would be expected to play a more significant role in determining the neuron's response, even if the neuron displays mixed pre- and postsynaptic specializations in other compartments. Consider, for example, this rule applied to a much-studied olfactory neuron, the projection neuron (PN). PNs receive most of their inputs in the antennal lobe (AL) and project to regions like the MB and lateral horn (LH), where they have mixed terminals. Our rule would lead to the AL inputs being evaluated separately and being considered stronger contributors to a PN's spiking outputs than any synaptic inputs in the MB and LH. As discussed further in a subsequent section, we will use this logic to define the 'modality' of most ring neurons, which innervate the BU and EB, by the anterior optic tubercle (AOTU) input they receive in the BU rather than by the inputs they receive in the EB. This logic also applies to the tangential neurons of the FB, which receive inputs mainly in regions outside the CX and have mixed terminals inside the FB. Note that some CX neuron types may not rely on spiking at all, and that our assumptions may not apply to such graded potential neurons. The situation is also somewhat different for some interneuron types, such as the PB-intrinsic Δ7 neurons, which have multiple arbors with postsynaptic specializations.

CX synapses are not all of the 'T-bar' type that is most common in the insect brain (*Frohlich, 1985*; *Meinertzhagen, 1996*; *Trujillo-Cenoz, 1969*). As discussed later, several CX neurons make 'E-bar' synapses (*Shaw and Meinertzhagen, 1986*; *Takemura et al., 2017a*), which we do not treat any differently in analysis. In addition, although many synapses are polyadic, with a single presynaptic neuron contacting multiple postsynaptic partners (Materials and methods, *Figure 1A*), it is notable that some neurons, such as ring neurons, make rarer convergent synapses in which multiple presynaptic ring neurons contact a single postsynaptic partner (Materials and methods, *Figure 1B*; *Martin-Pena et al., 2014*). The function of such convergences is, at present, unknown.

## Visual, circadian, mechanosensory, and motor pathways into the EB

All CX neuropils receive input from other parts of the brain. While there is overlap between the regions that provide input to different CX neuropils, each CX neuropil has a distinct set of input regions (*Figure 5A*, *Figure 5—figure supplement 1*). For instance, the EB receives input primarily from the BU, LAL and GA, and to a lesser extent from the CRE, the inferior bridge (IB), and the superior neuropils. The NO also receives inputs from the LAL, GA, and CRE, but gets additional inputs from the wedge (WED), epaulette (EPA), and vest (VES). To assess the information transmitted to the CX by different input pathways, we traced, when possible, these pathways back to their origin or 'source.' These source neurons were grouped into classes associated with particular brain regions, functions, or neuromodulators (legend in *Figure 5B*; also see Appendix 1—table 6 in *Scheffer et al., 2020*). We found that the CX receives inputs originating from a variety of neuron types, including visual projection neurons (vPNs), AL neurons, fruitless (Fru) neurons, MBON neurons, and neuromodulatory neurons (*Figure 5B*). Although the hemibrain volume did not permit us to trace pathways completely from the sensory periphery all the way into the CX, we tried to identify as many inputs as possible using previous results from light microscopy as our guide. We will begin by describing components of

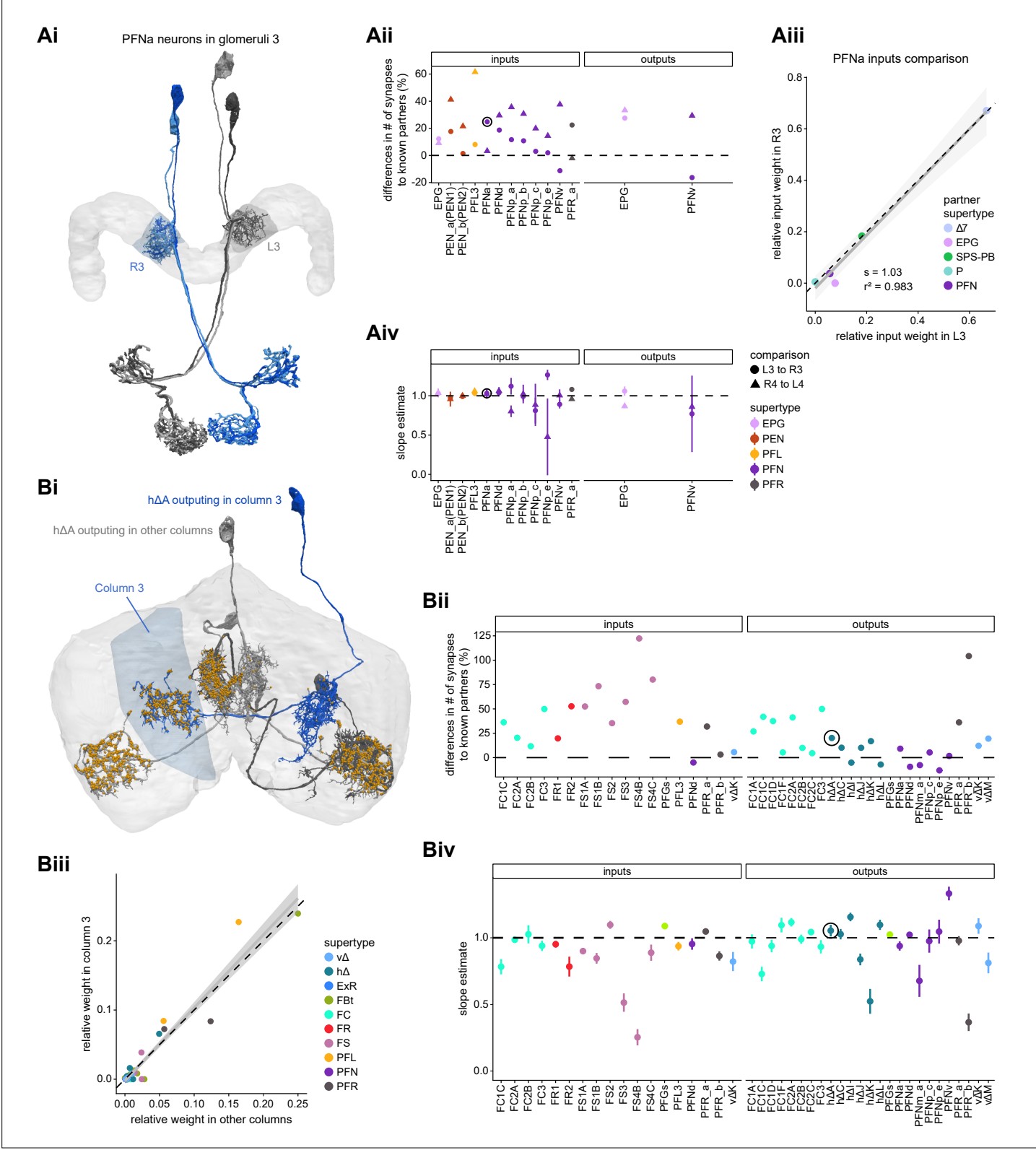

**Figure 4.** Differences in connectivity between compartments at different levels of tracing. (**A**) Differences in connectivity between mirror-symmetric protocerebral bridge (PB) glomeruli. We compare glomeruli that are densely proofread (L4/R3) or not (R4/L3). R or L refer to the right or left half of the PB, respectively. Each half of the PB is made up of nine distinct glomeruli, with glomerulus 1 the most medial and glomerulus 9 the most lateral. (**Ai**) Sample PFNa neurons that each arborize in a single PB glomerulus. Two arborize in L3, and the other two in its mirror symmetric glomerulus, the

*Figure 4 continued on next page*

Figure 4 continued

densely proofread PB glomerulus R3. (**Aii**) Percentage increase in input connectivity (left) and output connectivity (right) to known partners for neuron types innervating single glomeruli between R4 and L4 or L3 and R3. Types were selected if they had neuron instances that innervate all four of these glomeruli, with each instance having at least an average of 20 synapses per glomerulus and at least 80% of their PB synapses in the given glomerulus. For a given type, circles denote the L3-to-R3 comparison and triangles the R4-to-L4 comparison. Few output comparisons can be made because most columnar neurons mainly receive input in the PB. (**Aiii**) Comparison of input connectivity for the type shown in (**Ai**) in R3 and L3. Each point is the relative weight of a connection between that type and another neuron type. The color denotes the supertype of the partner. The gray line is a linear fit with 95% confidence intervals. The dashed line is the identity line. (**Aiv**) Slope of the linear fit (similar to the one in **Aiii**) with 95% confidence intervals for all types considered. (**B**) Differences in connectivity between a densely proofread section of the FB (denoted as 'column 3', or C3) and other parts of the FB. (**Bi**) Sample hΔA neurons. One (in blue) has almost all of its output synapses in C3. The other four avoid C3 altogether. Output synapses are in orange. (**Bii**) Comparison of the average number of synapses to known partners per type between neuron instances innervating the heavily traced C3 and instances innervating other columns. Types are selected as having instances innervating C3 with at least an average of 200 synapses of a given polarity in the fan-shaped body (FB) and having at least 80% of those synapses in C3. They are compared to neurons of the same type with no synapses in C3 (e.g., the hΔA neurons in gray in **Bi**, circled in black). Plotted are the percentage increases in input connectivity (left) or output connectivity (right) to known partners for neurons in FB C3 versus other columns, by type. (**Biii**) Comparison of output connectivity for the type shown in (**Bi**) between neuron instances innervating C3 and instances avoiding C3. Each point is the average relative weight of a connection between that type and another neuron type. The color denotes the supertype of the partner type. The gray line is a linear fit with 95% confidence intervals, the dashed line is the identity line. (**Biv**) Slope of the linear fit (similar to the one in **Biii**) with 95% confidence intervals for the types considered. hΔA neurons are circled in black.

The online version of this article includes the following figure supplement(s) for figure 4:

**Figure supplement 1.** Comparing protocerebral bridge (PB) connectivity in glomeruli with similar levels of tracing.

a prominent pathway from the fly's eyes to the EB. We will then trace a possible pathway for mechanosensory input to enter the CX and describe how sensory information is integrated in the EB. In a later section, we will describe a second input pathway to the CX via the NO.

## The anterior visual pathway: organization within the AOTU

The anterior visual pathway brings visual information from the medulla into the small subunit of the AOTU (AOTUsu, also called 'lower unit of the AOTU' in other insects), and thence to the BU's ring neurons (*Hanesch et al., 1989*), which deliver highly processed information to the EB (*Omoto et al., 2017*; *Timaeus et al., 2020*; *Figure 6A and B*, *Video 1*, *Video 2*). The ring neurons, which are called TL neurons in other insects (*Homberg et al., 1999*; *Muller et al., 1997*), are fed by multiple, developmentally distinguishable types of visually responsive neurons from the AOTUsu. Together, these tuberculo-bulbar or TuBu neurons compose the first part of the anterior visual pathway that is covered by the hemibrain volume. The full pathway comprises neurons that project from the photoreceptors to the medulla, the medulla to the AOTUsu, the AOTUsu to the BU, and the BU to the EB (*Figure 6A and B*; *Homberg et al., 2003*; *Omoto et al., 2017*; *Sun et al., 2017*). Across insects, some types of TuBu neurons (also called TuLAL1 neurons in some insects) are known to be tuned to polarized light e-vector orientations, spectral cues, and visual features (*el Jundi et al., 2014*; *Heinze et al., 2009*; *Heinze and Reppert, 2011*; *Omoto et al., 2017*; *Pfeiffer et al., 2005*; *Sun et al., 2017*), properties that they likely inherit from their inputs in the AOTU (*Hardcastle et al., 2020*; *Omoto et al., 2017*; *Sun et al., 2017*). In the fly, some TuBu neurons are known to respond strongly to bright stimuli on dark backgrounds (*Omoto et al., 2017*) or to the orientation of polarized light e-vectors (*Hardcastle et al., 2020*), consistent with the idea that these neurons may be part of a sky compass pathway, as in other insects (*el Jundi et al., 2018*; *Homberg et al., 2011*).

The hemibrain volume (light blue shaded region in *Figure 6A*) does not include areas of the optic lobe that would permit an unambiguous identification of AOTU inputs from the medulla (together called the anterior optic tract), thus only two broad subclasses of medulla columnar neurons can be distinguished. MC64 and MC61 (*Figure 6C*). However, following the schema employed in recent studies, we used the innervation patterns of different medulla columnar neuron types to delineate two distinct zones in the AOTUsu (*Figure 6Di*). These zones are consistent with previously characterized, finer-grained regions that receive input from different types of medulla columnar neurons (*Omoto et al., 2017*; *Timaeus et al., 2020*).

In other insects, some AOTU-projecting medulla columnar neurons (MeTus) are thought to be tuned to polarized light e-vector orientations (*Heinze, 2014*), and such information is known to be present in the *Drosophila* medulla as well (*Weir et al., 2016*). A more recent study has confirmed that some classes of fly AOTU neurons, as well as their downstream partners, respond to polarized

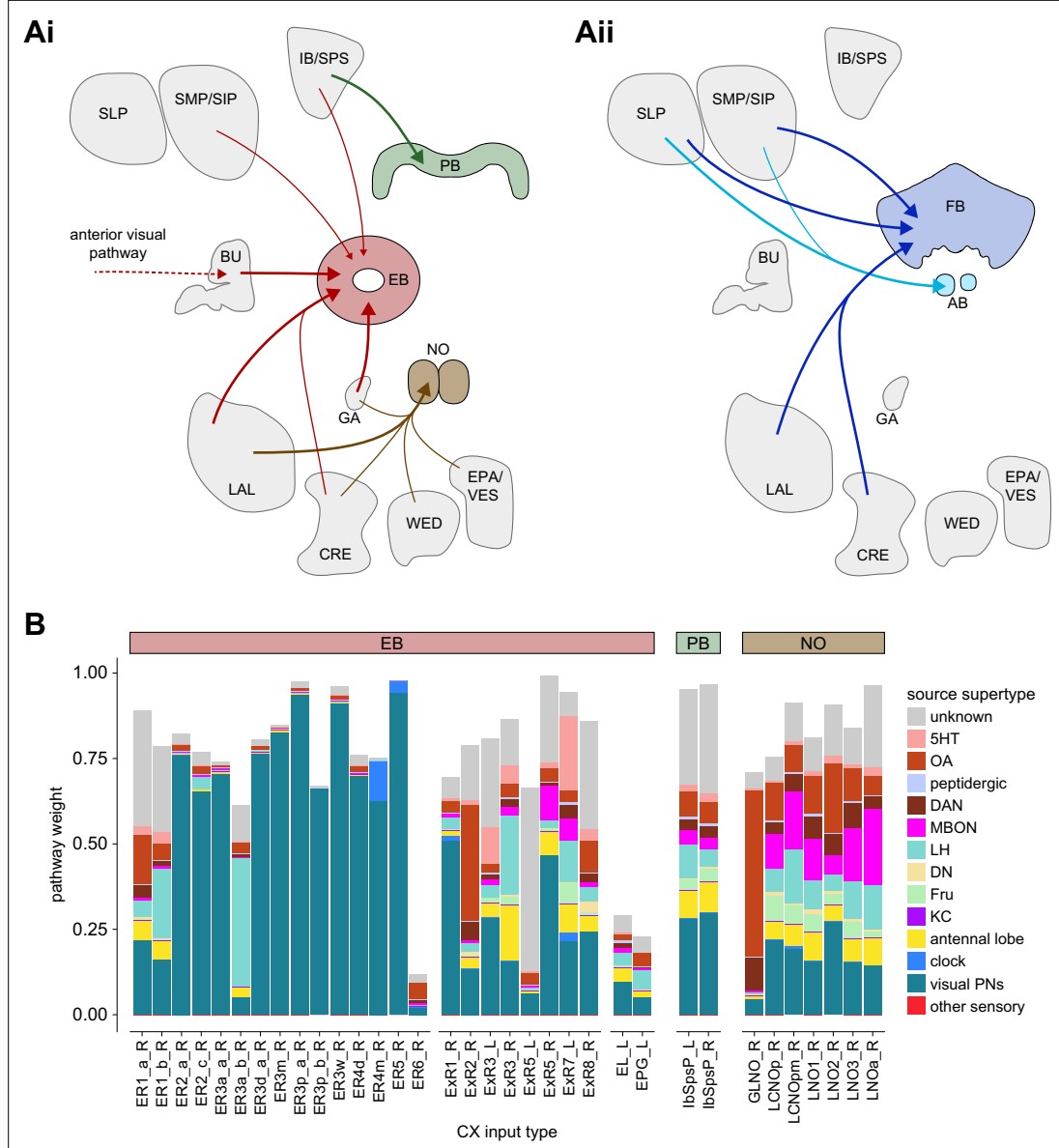

**Figure 5.** Overview of input pathways to the central complex (CX). (**A**) Schematic of input pathways, that is, pathways from non-CX brain regions, to the CX (see *Figure 5—figure supplement 1B*). (**Ai**) Input pathways to the ellipsoid body (EB) (red arrows), noduli (NO) (brown arrows), and protocerebral bridge (PB) (green arrows). (**Aii**) Input pathways to the fan-shaped body (FB) (blue arrows) and asymmetrical body (AB) (turquoise arrows). The width of the arrow is a qualitative indicator of the relative amount of input. (**B**) Input pathway classification for the EB, PB, and NO input neurons. Types are counted as inputs if they have at least 20 synapses of a given polarity outside of the CX and are the postsynaptic partner in at least one significant type-to-type connection outside of the CX. See *Appendix 1—figure 3* for an explanation of pathway weight. The corresponding data for FB and AB input pathways is presented in the FB section (*Figure 36—figure supplement 1C*, *Figure 40E*).

The online version of this article includes the following figure supplement(s) for figure 5:

**Figure supplement 1.** Additional information on input pathways to the central complex (CX).

light stimuli much like in other insects (*Hardcastle et al., 2020*). We believe that TuBu01 and TuBu06 neuron types are tuned to polarized light based on both their connectivity to ring neurons in the BU and their arborization patterns in the AOTU. TuBu01 is the only TuBu neuron that projects to the anterior BU and feeds the ER4m neuron type, which shows strong polarization tuning (*Hardcastle et al., 2020*), and TuBu06 appears to get input from the same population of MeTu neurons as TuBu01 in the AOTU (*Figure 6C and Dii*, top row, *Figure 7D*). However, a recent study reported that glomeruli in the

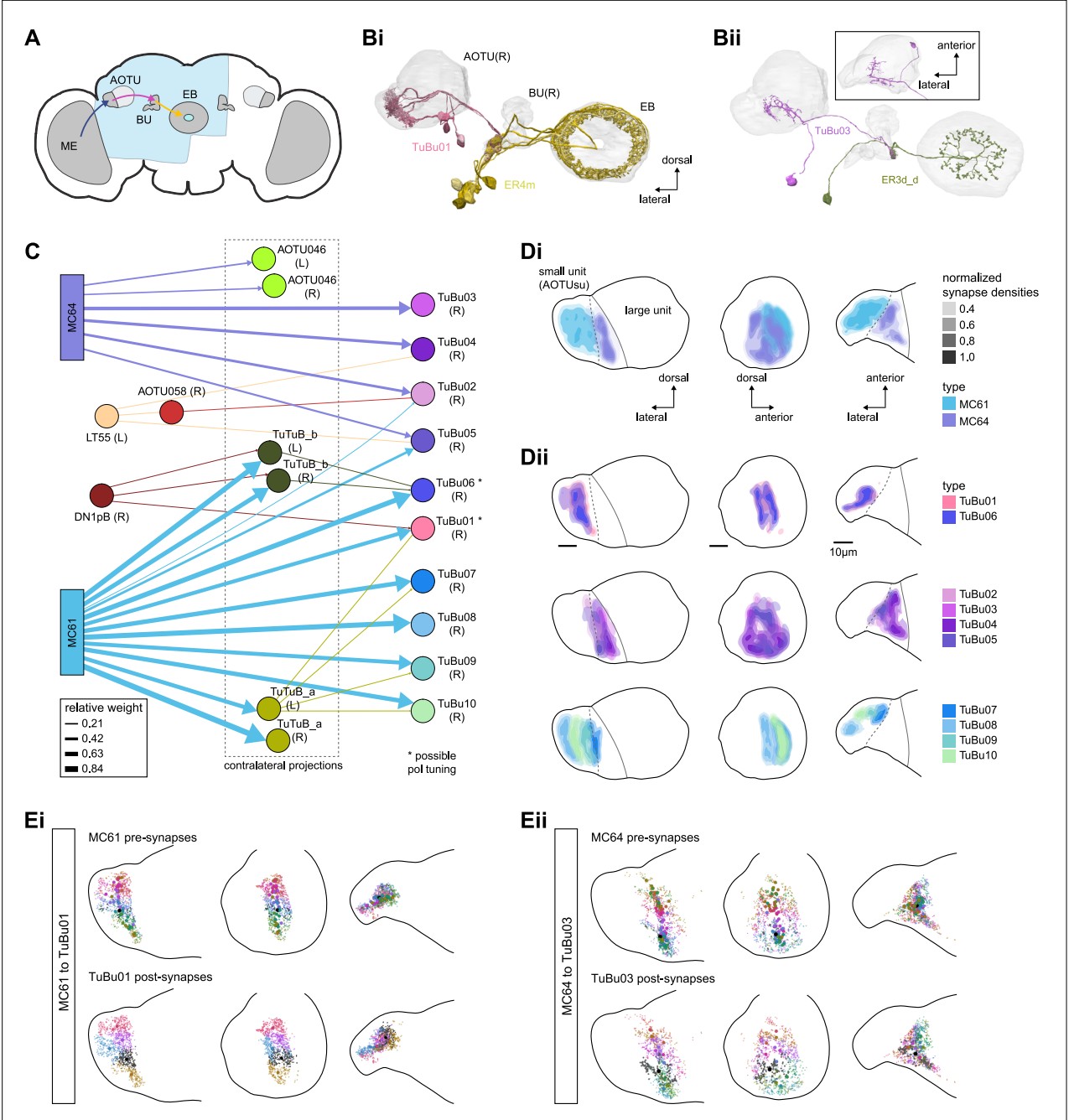

**Figure 6.** Overview of the anterior visual pathway and organization of the small unit of the anterior optic tubercle (AOTU). (**A**) Schematic of the fly brain indicating the neuropils that are part of the anterior visual pathway, which starts at the medulla (ME) and projects via the AOTU and the bulb (BU) to the ellipsoid body (EB). The anterior visual pathway only passes through the smaller subunit of the AOTU (AOTUsu). The light blue shaded region indicates the coverage of the hemibrain dataset. (**B**) Morphological renderings of a subset of neurons that are part of the anterior visual pathway. (**Bi**) and (**Bii**) highlight two of several parallel pathways. (**Bi**) TuBu01 neurons tile a subregion of the AOTUsu and project to the BU, where they form glomeruli and provide input to ER4m neurons. ER4m neurons project to the EB. All TuBu01 and ER4m neurons from the right hemisphere are shown. (**Bii**) TuBu03 neurons also arborize in the AOTU, but these neurons target different regions of both the AOTU and BU and form larger arbors in the AOTU than do TuBu01 neurons. TuBu03 also form glomeruli in the BU, where they connect to ER3d_d. Inset shows the TuBu03 arbor in the AOTU as seen from the ventral position. (**C**) Connectivity graph of the inputs to TuBu neurons in the AOTU (significant inputs were selected using a 0.05 [5%] cutoff for relative weight). AOTU046 neurons are included here as they provide input to TuBu neurons in the BU (see *Figures 7 and 8*). TuBu are colored from pink to green based on the regions they target in the BU (see *Figure 7*). The dashed rectangle marks neuron types that also project to the contralateral AOTU. An asterisk marks TuBu types with likely tuning to polarized light based on their morphology and connectivity (see text). (**D**) Projections of the normalized synapse densities for medulla columnar types (**Di**) and each TuBu type (**Dii**) along the dorsal-lateral (left), the dorsal-anterior (center), and

*Figure 6 continued on next page*

*Figure 6 continued*

the anterior-lateral (right) plane, respectively. The synapse locations of MC61 and MC64 define two subregions of the AOTUsu, which are marked with a dashed line. Projections for the 10 TuBu types were split up in subplots for ease of readability. Types that arborize in similar regions were grouped together. Note the columnar organization of TuBu01 and TuBu06-10 as opposed to the more diffuse projections of TuBu02-05. (**E**) Projections of individual synapse locations from medulla columnar to TuBu neurons. (**Ei**). Synapses from MC61 onto TuBu01 neurons. Projections are shown along the same planes as in (**D**). Synapse locations are color-coded by the identity of the presynaptic neuron (MC61, top) or the postsynaptic neuron (TuBu01, bottom). The large, black-outlined dots indicate the center of mass for synapses from an individual neuron. Note that there are many more MC61 than TuBu01 neurons. (**Eii**). Same as (**Ei**), but for synapses from MC64 to TuBu03. ME: medulla, AOTU: anterior optic tubercle, AOTUsu: small unit of the AOTU, BU: bulb, EB: ellipsoid body.

The online version of this article includes the following figure supplement(s) for figure 6:

**Figure supplement 1.** Connectivity motifs between MC and TuBu neurons in the AOTU.

dorsal part of the BUs were also polarization tuned (*Hardcastle et al., 2020*), suggesting that other TuBu types may also carry information about e-vector orientation.

TuBu neuron types arborize in subregions of the AOTU that respect the boundaries defined by medulla columnar inputs, and TuBu neurons form columns along the dorsoventral axis of their respective AOTUsu subregion (*Figure 6Dii*; *Omoto et al., 2017*; *Timaeus et al., 2020*). Both the medulla columnar neurons and the TuBu neurons tile each AOTUsu zone (*Figure 6Bi, E*), consistent with the TuBu neurons preserving a retinotopic organization from their columnar inputs (*Timaeus et al., 2020*). On average, there is a 40:1 convergence from medulla columnar neurons onto TuBu neurons (*Figure 6—figure supplement 1*), potentially increasing the size of the receptive fields of TuBu neurons compared to MeTu neurons. Although this tiling is well-organized for TuBu neurons that receive inputs from MC61 medulla neurons (*Figure 6Ei*), it becomes more diffuse for TuBu neurons that receive inputs from MC64 medulla neurons (*Figure 6Eii*), consistent with LM-based anatomical analysis (*Timaeus et al., 2020*). This differentiation of TuBu neuron types is also maintained in their downstream projections to the BU (next section). Note, however, that multiple TuBu neuron types can also receive their inputs from the same AOTUsu zone, for example, TuBu01 and TuBu06 (*Figure 6Dii*). It is possible that these different TuBu neuron types receive different medullary inputs in the same zone of the AOTU, but they are, regardless, easily distinguished both by the additional inputs they receive in the BU, as well as by their downstream partners in that structure (discussed below).

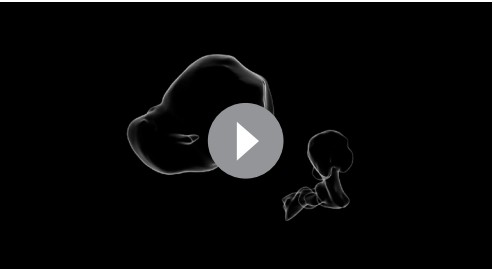

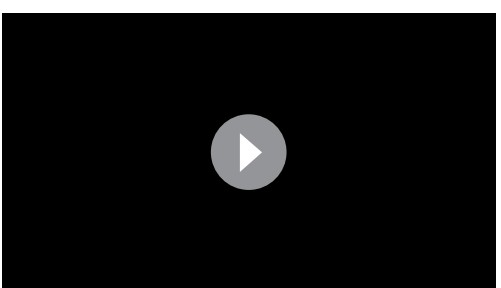

**Video 1.** Introduction to the central complex (CX), its neurons, and pathways. Movie showing meshes of the main CX neuropils along with the major CX-associated neuropils. In the second half, the movie uses morphological renderings of various CX neurons to trace a pathway that travels from the anterior visual pathway (BU to EB), through the compass network (EB and PB), to premotor neurons in the FB that target descending neurons in the LAL.

https://elifesciences.org/articles/66039/figures#video1

**Video 2.** Morphological rendering of two parallel pathways in the anterior visual pathway. The movie shows two of several parallel pathways in the anterior visual pathway. Meshes of the AOTU, BU and EB are shown. The first pathway consists of TuBu01 (shown in pink) and ER4m (shown in yellow). Initially, a single TuBu01 neuron and a single ER4m neuron are shown. They make a connection in the BU, where they form a glomerulus. The movie shows EM slices through the glomerulus. Later, complete populations of TuBu01 and ER4m neurons are shown. The second pathway presented in the movie involves TuBu03 (purple) and ER3d (teal). This movie is related to Figure 6B.

https://elifesciences.org/articles/66039/figures#video2

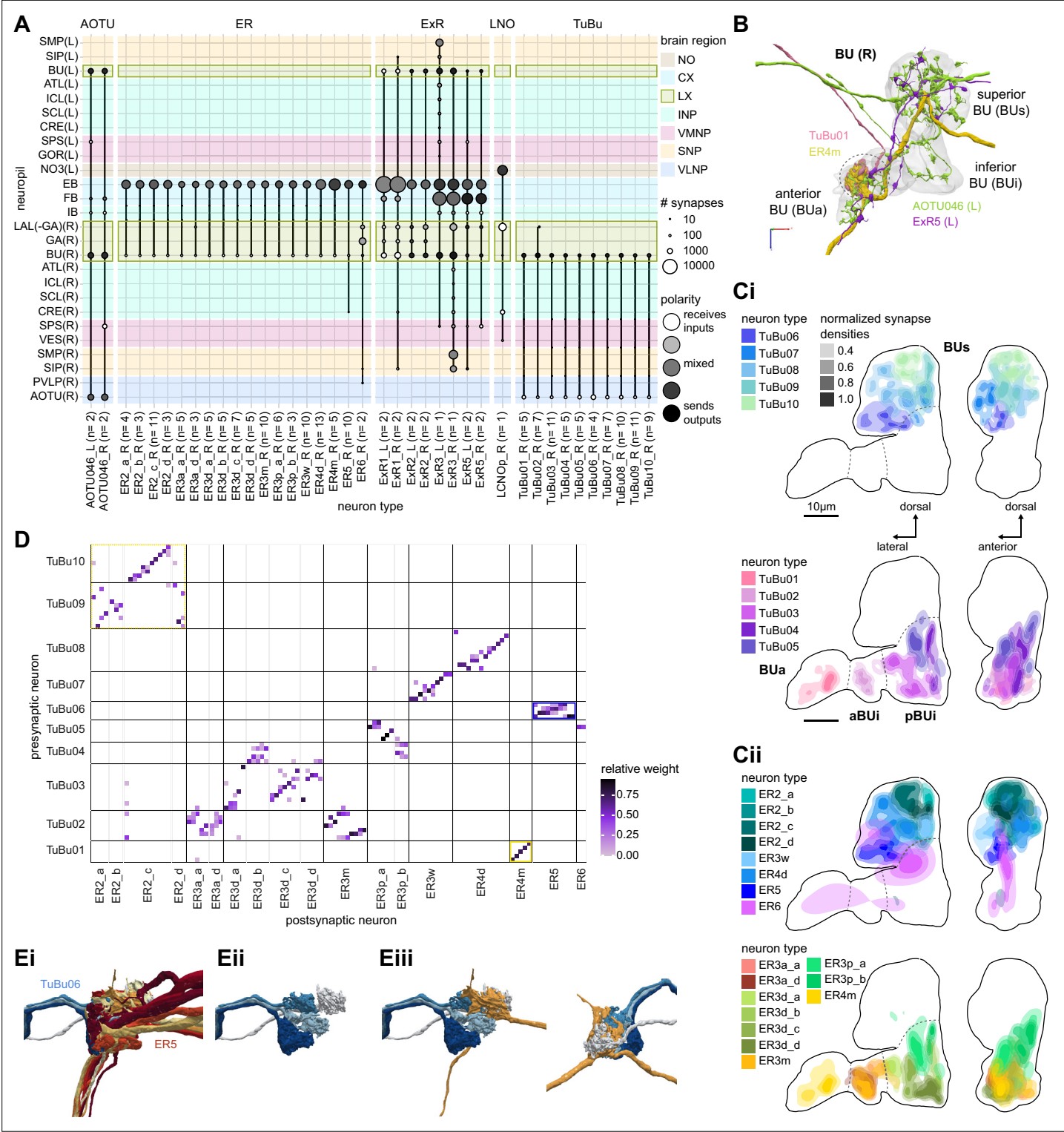

**Figure 7.** The bulb (BU) is more than just a relay station of visual information. (**A**) Region arborization plot of cell types that innervate the BU, showing average pre- and postsynaptic counts by region. The following types were excluded upon manual inspection based on their relatively small number of synapses in the BU. ExR7, SMP238, CRE013, LHCENT11, LHPV5l1. The LNO neuron (LCNOp) is an input neuron to the noduli (NO), which will be described in a later section. (**B**) Morphological rendering of processes from one AOTU046 and one ExR5 neuron, which both arborize widely within the BU, as well as one TuBu01 and one ER4m neuron, which form a glomerulus (dashed circle). Different anatomical zones of the BU are labeled. (**C**) Projections of the normalized synapse densities for TuBu types (**Ci**) and ER types (**Cii**) along the dorsal-lateral (left) and the anterior-lateral (right) planes of the BU, respectively. Borders between different anatomical zones are indicated with dashed lines. For readability, synapse densities of TuBu and ER

*Figure 7 continued on next page*

*Figure 7 continued*

types that arborize in the BUs (top) versus the BUi or Bua (bottom) are displayed separately. All populations of neurons, except ER6, form glomeruli. (**D**) Neuron-to-neuron connectivity matrix of connections from TuBu neurons to ER neurons. Neurons were grouped according to type and, within a type, ordered such that most connections lie on a diagonal. The yellow boxes mark connections between neurons (putatively) tuned to polarized light. The blue box marks connections of sleep-related neurons. (**E**) Morphological rendering of the glomeruli formed by TuBu06 and ER5. (**Ei**). All TuBu06 and ER5 neurons. (**Eii**). Same as (**Ei**) but just TuBU06 neurons. (**Eiii**) Same as (**Ei**), but with only one ER5 neuron shown to highlight how a single ER neuron can target multiple glomeruli. Top view shown on the right. BUs: superior bulb, BUi: inferior bulb; BUa: anterior bulb; pBUi: posterior inferior bulb, aBUi: anterior inferior bulb.

The online version of this article includes the following figure supplement(s) for figure 7:

**Figure supplement 1.** Connectivity motifs between TuBu and ER neurons in the BU.

## The anterior visual pathway: convergence and divergence largely within BU zones

The BU is primarily an input structure for the CX. Nearly every neuron type that receives most of its input in the BU has presynaptic specializations within one or more core CX structures (*Figure 7A*). The majority of cells in the BU are part of the anterior visual pathway. This pathway includes the TuBu neurons and their postsynaptic partners, the ring neurons (ER), which bring visual information to the EB (*Table 5*). Most neurons innervating the BU (e.g., the TuBu neuron types and the ring neurons [except ER6]) have spatially restricted, glomerular arborizations (*Trager et al., 2008*). Other neuron types arborize widely within the structure, such as the AOTU046 and extrinsic ring (ExR) neurons (*Figure 7B*). A recent study combining lineage-based anatomy and functional imaging has suggested that the BU may be organized into zones with similarly tuned TuBu neurons (*Omoto et al., 2017*). We used synapse locations of different types of TuBu neurons and their downstream partners, the ring neurons, to partition the BU into different zones (*Figure 7B and C*), each with numerous microglomeruli where TuBu neurons make synapses onto ring neurons (*Trager et al., 2008*). Each microglomerulus is formed by small arbors of up to five TuBu neurons from one type and their downstream ring neuron partners (see below and *Figure 7—figure supplement 1A and B*). Consistent with the functional and anatomical segregation suggested by *Omoto et al., 2017*, TuBu neurons originating from different parts of the AOTU do indeed segregate into different zones within the BU (superior, inferior, and anterior BU, *Figure 7Ci*), with MC61-fed TuBu neurons innervating the superior BU and MC64-fed TuBu neurons targeting the inferior BU (*Figure 6C*). The only exception is the polarization-tuned TuBu01, which arborizes in its own compartment, the anterior BU (*Figure 7Ci*). MC61-fed superior and anterior TuBu share a developmental origin, DALcl1, while the MC64-fed inferior TuBu neurons originate from DALcl2 (*Omoto et al., 2017*). Glomeruli in the superior BU tend to be smaller and more defined than glomeruli in the inferior BU. Except for ER6, an atypical ring neuron, ring neurons that receive their input in the BU also send their dendrites into a single zone of the BU, thereby maintaining some separation of visual pathways from the AOTU (*Figure 7Cii*).

The type-to-type mapping from TuBu neurons to ring neurons is largely one-to-one, with most ring neuron types receiving synaptic inputs from only a single TuBu type each, for example, TuBu01 to ER4m and TuBu06 to ER5 (*Figure 7D*, see yellow and blue-framed boxes). However, some TuBu types feed multiple ring neuron types, for example, most TuBu02 neurons project to both ER3a_a and ER3a_d ring neurons, and also to ER3m ring neurons (*Figure 7D*). Although the segregation of TuBu types is largely maintained at the type-to-type level, there is significant mixing at the level of individual TuBu to ring neuron connections. Most TuBu neurons feed several ring neurons of a given type, but the level of divergence varies between TuBu types (*Figure 7D*, *Figure 7—figure supplement 1B*). TuBu02 neurons, for example, make synapses onto multiple ER3a_a, ER3a_d and ER3m neurons (*Figure 7*, *Figure 7—figure supplement 1B*). There is also significant convergence, with many ring neurons receiving inputs from multiple TuBu neurons of the same type (*Figure 7D*, *Figure 7—figure supplement 1A*).

A particularly strong contrast can be observed between the mapping of TuBu01 to ER4m, which is strictly one-to-one, preserving receptive fields (but not polarotopy; *Hardcastle et al., 2020*), and TuBu06 to ER5, where multiple TuBu06 neurons contact a single ER5 neuron and single TuBu06 neurons project to multiple ER5 neurons (*Figure 7D and E*, *Figure 7—figure supplement 1B* and C). This is noteworthy as TuBu01 and TuBu06 receive input in the same region of the AOTU, but their downstream partners, ER4m and ER5 neurons, are known to have different functions. ER4m is tuned

**Table 5.** Known properties of ring neuron classes.

LAL: lateral accessory lobe; BUs: superior bulb; pBUi: posterior inferior bulb; aBUi: anterior-inferior bulb; BUa: anterior bulb; CRE: crepine.

| Neuron type | Tuning | Modality group | Input region | Reference |
|---|---|---|---|---|
| ER1_a | Mechanosensory? | Mechanosensory | LAL | *Okubo et al., 2020* |
| ER1_b | Mechanosensory (wind) | | LAL | |
| ER2_a | Visual with small (~45°) ipsilateral receptive fields; subset with polarization tuning | Ipsilateral visual + pol | BUs | *Hardcastle et al., 2020*; *Omoto et al., 2017*; *Seelig and Jayaraman, 2013*; *Sun et al., 2017* |
| ER2_b | | | BUs | |
| ER2_c | | | BUs | |
| ER2_d | | | BUs | |
| ER3a_a | Visual, large contralateral receptive fields and self-motion motor tuning; ER3a_b also wind tuning | Contralateral visual + motor (+ wind) | aBUi | *Okubo et al., 2020*; *Omoto et al., 2017*; *Shiozaki and Kazama, 2017* |
| ER3a_b | | | LAL+ CRE | |
| ER3a_c | | | LAL + CRE | |
| ER3a_d | | | aBUi + LAL | |
| ER3d_a | Control of sleep structure | Sleep | pBUi | *Liu et al., 2019*, Connectivity with ExR1 and ExR3 (EB *Figure 10F*, sleep *Figure 53*) |
| ER3d_b | | | pBUi | |
| ER3d_c | | | pBUi | |
| ER3d_d | | | pBUi | |
| ER3m | Visual, large contralateral receptive fields and self-motion motor tuning | Contralateral visual + motor | aBUi | *Omoto et al., 2017*; *Shiozaki and Kazama, 2017* |
| ER3p_a | Visual, large contralateral receptive fields? | Contralateral visual + motor | pBUi | *Omoto et al., 2017*; *Shiozaki and Kazama, 2017* |
| ER3p_b | | | pBUi | |
| ER3w | Assumed ipsilateral visual based on anatomy | Ipsilateral visual + pol | BUs | *Shiozaki and Kazama, 2017* |
| ER4d | Visual with small (~45°) ipsilateral receptive fields | Ipsilateral visual + pol | BUs | *Omoto et al., 2017*; *Seelig and Jayaraman, 2013*; *Sun et al., 2017* |
| ER4m | Polarized light tuning | Ipsilateral visual + pol | BUa | *Hardcastle et al., 2020* |
| ER5 | Sleep homeostasis | Sleep | BUs (sleep) | *Donlea et al., 2018*; *Liu et al., 2016* |
| ER6 | ? | - | BU + LAL + CRE | |

to polarized light (*Hardcastle et al., 2020*) and is likely involved in visual orientation, whereas ER5 neurons are involved in sleep (*Liu et al., 2016*). Both will be discussed in more detail in later sections.

Overall, this combination of divergence from individual TuBu neurons to multiple ring neurons and convergence from multiple TuBu neurons to individual ring neurons strongly suggests that we should expect receptive fields in the anterior visual pathway to expand, and sensory tuning to potentially become more complex from TuBu to ring neurons. Finally, it is important to note that there are several neuron types with wide arborizations within the BU that likely also influence processing in the structure. These include a subset of ExR neuron types (ExR1, ExR2, ExR3, and ExR5) and the AOTU046 neurons, which are discussed below.

## The anterior visual pathway: contralateral influences
Thus far, we have characterized the anterior visual pathway as a largely feedforward pathway of neurons with spatially localized arbors projecting from the early visual system to the AOTU and on to

the BU. Indeed, many TuBu neuron types in the superior BU are known to display prominent ipsilateral visual receptive fields consistent with ipsilateral inputs in the AOTU (*Omoto et al., 2017*; *Seelig and Jayaraman, 2013*; *Sun et al., 2017*). However, the connectome suggests that other, widely arborizing neuron types influence responses of neurons at different stages of the pathway (*Figure 8A and B*).

There is functional evidence that many of these neurons also receive large-field inhibitory input from the contralateral hemisphere, creating the potential for stimulus competition across the left and right visual fields (*Omoto et al., 2017*; *Sun et al., 2017*). The interhemispheric TuTuB_a neurons connect the right and left AOTU (*Figure 8Ai*). These neurons pool medullary input from the visual field of one hemisphere and synapse onto a subset of TuBu neurons on the contralateral side. Note that at least one of these TuTuB neurons is known to be tuned to polarized light e-vector orientation (*Hardcastle et al., 2020*). The interhemispheric AOTU046 neurons receive input from multiple brain regions including both AOTUs, the BU (although they primarily send outputs there), the FB, and the ipsilateral (to their soma) IB and superior posterior slope (SPS) (*Figure 8Aii,Ci, Di, and E*). AOTU046 and TuTu neurons are well positioned to mediate contralateral inhibition since both neuron types receive input from large areas of the contralateral AOTU (*Figure 8C*). Indeed, AOTU046 targets ring neuron types in the BU that show strong signatures of contralateral inhibition (*Sun et al., 2017*). Curiously, AOTU046 provides input to TuBu neurons in both the AOTU and BU, targeting somewhat different subsets (*Figure 8—figure supplement 1A*).

We did not find any obvious candidates that could provide TuBu neurons of the inferior BU with small-object-sized receptive fields in the contralateral hemisphere as has recently been suggested for a subset of these neurons (*Omoto et al., 2017*; *Shiozaki and Kazama, 2017*). Based on the connectome, one possibility is that these reported responses were not caused by small-field feature detectors related to the AOTU or BU, but rather by input from the ExR3 neurons (*Figure 8Aiii and Dii*, *Figure 8—figure supplement 1B*). The ExR3 neurons receive synaptic input from a variety of neurons in areas that are known to respond to optic flow and the fly's own movements (*Figure 8F*, note that inputs are mixed with outputs; see also later section on ExR neurons), both of which may have contributed to the reported response properties. A second possibility is that these responses were observed in the subset of ring neurons that form glomeruli in the BU, but also receive additional inputs in the LAL (ER3a_a and ER3a_d, *Figure 7A*, *Figure 10—figure supplement 5*). Finally, contralateral visual information may also reach these neurons from inter-medullary connections, a possibility that we could not investigate in the hemibrain dataset.

The widely arborizing neurons – ExR1, ExR2, ExR3, ExR5, and AOTU046 (*Figure 8A*) – have somewhat overlapping arbors in the BU (*Figure 8D*) but are selectively interconnected in the region (*Figure 8B*). The bilaterally projecting AOTU046 neurons receive ipsilateral inputs from ExR2 and ExR5 neurons and contralateral input from the ExR3 neurons (*Figure 8B*, the right AOTU046 receives input from ExR2 and ExR5, the left one from ExR3), and then provide input to ExR1 neurons in the same hemisphere (*Figure 8B*). The ExR3 neurons appear to be recurrently connected to TuBu02, TuBu03, and TuBu04 neuron types (*Figure 8B*, *Figure 8—figure supplement 1B*). The function of these external inputs is not yet known.

## Ring neurons that receive circadian input

In contrast to all the ring neuron types described above, most of which have been considered as bringing visual input into the EB, we note that the ER5 neurons have been primarily associated with conveying a signal related to sleep homeostasis and are thought to not be responsive to visual stimuli (*Donlea et al., 2018*; *Liu et al., 2019*; *Liu et al., 2016*; *Raccuglia et al., 2019*). These neurons are known to receive input from TuBu neurons (*Guo et al., 2018*; *Lamaze et al., 2018*; *Liang et al., 2019*; *Raccuglia et al., 2019*). The connectome allowed us to identify these neurons as TuBu06 (*Figures 7D and E and 8B*). TuBu06 neurons, in turn, receive input from the bilateral TuTuB_b neurons that receive clock circuit input through the DN1pB neurons (*Guo et al., 2018*; *Lamaze et al., 2018*; *Liang et al., 2019*; *Figure 6C*). These connections provide a link between sleep and circadian circuits within the CX, which we discuss in more detail in a later section. Circadian input from DN1pB neurons to ER4m via TuBu01 neurons (*Figure 6C*) may also affect neural responses in the polarized light e-vector pathway, consistent with findings in other insects; this could allow for polarized light responses to be adjusted based on the time of day to compensate for changes in the polarization pattern as the sun moves across the sky (*Heinze and Reppert, 2011*; *Pfeiffer and Homberg, 2007*; *Sauman et al., 2005*).

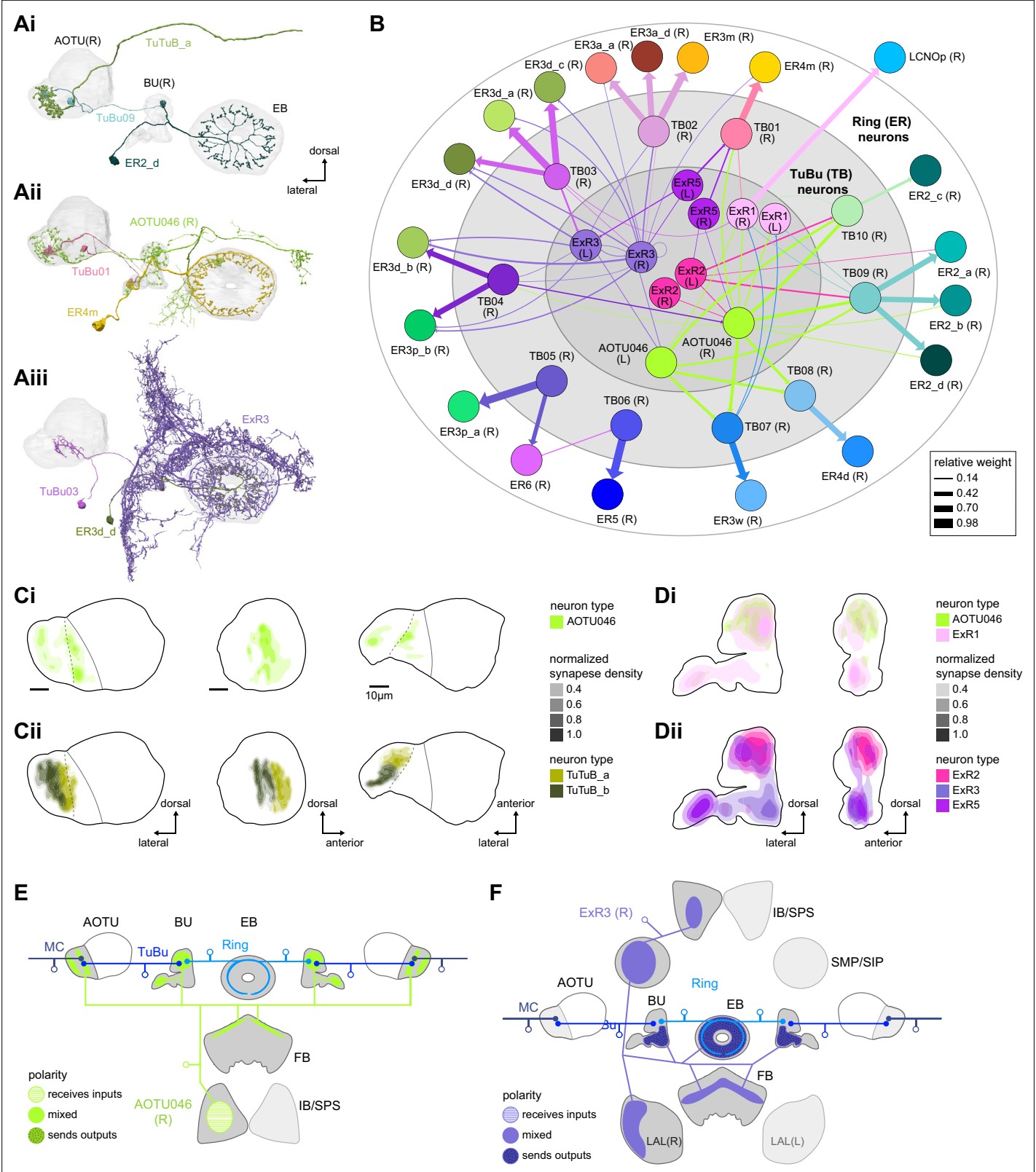

**Figure 8.** Source of contralateral visual information. (**A**) Morphological renderings of neurons in the anterior visual pathway together with neurons that connect to the contralateral anterior optic tubercle (AOTU) and/or bulb (BU). (**Ai**) TuBu09, ER2_d, and TuTuB_a. (**Aii**) TuBu01, ER4m, and AOTU046. (**Aiii**) TuBu03, ER3d_d, and ExR3. (**B**) Connectivity graph of TuBu and ER neurons as well as other neurons, ExR and AOTU046, that provide input to TuBu and ER neurons in the right BU. To highlight the organizational principles of connectivity in the BU, the nodes representing ER neurons are placed in an

*Figure 8 continued on next page*

Figure 8 continued

outer ring, those representing TuBu neurons (for brevity named TB here) in a middle ring, and nodes representing ExR and AOTU046 inside a central circle. (**C**) Projections of the normalized synapse densities of AOTU046 (**Ci**) and TuTuB (**Cii**) neurons in the right AOTU. Visualization as in *Figure 6D*. (**D**) Projections of the normalized synapse densities of AOTU046 and ExR neurons in the right BU. AOTU046 and ExR1 shown in (**Di**); ExR2, ExR3, and ExR5 shown in (**Dii**). Visualization as in *Figure 7C*. (**E**) Schematic of the projection pattern of a right AOTU046 neuron, piecing together innervations of the right AOTU046 neuron in the left hemisphere from the innervation of the left AOTU046 neurons in the right hemisphere, assuming mirror symmetric innervation patterns of the left and right neurons. Qualitative indication of input/output ratios per region is given based on region innervation plots shown in *Figure 7A*. (**F**) Schematic as in (**E**), but for the right ExR3 neuron.

The online version of this article includes the following figure supplement(s) for figure 8:

**Figure supplement 1.** Connectivity of AOTU046 and ExR3 with TuBu neurons.

## Mechanosensory (wind) input to ring neurons

In addition to the visual inputs described above, the EB also receives mechanosensory input, as demonstrated by intracellular recordings of wind-sensitive activity in the locust (*Homberg, 1994*) and by extracellular recordings in the cockroach CX demonstrating directionally tuned activity in response to mechanical stimulation of the antennae (*Bender et al., 2010*; *Guo and Ritzmann, 2013*; *Ritzmann et al., 2008*). In the fly, recent work has shown that mechanosensory information is carried by ER1 and a subset of ER3a ring neurons (*Okubo et al., 2020*). Unlike the majority of ring neurons that bring sensory information to the EB and receive visual input in the BU, these ring neurons gather much of their input from the LAL and WED (*Figure 9A and B*), a structure known to receive significant mechanosensory input from the antennal Johnston's organ (*Patella and Wilson, 2018*; *Suver et al., 2019*). ER1 neurons differ from ER3a cells most prominently in their arborizations in the EB, which we will cover in more detail in the next section. The CX connectome revealed that both of these neuron populations consist of multiple types. There are two ER1 types, ER1_a and ER1_b, and four ER3a types (*Figure 10—figure supplement 5*). Of the four ER3a types, ER3a_a and ER3a_d primarily receive inputs in the BU, whereas ER3a_b and ER3a_c receive inputs in the LAL (*Figure 10—figure supplement 5*). We found that only ER3a_b and ER1_b neurons are postsynaptic to neurons that we believe to be WPN neurons (*Figure 9A–C*) – neurons that have been characterized as being wind-direction-sensitive (*Suver et al., 2019*). These putative WPN neurons are themselves downstream of APN neurons (*Figure 9A and C*), which are known to be mechanosensory (*Patella and Wilson, 2018*; *Suver et al., 2019*). The ER1_b neurons also receive input in the LAL from the LAL138 (previously known as WL-L) neurons (*Franconville et al., 2018*; *Okubo et al., 2020*), which, recent work has suggested, tonically inhibit the ER1_b neurons (*Okubo et al., 2020*). We note that this study also showed evidence for WL-L neurons being gap-junction-coupled to ER1_b neurons. The hemibrain volume lacks the resolution to confirm this observation. In contrast to the ER1_a neurons, whose processes are uniformly distributed along the dorsal-ventral axis of the LAL (*Figure 9Di and Ei*), the ER1_b neurons receive inputs, especially from ExR7 neurons (discussed in later sections) in relatively distinct clusters arranged along the dorsal-ventral axis (*Figure 9Dii and Eii*). The latter spatial segregation is consistent with physiological reports of their distinct but overlapping tuning to wind direction (*Okubo et al., 2020*). Our analysis suggests that directional information from wind stimuli is conveyed to the EB by ER1_b and ER3a_b neurons. Functional properties of major inputs to the other LAL ring neurons have not been characterized, and it is therefore unknown what information is conveyed by these types. Because ER1_a, ER1_b, and ER3a_b neurons receive the majority of their input from unidentified neurons in the LAL, WED, and CRE (*Figure 9B*), the directional mechanosensory information that they receive may well be combined with other sensorimotor information from other presynaptic partners (*Figure 9—figure supplement 1*).

## The EB: architecture for a flexible, multimodal compass representation

The EB, also known as the lower division of the central body (CBL) in other insects (*Muller et al., 1997*), is a ring-shaped brain structure in the fly CX. Behaviorally, the EB has long been considered to play a key role in locomotion and navigation (*Bausenwein et al., 1994*; *Kong et al., 2010*; *Kottler et al., 2019*; *Kuntz et al., 2017*; *Liang et al., 2019*; *Neuser et al., 2008*; *Ofstad et al., 2011*; *Pan et al., 2009*; *Strauss, 2002*; *Wang et al., 2008*). More recently, functional studies have demonstrated that the population activity of EPG (compass) neurons, each of which arborizes in a single EB sector ('wedge'), represents the fly's orientation relative to the external world (*Seelig and Jayaraman,*

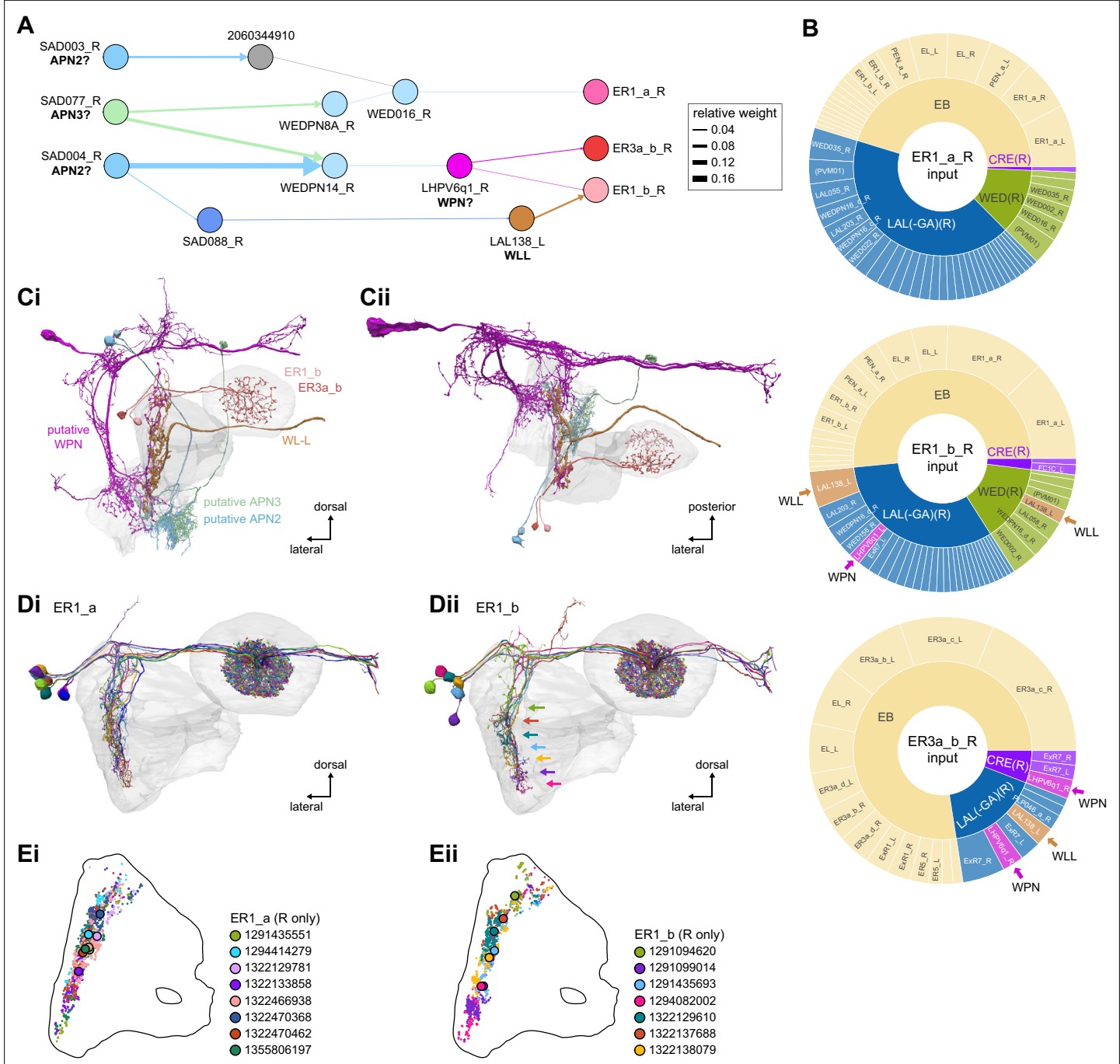

**Figure 9.** Mechanosensory input to the ellipsoid body (EB). (**A**) Connectivity graph of paths from putative APN2 and APN3 to ER neurons. Only pathways with a minimal total weight of 1E-05 and a maximum length of 5 were considered. APN: AMMC projection neuron; WPN: wedge projection neuron; WLL: wedge-LAL-LAL neuron. (**B**) Hierarchical pie charts showing the fraction of inputs from various neuron types separated by input region for ER1_a (left), ER1_b (center), and ER3a_b (right) neurons. The fractions represent the average per type (computed only for neurons from the right hemisphere). Arrows highlight inputs from WPN (LHPV6q1) and WL-L (LAL138). (**C**) Morphological renderings of putative APN2 (SAD003, SAD004), APN3 (SAD077), WPN (LHPV6q1), and WL-L (LAL138) neurons as well as ER1_b and ER3a_b. (**Ci**). Frontal view. (**Cii**). Top view. (**D**) Morphological renderings of ER1_a (**Di**) and ER1_b (**Dii**). Only neurons with cell bodies in the right hemisphere are shown. Individual neurons are colored differently. (**E**) Projections of synapse locations of the neurons shown in (**D**). Synapses are colored by neuron identity (see legend). Larger, black-outlined dots mark the mean synapse position (center of mass) of each neuron. Synapses of individual ER1_b neurons separate along the dorsal-ventral axis (**Eii**), whereas synapses of ER1_a neurons are more spatially mixed (**Ei**).

The online version of this article includes the following figure supplement(s) for figure 9:

**Figure supplement 1.** Organization of inputs to ER1 neurons in the lateral accessory lobe (LAL).

*2015*). This activity, which localizes to a contiguous 'bump' spanning the processes of EPG neurons in neighboring sectors, is required for flies to stably maintain arbitrary headings as they walk or fly (*Giraldo et al., 2018*; *Green et al., 2019*). We will not describe the fly compass function in detail here, and instead refer readers to recent reviews on the topic (*Green and Maimon, 2018*; *Hulse and Jayaraman, 2019*). Broadly, numerous experimental and computational studies suggest that the compass-like dynamics of the EPG population, which resemble those of a theorized network structure called a ring attractor (*Kim et al., 2017c*; *Turner-Evans et al., 2020*), are generated by the interaction of several neuron types (*Cope et al., 2017*; *Fisher et al., 2019*; *Green et al., 2017*; *Kakaria and de Bivort, 2017*; *Kim et al., 2019*; *Pisokas et al., 2020*; *Turner-Evans et al., 2017*; *Turner-Evans et al., 2020*). The neural compass relies on input from the anterior visual pathway to remain tethered to the fly's visual surroundings (*Fisher et al., 2019*; *Hardcastle et al., 2020*; *Kim et al., 2019*; *Turner-Evans et al., 2020*) and from mechanosensory ring neurons to tether to the direction of wind (*Okubo et al., 2020*). This representation of the fly's head direction is also updated by self-motion cues (*Green et al., 2017*; *Turner-Evans et al., 2017*) and persists even when the fly is standing still in darkness (*Seelig and Jayaraman, 2015*; *Turner-Evans et al., 2020*). Not surprisingly, the EB receives input from a large number of canonical ring neuron types. It also receives input from numerous columnar neurons that together bidirectionally link the EB to the PB, NO, GA, and LAL, as well as ExR neurons, which connect the EB to many brain structures outside the CX (*Figure 10A*, *Figure 5Ai and B*). In the sections that follow, we provide a high-level view of the anatomical organization of the EB, followed by analyses of the different neuron types innervating the structure, and of their interconnectivity within it, particularly in the context of tethering the fly's compass representation to sensory cues in its environment.

## The anatomical organization of the EB

Each ring neuron sends its processes around the entire circumference of the EB, forming a ring-shaped arbor. Different types occupy rings at specific depths within the anterior-posterior axis (*Hanesch et al., 1989*; *Lin et al., 2013*; *Omoto et al., 2018*; *Young and Armstrong, 2010b*) and make synapses within type-specific annuli along the radial axis (*Video 2*). *Figure 10B*, for example, shows the locations of ER4m synapses for three different projections (see *Figure 10—figure supplements 1–3* for synapse locations of other ring, columnar, and ExR neuron types, respectively). Ring neuron types also vary in the spatial extent of their arbors and the degree to which they overlap with the arbors of other ring neuron types. *Figure 10Ci* displays this overlap through the location of synapses in the anterior-radial plane (see *Figure 10—figure supplement 4A* for anterior-radial cross-sections along different parts of the EB). When neuron types are grouped by the subregions of structures outside the EB where they receive their inputs, synapses of different ring neuron classes appear to tile the anterior-radial cross-section of the EB (sample cross-section shown in *Figure 10Cii*). The only significant overlap between different ring neuron classes occurs between the BUs and BUa ring neurons, which receive inputs in similar parts of the AOTUsu (see previous section). Most columnar neurons, which are so named because they each only innervate single glomeruli in the PB and single sectors ('tiles' and 'wedges' [*Wolff et al., 2015*]) of the EB torus, spread their synapses across fairly large fractions of the anterior-radial cross-section of the structure (*Figure 10D*, *Figure 10—figure supplement 4B*, see *Figure 10—figure supplement 2C* for synapse locations of individual EB columnar types). In particular, the population of EPG and EL neurons makes synapses across most of the EB (*Figure 10—figure supplement 2*). EL neurons (called EB.w.AMP.s-Dga-s.b in *Wolff et al., 2015*) have never been characterized physiologically. The ExR neuron types are similar to ring neurons in arborizing across the entire circumference of the EB. However, in contrast to most ring neurons, they make synapses across fairly large fractions of the anterior-radial cross-section of the EB (*Figure 10E*, *Figure 10—figure supplement 3C*, *Figure 10—figure supplement 4C*). The networks created by connections between these different neuron types in the EB are thought to be involved in generating neural dynamics that maintain the fly's head direction representation and control sleep (*Figure 10F*).

## Ring neurons bring type-specific information into the EB

Most ring neurons are thought to be GABAergic (*Hanesch et al., 1989*; *Homberg et al., 2018*; *Homberg et al., 1999*; *Isaacman-Beck et al., 2019*; *Turner-Evans et al., 2020*) and therefore inhibitory. They bring diverse sensory information into the EB (see *Table 5*, morphological renderings in *Figure 10—figure supplement 5*). Visually tuned ring neurons with ipsilateral receptive fields (ER2a-d,

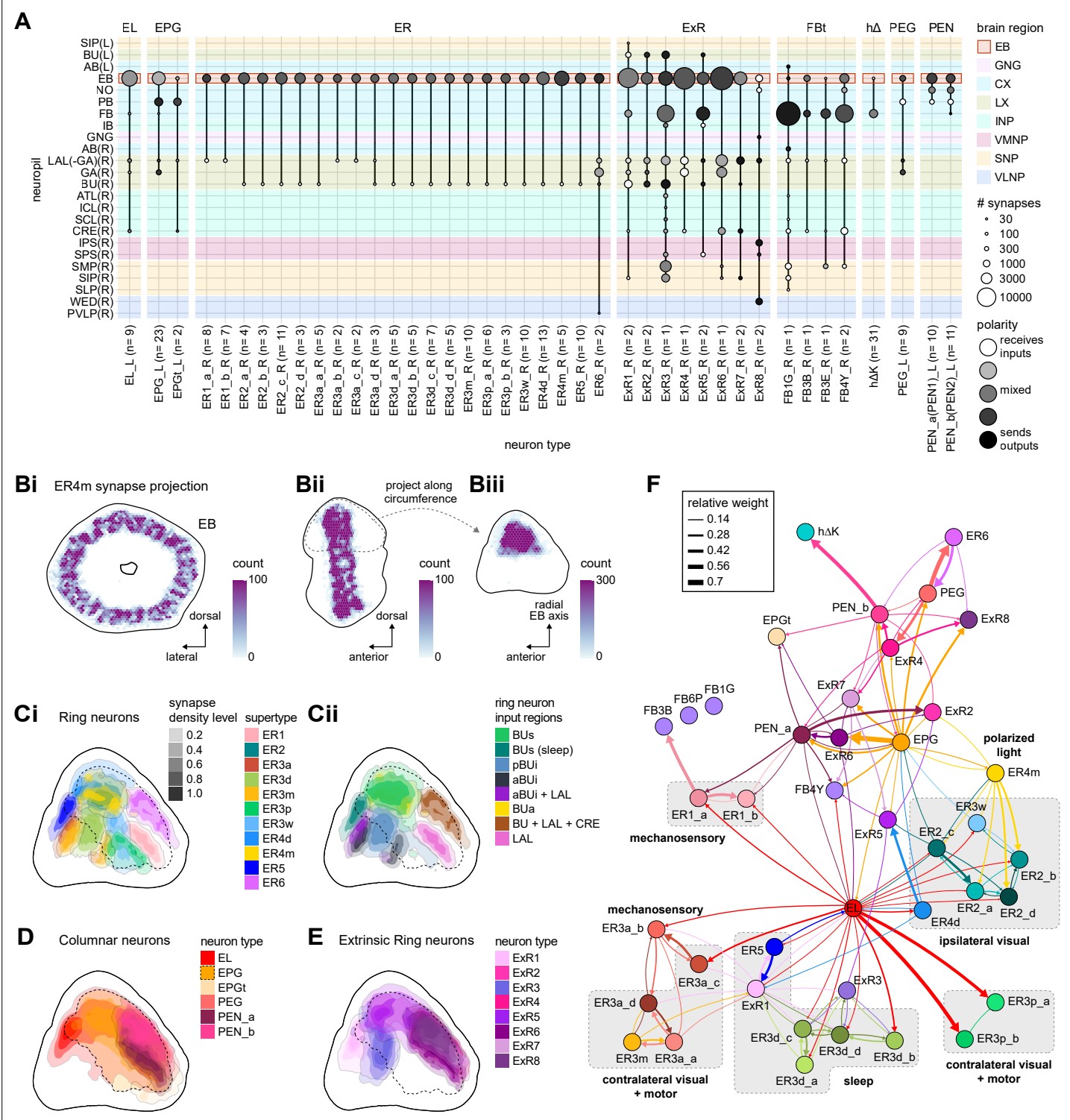

**Figure 10.** Overview of the organization of the ellipsoid body (EB). (**A**) Region arborization plot of neuron types that innervate the EB, showing average pre- and postsynaptic counts by region. For each neuron type, the number of cells from the right hemisphere is noted in the x-axis label. (**B**) Two-dimensional histograms of synapse counts of ER4m after projection onto the EB cross-sections along the dorsolateral (**Bi**), dorsoanterior (**Bii**), and anterior-radial axes (**Biii**). Note that for (**Biii**) anterior-radial cross-sections along the circumference of the EB were collapsed onto a single plane. The dashed line in (**Bii**) indicates one of the cross-sections that were collapsed in (**Biii**). The shapes of the anterior-radial cross-sections vary along the circumference of the EB, which is shown in *Figure 10—figure supplement 4*. (**C**) Normalized synapse densities of ring neurons onto the EB cross-section along the anterior-radial axes (see dashed outline in **Bii**, solid outline in **Biii**). (**Ci**). The synapse densities are color-coded by ring neuron type.

*Figure 10 continued on next page*

Figure 10 continued

(**Cii**). The synapse densities are color-coded by input regions. The dashed line indicates the outline of the EPG synapse density as seen in (**D**), for reference. (**D**) Same as in (**Ci**), but for columnar EB neurons. (**E**) Same as in (**Ci**), but for extrinsic ring (ExR) neurons. (**F**) Connectivity graph of neurons innervating the EB. Relative weight as measured on a type-to-type level has been mapped to the edge width. Gray shapes indicate groups of neuron types that likely share similar functional tuning based on existing literature. Only connections with a minimal relative weight of 0.05 (5%) are shown. Connections of a type to itself are omitted for simplicity.

The online version of this article includes the following figure supplement(s) for figure 10:

**Figure supplement 1.** Ring neuron synapse positions.

**Figure supplement 2.** Ellipsoid body (EB) columnar neuron synapse positions.

**Figure supplement 3.** Extrinsic ring (ExR) neuron synapse positions.

**Figure supplement 4.** Synapse projections onto the anterior-radial axis along the circumference of the ellipsoid body (EB).

**Figure supplement 5.** Morphological renderings of ring neurons.

**Figure supplement 6.** Morphological renderings of ring neurons.

**Figure supplement 7.** Morphological renderings of ring neurons.

**Figure supplement 8.** Morphological renderings of ring neurons.

**Figure supplement 9.** Morphological renderings of ring neurons.

ER4d) and ring neurons tuned to polarized light (ER4m) are well-characterized and the fly's head direction system is known to tether to such cues (*Fisher et al., 2019*; *Hardcastle et al., 2020*; *Kim et al., 2019*). ER1_b and ER3a_b are tuned to wind direction, but only ER1_b appears to directly influence the head direction representation (*Okubo et al., 2020*). Less is known about the tuning and function of other ER3 neuron types. Pan-neuronal imaging has suggested that a subset of ER3 neurons in the aBUi responds to contralateral visual information and motor actions (potentially the ER3a neurons, based on their anatomical position in the BU and their inputs in the LAL), but many ER3 neurons have been reported to be unresponsive to visual stimuli (*Omoto et al., 2017*; *Shiozaki and Kazama, 2017*). Based on their connectivity with ExR3 and ExR1, we hypothesize that ER3d ring neurons are involved in sleep-wake control (see section on sleep for details). One other ring neuron type, ER5, has been explicitly linked to tracking sleep need (discussed further in a later section). Finally, the unusual ring neuron type – the ER6 – receives most of its non-EB input in the GA (discussed in more detail in a later section), where it contacts EPG and PEG neurons.

## Generating a head direction representation from diverse sensory cues

The fly's compass system can tether to directional sensory cues carried by a variety of ring neuron types, forming an internal representation of head direction in a world-centered (i.e., allocentric) reference frame. Using the complete EB connectivity information of the hemibrain connectome, we found that most – but not all – ring neuron types from both hemispheres make direct synaptic contacts with EPG neurons from both sides of the brain (*Figure 11A*, third and fourth columns of the connectivity matrix, *Video 3*; *Video 4*; see also *Turner-Evans et al., 2020*). In the analyses that follow, we will consider the number and location of synapses that various visual and mechanosensory ring neurons make onto EPG neurons as a proxy for the influence that these different sensory cues exert on the EPG compass. For example, the physiological observation that ER3a_b neurons, which respond to wind direction, do not directly impact EPG dynamics (*Okubo et al., 2020*) is well accounted for by their lack of direct synaptic connections to EPG neurons (*Figure 11A*). The other ER3a neurons also make few connections to EPG neurons. Note that although the unusual ER6 neuron type does not synaptically contact EPG neurons in the EB (top two row blocks in *Figure 11A*), these neurons – and most of the ER3a as well as ER5 neurons – do make strong contacts with the EL columnar neuron types (first and second columns in the connectivity matrix in *Figure 11A*, morphological rendering in *Figure 11B*). The functional significance of EL neurons and of these connections is unknown. The stimulation of ER6 neurons triggers an inhibitory response in EPG neurons (*Franconville et al., 2018*), which might be mediated through their connections in the gall (see Figure 56) or indirectly through other columnar neurons (see below).

The diversity of sensory cues that the head direction representation tethers to indicates that the fly compass is flexible enough to function in a variety of multisensory settings. This flexibility of tethering

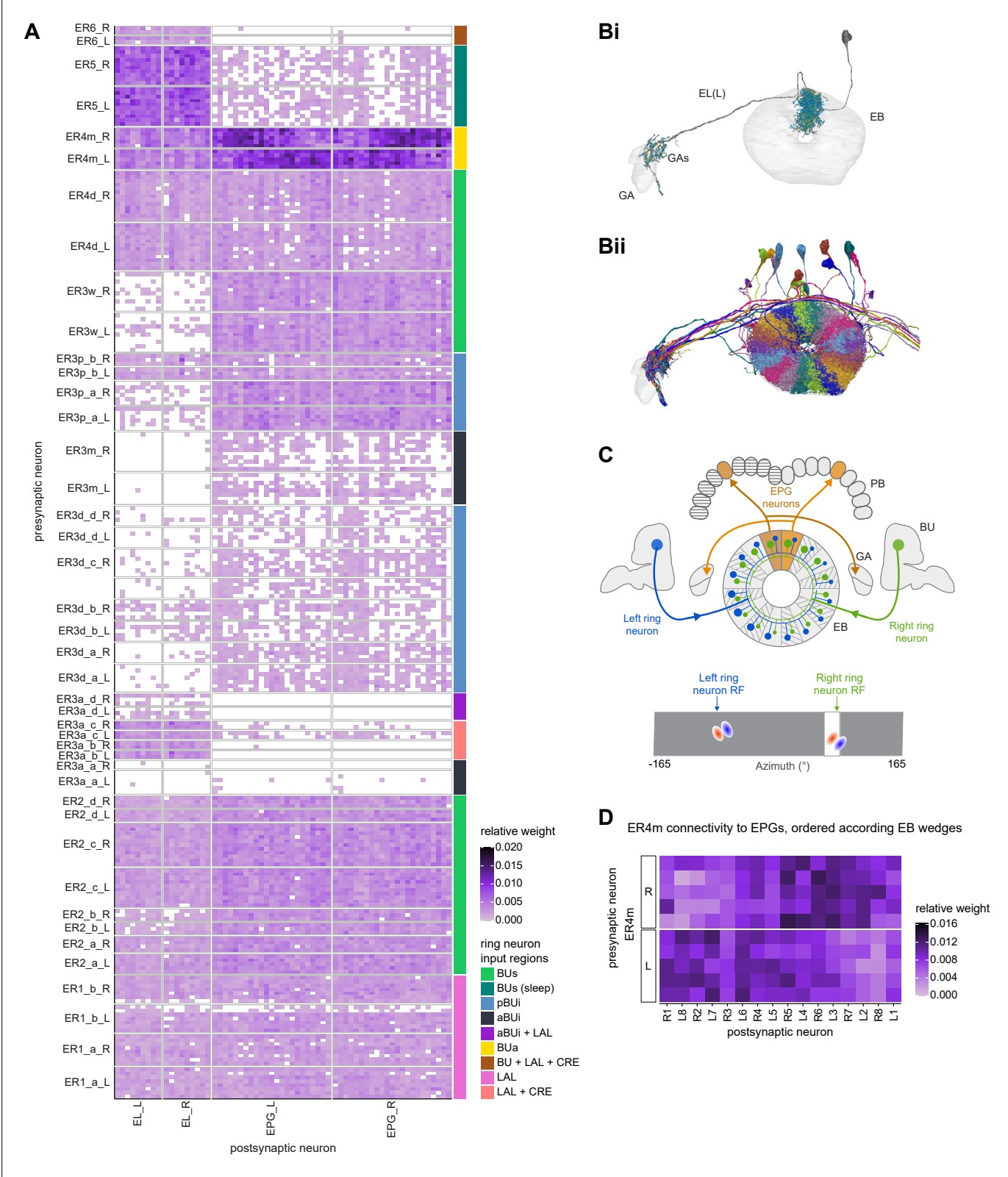

**Figure 11.** Ring neuron to columnar connectivity. (**A**) Neuron-to-neuron connectivity matrix for connections from ring neurons to EL and EPG neurons in the ellipsoid body (EB) on a single neuron level. The boxes on the right side are colored according to the ring neuron's input region. (**B**) Morphological renderings of EL neurons and renderings of innervated regions of interest (ROIs). Note that EL neurons target a small region next to the GA, called the gall surround (GAs). (**Bi**). Single left hemisphere EL neuron with blue dots marking the location of postsynaptic sites and yellow dots those of presynaptic

*Figure 11 continued*

sites. (**Bii**). Full population of EL neurons. (**C**) Schematic illustrating variation in synaptic strength in ring neuron to EPG connections due to neural plasticity. Top: connectivity between ring neurons and EPG neurons. Bottom. Illustration of receptive fields (RFs) of single-ring neurons. (**D**) Connectivity matrix of ER4m inputs to EPG neurons that have been sorted and averaged according to the EB wedge they innervate.

The online version of this article includes the following figure supplement(s) for figure 11:

**Figure supplement 1.** Wedge-specific modularity of inputs from ring neurons to EPG neurons.

the EPG compass to different cues is thought to be achieved by experience-dependent plasticity acting on the synaptic weights between ring neurons and EPG neurons (*Fisher et al., 2019*; *Kim et al., 2019*). These studies suggested that the strength of a visual ring neuron's synapses onto EPG neurons would, through sculpted inhibition, localize EPG activity to a specific sector of the EB for a given head direction in a specific visual setting (schematized in *Figure 11C*). Considering the strong correlation between synapse counts and the area of synaptic contact between neuron pairs in at least larval *Drosophila* (*Barnes et al., 2020*), and assuming that the functional strength of synapses depends on synaptic surface area (*Holler-Rickauer et al., 2019*), we asked whether we could detect any signatures of such plasticity-based sculpting in synapse counts between ring and EPG neurons (note that a change in synapse counts is only one of many potential signatures of plasticity-related changes in synaptic strength). Specifically, we examined the connectivity matrix between a variety of ring neurons – ER1_a, ER1_b, ER2_a-d, ER4d, and ER4m – and EPG neurons for any consistent nonuniformities. We conjectured that if synapse counts or relative weights reflect functional synaptic strengths, we should see specific patterns in the connectivity between ring neurons and EPG neurons. For example, we might expect that the ring neuron synaptic profiles onto multiple EPG neurons within a wedge (i.e., EPG neurons that share the same head direction tuning) would be more similar than their synaptic profiles onto EPG neurons with very different tuning (i.e., those from angularly distant parts of the EB) (see *Figure 11C*). We found no such correlations or modularity for most ring neuron types (*Figure 11—figure supplement 1*; *Table 6*). There were two exceptions: the polarization-tuned ER4m neurons (*Figure 11A and D*, *Figure 11—figure supplement 1Cviii*), and, to a lesser extent, the ER1_a neurons (*Figure 11—figure supplement 1Ci*). Thus, if synaptic weights from sensory ring neurons onto EPG neurons vary in consistent patterns as a result of a fly's experience, as we expect they must, this is not reflected in synapse counts for the vast majority of ring neuron types (note, however, that we know little of the sensory experience of this fly; see Materials and methods). The absence of obvious structural signatures of synaptic strength is perhaps unsurprising in the context of plasticity that acts at a timescale of tens of seconds to a few minutes, as is the case here (*Fisher et al., 2019*; *Kim et al., 2019*). In the case of the polarization-tuned ER4m neurons, we observed that synapse counts to EPG neurons varied smoothly along the circumference of the EB, but with mirror-symmetric profiles for ER4m neurons from the left and right hemisphere, respectively (*Figure 11D*),

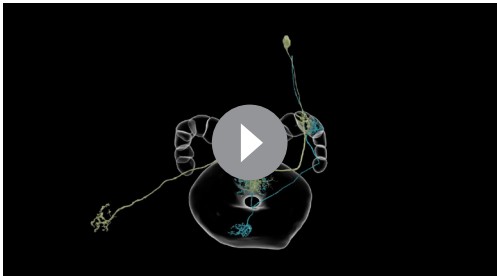

**Video 3.** Ring neurons and their connections to EPG neurons. Movie begins by showing morphological renderings of single TuBu, ring (ER), and compass neurons (EPG) to outline the anterior visual pathway. Later, all ring and EPG neurons are rendered to highlight the numerous parallel pathways that bring visual, circadian, mechanosensory and motor signals into the EB.

https://elifesciences.org/articles/66039/figures#video3

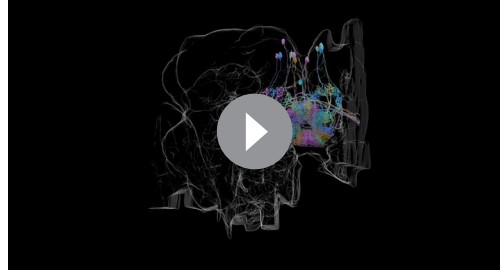

**Video 4.** EPG and PEN neurons. Movie begins by showing a morphological rendering of the entire EPG population. Next, individual EPG and PEN neurons are shown and their synaptic connections are highlighted in both the PB and the EB. Finally, pairs of EPG and PEN neurons are shown to highlight the PEN phase shift in the EB with respect to EPG neurons that innervate the same PB glomerulus.

https://elifesciences.org/articles/66039/figures#video4

**Table 6.** Significance values for wedge-specific modularity in connections between ring and EPG neurons.

| Neuron | Ring to EPG | Ring to EPG | EPG to ring | EPG to ring |
|---|---|---|---|---|
| name | synapse count | relative weight | synapse count | relative weight |
| ER1_a | 0.028 | 0.007 | 0.989 | 0.994 |
| ER1_b | 0.826 | 0.697 | 0.911 | 0.936 |
| ER2_a | 0.626 | 0.592 | 0.993 | 0.993 |
| ER2_b | 0.509 | 0.492 | 0.735 | 0.739 |
| ER2_c | 0.414 | 0.304 | 0.992 | 0.985 |
| ER2_d | 0.074 | 0.074 | 0.947 | 0.946 |
| ER4d | 0.916 | 0.879 | 0.269 | 0.274 |
| ER4m | 0 | 0 | 0.503 | 0.492 |

Each matrix was shuffled 1000 times, and modularity was computed for each shuffled matrix. When computing the modularity of ring neuron inputs to EPG neurons ('ring to EPG'), the connectivity was shuffled from each individual ring neuron to the set of EPG neurons; when computing the modularity of EPG neuron inputs to ring neurons ('EPG to ring'), the connectivity was shuffled from each individual EPG neuron to the set of ring neurons. Table entries indicate the fraction of shuffles for which the modularity of the shuffled data exceeded the modularity of the true (unshuffled) data. An entry of zero indicates a p-value of <0.001; all other entries directly indicate the p-value.

resulting in stronger connections from the left hemisphere for EB wedges on the right half of the EB and stronger connections from the right hemisphere to the left half of the EB. Given the mirror symmetry of the represented axis of polarization (*Hardcastle et al., 2020*), this connectivity pattern may allow the fly to generate a complete, 360° head direction representation from twofold symmetric polarized light input, an idea that we return to in 'Discussion.' Although a recent study reported wind-direction-tuned responses in ER1 neurons (*Okubo et al., 2020*), we believe that the neurons characterized in that study to be only ER1_b neurons – whether the ER1_a neuron type is also mechanosensory is as yet unknown.

## Distributions of ring neuron synapse locations on EPG arbors differ by type

The same plasticity in the EB that enables the fly's internal compass to tether to visual cues in the animal's surroundings (*Fisher et al., 2019*; *Kim et al., 2019*) likely also enables the flexible tethering of the head direction representation to other sensory cues. What if the different sensory streams do not provide a consistent estimate of head direction? This kind of conflict can, for example, arise during translational movement when local visual landmarks provide a less stable estimate of head direction than global cues like wind direction and the polarization pattern of the sky. We asked if the spatial connectivity pattern of ring neurons onto EPG neurons holds any clues about how such conflicts are resolved. Specifically, we considered the electrotonic distances of synapses to the spike initiation zone of a neuron, which is often correlated with the influence that the input can exert on the neuron's ability to fire (or not fire) a spike. We asked if we could extract an expected weighting of head direction-tethering cues conveyed by ring neurons based on the electrotonic distance of their synapses to the estimated spike initiation zone of their EPG neuron targets.

Although little is known about spike initiation zones in fly neurons, the best estimates thus far have come from studies in olfactory PNs (*Gouwens and Wilson, 2009*). In these multipolar neurons, spikes are thought to be initiated near the base of their major input arbors in the AL. These results are also consistent with the subcellular localization of voltage-gated sodium channels in a broader class of central brain neurons (*Ravenscroft et al., 2020*). We thus made the assumption that spikes in the EPG neurons are initiated at the root of their arbors in the EB, near the intersection of these processes with processes that travel to the GA and PB (*Figure 12A*). We found that these putative spike initiation points (yellow circles in *Figure 12A and C*, left) tend to cluster at a certain depth and radial position of the EB. Specifically, EPG neurons tend to enter the EB near a central annulus and from the posterior side. We expected some systematic variation in ring neuron synapse locations by type because different ring neuron types arborize at different depths and radial positions in the EB (*Figure 10—figure supplement 1C*). By grouping the ring neuron types according to their sensory modality, inspired by their overall connectivity in the EB and their known sensory tuning (*Figures 10F and 12B*), we found that the locations of synapses from the ring neurons to the EPG neurons tend to cluster by modality group (*Figure 12C*, right, *Figure 12—figure supplement 1A*, compare also to *Figure 10Cii*). Further, an analysis of the synapses of the ring neurons in these different sensory modality groups onto the EPG neurons revealed a clear trend in their relative electrotonic distances

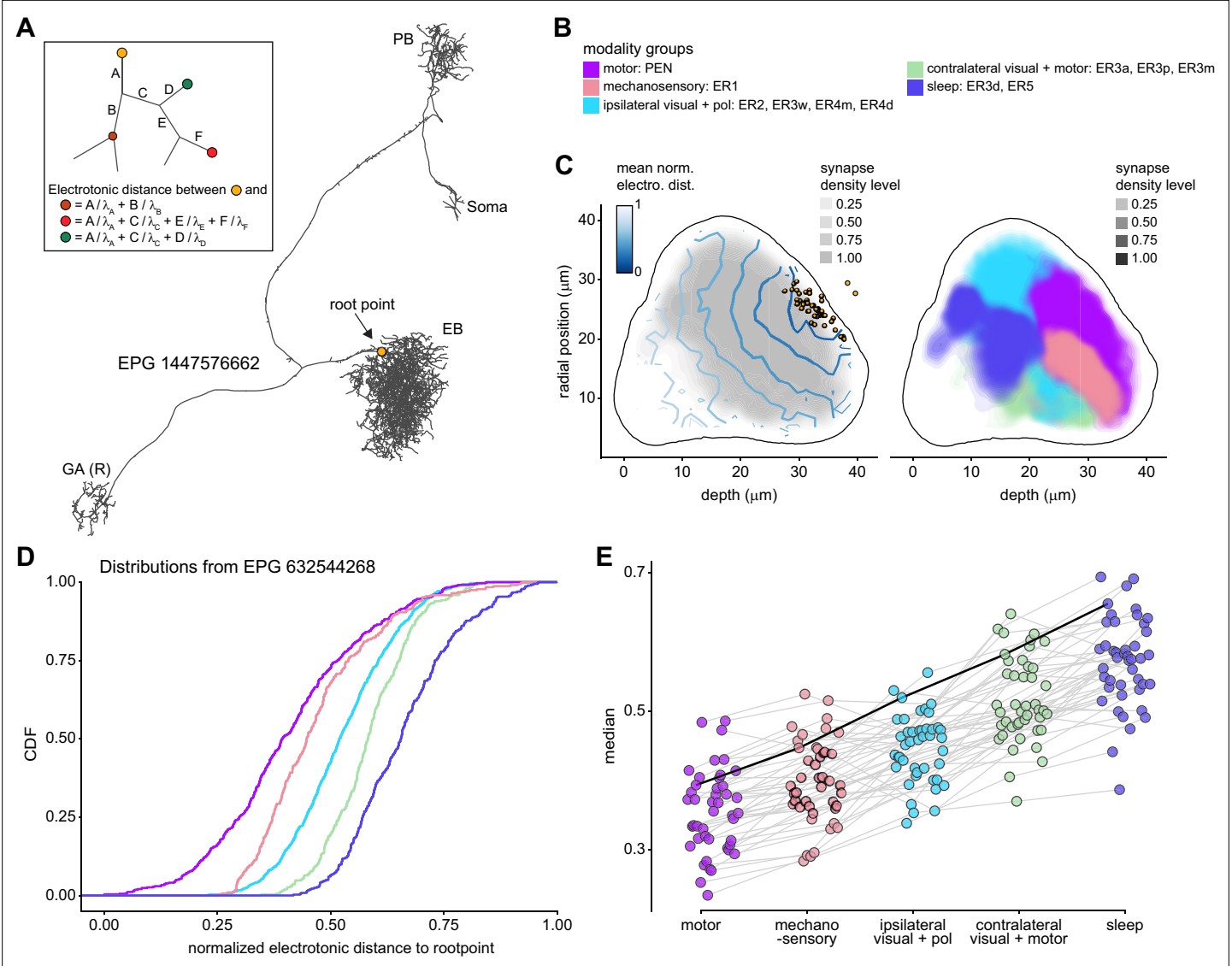

**Figure 12.** Morphology analysis of ring neuron connectivity to EPG neurons. (**A**) Skeleton of a single EPG (id. 1447576662) with the selected root point indicated in yellow. Inset: schematic indicating how the electrotonic distance from a point on the skeleton to the root point is calculated. The Euclidean metric is used to calculate the length of each segment (**A–F**) and λ (for = **A, B, C, D, E, F**) represents the length constants of the edges (see Materials and methods). (**B–E**) Localization of synaptic inputs to EPGs in the ellipsoid body (EB) along the dendritic tree, split by modality group. (**B**) The modality groups, the neuron types that fall into these groups, and the colormap that is used for modality groups for the rest of the panels in this figure. (**C**) Density of synapse locations onto EPGs in the radial vs. depth plane for all EPGs included in the analysis (n = 44). The black outline approximates the EB outline in this plane. Left: synapse locations are shown in gray (included here are synapses from partner types ER, ExR, PEG, PEN, EPG, EPGt). Overlaid contour lines indicate the distribution of the mean of the normalized electrotonic distance from the root. The yellow points indicate where the root points of the EPGs are located in this plane. Right: synapse locations from selected inputs separated and color-coded based on input modality (see **B** for input assignment to modality). (**D**) Cumulative density function (CDF) of the distribution of the normalized electrotonic distance to root for synapses separated by input modalities for a single EPG (id. 632544268). (**E**) Medians of the normalized electrotonic distance distributions grouped by modality. The connecting lines indicate the points corresponding to each individual EPG (n = 44), with the black line corresponding to the EPG whose CDFs are shown in (**D**).

The online version of this article includes the following figure supplement(s) for figure 12:

**Figure supplement 1.** Additional information on the analysis of electrotonic distances of synapse locations of different ring neuron types onto EPG neurons.

**Figure supplement 2.** Comparison of EPG synapse locations by ring neuron type.

**Figure supplement 3.** Morphology analysis of ring neuron connectivity to EL neurons.

to the root node (*Figure 12D and E*). Broadly, as seen in the cumulative distribution function (CDF) of an example EPG (*Figure 12D*), the electrotonic distance between different ring neuron types and the root of each of their postsynaptic EPG neurons suggests an ordering in expected influence on EPG activity, with the mechanosensory neuron synapses being the closest, the different visual neurons being next, and the sleep-related ring neurons potentially exerting the least impact on the EPG neuron's ability to fire a spike. We then compared the locations of ring neuron synapses to those of PEN synapses (labeled 'motor' in *Figure 12B–E*). These self-motion inputs to the fly's head direction system (discussed in later sections) were electrotonically closer to the root of the EPG arbors in the EB than any of the ring neuron inputs (compare left plot with purple region in right plot in *Figure 12C*, *Figure 12—figure supplement 1A*). While there is some variability, these broad trends are consistent across neuron types within the different modality groups (*Figure 12—figure supplement 2*). The rank ordering is fairly consistent across EPG neurons, as can be seen by comparing the medians of the distributions of the synapse locations of ring neurons from each modality group (*Figure 12E*, *Figure 12—figure supplement 1B, C*; see Materials and methods). These same trends were also observed when analyzing the physical distance along the arbor between synapses and the postsynaptic EPG root (*Figure 12—figure supplement 1D, E*). Most synapses were found to occur within 50 μm of the root (*Figure 12—figure supplement 1F*). A similar picture emerged for the EL neuron type, the second columnar neuron type that receives inputs from many ring neurons (*Figure 12—figure supplement 3*; the higher variability across individual ELs is likely due to a greater variation in arbor shape along the circumference of the EB relative to the EPG neurons as seen in *Figure 10—figure supplement 4B*). In summary, our analysis of the electrotonic distances of synapses from different ring neuron types to EPG neurons in the EB suggests a consistent prioritization of sensory inputs to EPG neurons that may reflect the relative importance of different sensory stimuli in driving the fly's head direction estimate. Sculpted inhibition from different sensory ring neurons appears largely focused on the intermediate arbors of the EPG neurons, and excitatory self-motion inputs that update the head direction representation during turns are closer to the root of the EPG neuron arbors in the EB.

## A suppression hierarchy of ring neuron types defined by all-to-all inhibition

Past studies have classified ring neurons based on their morphology and their developmental origins (*Hanesch et al., 1989*; *Omoto et al., 2017*; *Omoto et al., 2018*; *Renn et al., 1999*; *Young and Armstrong, 2010a*). The connectome permitted us to refine the classification of ring neurons based on their pre- and postsynaptic connectivity patterns in different brain regions (*Scheffer et al., 2020*). In this connectivity-based classification of ring neurons into types, a key distinguishing feature was their dense 'within-type' connectivity. Ring neurons are strongly and consistently connected to other ring neurons of the same type. A recent study combined optogenetics with electrophysiology to show that ring neurons inhibit other ring neurons of the same type (*Isaacman-Beck et al., 2019*). Although these data were collected in a single-ring neuron type, the connectome suggests that all-to-all inhibition among cells of a single type is likely a feature of nearly all ring neuron types (blocks around the diagonal in *Figure 13A*). As we outline in 'Discussion,' such all-to-all inhibition may help to minimize the influence of noise on the compass system and effectively enhance the influence of the most prominent sensory cues on the angular position of the EPG bump in the EB.

Notably, some ring neuron types show all-to-all connectivity not just to neurons of their type, but also to neurons of other types. Examples of types that show all-to-all connections include the ER4m, ER2a-d, and ER3w types (off-diagonal blocks in *Figure 13A*). A different heavily connected set includes the ER3_m and all ER3a neurons. In both cases, most – but not all – type-to-type connections are reciprocal. While the first set of ring neuron types synapse directly onto EPG neurons (*Figures 11A and 13B*), the second set mostly interact with EPG neurons indirectly – in the case of the ER3a neurons, through their impact on ER3m neurons (*Figures 11A and 13A*) and EL neurons (*Figure 13B*). To assess the potential impact of across-type connections between ring neurons, we generated a graph for connections between ring neurons, which showed that ring neuron types with similar tuning formed highly interconnected clusters (*Figure 13C*, *Table 5*). We ranked individual types by the numbers of synapses that they make onto EPG neurons – presumably contributing to the generation and maintenance of the head direction signal (*Fisher et al., 2019*; *Kim et al., 2019*; *Figure 13C*, see Materials and methods). For groups of ring neuron types that provide strong input to the EPG neurons, this revealed a second, implicit hierarchy between different ring neuron types

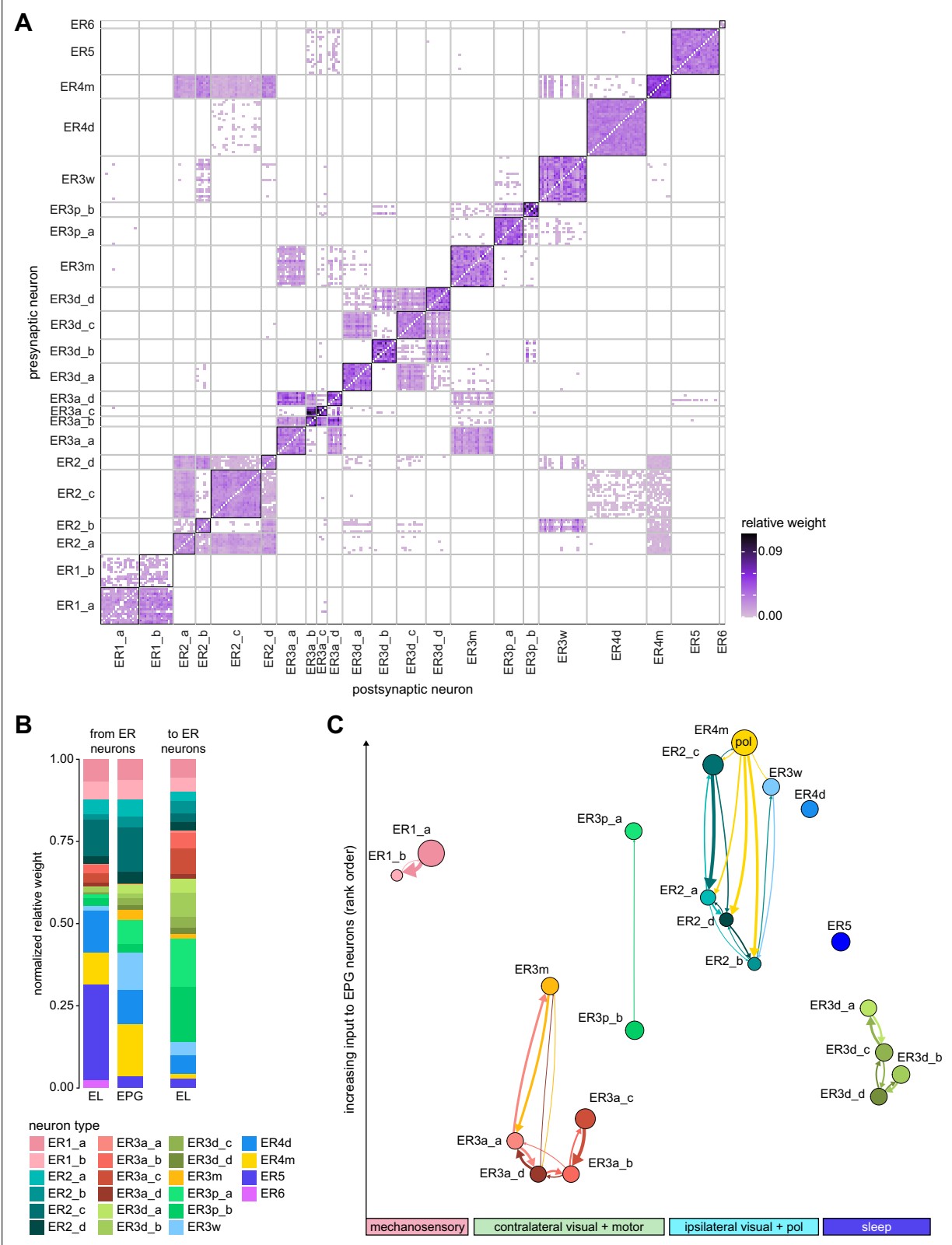

**Figure 13.** Inter-ring neuron connectivity. (**A**) Connectivity matrix for connections between ring neurons in the ellipsoid body (EB) on single neuron level. Connections between neurons of the same type are highlighted with black boxes. (**B**) Normalized contributions of different ring neuron types to EL and EPG neurons (left) vs. normalized contributions of EL neurons to different ring neuron types (right, EPGs make very few synapses to ring neurons, see *Figure 13—figure supplement 1B*). (**C**) Connectivity graph of connections between ring neurons. The graph nodes are arranged along the x-axis to

*Figure 13 continued on next page*

*Figure 13 continued*

group ring neuron types with putatively similar tuning. Vertices are ordered on the y-axis according to their rank-ordered connectivity strength to EPG neurons. Vertex size is scaled by the ratio of the sum of all outputs divided by the sum of all inputs. Only connections with a relative weight of at least 0.05 (5%) are shown. Furthermore, connections between neurons of the same type are not shown.

The online version of this article includes the following figure supplement(s) for figure 13:

**Figure supplement 1.** Connectivity between ellipsoid body (EB) columnar neurons and ring neurons.

(*Figures 11A and 13B*). Prominent among these ring neuron types are those that – based on their TuBu inputs and recent neurophysiological evidence (*Hardcastle et al., 2020*) – we believe to be associated with the sky compass pathway. ER4m, ER2_a-d, and possibly ER3w. All of these types are part of a highly interconnected cluster. Within this cluster, connections between inhibitory ring neurons define a suppression hierarchy. For example, the ER4m neurons, which are most strongly tuned to polarized light e-vector orientation (*Hardcastle et al., 2020*), make a significant number of synapses onto all other neuron types in the cluster (*Figure 13A and C*), and do not receive nearly as many in return (as indicated by the size of the dot representing the neuron type), placing this neuron type at the top of the hierarchy within this cluster. The ER2_c type appears next, inhibiting ER2_a and ER2_d neuron types. ER4d neurons, which likely convey information about a broader range of visual features (*Seelig and Jayaraman, 2013*), are not inhibited by any of the other ring neuron types and make a large number of synapses onto EPG neurons themselves (*Figure 13C*). Other high-ranking ring neuron types include the ER3p_a, which is one of the primary target types of the EL neurons in the EB (*Figure 13B*, *Figure 13—figure supplement 1B*), and the ER1_a and ER1_b neurons (*Figure 12B*), at least one of which is mechanosensory and brings information about wind direction to the EB (likely ER1_b, *Figure 9*; *Okubo et al., 2020*). *Figure 13B* shows the relative contributions of all ring neuron types to tethering the EPG compass in terms of the relative strength of the respective ring neuron inputs to EPG neurons. On top of this, the suppression hierarchy circuit motif could help with selecting a single-ring neuron type for updating the compass when multiple-ring neuron populations are activated, thus effectively establishing a preference for certain sensory compass cues over others.

A few ring neuron types appear to be privileged in also receiving feedback from the EPG neurons (*Figure 13—figure supplement 1B*). This privileged set of ring neuron types includes, once again, the ER4m neurons, which carry polarized light e-vector information. This feedback may serve to amplify their impact on tethering the head direction representation to sensory input, while potentially reducing the influence of ring neurons that carry other types of sensory information.

In summary, we found evidence for several mechanisms by which different sensory inputs to EPG neurons could be integrated, and potential conflicts between cues resolved. First, the strength of connections from various ring neurons onto EPG neurons, as measured by relative weight, varies between types. Polarization-sensitive ER4m neurons provide the strongest input to EPG neurons and are also privileged in receiving feedback from EPG neurons. Consistent, but weaker, ring neuron-to-EPG connections are made by other visually tuned ring neurons and those that bring in mechanosensory information (*Figures 11A and 13B*). Second, the position of ring neuron synapses along EPG neuron arbors varies systematically, not only with ring neuron type but more generally with the type of sensory information carried by different types (*Figure 12E*). This is also reflected in the organization of ring neuron arbors in the EB (*Figure 10Cii*). Based on the electrotonic distance of synapses from the putative spike initiation zone of EPG neurons, self-motion signals from PEN neurons are in a privileged position to excite the compass neurons, while ring neurons carrying mechanosensory, ipsilateral visual, and polarization e-vector cues provide the strongest inhibitory influences (*Figure 12E*). However, the ordering of ring neuron influence on the EPG compass based on synapse location differs slightly from the ranking based on relative connection weights for mechanosensory ER1 neurons and the likely polarization-sensitive ER4m, ER2c, and ER3w neurons (compare *Figure 12E* to *Figure 13C*). Finally, within modality groups, inhibitory ring neurons form suppression hierarchies that may help to select strong guidance signals when multiple sensory cues are present. These suppression hierarchies occur between types that carry potentially related compass information such as the position of the sun (potentially ER2 types) and the celestial polarization pattern (likely ER4m) (*Figure 13C*). Taken together, these circuit motifs may help to resolve guidance cue conflicts and ensure a stable head direction representation.

## Ring neuron connectivity with other neuron types

Some ring neuron types synapse onto some of the other columnar neuron types in the EB. These connections are summarized in *Figure 13—figure supplement 1*. The strongest connections are made by ER6 neurons (already noted above as contacting EL rather than EPG neurons in the EB, *Figure 11A*), which also make a large number of synapses onto PEG and PEN_b (formerly called PEN2) (*Figure 13—figure supplement 1A*). In addition, ER1_a and ER1_b neurons contact PEN_a (formerly PEN1) neurons from both sides of the brain although those connections are rather weak (*Figure 13—figure supplement 1A*). Some ring neuron types receive presynaptic input in the EB from columnar neurons (*Figure 13—figure supplement 1B*). Most of the columnar-to-ring-neuron connections are weak or inconsistent across neurons of the same type (e.g., inputs from PEN_a to ER1_a and ER1_b neurons, or inputs from EPG and PEN_b neurons to ER4m). A notable exception is the strong PEG-to-ER6 neuron connectivity. Most columnar feedback to ring neurons comes from EL neurons, which themselves receive input from several ER neuron types. The EL neurons make synaptic contacts onto most ring neuron types, most strongly to ER3p neurons. We do not yet know the function of these columnar-to-ER connections.

## 'Head direction' versus 'heading direction'

In contrast to mammals, the function of the fly compass has not yet been monitored in head-free animals (*Hulse and Jayaraman, 2019*; *Rubin et al., 2014*; *Taube et al., 1990a*; *Taube et al., 1990b*). This motivated us and others to refer to the representation as being of 'heading direction' rather than 'head direction' until the issue can be conclusively resolved with direct evidence. However, considering that diverse sensory cues are all communicated to the EB in head-centered coordinates, we will now employ the term 'head direction' to refer to the EPG population representation. Note that this is consistent with the terminology that has long been used in studies of the CX in many other insects (*Homberg, 2004*; *Varga and Ritzmann, 2016*), but awaits experimental confirmation in those animals as well.

## ExR neurons connect the EB with numerous other brain regions

In addition to the ring neuron types, which bring information from outside the CX to the EB, another class of neurons termed 'extrinsic ring' (ExR) neurons also form ring-shaped arbors along the circumference of the EB (*Hanesch et al., 1989*). In contrast to most of the ring neurons, ExR neurons are comprised of only one or two neurons per hemisphere and arborize in multiple regions outside the CX (*Figure 14A*). Only a subset of the ExR neurons have been described before and little is known about their function. We identified eight types of ExR neurons, each type targeting distinct sets of brain regions (*Figure 14A*, *Figure 14—figure supplement 1*, *Figure 14—figure supplement 2*). Consistent with their diverse morphologies and projection patterns outside the CX, the inputs and outputs of different ExR types are largely distinct (*Figure 14B*, *Figure 14—figure supplement 3A*). While all ring neurons serve as inputs to the CX, only a subset of the ExR neurons appear to be inputs. Most ExR neuron types have mixed polarity in the EB, and two ExR neuron types are CX output neurons (*Figure 14A*, bottom). The CX input neurons are ExR1 and ExR4, which convey information primarily from the BU, LAL, and GA (*Figure 14A*, *Figure 14—figure supplement 3B*). For a more detailed descriptions of ExR connectivity in the BU, see also *Figure 7B*. The CX output neurons, ExR7 and ExR8, will be discussed in more detail in the output section. The remaining ExR neurons – ExR2, ExR3, ExR5, and ExR6 – have both inputs and outputs outside of the EB, although some do have clear input or output regions (e.g., ExR5, *Figure 14A*).

In the EB, ExR neurons make connections with three groups of neuron types: ring neurons, columnar neurons, and other ExR neurons (*Figure 10F*, *Figure 14—figure supplement 3B*). Most ExR neurons arborize in the posterior part of the EB, with the exception of ExR1 and ExR3, which target more anterior shells of the EB and form a connectivity cluster with sleep-related ring neurons (*Figure 10E and F*, *Figure 14—figure supplement 1*, *Figure 14—figure supplement 2*). Indeed, ExR1 neurons, also called Helicon cells, have been linked to the control of sleep homeostasis (*Donlea et al., 2018*). We will cover ExR1 and ExR3 neurons in more detail in the sleep section. Besides ExR1 and ExR3, ExR5 also receives a large fraction of its EB inputs from ring neurons, specifically from the ipsilateral visual ER4d neurons (*Figure 14—figure supplement 3B*, *Figure 10F*). All ExR neurons make connections to columnar EL and EPG neurons, suggesting a role in modulating the function of the fly's head direction

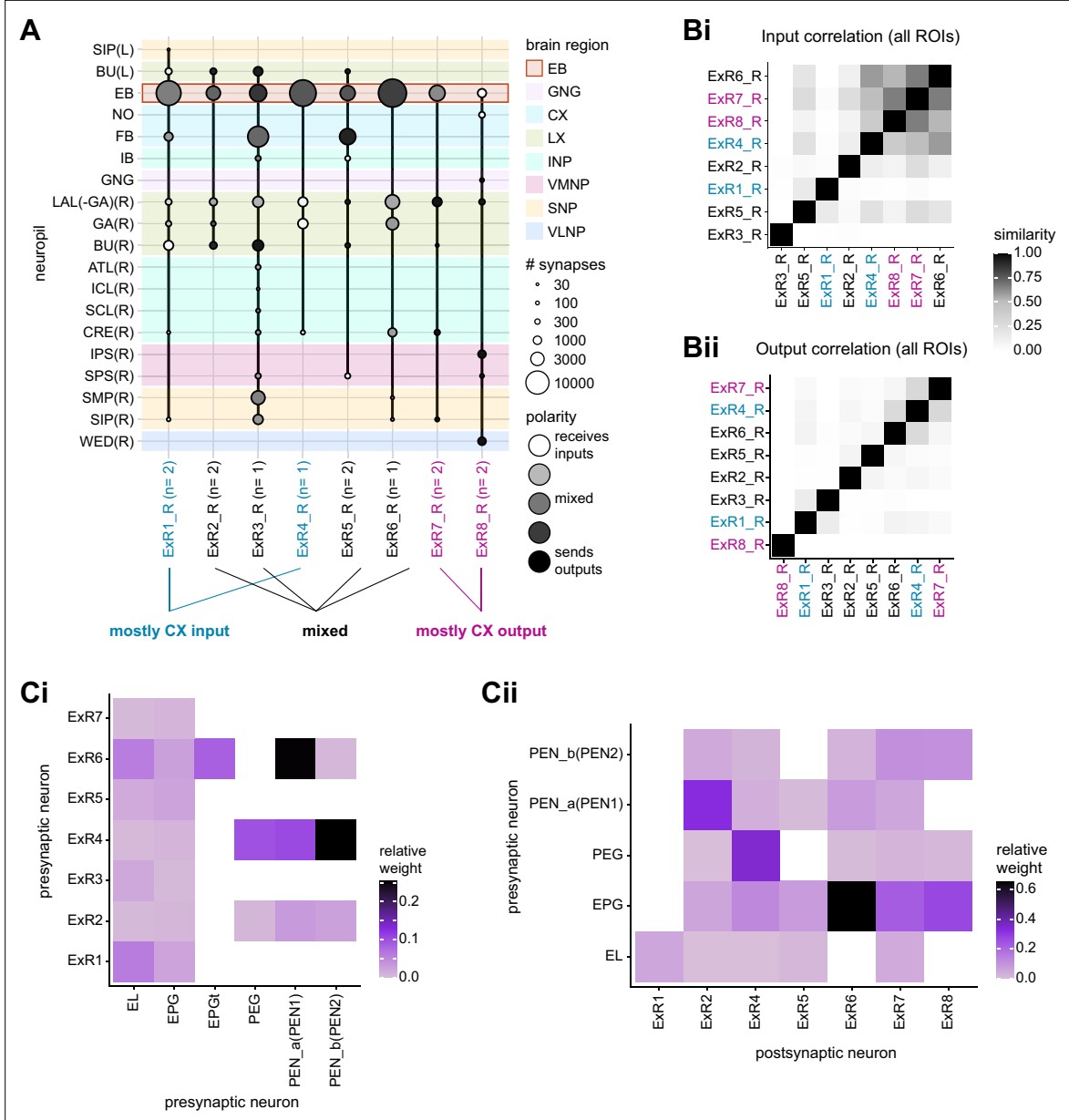

**Figure 14.** Overview of extrinsic ring (ExR) neurons. (**A**) Region arborization plot of all ExR types from the right hemisphere, showing average pre- and postsynaptic counts by region. Indicated below the plot is a qualitative categorization into three groups: mostly input to the ellipsoid body (EB) (blue), mostly output from the EB (pink), and mixed (black). (**B**) Similarity matrices (see Materials and methods) for ExR neurons based on all their inputs (**Bi**) and outputs (**Bii**). ExR-type labels are colored according to groups in (**A**). (**C**) Type-to-type connectivity matrix of ExR to EB columnar neurons (**Ci**) and EB columnar to ExR neurons (**Cii**).

The online version of this article includes the following figure supplement(s) for figure 14:

**Figure supplement 1.** Morphological renderings of all ExR types: ExR1, ExR2, ExR3, and ExR4.

**Figure supplement 2.** Morphological renderings of ExR type. ExR4, ExR5, ExR6, ExR7, ExR8.

**Figure supplement 3.** Comparison of inputs and outputs of extrinsic ring (ExR) neurons.

**Figure supplement 4.** Neuron-to-neuron connectivity matrices for connections between extrinsic ring (ExR) and columnar neurons in the ellipsoid body (EB).

representation (*Figure 14Ci*, *Figure 14—figure supplement 4A*). Connectivity with EPGt, PEG, and PEN neurons is sparser, with ExR4 providing strong input to PEG and both PEN neuron types, and ExR6 selectively contacting PEN_a and EPGt neuron types (*Figure 14Ci*, *Figure 14—figure supplement 4A*). Columnar neurons also feedback onto ExR neurons (*Figure 14Cii*, *Figure 14—figure supplement 4B*).

Many ExR neurons make direct and indirect connections to the same partner types in multiple brain regions, suggesting that they do more than just act as input and output pathways for the EB. We analyzed these different connectivity motifs by comparing ExR connections within the EB to their direct and indirect connections in other regions (*Figure 15A*). We restricted our analysis to downstream partners of ExRs and focused on three specific connectivity motifs: 'parallel connections,' 'canonical feedback,' and 'linked targets' (see schematic in *Figure 15A*). 'Parallel connections' describe a motif in which ExR neurons make direct or indirect connections outside the EB to the same neuron type that they also contact in the EB (red arrow in *Figure 15A*, left). The 'canonical feedback' motif covers cases in which ExR neurons directly or indirectly connect outside the EB to neurons that feed back onto ExR neurons in the EB (yellow arrow in *Figure 15A*, middle). The third motif captures cases where downstream partners of ExR neurons are themselves connected in the EB, making them 'linked targets' (green arrow in *Figure 15A*, right). Not all ExR neuron types engage in these motifs. ExR1 and ExR4, because they are primarily input neurons to the EB, and ExR8, because it is purely an output neuron of the EB. Among the others, the linked targets motif is most commonly observed, potentially allowing these ExR neurons to link the activity of related circuitry in multiple brain regions (*Figure 15B*). The majority of motifs are formed through ring neurons (*Figure 15C*), suggesting that ExR neurons form a feedback loop onto one of the primary input pathways to the EB.

We took a closer look at the two large ExR neurons that participate in the strongest out-of-EB connection motifs: ExR2 and ExR3 (*Figure 15D*). ExR2 are DANs of the PPM3 cluster and have been linked to ethanol-induced hyperactivity and the control of circadian activity peaks (*Kong et al., 2010*; *Liang et al., 2019*; *Nässel and Elekes, 1992*; *Omoto et al., 2018*). Outside of the EB, ExR2 neurons receive input in the LAL and EB and send outputs primarily to the BU (*Figure 14A*, *Figure 14—figure supplement 3B*). Through these BU connections, ExR2 neurons directly and indirectly target many of the same ring neurons that they also contact in the EB, specifically ER neurons from the ipsilateral visually-responsive cluster (*Figure 15Ei and Fi*, *Figure 8B*). Thus, ExR2 neurons may modulate the fly's motor activity by regulating visual inputs to the fly's head direction system. The putatively serotonergic ExR3 neurons form parallel connections and feedback connections with three sets of highly interconnected ring neuron populations (*Figure 15C and Eii*). One group of ring neuron partners is the ipsilateral visually tuned cluster of ring neurons, which is also targeted by ExR2. A second group is the ER3a/ER3m cluster, with which ExR3 forms parallel and a few feedback connections in the EB. The third and final group is the ER3d cluster, which may play a role in sleep-wake control (see Figure 53). ExR3 neurons contact ER3d neurons directly in both the EB and the BU, and indirectly through TuBu neurons, forming parallel and feedback connections with each ER3d neuron in the EB (*Figure 15Eii and Fii*, *Figure 8B*).

Taken together, our analysis suggests that ExR neurons are highly diverse but can be broadly grouped into types that primarily provide input to the EB, those that serve as output pathways from the EB and those that appear to be mixed input and output neurons; ExR neurons in the last group modulate input pathways to the EB through connections both inside and outside the EB.

## A ring attractor network formed by recurrent loops between the EB and PB

The architecture of the fly's head direction system has already been described in some detail in a series of experimental studies. These studies identified network motifs that likely underlie the generation, maintenance, and updating of the fly's head direction representation (*Green et al., 2017*; *Green and Maimon, 2018*; *Hulse and Jayaraman, 2019*; *Turner-Evans et al., 2017*; *Turner-Evans et al., 2020*). These motifs, the network's dynamics, and the system's responses to perturbation strongly suggest that the network implements a type of ring attractor (*Kim et al., 2017c*; *Turner-Evans et al., 2017*; *Turner-Evans et al., 2020*). In this section, we describe the connectivity of columnar neurons that form the core of the network. These four classes of EB-PB columnar neurons arborize in single glomeruli in the PB and in localized regions of the EB, while also sending processes to a third structure (either the

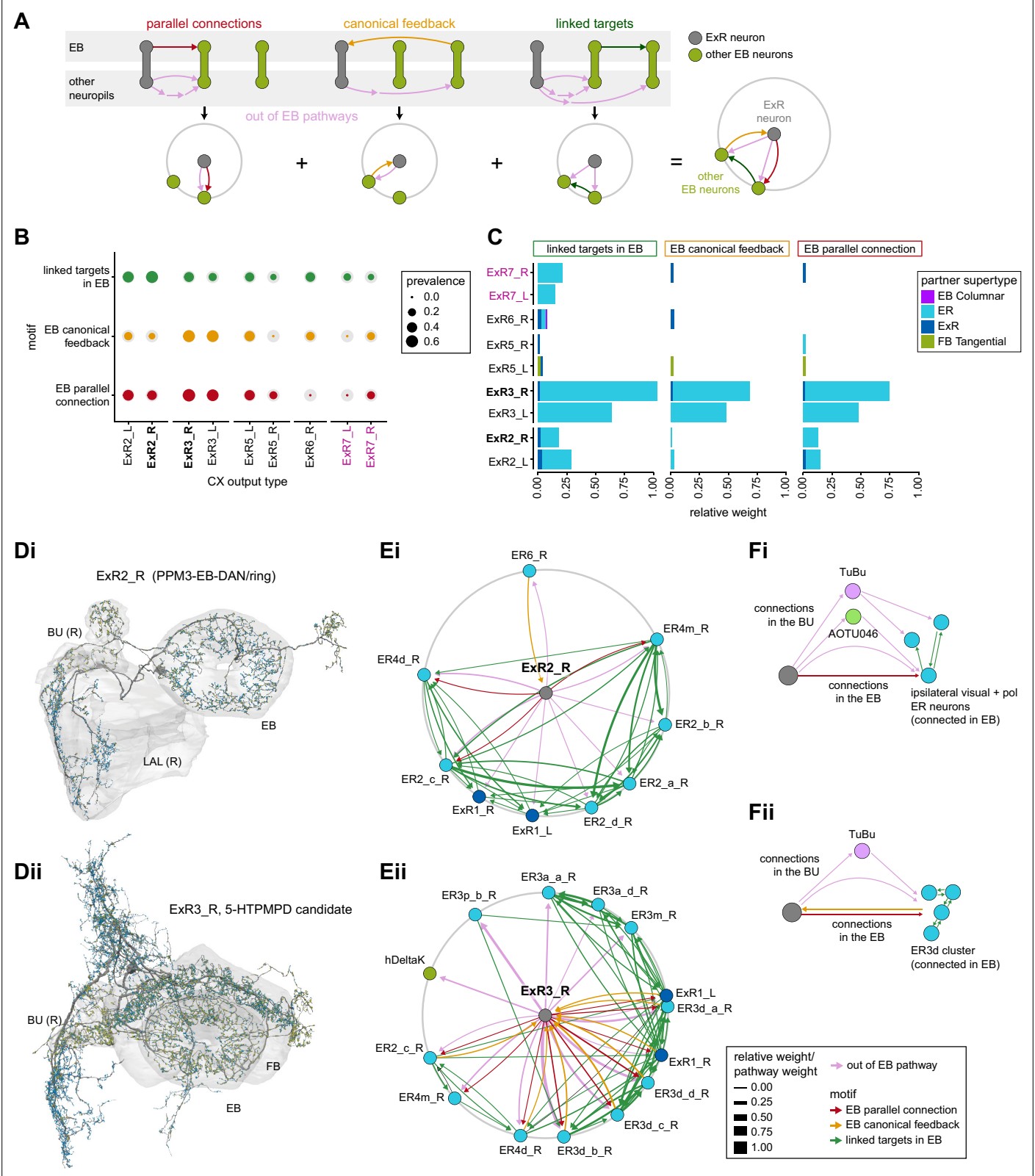

**Figure 15.** Extrinsic ring (ExR) connectivity motifs. (**A**) Schematic explaining the ExR connectivity motif analysis, which compares connectivity within the ellipsoid body (EB) to connectivity outside the EB. The top row shows the three circuit motifs that were considered, and the bottom row their equivalent representation in a compact circular network plot. Here we compare connections from ExR to other EB neurons outside and inside the EB. We only consider out-of-EB pathways for ExR neurons. The out-of-EB pathways can be direct or indirect connections (pink arrows) to other EB neurons (in green).

*Figure 15 continued on next page*

*Figure 15 continued*

'Parallel connections' occur when the source neurons also contact the pathway target neuron inside the central complex (CX) (in red). The 'canonical feedback' motif describes the case where the target of the pathway contacts the source type in the CX (in yellow). 'Linked targets' are neurons connected in the CX that are targets of the same neuron outside of the CX (in green). (**B**) Summary of motif prevalence across different ExR types. The colored circles represent the prevalence of each specific motif, whereas the gray circles represent the total number of all the motifs of the same type that could form given that type's partners outside of the CX (normalized per type and motif). (**C**) Bar graph showing the contribution (measured by relative weight) of ExR partners in the EB to the observed connectivity motifs. The sum of the relative weights of each connection for an ExR to its partner is shown, separated by motif and partner type. (**D**) Morphological rendering of one ExR2_R (**Di**) and ExR3_R (**Dii**). Some of the innervated brain regions are shown in gray. Blue dots mark postsynaptic sites, and yellow dots mark presynaptic sites. (**E**) Graphical representation of connectivity motifs as depicted in (**A**) for ExR2_R (**Ei**) and ExR3_R (**Eii**). (**F**) Schematic relating groups of connectivity motifs in ExR2 (**Fi**) and ExR3 (**Fii**) to the anatomical location of the connections that are involved.

GA or the NO). We do not describe ring attractor theories in any detail here, but rather encourage the reader to refer to prior work for more information on the significance of attractor networks.

## EB-PB columnar neurons: recurrent partners that maintain and update the compass

The activity in the EPG neurons represents the fly's head direction. As shown in *Figure 16A*, each EPG neuron connects an EB wedge with a single glomerulus in the PB, while also innervating the GA (discussed in later sections). The PB is mirror symmetric in appearance, with a left half and a right half. The 16 EPG wedges in the EB alternate so that half go to the right PB while half go to the left. In this

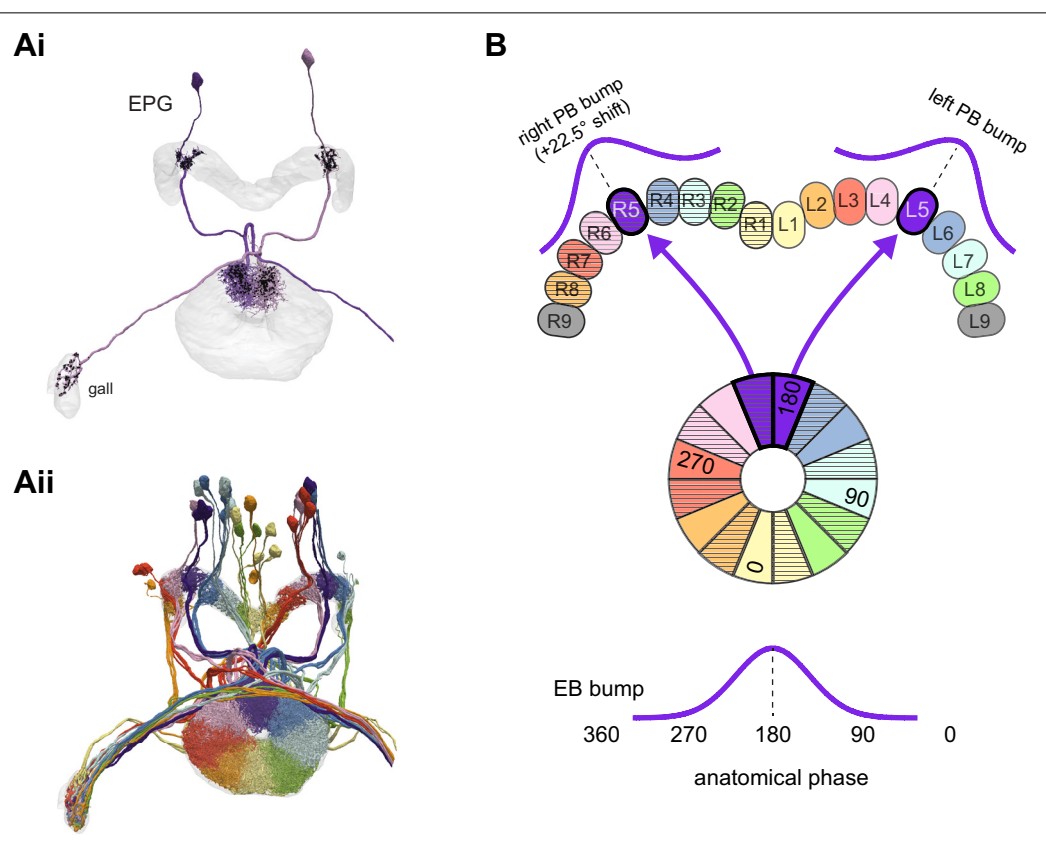

**Figure 16.** EPGs connect the ellipsoid body (EB) to the protocerebral bridge (PB). (**A**, **Ai**) A morphological rendering of two EPG neurons. Black dots are presynaptic sites. (**Aii**) A morphological rendering of the entire population of EPG neurons, color-coded by PB glomerulus. (**B**) Schematic showing where the EPG processes arborize in the EB and in the PB. The EPG neurons map the different locations around the ring of the EB to the right and the left PB. A fictive bump of activity in the EB will therefore split into both a right and a left bump of activity in the PB. Note that the bumps in the PB are slightly shifted with respect to one another due to the 22.5° offset between the right- and left-projecting wedges in the EB.

way, the EPG neurons map the different locations around the ring of the EB to both the right and the left PB (*Figure 16B*). The EPG neurons bring the head direction signal to the PB from the EB. A bump of activity in the EPG neurons in the EB will therefore manifest as two bumps in the PB, with one on either side. Due to the alternating left and right projections of EB wedges to either the right or the left PB, the bump on the right side will be shifted 22.5° (360°/16 wedges) with respect to the bump on the left (*Wolff et al., 2015*), as has been observed experimentally (*Lyu et al., 2020*).

The remaining three types of columnar neurons linking the EB and PB – the PEN_a, PEN_b, and PEG neurons – are mainly postsynaptic in the PB and complete direct and indirect recurrent loops with the EPG neurons in the EB. These loops update and maintain the head direction representation, as will be described in more detail in the following section (*Green et al., 2017*; *Green and Maimon, 2018*; *Hulse and Jayaraman, 2019*; *Turner-Evans et al., 2017*; *Turner-Evans et al., 2020*). As noted above, the hemibrain dataset does not contain the previously identified 'canal' cell (*Wolff and Rubin, 2018*).

## Updating head direction by integrating angular velocity input

We recently used EM reconstruction to identify the network motifs underlying the interaction of the PB-EB columnar neurons (*Turner-Evans et al., 2020*). However, this study, which was based primarily on the Full Adult Fly Brain (FAFB) dataset (*Zheng et al., 2018*), relied on manual reconstruction of only a fraction of the circuit. The completeness of the hemibrain connectome has now allowed us to more thoroughly examine the compass subnetwork and exposed additional structure-function relationships within the compass circuit, which we discuss below.

The EPG activity bump moves around the EB either clockwise (CW) or counterclockwise (CCW), depending on which direction the fly turns (head movements are also likely to update the EPG bump, but this has yet to be established). The angular velocity input that moves the bump in the EB is thought to primarily come from the PEN neurons (*Green et al., 2017*; *Turner-Evans et al., 2017*; *Turner-Evans et al., 2020*). There are two types of PEN neurons, PEN_a and PEN_b, both of which are synaptically connected in recurrent loops with EPG neurons (*Turner-Evans et al., 2020*), are conjunctively tuned to head direction and angular velocity (*Green et al., 2017*; *Turner-Evans et al., 2017*; *Turner-Evans et al., 2020*), and likely receive angular velocity inputs in the NO (discussed in a later section). These recurrent loops have either a CW or CCW shift in their projection patterns. Considering that the EPG head direction representation spans 360° of angular space, we can describe these projection patterns as 'anatomical phase shifts' relative to the EPG population (*Figure 17A*, schematized in *Figure 17B*). That is, PEN neurons receive their head direction inputs from a given EPG neuron in the PB. They then send their projections to an EB tile that is shifted CW or CCW from the EB wedge innervated by the presynaptic EPG neuron. The functional consequence of this phase shift is that the phase (angle) of the PEN activity bump is likely to be shifted relative to the phase (angle) of the EPG activity bump. This phase shift is in opposite directions for PEN neurons with arbors in the right vs. left side of the PB (see sample images of EPG and PEN_a neurons in *Figures 16A and 17A*; connectivity matrices in *Figure 17C and D*). As discussed in later sections, PB-FB neurons also have anatomical phase shifts.

A key assumption behind fly compass models is that recurrent loops involving EPG and PEN neurons allow activity to propagate all the way around the EB and in either direction. Consistent with these models, we found that PEN neurons from each PB glomerulus contact EPG neurons in every wedge of the EB (*Figure 17C*). EPG neurons then complete the loop by synapsing onto PEN neurons in all but the most lateral glomeruli on each side of the PB. The apparent disconnect in connectivity at the edges of the PB is likely addressed by an additional cell type, as discussed below. Importantly, even without this bridging cell type, EPG and PEN_a neurons are so densely interconnected within the EB that a network graph of their subnetwork forms a ring (*Figure 17E*; see Materials and methods). The PEN_b neurons' connectivity to the EPG neurons looks much like the PEN_a neurons' connectivity to the EPG neurons in both the EB and PB. This similarity in connectivity does not extend to all partners of the PENs, however. Indeed, each type has unique partners in both brain regions (*Figure 17—figure supplement 1*).

The PEN neurons in the outermost (ninth) glomeruli do not receive any EPG input from the EPG neurons that project to the eighth glomerulus. Instead, a key neuron type, EPGt, arborizes in that outer glomerulus and appears to fill the 'gap' (*Wolff et al., 2015*; *Figure 18A*). While the EPGt neurons have very similar connectivity to the EPG neurons in the PB (*Figure 18B*), they receive far fewer synaptic inputs in the EB (*Figure 18C*), with a striking sparsity of ring neuron inputs (*Figure 18D*). However,

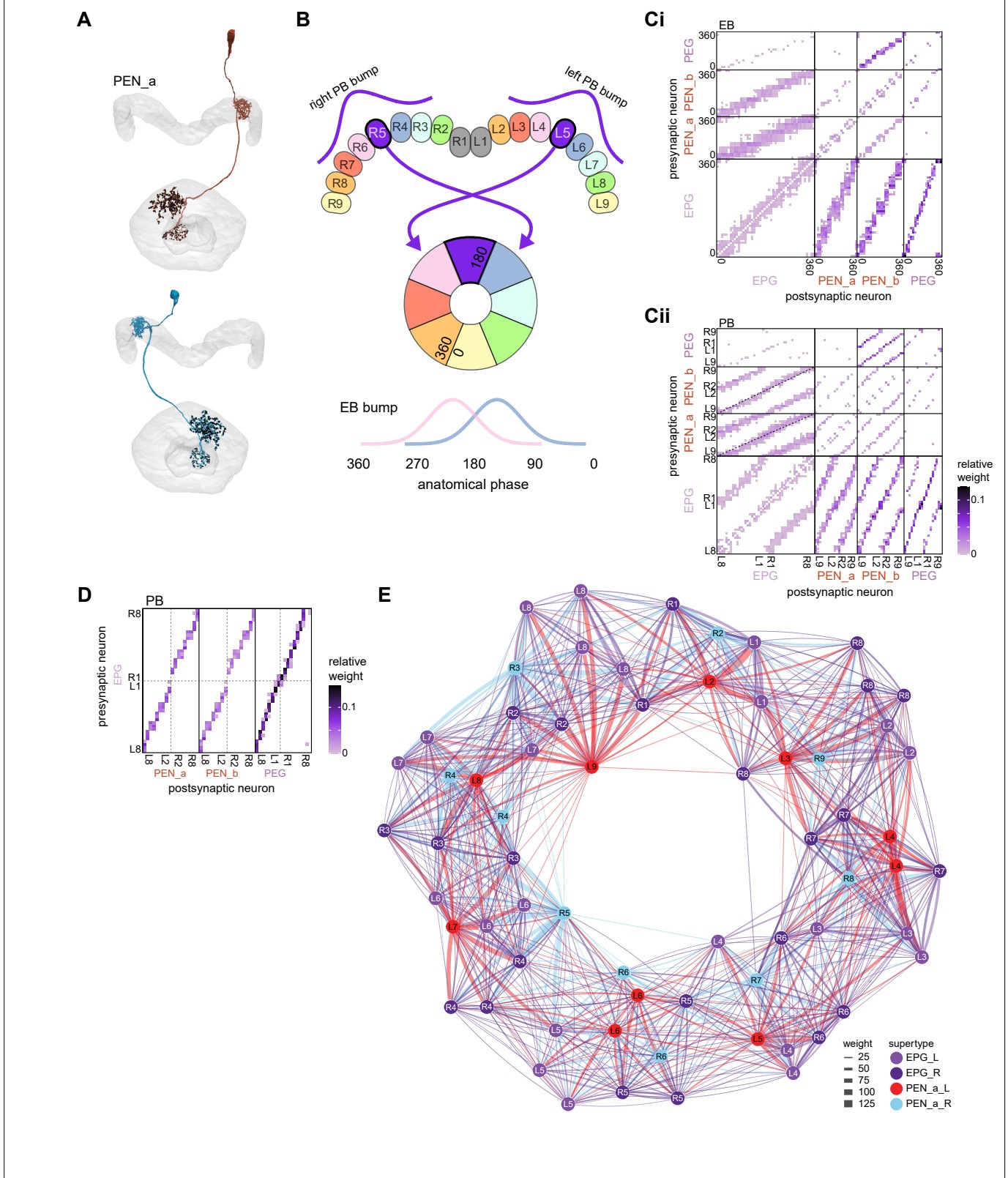

**Figure 17.** PEN_a neurons connect the protocerebral bridge (PB) back to the ellipsoid body (EB), with a shift, forming feedback loops with the EPG neurons. (**A**, top) PEN_a neurons on the left side of the PB send projections to the EB that are counterclockwise shifted with respect to the EB processes of their EPG inputs in the PB (see *Figure 16*). (Bottom) PEN_a neurons on the right side of the PB send projections to the EB that are clockwise shifted with respect to the EB processes of their EPG inputs in the PB. Black dots are presynaptic sites. (**B**) Schematic showing where the PEN_a processes

*Figure 17 continued on next page*

*Figure 17 continued*

arborize in the EB and in the PB. The processes in the right PB project to different locations in the EB than the processes in the matched glomerulus in the left PB. A bump of activity at the same location in the right and left PB will therefore form two shifted bumps of activity in the EB. The EB processes of the PEN_a neurons form eight equiangular tiles, each of which covers two of the EPG wedges. (**C**) Neuron-to-neuron connectivity matrix for EPG, PEN_a, PEN_b, and PEG neurons in the EB. The neurons are arranged according to their angular position in the EB (**Ci**) or according to their arrangement in the PB (**Cii**). Dotted lines are overlaid on the diagonal of the PEN to EPG quadrants to emphasize the offset in connectivity. Though not represented in the axis labels, multiple neurons often cover the same angle or arborize in the same PB glomerulus. (**D**) Neuron-to-neuron connectivity matrix for EPG, PEN_a, PEN_b, and PEG neurons in the PB. The EPG neurons directly connect to the PEN_a, PEN_b, or PEG neurons in glomeruli where they both have processes (L2–L8 for the PEN neurons and L1–L8 for the PEG neurons). The EPG neurons also occasionally synapse onto partners in neighboring glomeruli. As in (**C**), multiple neurons often cover the same glomerulus. (**E**) A force-directed network layout of the EPG and PEN_a connections. Weight refers to the number of synapses between partners.

The online version of this article includes the following figure supplement(s) for figure 17:

**Figure supplement 1.** PEN_a and PEN_b connectivity.

---

EPGt neurons do receive PEN input in the EB. The wedges at the bottom of the EB are innervated by both the EPGt neurons from glomerulus nine and by the EPG neurons that project to glomerulus 1, potentially completing the loop (*Figure 18E*).

The PEG neurons also appear to map the nine glomeruli of the PB to eight tiles in the EB though they do so without a phase shift. The PEG neurons form a recurrent loop with the EPG neurons, as reported previously (*Turner-Evans et al., 2020*). Briefly, they receive input from the EPG neurons in the PB (*Figure 17D*, right columns) and synapse onto the PEN_b neurons in the EB (*Figure 17C*, top rows, second group of columns to the right). The PEN_b neurons synapse onto the EPG neurons in the EB, thereby completing the loop (*Figure 17C*, second group of rows from the top, first group of columns). Although some PEG neurons do synapse directly onto EPG neurons in the EB, these connections are sparse and feature only a few synapses. By contrast, the PEG-to-PEN_b and PEN_b--to-EPG connections are both strong and consistent around the ring. There are nine PEG neurons on each side of the PB, one for each glomerulus. Each connects to one of the eight PEN_b neurons in the EB, with PEG neurons from both glomerulus 1 and glomerulus 9 connecting to PEN_b neurons from PB glomerulus 2. This connectivity pattern matches the connectivity of EPG neurons from glomerulus 1 and EPGt neurons from glomerulus 9 to PEN_b neurons from glomerulus 2.

## The PB: reshaping the compass signal for navigational computations

The handlebar-shaped PB is conserved across hexapods and some crustaceans (*Bullock and Horridge, 1965*; *Homberg, 2008*; *Strausfeld, 1976*; *Strausfeld, 2012*), including species as distant as locusts (*Homberg, 1991*), flies (*Lin et al., 2013*; *Phillips-Portillo, 2012*; *Wolff et al., 2015*), and crayfish (*Sandeman et al., 1990*). This structure has been associated with a wide range of locomotor behaviors (*Harley and Ritzmann, 2010*; *Krause et al., 2019*; *Poeck et al., 2008*; *Strauss et al., 1992*), and neural activity in the PB is known to carry the head direction signal (*Bockhorst and Homberg, 2015*; *Giraldo et al., 2018*; *Green et al., 2017*; *Heinze and Homberg, 2007*; *Pegel et al., 2018*; *Turner-Evans et al., 2017*; *Turner-Evans et al., 2020*; *Zittrell et al., 2020*). The PB, in fact, is where sensory-driven activity with a compass-like anatomical organization was first reported in the insect brain (*Heinze and Homberg, 2007*).

Above, we discussed the EB-PB recurrent network, which supports the function of the head direction system through the interaction of EPG, EPGt, PEG, PEN_a, and PEN_b neurons. EPG and EPGt neurons bring columnar input from the EB into the PB, and the other neuron types take information from the PB (and accessory regions) to the EB (*Figure 19A*). As noted earlier, the hemibrain dataset does not contain one type of previously identified PB-EB columnar neuron, the 'canal' cells (*Wolff and Rubin, 2018*).

The PB also connects to other brain regions. Consistent with descriptions based on light microscopy (*Lin et al., 2013*; *Wolff et al., 2015*), the hemibrain connectome shows that the PB receives much of its non-CX input from three accessory structures. the IB, SPS, and inferior posterior slope (IPS) (*Figure 19Ai*). The FB provides one final source of PB input through a single PB-FB columnar neuron type, to be discussed later. Many other PB-FB columnar neuron types convey information in the other direction, sending outputs to the FB, CRE, and LAL from the PB (*Figure 19Aii*). Notably, the LAL is innervated by DNs. The completeness of the hemibrain connectome in the CX allowed us to

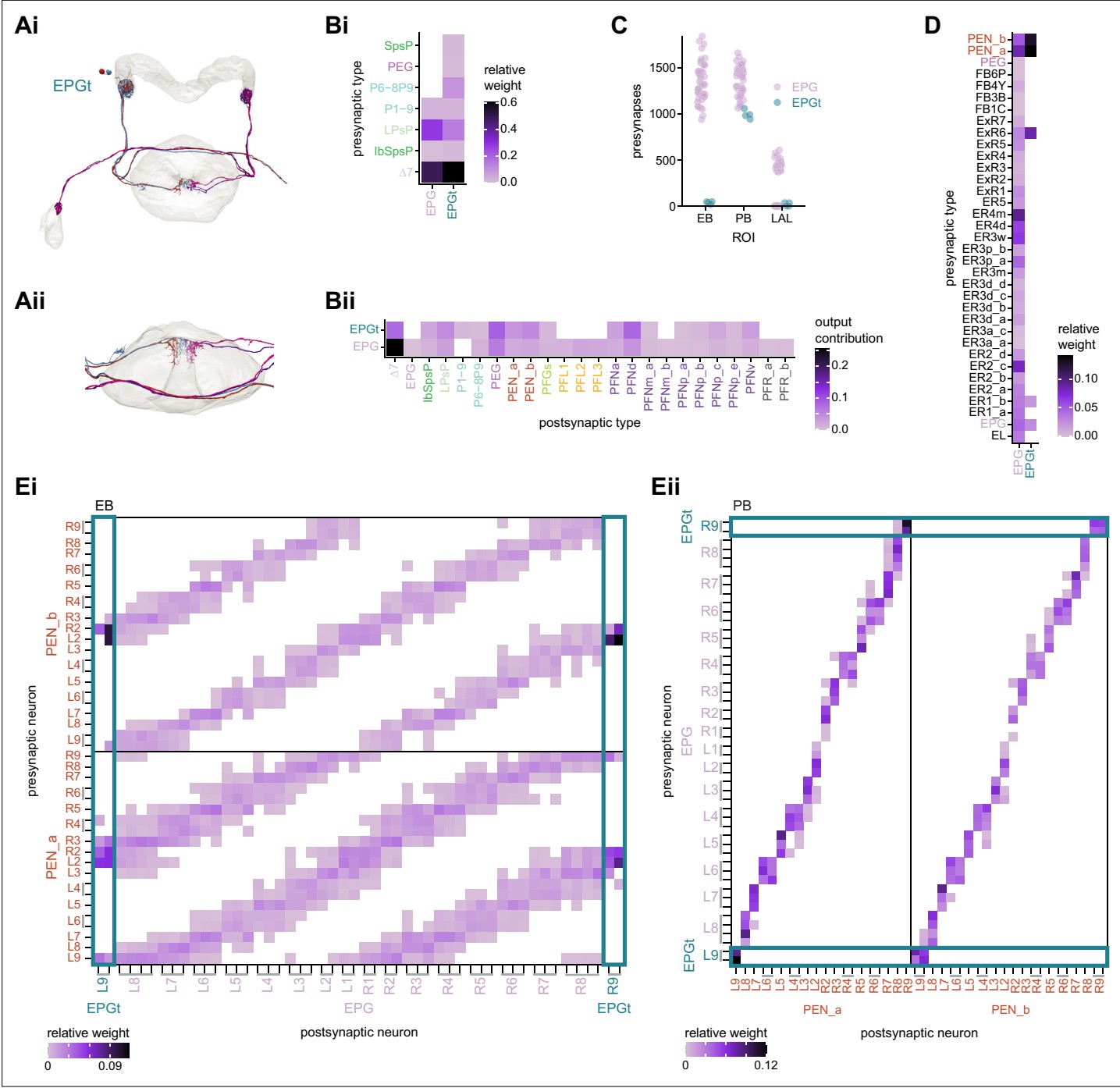

**Figure 18.** EPGt neurons extend EPG-like connectivity. (**A**) Morphological renderings of all EPGt neurons. The EPGt neurons arborize only in glomeruli L9 and R9 in the protocerebral bridge (PB) and, in the ellipsoid body (EB), their arbors line the canal at the bottom of the torus (**Ai**). A side view of the EB shows the position of EPGt processes in the EB (**Aii**). (**B**) Type-to-type connectivity matrix showing the inputs (**Bi**) and outputs (**Bii**) for the EPG and EPGt neurons in the PB. (**C**) Total number of presynaptic sites for the EPG and EPGt neurons by brain region. (**D**) Type-to-type connectivity matrix showing the inputs to the EPG and EPGt neurons in the EB. (**E**) Neuron-to-neuron input connectivity from the PEN_a and PEN_b neurons to the EPG and EPGt neurons in the EB (**Ei**) and outputs from the EPG and EPGt neurons to the PEN_a and PEN_b neurons in the PB (**Eii**).

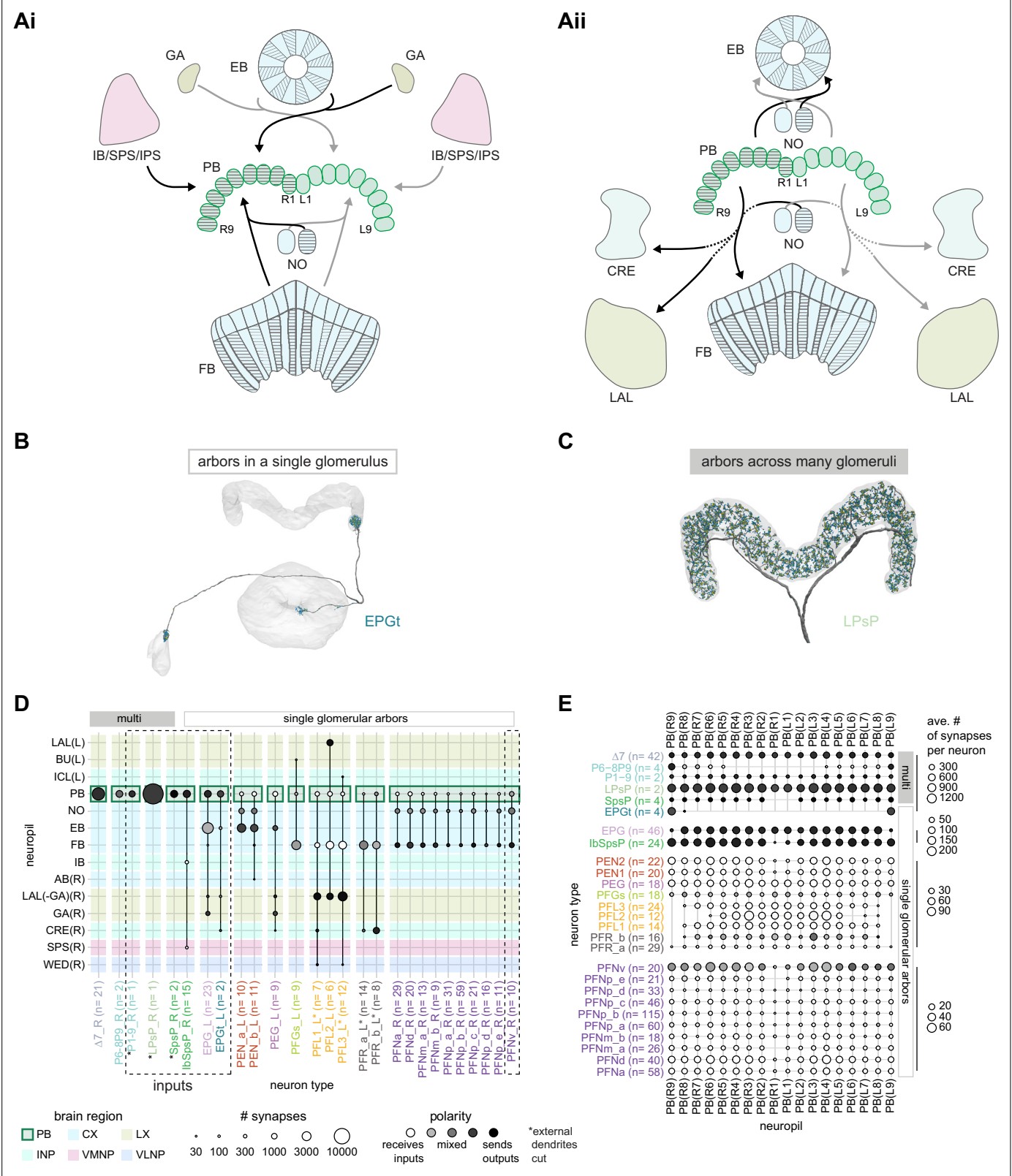

**Figure 19.** An overview of the protocerebral bridge. (**A**) A diagram of the input (**Ai**) and output (**Aii**) pathways for the protocerebral bridge (PB). Connected brain regions include the ellipsoid body (EB), the inferior bridge (IB), the superior posterior slope (SPS), the posterior slope (PS), the crepine (CRE), the lateral accessory lobe (LAL), the fan-shaped body (FB), and the noduli (NO). (**B**) Morphological rendering of an EPGt neuron, which only arborizes in a single glomerulus in the PB. Yellow dots mark presynaptic site. Blue dots mark postsynaptic sites. (**C**) Morphological rendering as in (**B**) of

*Figure 19 continued on next page*

Figure 19 continued

an LPsP neuron, which has arbors throughout the PB. Yellow dots mark presynaptic site. Blue dots mark postsynaptic sites. (**D**) Region arborization plot for each neuron type that contains arbors in the PB. Neuron types that provide input to the PB are denoted by the dashed vertical boxes. The horizontal boxes at top indicate which neurons arborize in multiple glomeruli (filled gray boxes) and which arborize in single glomeruli (gray outline). (**E**) The average number of synapses per neuron in each PB glomerulus for each neuron type that contains arbors in the PB.

split a few of the previously identified PB-FB columnar neuron types into multiple new, distinct types, including some subtypes that were distinguishable less by their morphology than by their connectivity patterns.

Although most PB neurons are columnar and arborize in only a single PB glomerulus (*Figure 19B*), the region also contains several multi-glomerular neuron types (*Figure 19C*), including two types of interneurons (*Figure 19D*). One of the PB interneuron types, the Δ7 neurons, innervates all PB glomeruli, as do the dopaminergic LPsP neurons and the octopaminergic P1-9 neurons (*Figure 19E*, see rows labeled 'multi' at right) (*Wolff et al., 2015*). The other multi-glomerular neuron types innervate at least four glomeruli each. The columnar neuron types, on the other hand, vary in their coverage of PB glomeruli. Individual columnar neurons primarily innervate one glomerulus. Across the population, the columnar neurons of most types innervate contiguous subsets of glomeruli and exclude either the innermost or outermost glomeruli (*Figure 19E*, see rows labeled 'single glomerular arbors' at right), thereby sampling the full 360° of the HD representation in the left and right PB.

## The Δ7 neurons: sinusoidal reformatting of the head direction signal

The direct connection from EPG neurons to PEN neurons in the PB forms a key part of the EB-PB recurrent loop, which updates the fly's head direction representation with self-motion input. However, this EPG input is also transformed by a population of multi-glomerular interneurons, the glutamatergic Δ7 neurons (*Figure 20A*; *Daniels et al., 2008*; *Turner-Evans et al., 2020*). The Δ7 neurons provide a strong, albeit indirect, link from EPG neurons to all other columnar neurons.

The PB-spanning Δ7 neurons and their homologs in other insects have long been believed to play a role in navigational computations. Their responses to polarized light e-vector stimuli suggested that they were organized in map-like, polarotopic fashion in the locust brain (*Heinze and Homberg, 2007*). More recently, imaging and perturbation experiments in the fly have confirmed the importance of these glutamatergic neurons to the function of the fly compass circuit (*Turner-Evans et al., 2020*). Although this recent work also used EM to identify synaptic connections between Δ7 neurons and EPG, PEG, PEN_a, and PEN_b neurons, it reconstructed only a subset of the processes in a few Δ7 neurons. The hemibrain connectome allowed us to identify the complete set of inputs and outputs of Δ7 neurons across the entire PB. The picture that emerges from this analysis is of a neuron type that is the central hub of the PB. all neurons downstream of EPG neurons receive head direction input that is also processed by theΔ7 neurons (*Figure 20B*, *Figure 20—figure supplements 1 and 2*).

A key question raised by the hub-like connectivity of the Δ7 neurons is how these neurons pass on the EPG head direction signal. Many navigational algorithms combine head direction signals with information about the animal's movements to compute a vector representation of the animal's position. In these models, the vector representation is stored in sinusoidal activity patterns, where the phase of the sine wave encodes the angle of the vector and the amplitude of the sine wave represents its length (*Pisokas et al., 2020*; *Stone et al., 2017*; *Touretzky et al., 1993*; *Wittmann and Schwegler, 1995*).

We therefore asked if the Δ7 neurons transform the local EPG head direction signal into sinusoidal activity patterns in the Δ7 neurons' downstream partners. Each Δ7 neuron receives smoothly varying input from EPG neurons across the PB (*Figure 20C*, upper-left quadrant). Aligning and averaging these input profiles across all Δ7 neurons revealed that the EPG input to Δ7 neurons was in fact well fit by a cosine (*Figure 20D*, see Materials and methods) (similar observations are made by a parallel study [*Lyu et al., 2020*]). Each Δ7 neuron sends output to specific PB glomeruli that are seven glomeruli apart (*Figure 20A and C*, lower-right quadrant). Since the output is highly targeted and the Δ7 neurons as a population continuously cover the entire PB, the sinusoidal input profile is unlikely to be further shaped by the Δ7 neurons' outputs.

To further test if any input from the EPG population will be reformatted into a sinusoid, we simulated EPG activity propagating through the Δ7 neurons to their outputs. We began by assuming

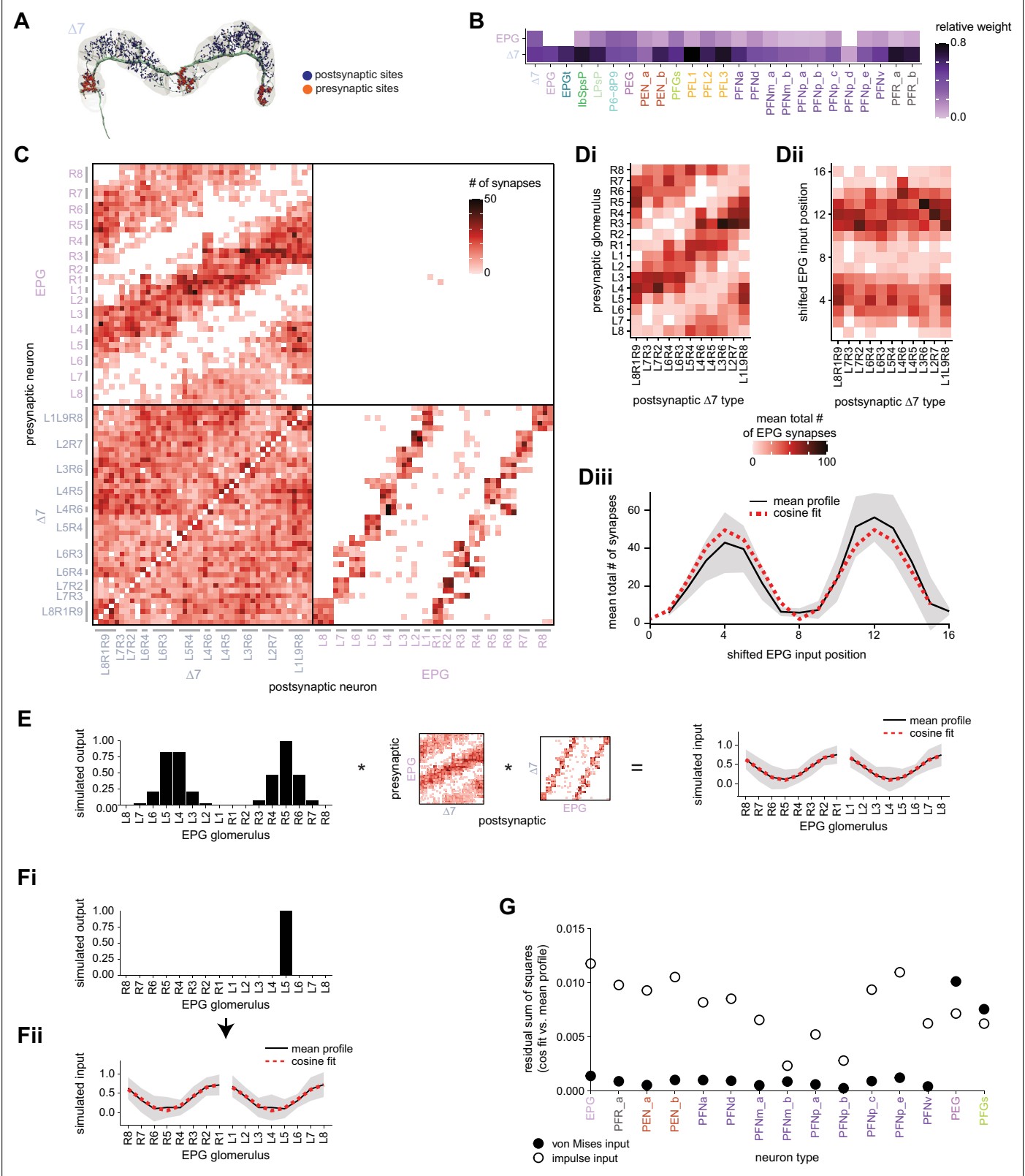

**Figure 20.** E-PG to Δ7 connectivity forms a cosine-like profile. (**A**) A morphological rendering of a Δ7 neuron that outputs to glomeruli R8, L1, and L9. (**B**) Type-to-type connectivity table from EPG and Δ7 neurons to themselves and to all other protocerebral bridge (PB) neurons. (**C**) Synaptic connectivity matrix between EPG and Δ7 neurons. (**D, Di**) The EPG to Δ7 synapses were added together within each EPG glomerulus for each Δ7 neuron. The total synapse counts were then averaged across all Δ7 neurons that have the same arborization pattern. (**Dii**) Each column in the EPG to Δ7 connectivity

*Figure 20 continued on next page*

*Figure 20 continued*

matrix in (**Di**) was circularly shifted to align the peaks. (**Diii**) The mean and standard deviation across aligned Δ7 neurons. A cosine fit to the mean profile is shown with the dotted red line. (**E**, left) A simulated von Mises bump profile in the ellipsoid body (EB) leads to von Mises profiles in the right and left PB. (Middle) The profile is multiplied by the EPG to Δ7 synaptic connectivity and then by the Δ7 to EPG connectivity to simulate the Δ7 input onto the EPG neurons. (Right) The normalized mean Δ7 to EPG input profile in the right and left PB, averaged across all possible bump positions and assuming a von Mises input. The standard deviation is shown in gray, and a cosine fit to the right or to the left mean is shown with the dotted red curve. (**F**) A simulated impulse profile to one glomerulus in the PB (**Fi**) and the resulting simulated activity profile in the Δ7 neurons (**Fii**). The procedure follows that used in (**E**). (**G**) The residual sum of squares error between a cosine and the mean Δ7 input to a given neuron type assuming either a von Mises (black outline) or an impulse (black fill) input from the EPG neurons. The error is averaged across the fits to the right and to the left PB.

The online version of this article includes the following figure supplement(s) for figure 20:

**Figure supplement 1.** EPG and Δ7 neuron-to-neuron connectivity to PEG, PEN, PFGs, PFL, and PFR neurons.

**Figure supplement 2.** EPG and Δ7 neuron-to-neuron connectivity to PFN neurons.

**Figure supplement 3.** The Δ7 neurons get input in glomeruli that represent angles ~180° offset from their output glomeruli.

that the EPG neurons have an activity bump, similar to that observed experimentally (*Seelig and Jayaraman, 2015*). We also assumed that the EPG to Δ7 and Δ7 to EPG connection weights are approximately proportional to their synaptic counts. Our simulations revealed that propagating this activity across the neuron types led to an activity profile that could be even better fit by a sinusoid than the synaptic connectivity profile alone (*Figure 20E*). Since the EPG to Δ7 connectivity profile is already sinusoidal, this result is relatively independent of the shape of the input activity. For example, if we assumed an impulse (delta) function input from the EPG neurons to one PB glomerulus, the resulting signal was still shaped like a sinusoid (*Figure 20F*). We note, however, that these calculations ignore any nonlinearities, as well as the large number of synapses between individual Δ7 neurons. Considering that the Δ7 neurons synapse onto nearly all PB neurons in their specific output glomeruli (*Figure 20C*, *Figure 20—figure supplements 1 and 2*), we would expect this transformation of the bump to apply to those neurons as well. By similarly multiplying the EPG-Δ7 synaptic profile with the output profile from the Δ7 neurons onto each of the other neuron types, we found that the sinusoidal reshaping applied to nearly all columnar neurons (*Figure 20G*). The unique spacing of the Δ7 neurons' inputs and outputs further suggests that this sinusoidal profile will be passed most strongly to glomeruli that represent head direction angles that are 180° shifted from the glomeruli where the EPG activity is strongest (*Figure 20—figure supplement 3*).

Note that just as the EPGt neuron type may expand the influence of the head direction signal to the outermost PB glomerulus (discussed in an earlier section), the P6-8P9 neuron type may extend the

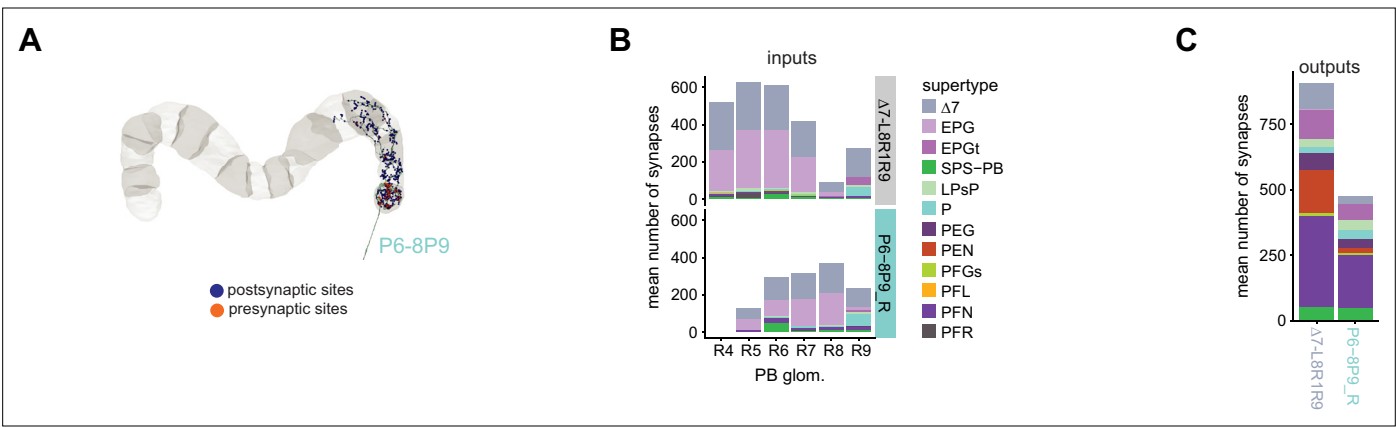

**Figure 21.** P6-8P9 neuron morphology and connectivity resembles that of the Δ7 neurons that arborize in the outer glomeruli. (**A**) A morphological rendering of a P6-8P9 neuron. There are two P6-8P9 neurons on each side of the protocerebral bridge (PB), both of which are presynaptic in glomerulus 9. (**B**) Both Δ7_L8R1R9 (top) and P6-8P9 (bottom) neurons get input in PB glomeruli 5–9. Both output in PB glomerulus 9 (not shown here). P6-8P9 neurons have the highest number of input synapses in glomerulus 8, while the Δ7_L8R1R9 neurons have the highest number of input synapses in glomeruli 5 and 6. The left PB is not considered as one of the two P6-8P9_L neurons was not able to be fully connected due to a hot knife error. (**C**) The mean number of output synapses from each Δ7_L8R1R9 neuron (left) or P6-8P9_R neuron (right) in PB glomerulus R9. The color code is identical to that in (**C**).

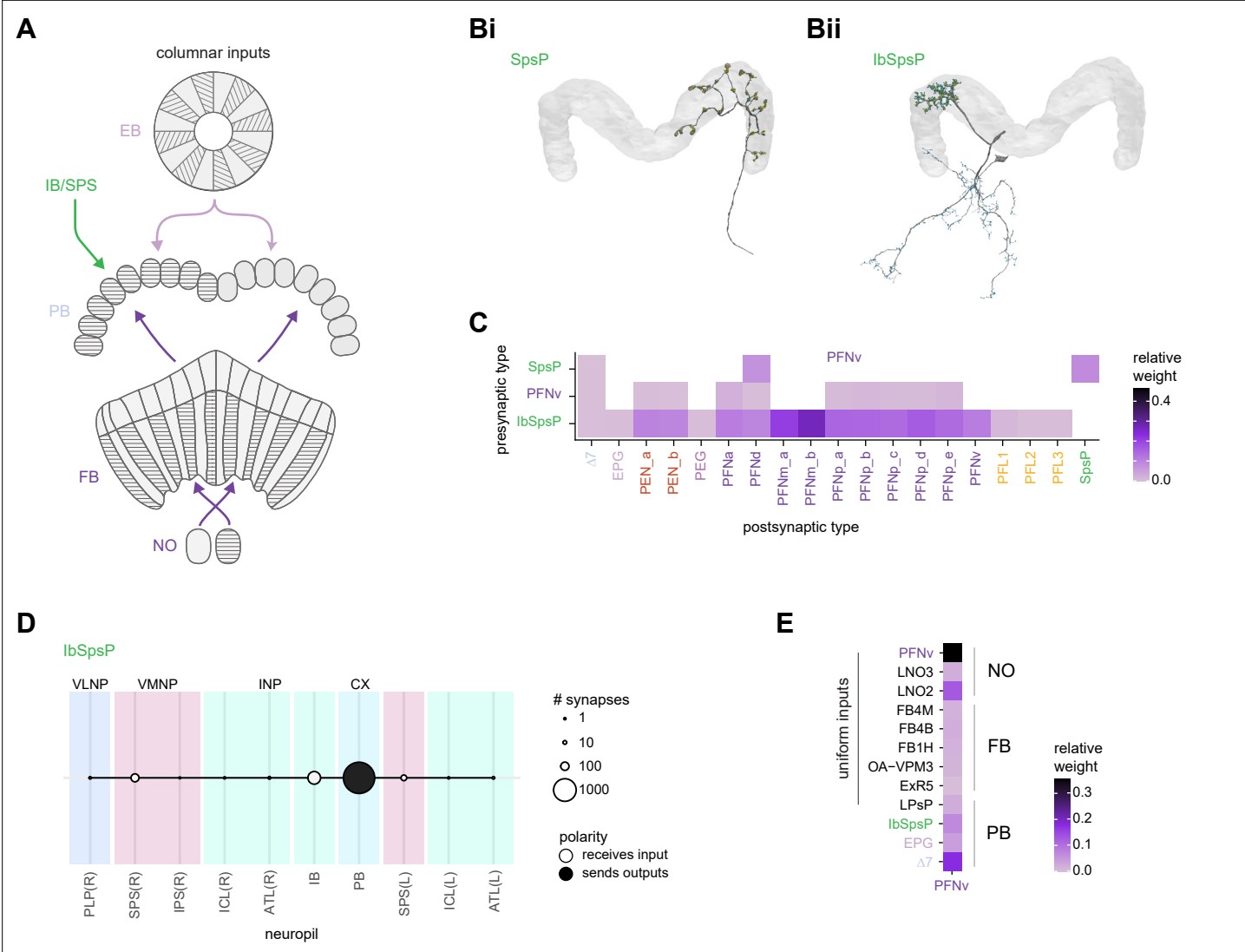

**Figure 22.** Protocerebral bridge (PB) input and inner neuron connectivity to output neurons. (**A**) Schematic depicting the neuropil that bring input to the PB via columnar neurons that target single PB glomeruli. (**B**) Morphological renderings of single SpsP (**Bi**) and IbSpsP (**Bii**) neurons. (**C**) Type-to-type connectivity matrix from select PB inputs (IbSpsP, PFNv, and SpsP neurons) to PB output neurons. The SpsP neurons also connect to themselves. (**D**) Region arborization plot for the right IbSpsP neurons. The left IbSpsP neurons were not fully contained in the imaged volume. (**E**) Type-to-type inputs to the PFNv neurons, separated by neuropil region.

The online version of this article includes the following figure supplement(s) for figure 22:

**Figure supplement 1.** Presynaptic partners of the IbSpsP neurons, outside of the protocerebral bridge (PB).

sinusoidal shaping of the signal to the entire PB. The P6-8P9 neurons have a similar morphology to the Δ7 neurons, receiving input across multiple PB glomeruli while outputting in only one glomerulus (***Figure 21A***). The P6-8P9 neurons also have similar input partners to the Δ7 neurons across glomeruli 6–9 (***Figure 21B***), and similar output partners in glomerulus 9 (***Figure 21C***), though the number of input and output synapses per glomerulus differs across the two cell types. Overall, the similarities between the P6-89 and the Δ7 neurons suggest that the P6-8P9 neurons perform a similar, albeit far more localized, function as the Δ7 neurons.

## Inputs to the PB from within and outside the CX

Although the Δ7 neurons shape the activity profile of columnar PB neurons, the amplitude of this activity is likely influenced by inputs to the PB (***Figure 22A***). External (non-CX) input comes from the IB and SPS through the IbSpsP neurons and SpsP neurons (***Figure 22B***; *Wolff et al., 2015*). The IbSpsP

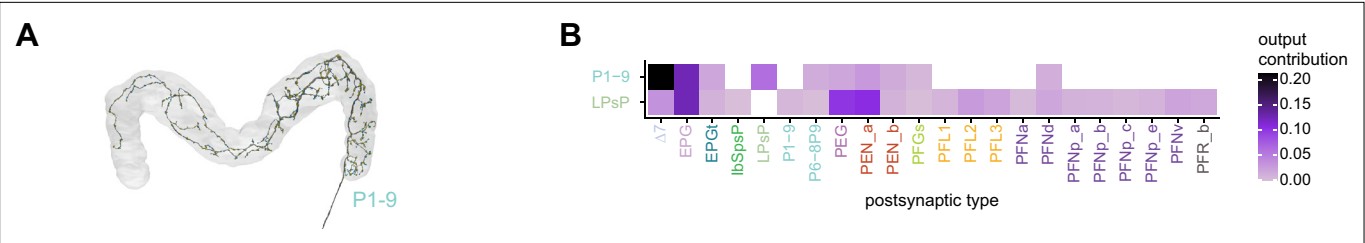

**Figure 23.** Neuromodulatory neurons in the protocerebral bridge (PB) output broadly across types. (**A**) A morphological rendering of a putative octapaminergic P1-9 neuron. (**B**) Type-to-type connectivity matrix for the outputs of the P1-9 neurons and the putative dopaminergic LPsP neurons in the PB.

neurons are columnar, projecting to one, two, or three often-adjacent glomeruli. They target a broad range of other columnar neurons including the PEN and PFN supertypes (*Figure 22C*). The pair of SpsP neurons, which are easily recognized by arbors that span exactly one half of the PB each, are very selective and target only the PFNd neurons (*Figure 22C*). The SPS is known to be innervated by DNs (*Schnell et al., 2017*) and can thus provisionally be considered a premotor area, but less is known about the IB (though a number of CX outputs project to the IB, as discussed later). We speculate in a later section about the potential navigational role of these PB input neurons. The IbSpsPs inputs are varied and largely unknown (*Figure 22D*, *Figure 22—figure supplement 1*). The SpsP neurons are incomplete in the hemibrain, and thus their inputs in the SPS are at present unknown. The PB also receives columnar, glomerulus-specific input from the PFNv neurons, which have recently been characterized physiologically (*Figure 22C*; *Lu et al., 2020b*; *Lyu et al., 2020*). These neurons receive relatively uniform input in the NO and FB (*Figure 22E*).

## Sources of neuromodulation in the PB

The PB receives neuromodulatory input from both DANs (LPsP, *Figure 19C*) and octopaminergic neurons (P1-9, *Figure 23A*). The dopaminergic LPsP neurons receive PB input from the Δ7 neurons and the EPG neurons (*Figure 20B*). They also make synaptic contact with a wide range of other neurons, including the octopaminergic P1-9 neurons (*Busch et al., 2009*; *Wolff and Rubin, 2018*; *Figure 23A*), from which they also receive input (*Figure 23B*). Unfortunately, the processes of both the P1-9 and the LPsP neurons are cut off outside the PB, where they putatively receive most of their input (*Wolff et al., 2015*). The P1-9 neurons project throughout the brain while the LPsP neurons get input in the LAL, the SPS, and the IPS. The function of these two types is unknown, though they may control plasticity in the PB. Plasticity in the PB has been implicated in the fly's ability to estimate the size of gaps that it can safely cross based on its own body size (*Krause et al., 2019*).

## Symmetric nonuniformity in the number of PB columnar neurons in each glomerulus

A representational system for a circular variable, like head direction, might be expected to have the same number of elements to represent each angle. Thus, we might expect each columnar neuron type to have the same number of neurons in each PB glomerulus. In addition to the columnar PB-EB neurons, this would also include the many types of PB-FB columnar neurons that receive input from the EPG and Δ7 neurons (*Figure 20B*). While the PEG, PFGs, and PFL1 neurons have one neuron in each glomerulus in their PB domain (*Figure 24A and B*, blue bars), these uniform distributions are the exception rather than the rule. Deviations from uniformity tend to be mirror symmetric across the right and left PB, suggesting that they are not just the result of developmental variability. The EPG neurons, for example, have three neurons in glomeruli R/L 3–7, four neurons in glomeruli R/L 8, and two neurons in glomeruli R/L 1 and 2 (*Figure 24A*, purple bars). In contrast, the PEN neurons and PFN neurons each have roughly double the number of neurons in glomeruli 4 and 6 as they have in the other glomeruli; for example, PEN_a neurons have two neurons in R4, R6, L4, and L6, but one in each of the other glomeruli where they arborize (*Figure 24A and C*, magenta bars). Finally, the PFL and PFR neurons each have their own unique distributions of neuron numbers across the PB (*Figure 24B*). We outline some potential origins and roles for such numeric differences in 'Discussion.' We also

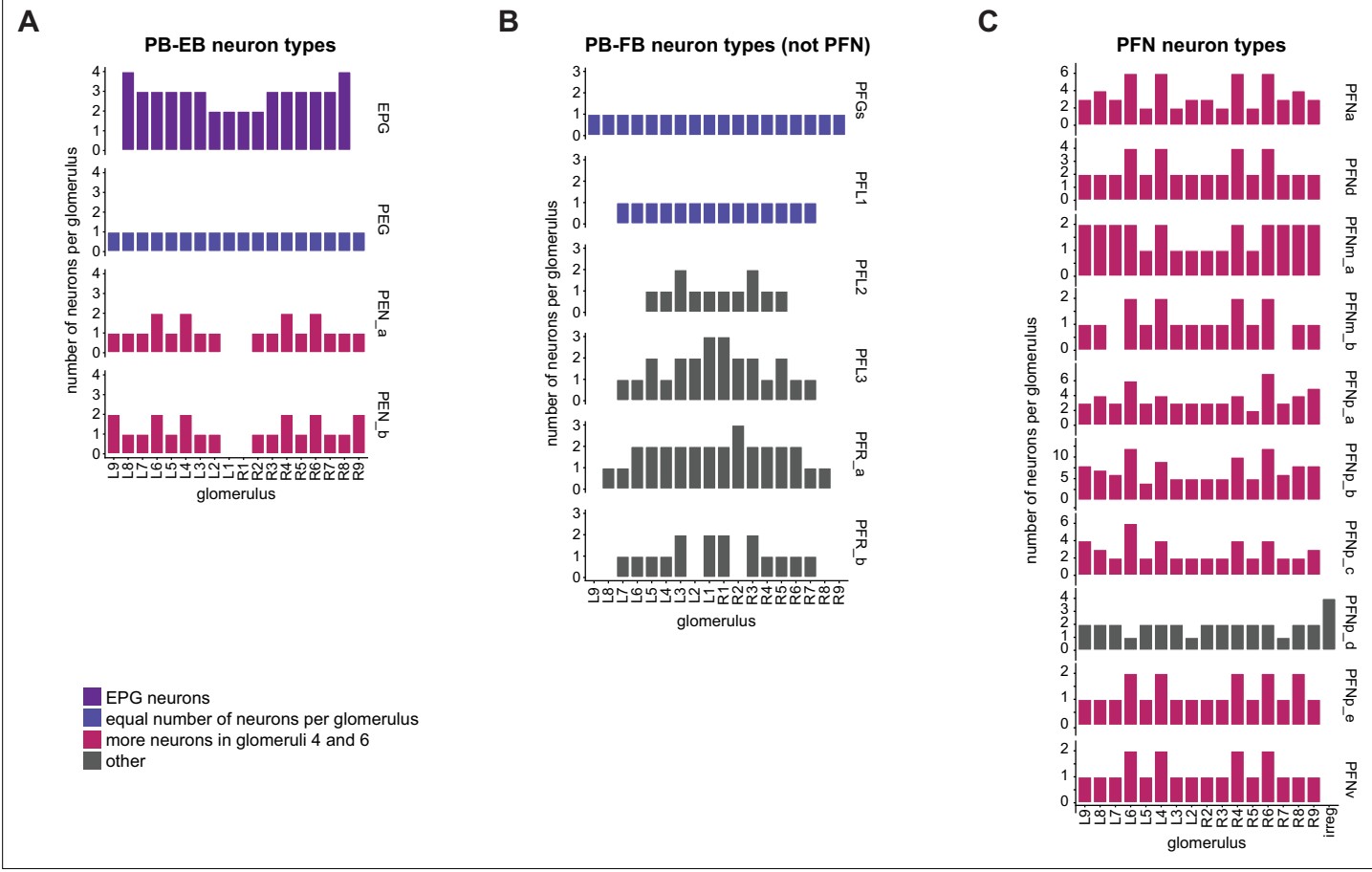

**Figure 24.** The number of neurons per glomerulus varies for each columnar neuron type. (**A**) Number of neurons per protocerebral bridge (PB) glomerulus for each of the PB-EB neuron types. (**B**) As in (**A**), for the PFGs, PFL, and PFR neurons. (**C**) As in (**A**), for the PFN neurons. The irregular PFNp_d neurons have minimal arborizations in the PB.

The online version of this article includes the following figure supplement(s) for figure 24:

**Figure supplement 1.** Neuron types with more instances in a glomerulus have fewer total input or output synapses per region of interest (ROI).

note that neurons with more instances in a given glomerulus often have fewer total input and output synapses across the different regions in which they arborize (*Figure 24—figure supplement 1*). These connectivity differences may compensate for the differences in the number of neurons per glomerulus, potentially allowing the net activity in different PB glomeruli to be similar even if those glomeruli have a different number of neurons.

## The NO: input for navigational computations in the EB and FB

The connectome establishes the bilaterally symmetric, paired NO as an input structure to the CX – particularly for CX columnar neurons (*Figure 25A–C*). Structurally, both the left and right NO are divided into three subcompartments. NO1, NO2, and NO3 (*Wolff and Rubin, 2018*; inset in *Figure 25B*). This compartmentalization is respected by most neuron types, as is reflected in the connectivity, which shows clusters of neurons corresponding to the NO region that they innervate (*Figure 25B*). The NO receives inputs from LNO neuron types (see Figure 27B for morphological renderings) that innervate the LAL, GA, and CRE (*Figure 25*). These likely inhibitory neurons (*Franconville et al., 2018*; *Lu et al., 2020b*; *Lyu et al., 2020*) provide input to and receive feedback from CX columnar neurons (*Figure 25B and C*). The only CX columnar neurons that lend some credence to the notion of the NO being an output structure of the CX are the PEN_b neurons, which provide strong inputs to the ExR8 neurons (discussed further in the CX output section) (*Figure 25B*). Finally, most tangential FB (FBt) neurons, which each arborize in FB layers that span most columns of the FB, also

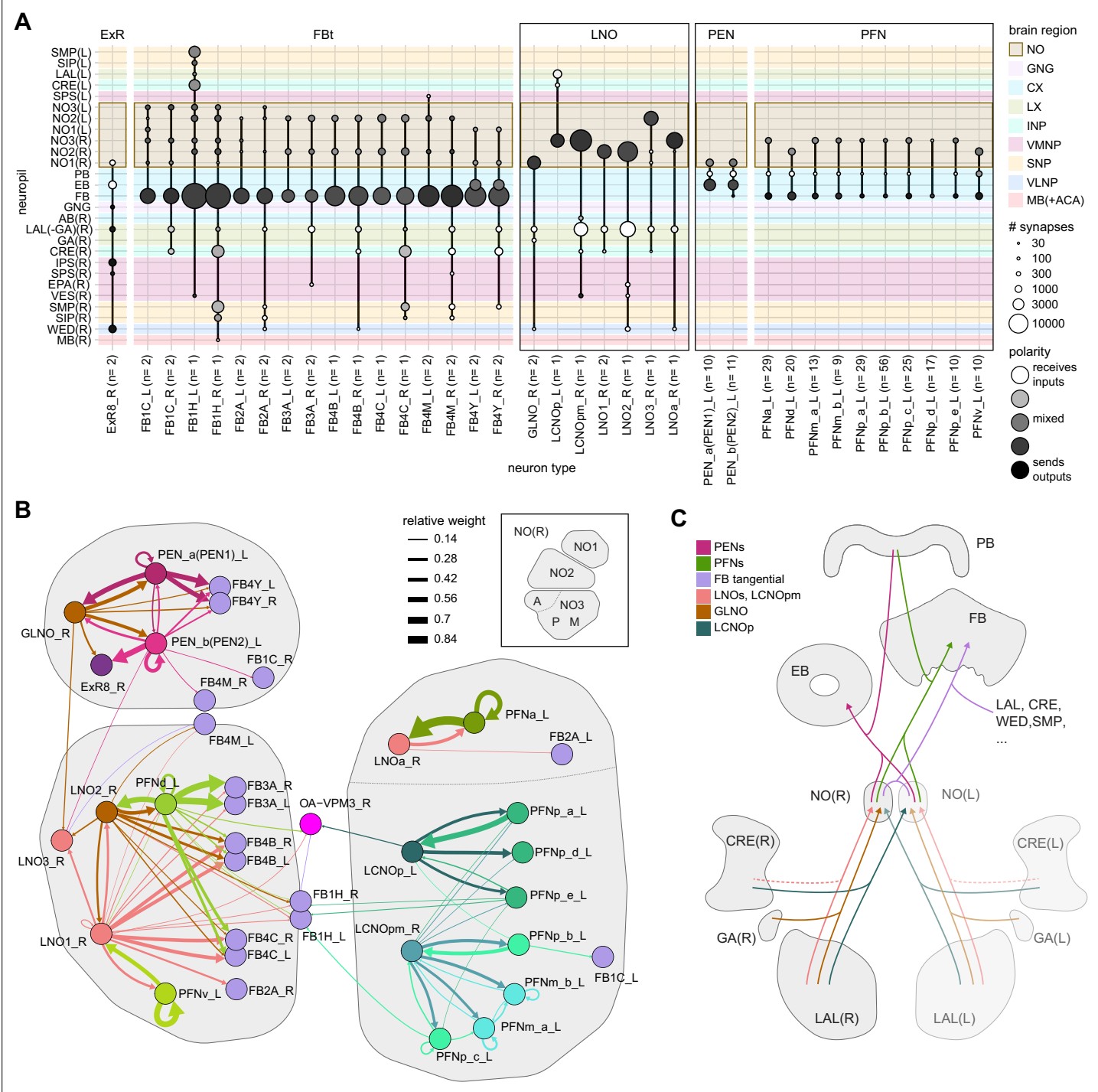

**Figure 25.** Overview of the noduli and illustration of separate compartments. (**A**) Region arborization plot summarizing all cell types that innervate the noduli (NO), showing average pre- and postsynaptic counts by region. Boxes mark groups of neuron types that will be described in more detail in this section. (**B**) Connectivity graph of all neuron types in the right NO, highlighting clusters that approximately correspond to anatomically defined subcompartments (see inset). The line thickness corresponds to the relative weight of a given type-to-type connection. Only connections with a relative weight of at least 0.05 (5%) are shown. (**C**) Schematic of how the NO connect to other brain regions.

make weak reciprocal connections with LNO neurons and columnar neurons in the NO (*Figure 25B*, see also *Figure 26—figure supplement 1B*). The function of these FBt neurons is presently unknown. Thus, with the exception of the PEN_b neurons, the vast majority of NO outputs of CX columnar neurons are to other CX columnar neurons (usually of the same type), or to LNO neurons that then provide input to the CX columnar neurons.

Besides their interconnections in the NO, CX columnar neurons receive input in individual glomeruli of the PB and arborize in either FB columns (PFN types) or EB tiles (PEN types) (morphological renderings in *Figure 26A*), where they make the majority of their outputs and, to a lesser degree, receive further inputs (*Figure 26B*, *Figure 26—figure supplement 1A*). None of the CX columnar neuron types that target the PB and NO send arbors to both the EB and FB. Broadly, the FB and EB columnar neurons are known to be sensitive to the fly's head direction and to self-motion signals, likely enabling these neurons to update navigational representations in the EB and FB (*Currier et al., 2020*; *Green et al., 2017*; *Shiozaki et al., 2020*; *Turner-Evans et al., 2017*; *Turner-Evans et al., 2020*). These neurons likely receive their head direction input in the PB (see previous section and *Figure 26B*, *Figure 26—figure supplement 1C*). While all columnar neurons receive similar inputs in the PB, their inputs in the NO differ significantly (*Figure 26C*, *Figure 26—figure supplement 1B and C*), which we describe in more detail below. Much like the other CX structures, the NO is characterized by dense recurrence. All columnar neurons are recurrently connected to the LNO neurons that they receive input from (*Figures 25B and 26B*). Additionally, most of the columnar neurons – with the notable exception of the PFNp neurons (whose function is unknown) – are also strongly recurrently connected to other neurons of their type (*Figures 25B and 26D*). These recurrent within-type connections account for a large fraction of the NO outputs of EB and FB columnar neurons (*Figure 26—figure supplement 1A*). The functional relevance for this recurrence is unknown.

The primary input from outside the CX to the NO is from different subtypes of LNO neurons, which, in turn, receive most of their inputs in the LAL and other accessory structures (*Figure 27A and B*). One potential organizing principle for the LNO-columnar connections is suggested by the inputs that the different types of LNO neurons receive. Clustering the LNO types by their inputs neatly divides them into classes that synapse onto specific groupings of CX columnar neurons (e.g., the GLNO neurons that provide input to PEN neurons, while LNOa_R cells contact the PFNa_L neurons, *Figure 27C*, *Figure 27—figure supplement 1A*). Note that although a single LNO type may provide input to multiple classes of CX columnar neurons (*Figure 27C*), an LNO that is connected to an EB columnar neuron never contacts an FB columnar neuron and vice versa. Combined with the fact that most columnar neurons receive relatively similar inputs in the PB (*Figure 26C*), this LNO-based segregation suggests that the information that each grouping of CX columnar neurons receives in the NO may differ substantially, even within a single NO compartment. Thus, these NO inputs may largely determine differences between response properties of different CX columnar neuron types.

What kind of information may be conveyed by LNO neurons? Recordings from one LNO type in the sweat bee suggest that such neurons may bring optic flow-based self-motion information into the NO and to the columnar neurons (*Stone et al., 2017*). This is consistent both with recent imaging experiments in fly LNO neurons (*Lu et al., 2020b*; *Lyu et al., 2020*) and with the observed tuning to self-motion in *Drosophila* CX columnar neurons (*Green et al., 2017*; *Shiozaki et al., 2020*; *Turner-Evans et al., 2017*; *Turner-Evans et al., 2020*). Most of the direct inputs to LNO neurons have not yet been characterized (*Figure 27—figure supplement 1B*), but our analysis of inputs to the CX (*Figure 5B*) suggests that LNO neurons, especially those that provide input to FB columnar neurons, may be tuned to a diverse set of sensory information. Indeed, a recent study found that LNOa neurons are tuned to wind direction and that this tuning is inherited by PFNa neurons (*Currier et al., 2020*). Consistent with this observation, we found a connection between putative WPN neurons (LHPV6q1), which are tuned to wind direction, and LNOa (*Figure 27—figure supplement 2*). In addition, LNO neurons may carry efference signals. LCNOp neurons get input from PFL cells, one of the CX output neuron types (*Figure 27A*, *Figure 27—figure supplement 1B*, discussed further in the output section). Thus, inhibitory LNO neurons may carry information beyond self-motion signals, and different types of NO inputs may be used for different navigational computations (a topic we return to in the FB section and again in 'Discussion').

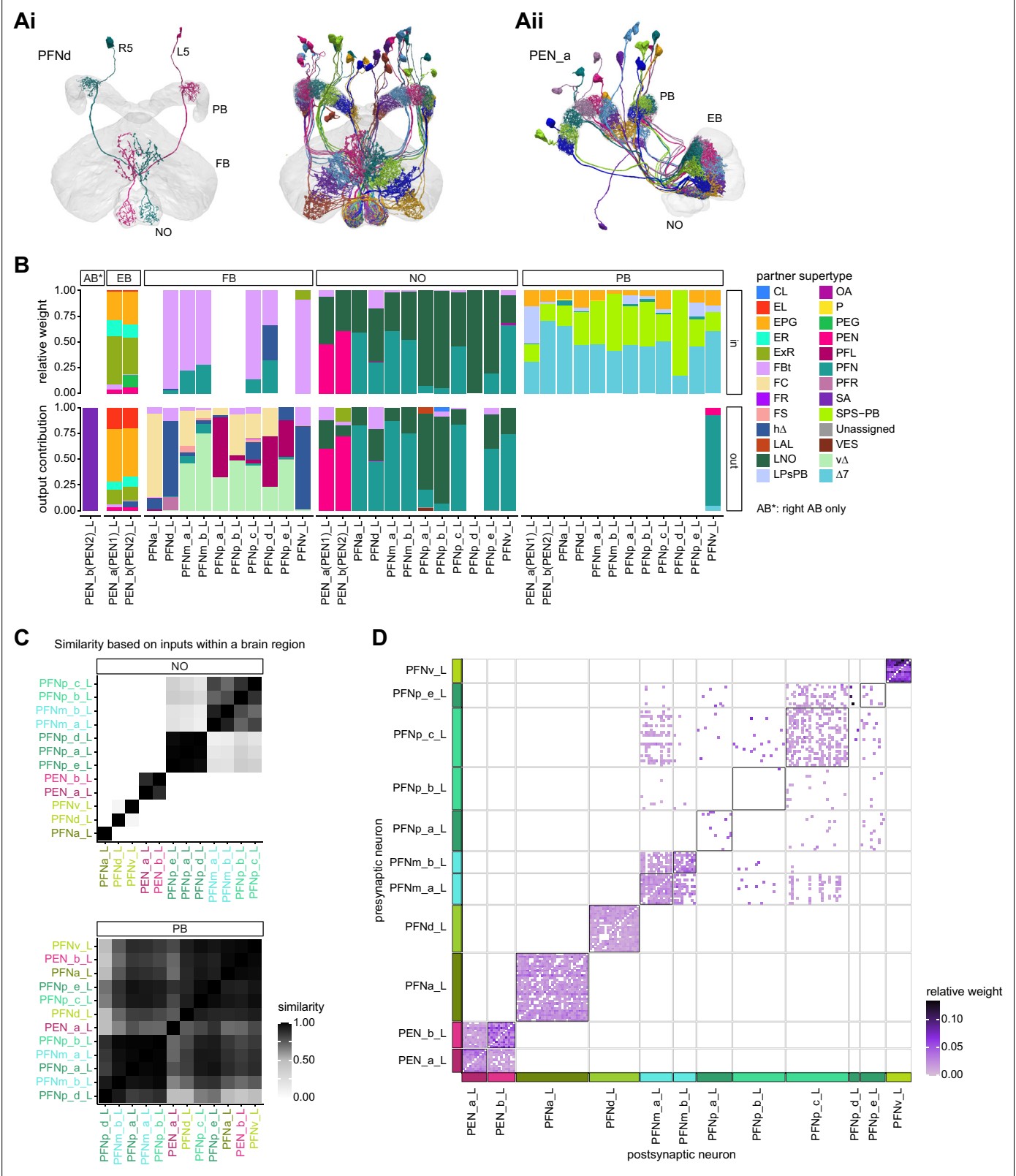

**Figure 26.** Columnar neurons in the noduli. (**A**) Morphological rendering of columnar neurons. (**Ai**) PFNd neurons. Left: two example neurons from the left and right PFNd population. Right: complete population of PFNd neurons. (**Aii**) PEN_a neurons. (**B**) Stacked bar graph illustrating the fraction of inputs and outputs to PFN and PEN partners grouped into supertypes and separated by brain region. Inputs and outputs are normalized per neuron type and brain region. The connectivity strength for inputs and outputs is measured by relative weight and output contribution, respectively.

*Figure 26 continued on next page*

*Figure 26 continued*

(**C**) Similarity matrices (see Materials and methods) for columnar noduli (NO) neurons based on their inputs in the NO (top) and protocerebral bridge (PB) (bottom). (**D**) Neuron-to-neuron connectivity matrix for columnar neurons in the right NO. Connections between neurons of the same type are highlighted with black boxes.

The online version of this article includes the following figure supplement(s) for figure 26:

**Figure supplement 1.** Comparison of PFN inputs and outputs.

Overall, the picture of the NO that emerges from these collected physiological and anatomical observations is of an important hub for self-motion information, but it may also receive contextual or directional sensory information, which is employed for navigational computations in the EB and FB.

## The FB: a structured recurrent network for context-dependent navigation and sleep

The FB, referred to as the upper division of the central body (CBU) in other insects (*Pfeiffer and Homberg, 2014*; *Strausfeld, 1976*; *Strausfeld, 2012*), is the largest and most complex structure in the CX. In *Drosophila*, the FB is composed of over 200 distinct neuron types that form a dense and highly recurrent network, described in detail below. In all insects examined to date (*de Vries et al., 2017*; *el Jundi et al., 2018*; *Heinze et al., 2013*; *Pfeiffer and Homberg, 2014*; *Strausfeld, 2012*; *Wolff et al., 2015*), the FB is organized into columns along its medial-lateral axis and layers along its dorsal-ventral axis (*Figure 28A*).

Previous experimental work has implicated FB circuits in a variety of behaviors that require directed movements, including operant visual learning (*Liu et al., 2006*), obstacle avoidance (*Harley and Ritzmann, 2010*), nociceptive avoidance (*Hu et al., 2018*), and head optomotor responses (*Akiba et al., 2020*). Further, recent physiological recordings have demonstrated that PB-FB columnar neurons convey the head direction representation from the left and right PB to the FB (*Shiozaki et al., 2020*), and that, similar to the PEN neurons, PFN activity is likely modulated by self-motion inputs received in the NO (*Currier et al., 2020*; *Lu et al., 2020b*; *Lyu et al., 2020*; *Shiozaki et al., 2020*; *Stone et al., 2017*). Sensorimotor information may also enter the FB through tangential neuron types whose activity is gated by behavioral state (*Weir and Dickinson, 2015*; *Weir et al., 2014*). Importantly, the FB is also home to a prominent class of columnar output neurons, known as PFL neurons in *Drosophila* (*Wolff et al., 2015*) and CPU1 neurons in other insects (*de Vries et al., 2017*; *el Jundi et al., 2015*; *el Jundi et al., 2018*; *Heinze et al., 2013*; *Heinze and Homberg, 2007*; *Heinze and Homberg, 2008*; *Heinze and Reppert, 2011*; *Stone et al., 2017*). These neurons, whose activity has been linked to directed movement (*Skutt-Kakaria et al., 2019*), send projections to the LAL, where they contact DNs involved in steering (*Rayshubskiy et al., 2020*). Consistent with this view, some CX neurons in cockroaches have activity that predicts future forward and rotational velocity, and electrical stimulation of the CX evokes stereotyped locomotor responses (*Martin et al., 2015*). Together, these studies support the view that FB circuits implement head-direction-based navigational behaviors such as straight-line orientation, long-range migration, and visual route following (reviewed in *Honkanen et al., 2019*). It remains largely unknown how the FB network may support these navigational functions, but a recent study proposed a network model inspired by physiology and anatomy in the bee that could perform vector-based path integration (*Stone et al., 2017*).

Navigational functions have mostly been associated with ventral FB circuits. In contrast, more dorsal layers have primarily been studied in the context of sleep-wake control (reviewed in *Dubowy and Sehgal, 2017*). Prominent among these are a population of dorsal FB tangential neurons whose intrinsic excitability tracks sleep need and whose activation induces sleep (*Donlea et al., 2011*; *Pimentel et al., 2016*). FB tangential neurons also receive contextual input from the MB (*Dag et al., 2019*; *Li et al., 2020*; *Scaplen et al., 2020*), an important center for learning and memory (reviewed in *Modi et al., 2020*). Together, the above evidence suggests that context- and state-dependent action selection, including initiating periods of behavioral quiescence, may be governed by the FB.

Compared to other CX regions, much less is known about the overall structure and connectivity of the FB network. Considering the sheer complexity of the structure's recurrent circuits, we devote many of the following sections to describing the FB's columnar organization before we delve into intra-FB connectivity patterns. We then describe the plethora of FB tangential neuron types that form

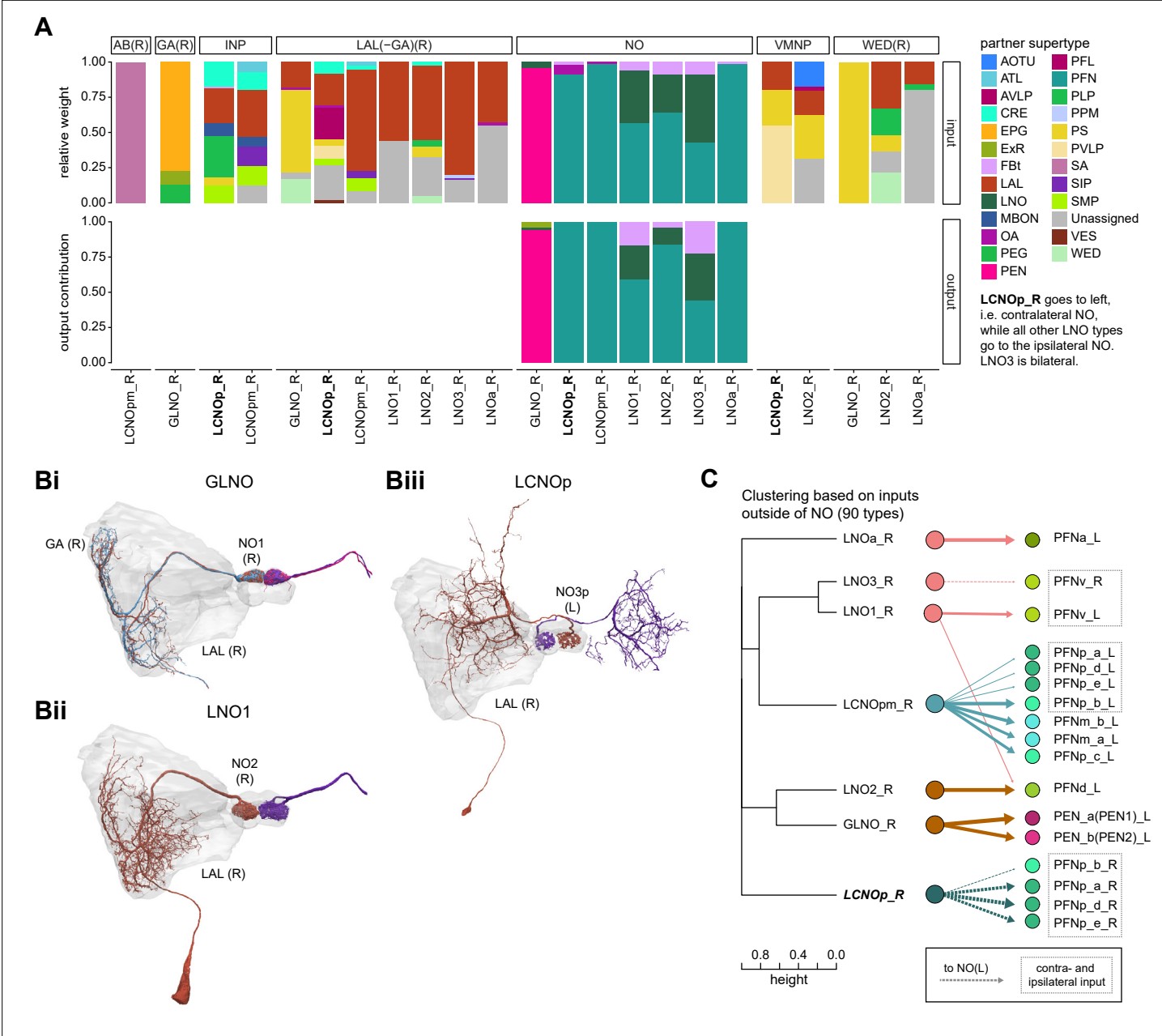

**Figure 27.** Comparison of LNO neurons, which provide input to columnar neurons. (**A**) Stacked bar graph illustrating the fraction of inputs and outputs of LNO to partners grouped into supertypes and separated by brain region. Inputs and outputs are normalized per neuron type and brain region. The connectivity strength for inputs and outputs is measured by relative weight and output contribution, respectively. (**B**) Morphological rendering of LNO neurons. (**Bi**) GLNO, (**Bii**) LNO1, and (**Biii**) LCNOp. Note that LCNOp crosses the midline and arborizes in the contralateral noduli (NO). Additional morphological renderings. LCNOpm, LNO2, LN03, LNa. (**C**) Illustration of how similarity between LNO neuron types relates to their connectivity to columnar NO neurons. Left: dendrogram depicting the similarity between GLNO, LNO, and LCNO neuron types based on their inputs outside of the NO (i.e., excluding feedback connections from PFN or PEN neurons). The branch height in the dendrogram indicates the normalized distance between types within the similarity space. Right: connectivity from GLNO, LNO, and LCNO neurons onto columnar NO neurons, visualized as in the connectivity graph in *Figure 25B*. Note that LCNOp and LNO3 neurons project to the ipsilateral NO and therefore target the right-side population of certain PFN types. PFN types that receive inputs from ipsi- and contralateral LNO and LCNO types are highlighted with dashed boxes.

The online version of this article includes the following figure supplement(s) for figure 27:

**Figure supplement 1.** Comparison of LNO inputs and outputs.

**Figure supplement 2.** Connectivity graph of paths from putative directionally tuned wind sensitive neurons (putative WPN neuron and WLL neuron) to any of the LNO neurons.

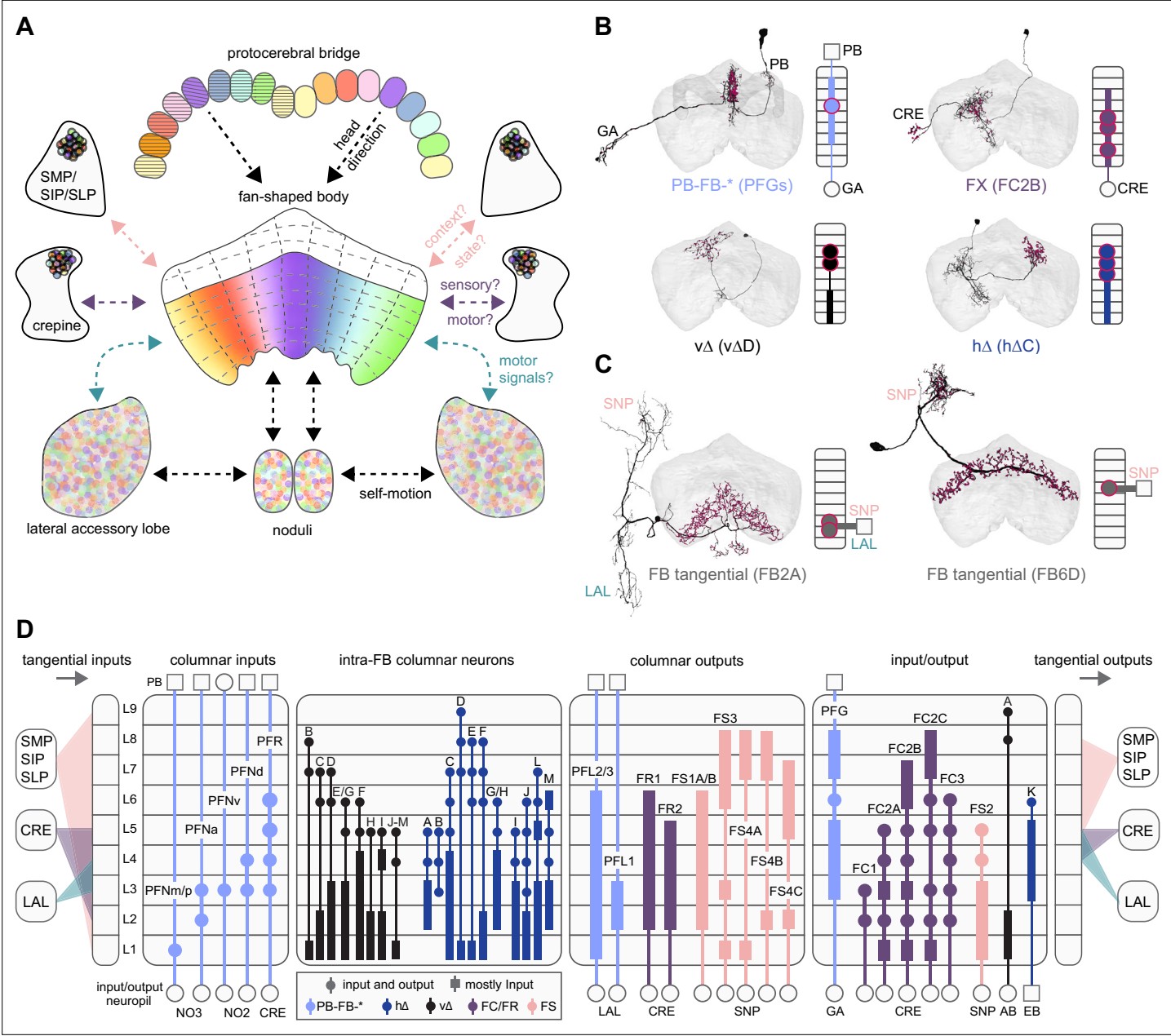

**Figure 28.** Fan-shaped body (FB) overview. (**A**) Schematic showing the FB, its main associated input and output neuropil, and the general types of information thought to be conveyed. Here, the FB is divided into nine vertical columns defined by protocerebral bridge-FB (PB-FB) neurons (see *Figure 29*), which map the nine glomeruli in the left and right PB to columns in the FB, as indicated by the color of each glomerulus/column (see *Figure 30*). However, unlike in the PB, the number of FB columns is not rigidly set, but depends on cell type. In addition to columns, FB tangential cells divide the structure into nine horizontal layers. The ventral FB (layers ~ 1–6) receives columnar input from the PB while the dorsal FB (layers ~ 7–9) does not. (**B**) Morphological renderings of individual columnar neurons (shown in black; red circles are presynaptic sites) from each of the four broad columnar neuron classes. PB-FB-*, FX, vΔ, and hΔ (where X and * stand for an additional, neuron type-specific neuropil). Each class contains many distinct neuron types. The population of neurons comprising each neuron type innervates all columns of the FB, but in a layer-restricted manner. As shown next to each anatomical rendering, a neuron type's morphology can be summarized by illustrating the location of dendritic (rectangle) and axonal (circles) compartments for the nine FB layers and any associated neuropil. Here, each neuron type is colored according to its class (see legend in **D**). (**C**) Same as in (**B**), but for 2 of the 145 types of FB tangential cells. (**D**) Schematic showing the innervation pattern of every FB columnar neuron type, each illustrated as in (**B**). Columnar neurons can be roughly grouped into four putative functional groups: those that convey information from outside the FB to specific FB layers (columnar inputs; subset of PB-FB-* neurons), those that convey information between layers of the FB (intra-FB columnar neurons; vΔ and hΔ neurons), those that convey information out of the FB (columnar outputs; PB-FB-* and FX), and those that could perform a mixture of these functions (input/output). Columnar inputs have axons in every FB layer they innervate, intra-FB columnar neurons have processes confined to

*Figure 28 continued on next page*

*Figure 28 continued*

the FB, and columnar outputs have dendrites (and very few axons) in every FB layer they innervate. Note that while some columnar types are grouped (e.g., PFNm and PFNp), these types can be distinguished by their connectivity both within and outside of the FB (e.g., PFNm and PFNp receive distinct noduli [NO] inputs). In addition, tangential cells innervating the superior protocerebrum (SMP/SIP/SLP), crepine (CRE), and/or lateral accessory lobe (LAL) (and additional structures) provide input to (left panel) and output from (right panel) specific FB layers. Tangential cells in many different layers send processes to the SMP/SIP/SLP, CRE, and/or LAL, but only consistently target these regions in most cell types in the layers that are shown. See *Figure 28—figure supplement 1* for average pre- and postsynaptic counts by region and columnar neuron type.

The online version of this article includes the following figure supplement(s) for figure 28:

**Figure supplement 1.** Fan-shaped body (FB) regional connectivity.

the structure's layers and that likely provide contextual and state information to the columnar network. We end with two more focused sections, one on the sleep-wake network of the dorsal FB and the other on pathways from the MB to the FB. In 'Discussion,' we build on this detailed structural description to propose hypotheses for the roles of different neuron types in FB circuit function.

## Overview of FB structure, neuron types, and major input/output pathways

The FB is coarsely divided into columns along its medial-lateral axis by four large classes of columnar neurons: PB-FB-*, FX, vΔ, and hΔ (*Figure 28A and B*), where '*' and 'X' refer to accessory regions of the CX. In addition to these ~60 columnar neuron types, ~150 types of FB tangential neurons divide the FB into nine layers along the dorsal-ventral axis (*Figure 28A and C*). Each class of columnar neuron contains many distinct neuron types. Within each type, individual columnar neurons form spatially restricted arbors that innervate type-specific FB layers and, as a population, tile FB columns (*Figure 28B and D*, *Figure 28—figure supplement 1*). Compared to the PB, with its spatially segregated glomeruli (nine in each hemisphere), FB columns are not as clearly defined, nor are they clearly visible in light-level images. Instead, FB columnar neurons form a type-specific number of columns (from 6 to 12) and there is considerable variability in how evenly each type tiles the FB. Similarly, FB layers lack clear boundaries, much like the EB's annuli (i.e., along the radial axis).

Columnar and tangential neurons also project to regions outside the FB, providing pathways for information exchange with accessory neuropils such as the NO, LAL, CRE, and SMP/SIP/SLP (*Figure 28A*). Based on their input from the EPG neurons and the polarity of their arbors, PB-FB-* neurons likely convey head-direction-related information from the PB to the FB (PFN, PFR, and PFGs types; *Shiozaki et al., 2020*), and also from the FB to the LAL (PFL types; *Rayshubskiy et al., 2020*). Two classes of interneurons, the vΔ and hΔ types, are composed of neurons whose arbors are largely confined to the FB, relaying information across layers and columns (*Figure 28B and D*). FX neurons are a heterogeneous columnar neuron class whose types primarily arborize in the FB and either the CRE or SMP/SIP/SLP. Similar to FX types, FB tangential neurons heavily innervate the CRE, SMP/SIP/SLP, and LAL (*Figure 28C and D*), but often contain additional arbors in type-specific neuropils, such as the NO, EB, or BU, which we detail in later sections.

## The columnar structure of the FB as defined by PB-FB-* neuron types

Most PB-FB-* columnar types divide the FB into approximately nine columns (*Figure 29*, *Video 5*), thereby linking the nine glomeruli in the left and right PB to corresponding regions the FB. PFN neurons that innervate layer 1, such as PFNp_a, project to one of nine distinct tooth-shaped structures (*Figure 29A*). This is consistent with previous light-level anatomy, which described the ventral margin of layer one as being composed of seven distinct teeth plus two 'cryptic teeth' (*Wolff et al., 2015*), together accounting for the nine clusters observed here. Columnar types with arbors in intermediate FB layers, such as the PFNa neurons, are less well clustered, and instead show a more continuous tiling of the FB, but their innervation pattern is also consistent with the existence of approximately nine columns (*Figure 29A*). One type, the PFGs neurons, has 18 neurons in total, which fairly evenly tile the FB, dividing it into 9 columns or approximately 18 'demi-columns,' roughly 1 demi-column per neuron (*Figure 29B*; see legend for exceptions to this pattern). Moreover, individual PB-FB-* neurons have neuronal arbors whose width is slightly less than 1/9th of the layer's width (*Figure 29D*), and the distance between neurons in adjacent columns is 1/9th the layer width on average (*Figure 29E*). One notable exception to this pattern is the PFNd type, which clearly forms eight columns. *Figure 29— figure supplement 1* shows morphological renderings and mean column locations for all PB-FB-*

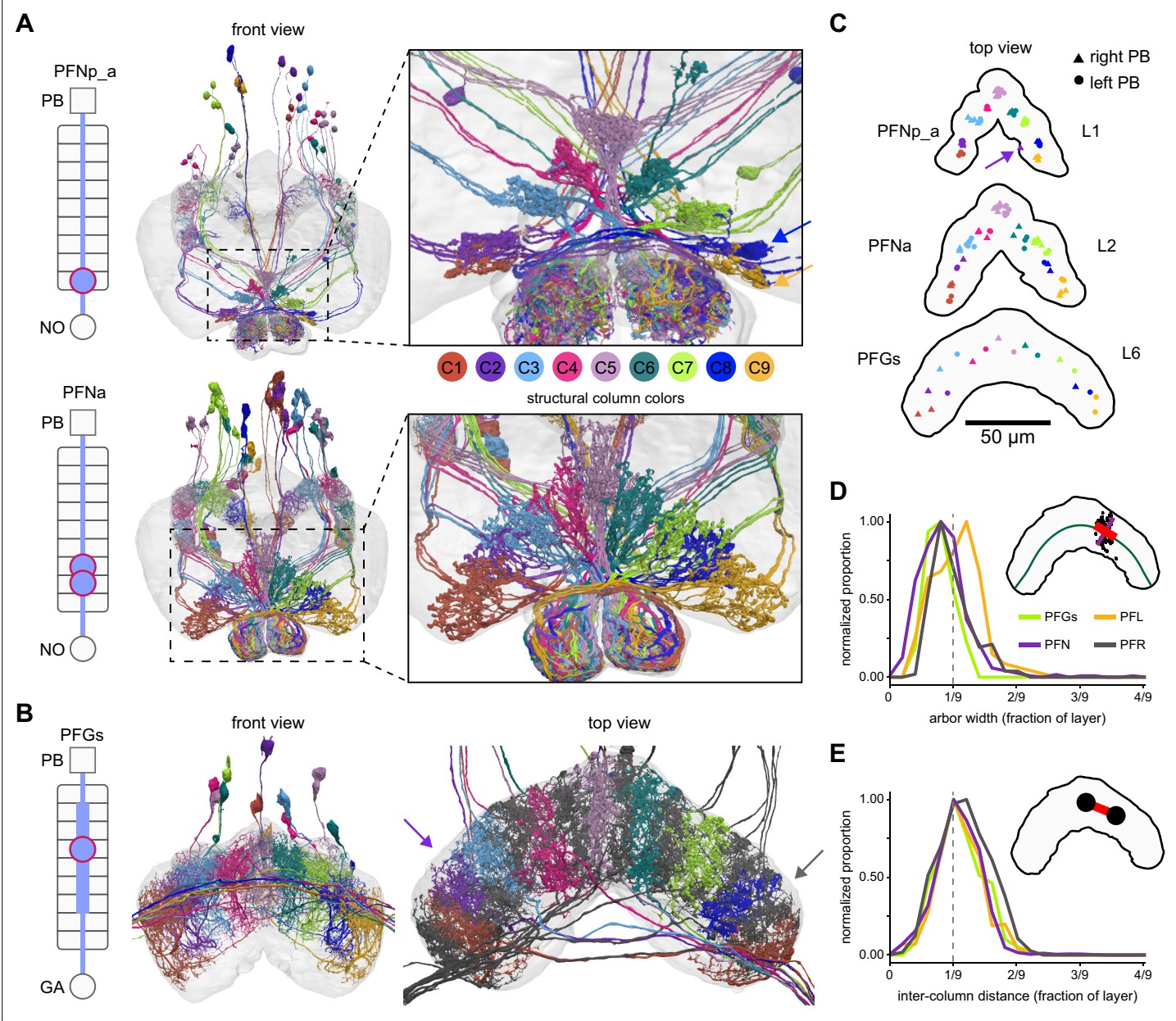

**Figure 29.** Most PB-FB-* neurons form nine columns in the fan-shaped body (FB). (**A**) Morphological renderings of PFNp_a and PFNa populations, colored by column (C1–C9). Left schematic shows location of dendritic (rectangle) and axonal (circles) compartments for the nine FB layers and any associated neuropil. Right panels show zoomed-in views of the FB, revealing a nine-column structure. Notice that PFNp_a columns are clustered while PFNa columns tile the FB more evenly. Note also that neurons innervating the same column often share the same fiber tract. The blue and yellow arrows in the top-right panel mark columns C8 and C9, which are closely spaced but show a clear spatially offset, as shown in panel (**C**). (**B**) Morphological rendering of the 18 neurons composing the PFGs population. Schematic on left as described in (**A**). Left panel shows front view (note that not all cell bodies are visible in this view). Arbor width is variable between cells. In addition, there is more substantial overlap in the dorsal FB arbors. The nine columns defined by this cell type are therefore more distinguishable in the ventral arbors. Right panel grays the nine neurons that project to the right gall, revealing that each column comprises two neurons, one of which projects to the left and the other to the right gall-surround, and that these right- and left-projecting neurons alternate in the FB. This projection pattern breaks the nine columns into ~18 'demi-columns,' one neuron per demi-column, with two exceptions (purple and gray arrows). The purple arrow marks a demi-column which lacks separation from adjacent demi-columns. Similarly, the gray arrow marks a demi-column containing two neurons, whereas all other demi-columns contain one neuron. Whether these are the result of wiring errors requires further investigation. (**C**) Top-down view showing every neuron's median location for all individual neurons in the PFNp_a, PFNa, and PFGs populations. Notice that while PFNp_a forms nine clear clusters, PFGs tile space more evenly. The distinct clustering seen in the PFNp_a arbors is reflected by the unique, scalloped morphology of layer 1 of the FB. The arrow in the PFNp_a panel points to a neuron that innervates both C1/C2 (assigned to C2) and C9, which is why its synapse location ended up outside of either cluster. (**D**) Distribution of neuronal arbor widths for PFGs, PFN,

*Figure 29 continued on next page*

PFL, and PFR neurons. As shown in the inset, the width (red line) of synaptic point clouds (black dots) from individual neurons was measured along a direction locally tangent to a line bisecting the FB layer (green). To account for differences in layer size, the raw width (red line) was normalized by dividing the length of the layer (green line). Each distribution was normalized to have a peak of 1. The vertical dashed line in the graph marks 1/9th of the layer width, the arbor width that would result from nine evenly spaced columns that have minimal overlap and collectively tile the layer. Notice that most neurons take up slightly less than 1/9th of the layer. Importantly, this measure is independent of the neuron's column assignment. (**E**) Distribution of inter-column distance, expressed as a fraction of the layer width, as in (**D**). Inter-column distance was measured by calculating the distance between the mean location of pairs of neurons in adjacent columns (as shown in inset), normalized to the length of the layer.

The online version of this article includes the following figure supplement(s) for figure 29:

**Figure supplement 1.** Columnar structure of PB-FB-* neuron types.

types, with the exception of PFL neurons, which we cover in later sections. Overall, the FB innervation of most PB-FB types is consistent with there being approximately nine columns, but there is considerable variability in how evenly each type tiles the FB. Variation in arbor width and spacing determines how much adjacent columns overlap.

## Anatomical phase shifts of PB-FB-* neurons between their PB and FB innervations

As described in previous sections (*Figure 16*), EPG neurons project from wedges in the EB to corresponding glomeruli in the left and right PB. This anatomical mapping conveys the activity bump in the EB to both the left and right PB, generating two bumps that get inherited by CX columnar neurons (*Green et al., 2017*; *Turner-Evans et al., 2017*). From this mapping, each PB glomerulus can be assigned an approximate 'anatomical phase' that indicates its preferred directional tuning, as defined by its EPG input. PB-FB neurons connect glomeruli in the left and right PB to columnar regions of the FB. However, their projection patterns have not been systematically characterized. Here we take a discrete view of this projection pattern and describe the various ways in which PB-FB types link PB glomeruli to FB columns.

The projection pattern of two EB columnar types – PEG and PEN – provides a framework for identifying similar patterns in PB-FB columnar neuron projections. First described at the light level (*Wolff et al., 2015*), PEG neurons project from PB glomeruli back to the same regions of the EB that provided their input, establishing a 'default pattern' where activity bumps from the left and right PB will approximately overlap in the EB. In contrast, PEN neurons project from PB glomeruli back to EB tiles with an 'anatomical phase shift' (*Figure 17*). This phase shift conveys the activity bumps from the left and right PB to regions approximately ±45° from the EPG bump, a motif that is responsible for updating the EPG bump position (*Green et al., 2017*; *Turner-Evans et al., 2017*).

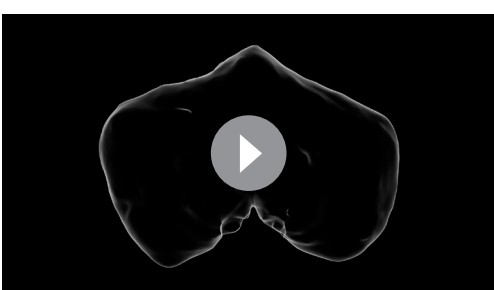

**Video 5.** Morphological renderings of the PB-FB columnar neurons. Each of the PB-FB columnar cell types is shown in order as follows: the PFGs, PFL1, PFL2, PFL3, PFNa, PFNd, PFNm_a, PFNm_b, PFNp_a, PFNp_b, PFNp_c, PFNp_d, PFNp_e, PFNv, PFR_a, and PFR_b neurons. Each neuron has been assigned to one of nine (loosely defined) FB columns, and is color coded accordingly. For each cell type, example neurons are shown first, followed by the entire population.
https://elifesciences.org/articles/66039/figures#video5

To investigate the mappings from PB glomeruli to FB columns, we first focused on neurons innervating glomeruli R5 and L5. These glomeruli receive input from adjacent wedges in the EB and, therefore, have similar allocentric head direction tuning (22.5° difference; *Figure 16*), ensuring that when a bump is centered at L5 in the left PB there will be a second bump centered between R5/R4 in the right PB (*Figure 30Ai*; *Green et al., 2017*; *Lyu et al., 2020*; *Turner-Evans et al., 2017*). Next, we compared the columnar position of the R5 and L5 projections in the FB (*Figure 30Ai and Aii*). Two neuron types – PFGs and PFR_a – were found to map R5 and L5 to the same FB column, C5, consistent with the absence of any phase shift since these similarly tuned neurons project to overlapping regions in the FB. When we extended the PB-FB mapping to the other neurons in these types, we found that this projection pattern was circularly symmetric, leading to

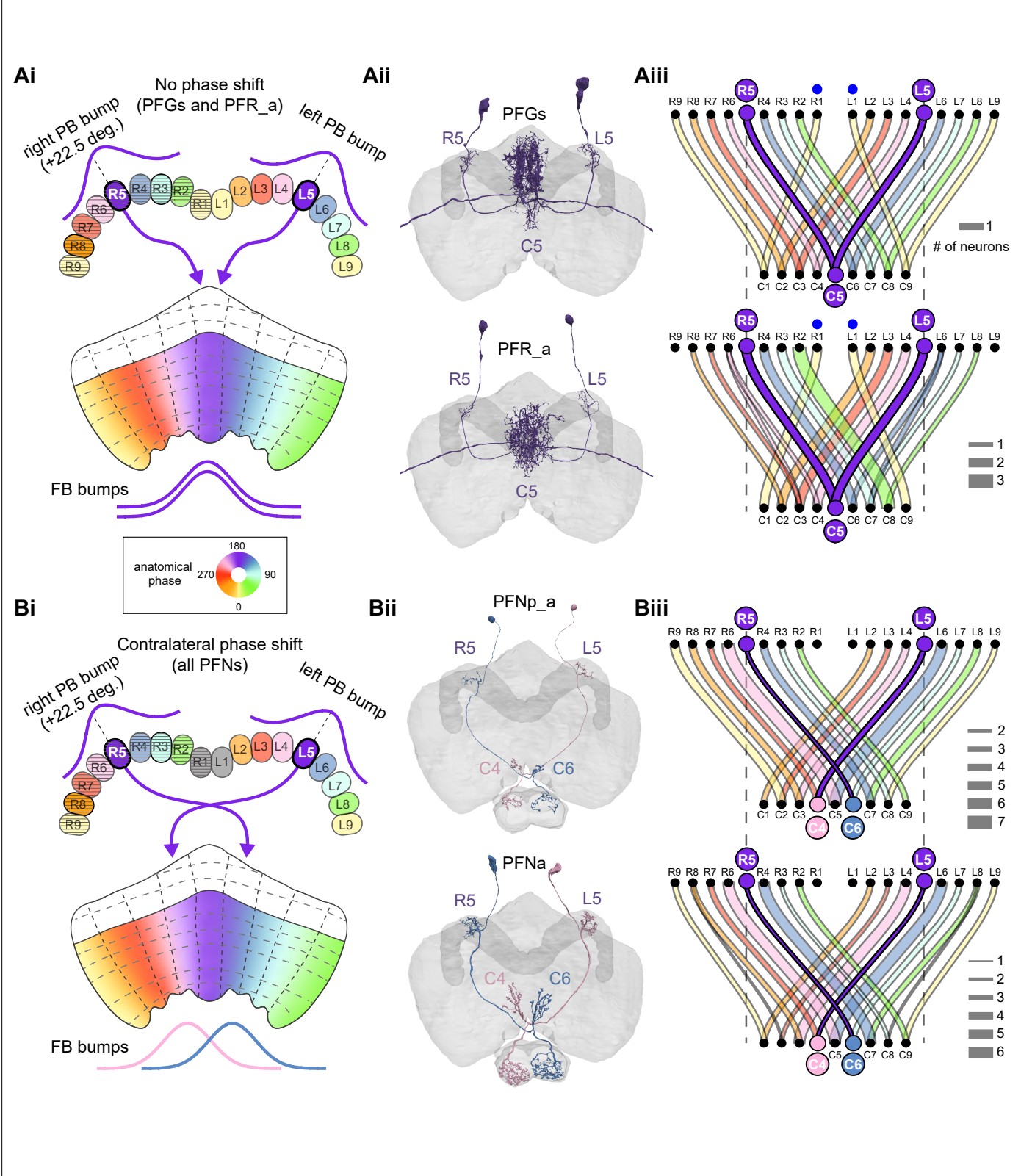

**Figure 30.** PB-FB-* neurons have type-specific phase shifts in fan-shaped body-to-protocerebral bridge (PB-to-FB) projections. (**A**) PFGs and PFRa neurons connect PB glomeruli to FB columns with no phase shift. (**Ai**) Schematic of a PB-to-FB projection pattern with no phase shift. PB glomeruli and FB columns are colored according to anatomical phase. Based on EB-to-PB columnar neuron projection patterns (EPG neurons, see *Figure 16*), when a bump is centered at L5 in the left PB, a second bump will be centered between R5/R4 in the right PB (both marked in purple). With no phase shift

*Figure 30 continued on next page*

*Figure 30 continued*

in their projection pattern, neurons innervating R5/L5 both project to C5 in the FB. This pattern, repeated across glomeruli/columns (see **Aiii**), would bring the two bumps in the PB to approximately the same FB location. (**Aii**) Morphological renderings of single neurons innervating R5 and L5, from the PFGs (top panel) and PFR_a (bottom panel) populations. Neurons are colored according to their FB column. Notice that the R5/L5 neurons end up at matching locations (C5) in the FB. (**Aiii**) Graphs showing the projection pattern from PB glomeruli to FB columns for all neurons in the PFGs (top panel) and PFR_a (bottom panel) populations. R5 and L5 projections have been highlighted as in (**Ai**). Lines connecting PB glomeruli to FB columns are colored according to PB glomerulus (i.e., anatomical phase). Blue dots mark glomeruli R1 and L1, whose neurons project to the opposite hemisphere (GAL for PFGs; ROB for PFR_a) than the other neurons in their half of the PB, the functional significance of which is unknown. (**B**) PFN types have one-column contralateral phase shifts in their PB-to-FB projection pattern. (**Bi**) Schematic of a PB-to-FB projection pattern, as in (**Ai**), but now showing a one-column contralateral phase shift. Notice that R5 projects to C6, and L5 projects to C4. This pattern, repeated across glomeruli/columns (see **Biii**), would cause PB bumps centered at R5 and L5 to end up at different locations in the FB. PFN neurons do not innervate glomeruli R1 and L1, as indicated by the gray shading. (**Bii**) Morphological renderings of single neurons innervating R5 and L5, as in (**Aii**), but now for PFNp_a and PFNa. Notice that the R5 neurons project to C6 and the L5 neurons project to C4. (**Biii**) Graphs showing the projection pattern from PB glomeruli to FB columns, as in (**Aiii**), but for PFNp_a and PFNa. Edges beginning at R5 and L5 have been highlighted, as in (**Bi**). Lines are colored according to PB glomeruli (i.e., anatomical phase).

a consistent, approximately 0° phase shift across the populations (*Figure 30Aiii*, *Video 6*). Thus, for PFGs and PFR_a neuron types, we would expect activity bumps carried by the left and right PB populations to overlap in the FB (see bottom of *Figure 30Ai*), regardless of bump location (*Figure 30Aiii*). As described previously (*Figure 19E*), each PB-FB type innervates a variable number of PB glomeruli across the population. PFGs innervate all nine glomeruli in the left and right PB, and their 0° phase shift provides one means of assigning an approximate phase to each FB column. Importantly, PFGs map glomeruli R1/R9 and L1/L9 to columns C1 and C9, respectively, suggesting that these outer FB columns share a similar anatomical phase (as shown in *Figure 30Ai and Aii, Bi and Bii*), consistent with a circular representation. The projection pattern of PFR_a neurons (*Figure 30Aiii*), which only innervate the medial eight glomeruli in the left and right PB, also supports this notion.

Unlike the PFGs and PFR_a types, all PFN neuron types have nonzero, contralateral phase shifts between their PB glomeruli and FB columns, much like PEN neurons in the PB-EB network. Specifically, as exemplified by the PFNp_a and PFNa types in *Figure 30B*, PFN neurons that innervate R5 project to C6 and neurons that innervate L5 project to C4. That is, PFN neurons connect PB glomeruli to FB regions using a one-column contralateral phase shift. This phase shift implies that PFN populations from the left and right PB would generate spatially shifted activity bumps in the FB that will be separated by ~90° (see bottom of *Figure 30Bi*). Plotting the full PB-FB mapping revealed that these one-column contralateral phase shifts are largely consistent across all PB glomeruli for these PFN neuron types (*Figure 30Biii*, *Video 7*). Notably, one distinct neuron type, the PFR_b, has a one-column ipsilateral phase shift.

Here we have provided a discrete view on PB-FB-* phase shifts, a useful description given the CX's strong topographic organization. However, these phase shifts are also apparent when viewed

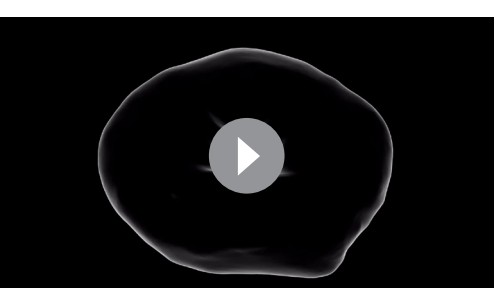

**Video 6.** PFGs phase shifts. Movie begins by showing morphological renderings of an individual EPG neuron that contacts an individual PFGs neuron in the PB. Later, PFGs pairs that innervate the left or right PB and share similar directional tunings are shown. Notice that these PFGs pairs project to similar regions of the FB, where their fibers partially overlap. This zero-degree phase shift establishes an approximate default mapping from PB glomeruli to EB columns. Related to Figure 30A.

https://elifesciences.org/articles/66039/figures#video6

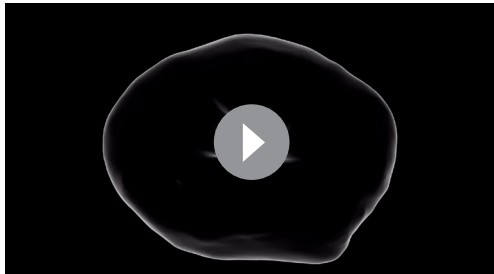

**Video 7.** PFNa phase shifts. Similar to *Video 6*, but now for PFNa neurons. Notice that, in the second half of the video, the PFNa pairs that share similar directional tuning project to spatially offset columns in the FB, generating a +/−45° phase shift. Related to Figure 30B.

https://elifesciences.org/articles/66039/figures#video7

continuously in anatomical space (*Videos 6 and 7*). As quantified further below (Figure 34), the precise magnitude of PB-FB phase shifts and FB column phase is also continuous and can be estimated using PB-FB connectivity since these phase shifts will depend on how postsynaptic neurons in the FB sample from their PB-FB inputs. As with the PEN neurons, whose phase shifts appear strongly linked to their function of shifting the EPG bump in the EB (*Green et al., 2017*; *Turner-Evans et al., 2017*; *Turner-Evans et al., 2020*), we believe that the PB-FB phase shifts offer insights as to the likely navigational function of each of these neuron types (outlined in 'Discussion'). Next, we describe the columnar structure of hΔ and vΔ types, which are the main downstream target of FB columnar types in the FB.

## Intra-FB columnar neurons: the vΔ and hΔ types

Previously referred to as 'pontine neurons' (*Hanesch et al., 1989*; *Heinze et al., 2013*; *Heinze and Homberg, 2008*; *Hensgen et al., 2021*; *Homberg, 1985*; *Siegl et al., 2009*; *Stone et al., 2017*), the FB's many interneuron types create an intricately structured scaffold for intercolumnar and inter-layer communication within the FB. We renamed these neurons vΔ and hΔ. vΔ refers to the predominantly 'vertical,' layer-skipping morphology of the constituent neurons (*Figure 31A*). hΔ refers to the predominantly 'horizontal,' column-skipping morphology of the constituent neurons (*Figure 31B*).

Two of these pontine neuron types are not FB interneurons, but rather are neurons that bring information into the FB from other CX structures. The vΔA_a neurons (*Figure 31Ai*) have mainly post-synaptic specializations in the AB, a mysterious structure at the base of the FB (*Jenett et al., 2012*; *Pascual et al., 2004*; *Wolff and Rubin, 2018*) that we discuss later, and send arbors up throughout the dorsal most layers of the FB. The hΔK neurons (*Figures 31Bi and 10F*), by contrast, each innervate wedge-shaped regions of the EB, which vary in size and coverage, and columns in intermediate FB layers before projecting to more dorsal layers of the FB. These neurons could carry head direction information directly from the EB to the FB, bypassing the PB, but their connectivity in the FB suggests that they may also be related to behavioral state, a topic we return to in a section focused on sleep circuits (Figure 53). With the exception of the vΔA_a, a subset of vΔA_b that innervates the AB, and hΔK neurons, all other vΔ and hΔ neuron types are interneurons whose arbors are restricted to the FB.

There are many different types of vΔ neurons, each distinguished by the layer(s) that they inner-vate, by the vertical distance between their multiple (usually two) arbors, and by the spread of each arbor. These different vΔ types are shown in *Figure 31—figure supplement 1* and *Video 8*. Note that although some neuron types, such as the vΔH, form nine FB columns, most types vary in how cleanly, as a population, they tile each layer that they innervate. Similar to PB-FB columnar neurons, even types with a disorganized columnar structure can be assigned to the discrete nine-column scheme based on the average location of their arbor (see Materials and methods), which provides a conve-nient means to assess their relative columnar positions. Interestingly, all vΔ types also contain one or more individual neurons that project bilaterally to both C1 and C9, which we refer to as 'C0' (see *Figure 31Ai*). These neurons provide further support that columns C1 and C9 correspond to similar anatomical phases since they receive shared into from individual C0 neurons.

All hΔ neurons have two horizontally separated arbors within the FB, one of which is predominantly presynaptic (*Figure 31B*). Each hΔ type has matching left- and right-projecting neuron pairs whose two arbors largely overlap. As a population, both the right- and left-projecting populations tile all FB columns. The many hΔ types are distinguished by the layer(s) that they innervate and by the horizontal spread of their arbors. These different types are shown in detail in *Figure 31—figure supplement 2* and *Video 9*. By contrast with the nine-column tiling of the vΔ neurons, some hΔ types divide the layers they innervate into 12 columns, such as the hΔA and hΔL neurons, others into 8 columns, such as the hΔD and hΔH neurons, and one, the hΔF, into just 6 columns (*Figure 31—figure supplement 2*). In most cases, the population of neurons within each type neatly tile the layers that they innervate (see, e.g., the hΔH and hΔM neurons). Overall, the structure of the hΔ and vΔ backbone provides an avenue to determine the direction of information flow through the intra-FB columnar network, which we discuss in a later section (Figure 37).

## FX types: novel FB pathways to and from putative premotor and contextual centers

In addition to the PB-FB-*, vΔ, and vΔ columnar classes described above, the connectome revealed ~20 columnar types – including many novel types – belonging to the FX class, which innervate either

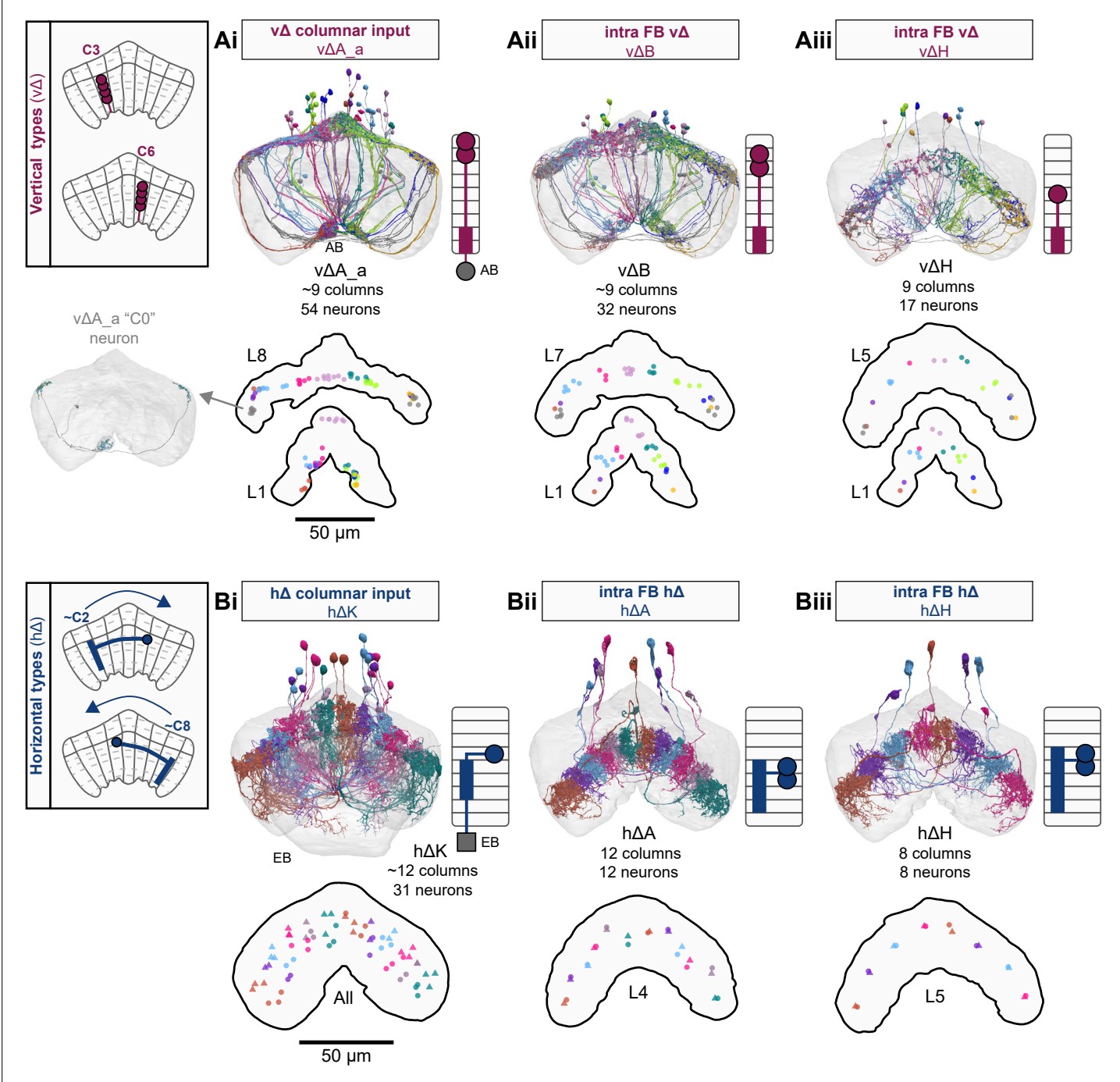

**Figure 31.** Overview of vΔ and hΔ columnar structure. (**A**) Vertical columnar interneurons – the vΔ neuron types – have individual neurons with processes centered around one fan-shaped body (FB) column. Schematic on left shows two schematized neurons with arbors centered on C3 and C6. (**Ai**) Morphological rendering of the vΔA_a population, along with their schematized innervation pattern. Individual neurons are colored by FB column (from C1 to C9). In addition to innervating the FB, vΔA_a neurons (and some vΔA_b) are unique among vΔ neurons in that they innervate an extra-FB area, the asymmetric body (AB). Also notice the high degree of overlap of processes in the dorsal FB and the messy columnar structure of the population. Inset to the left shows a 'C0' neuron, which has arbors in both C1 and C9. (**Aii**) Same as in (**Ai**), but for the vΔB population. As with all other vΔ types, these neurons have processes restricted to the FB and receive most of their input in ventral layers while sending most of their output to more dorsal layers. (**Aiii**) Same as in (**Ai**), but for the vΔH population. (**B**) Horizontal columnar interneurons – the hΔ types – have individual neurons with processes centered on two distant FB columns, as shown in the illustration for two generic hΔ neurons. In particular, each hΔ neuron has a dendritic compartment that is ~180° away from its axonal compartment (i.e., separated by half the FB's width). Half of the population has dendrites in right FB columns and project to left FB columns, while the other half of the population does the opposite. Individual hΔ neurons are assigned to columns based on the location of

*Figure 31 continued on next page*

*Figure 31 continued*

their dendritic compartment. (**Bi**) Morphological rendering of the hΔK population, along with their schematized innervation pattern. Individual neurons are colored according to FB column, with paired columns given matching colors. To achieve the ~180° phase shift, all hΔ types form an even number of columns. In this case, 12 columns (marked with six colors). In addition to innervating the FB, hΔK neurons are unique among hΔ neurons in that they innervate an extra-FB area, the EB. (**Bii**) Same as in (**Bi**) but for the hΔA population, which also forms 12 columns. Like most hΔ neurons, hΔA receives most of its input in ventral FB layers and provides most of its output to more dorsal FB layers. (**Biii**) Same as in (**Bi**) but the for the hΔH population, which forms eight columns instead of 12. Note the highly columnar structure of hΔ neuron types compared to the vΔ neuron types from (**Ai**) to (**Aiii**).

The online version of this article includes the following figure supplement(s) for figure 31:

**Figure supplement 1.** Columnar structure of vΔ neuron types.

**Figure supplement 2.** Columnar structure of hΔ neuron types.

the round body (ROB), SMP/SIP/SLP, or CRE (*Figure 32*). These types do not have arbors in the PB. Each type is composed of individual neurons that collectively tile all columns of the FB (*Figure 32—figure supplements 1–2*, *Video 10*), and the various types can be distinguished by the layer(s) of the FB they innervate, their overall columnar structure, and the extra-FB region that they project to. FR types send projections to the ROB; FS types send projections to the SMP/SIP/SLP regions; and FC types send projections to the CRE. With few exceptions, FR and FS types have primarily postsynaptic arbors in the FB and presynaptic specializations in their extra-FB neuropil (*Figure 28—figure supplement 1*), potentially forming additional columnar output pathways, as discussed in the CX outputs sections. FC types have mixed arbors both within and outside the FB (*Figure 28—figure supplement 1*), perhaps providing bidirectional communication between the FB and CRE.

## Intra-FB connectivity of the FB columnar neurons

In the sections above, we described the major FB neuron types, with a particular focus on their morphology, columnar structure, and extra-FB innervations. We now turn our focus to connectivity within the intra-FB network and describe how columnar information may flow from PB-FB inputs to columnar outputs.

Much of the FB's columnar input comes from the PFN neurons (*Figure 33A*, *Figure 33—figure supplement 1*, see also *Figures 19 and 25–26*), which have postsynaptic specializations in both the PB and NO. Some of the PFN types also make synapses within their type in the FB columns that they innervate – much like the PEN neurons in the EB. Though there are direct connections between PFN neurons and columnar FB output neurons in the FB, the majority of PFN synapses are to vΔ, hΔ, and FC neurons (*Figure 33A*, *Figure 33—figure supplement 1*). Thus, much of the input to the FB passes into a multilayer, multicolumn intra-FB network.

**Video 8.** Morphological renderings of the vΔ neurons. Each of the vΔ cell types is shown in order as follows: the vΔA_a, vΔA_b, vΔB, vΔC, vΔD, vΔE, vΔF, vΔG, vΔH, vΔI, vΔJ, vΔK, vΔL, and vΔM neurons. Each neuron has been assigned to one of nine (loosely defined) FB columns, and is color coded accordingly. For each cell type, example neurons are shown first, followed by the entire population. Each cell type also has neurons that arborize in both column 1 and column 9. These neurons are shown in gray, and an example of one multi-columnar neuron from each population is shown after the entire population is displayed.
https://elifesciences.org/articles/66039/figures#video8

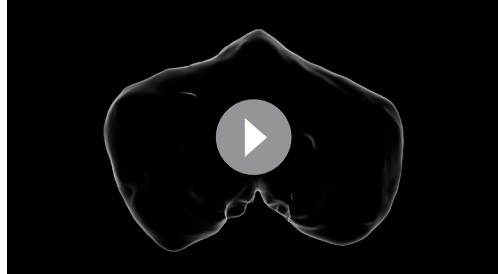

**Video 9.** Morphological renderings of the hΔ neurons. Each of the PB-FB columnar cell types is shown in order as follows: the PFGs, PFL1, PFL2, PFL3, PFNa, PFNd, PFNm_a, PFNm_b, PFNp_a, PFNp_b, PFNp_c, PFNp_d, PFNp_e, PFNv, PFR_a, and PFR_b neurons. Each neuron has been assigned to one of nine (loosely defined) FB columns, and is color coded accordingly. For each cell type, example neurons are shown first, followed by the entire population.
https://elifesciences.org/articles/66039/figures#video9

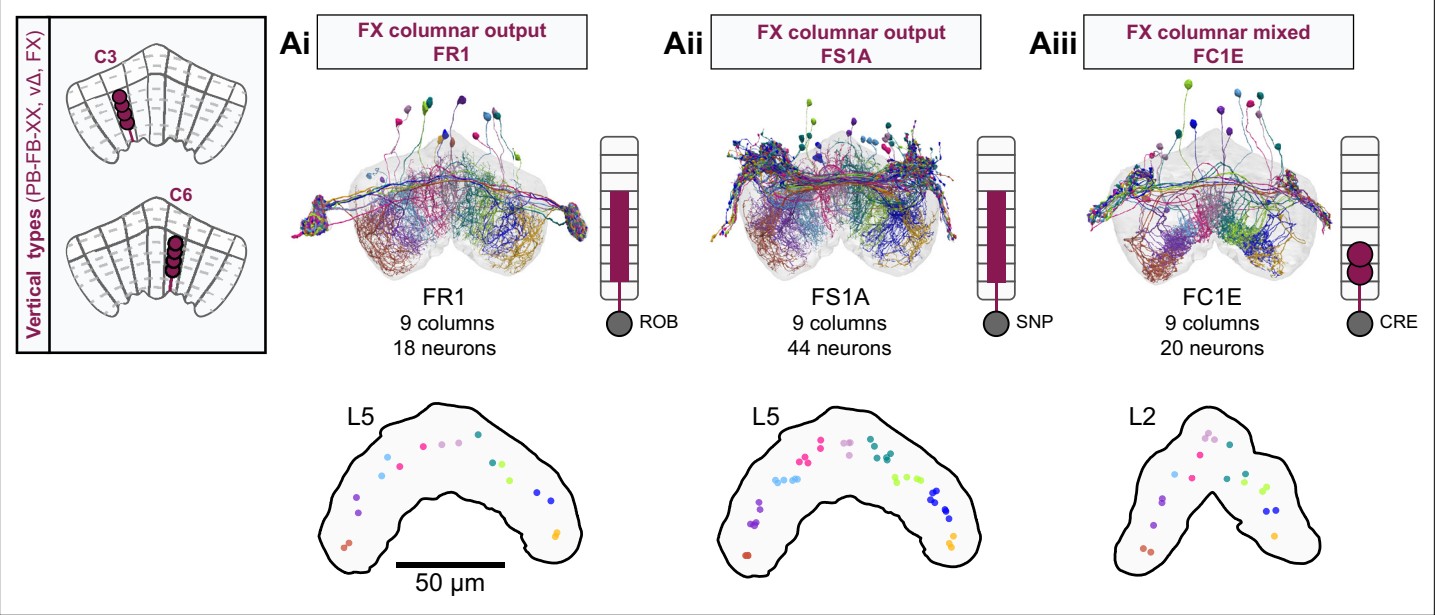

**Figure 32.** Overview of FX columnar structure. (**A**) FX neurons types all have a vertical morphology, with processes centered around one fan-shaped body (FB) column. Schematic on left shows two schematized neurons with arbors centered on C3 and C6. (**Ai**) Morphological rendering of the FR1 population, along with their schematized innervation pattern. Individual neurons are colored by FB column (from C1 to C9). In addition to innervating the FB, FR types innervate the ROB. (**Aii**) Same as in Ai, but for the FS1A population. FS types innervate both the FB and the SMP/SIP/SLP. (**Aiii**) Same as in Ai, but for the FC1E population. FC types innervate both the FB and the CRE.

The online version of this article includes the following figure supplement(s) for figure 32:

**Figure supplement 1.** Columnar structure of FR and FS neuron types.

**Figure supplement 2.** Columnar structure of FC neuron types.

To gauge the 'depth' of the intra-FB network, we quantified the different 'path lengths' from PFN inputs to columnar FB output neurons (*Figure 33Bi*) (see Materials and methods). For example, PFNa neurons have direct connections to PFL1 neurons. This is a path length of 1. PFNa neurons also have indirect connections to PFL2 neurons through first hΔJ and then hΔH neurons (*Figure 33Bi and Bii*). This is a path length of 3. We found that direct connections from PFN neurons to output neurons are relatively sparse, while many more two- or three-step pathways can be traced through hΔ and/or vΔ neurons (*Figure 33Bii*). While highly recurrent, the intra-FB network can therefore roughly be thought of as being 3–4 layers deep, though we note that this analysis does not include the FB tangential neurons.

We next sought to characterize the structure of the intra-FB network connectivity. The connectivity shows clear indications of preferred pathways and subnetworks. For example, the PFNa/FC1/PFL1 types are primarily connected to one another, with only sparse connections to other FB neuron types. These types therefore form their own subnetwork within the FB (below the line in *Figure 33A*, *Figure 33—figure supplement 1*). Examining this subnetwork at the individual neuron level (*Figure 33C*), we found that the connectivity patterns in the subnetwork largely

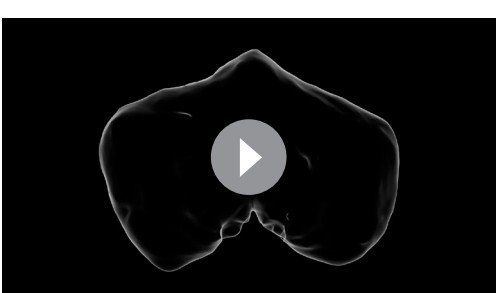

**Video 10.** Morphological renderings of the FR, FC, and FS neurons. Each of the FR, FC, and FS cell types is shown in order as follows: the FR1, FR2, FC1A, FC1B, FC1C, FC1D, FC1E, FC1F, FC2A, FC2B, FC2C, FC3, FS1A, FS1B, FS2, FS3, FS4A, FS4B, and FS4C neurons. Each neuron has been assigned to one of nine (loosely defined) FB columns, and is color coded accordingly. For each cell type, example neurons are shown first, followed by the entire population. After each individual population is shown, the populations are then each given a unique color and displayed rapidly one-by-one before they are all shown simultaneously.
https://elifesciences.org/articles/66039/figures#video10

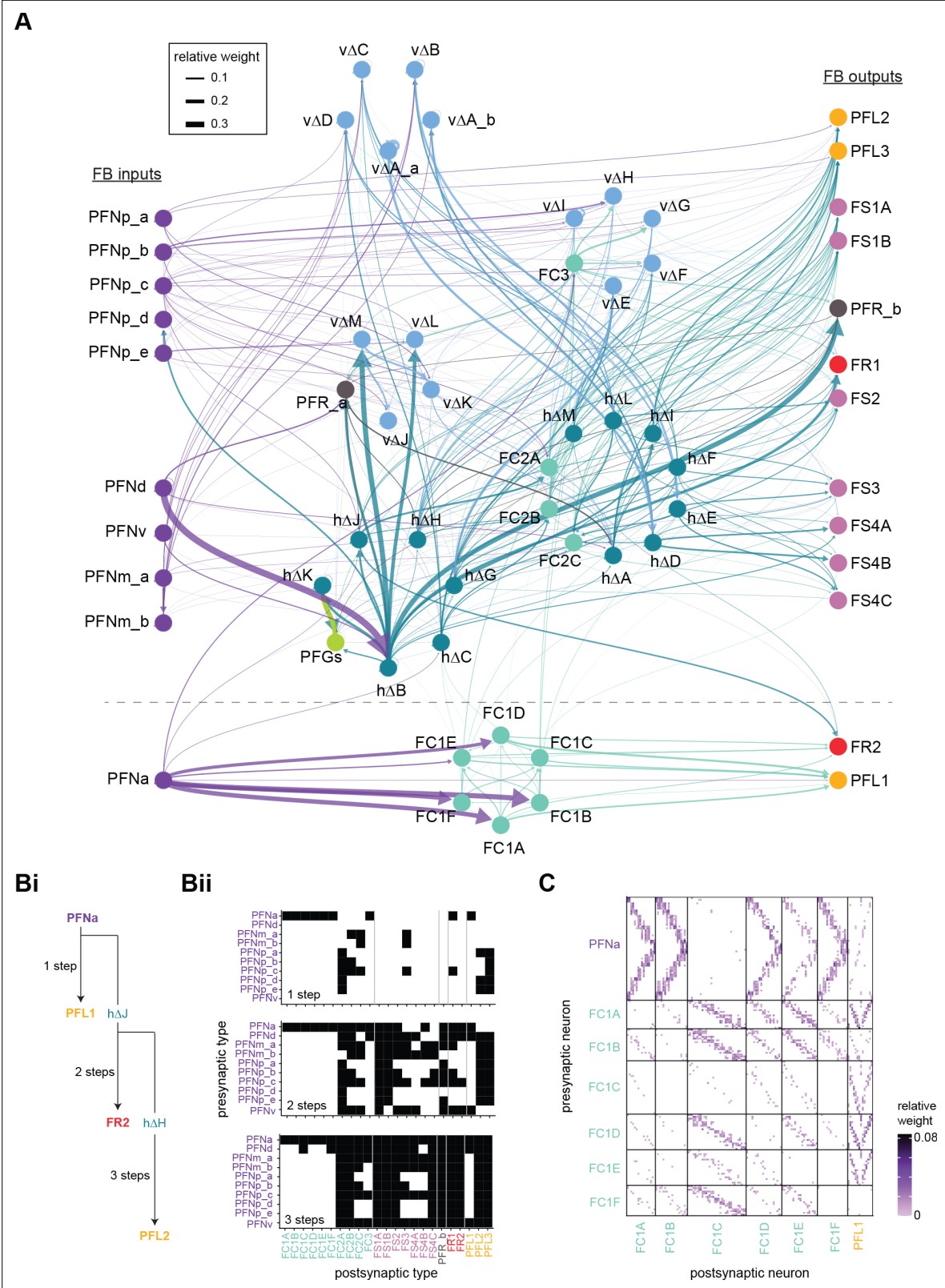

**Figure 33.** Fan-shaped body (FB) columnar type to columnar type connectivity. (**A**) The type-to-type connectivity between FB columnar neuron types arranged in a three-layer network diagram. FB inputs are shown at far left while FB outputs are shown at far right. Neuron nodes are color-coded by that neuron's class. Only connections where most of the presynaptic neurons connect to a postsynaptic neuron of the given type are shown (more than 2/3 of the columns must connect across types). (**B**) The number of steps between columnar FB inputs and columnar FB outputs through other columnar FB

*Figure 33 continued on next page*

*Figure 33 continued*

neurons. (**Bi**) While PFN neurons directly connect to a few of the FB columnar output neurons in the FB (top), the pathways between PFN neurons and columnar outputs are often longer, traveling through one (middle), two (bottom), or more intermediate columnar neurons. (**Bii**) Direct (top), two-step (middle), and three-step (bottom) connections between PFN and FB columnar output neurons are shown in black. (**C**) Neuron-to-neuron connectivity matrix for the PFNa, FC1, and PFL1 neurons. Type-to-type connections between these neurons are shown below the dotted horizontal line in (**A**).

The online version of this article includes the following figure supplement(s) for figure 33:

**Figure supplement 1.** Type-to-type connectivity matrix between fan-shaped body (FB) columnar neurons.

**Figure supplement 2.** Clustering by upstream and downstream partners.

**Figure supplement 3.** The vΔF, G, H, and I subnetwork.

matched the columnar overlap of the different constituent neurons. However, we found that direct connections from individual PFNa neurons to PFL1 neurons were not as strong or as consistent across neurons as direct connections from PFNa neurons to FC1 neurons. FC1 neurons themselves send their outputs to a putative motor area (CRE), to PFL1 neurons within their column, and to other FC neurons. Thus, the indirect pathways from PFNa to FC1 to PFL1 may contribute more to PFL1 activity than the direct PFNa to PFL1 connections. These indirect pathways point to the importance of depth in FB networks.

To parse additional subnetworks in the FB, we next grouped neuron types based on common connectivity patterns – specifically, their upstream and downstream partners (*Figure 33—figure supplement 2*, and Materials and methods). We expected that types within a subnetwork would share common upstream and downstream partners. Indeed, the FC1 neurons clustered together based on both their inputs and outputs, a clear signature of the PFNa/FC1/PFL1 subnetwork. The vΔF, vΔG, vΔH, and vΔI neurons also clustered together (*Figure 33—figure supplement 2*, *Figure 33—figure supplement 3*). However, most neuron types that share an upstream cluster split into multiple downstream clusters (and vice versa). This splitting suggests that the intra-FB network cannot, through columnar connectivity alone, be cleanly split into multiple subnetworks. Instead, information is propagated throughout the FB, across layers and columns. However, we note that FB tangential neurons may 'gate' some of these pathways, allowing for a clean functional separation of subnetworks. We return to this topic in the 'Discussion.'

## PB-FB columnar connectivity: preserved phase shifts without functional lateralization

The anatomical projection pattern of PB-FB neuron types suggests that they convey activity bumps from the left and right PB to the FB, but with type-specific phase shifts, as described above (*Figure 30*). In particular, the projection pattern of PFGs and PFR_a types suggests that they convey the activity bumps from the left and right PB to overlapping columnar locations in the FB, establishing a 'default' mapping between PB glomeruli and FB columns. In contrast, all PFN types have a one-column contralateral phase shift, predicting that the left and right activity bumps will end up ~90° apart in the FB (*Figure 34A*). Consistent with this notion, these anatomical phase shifts impact PB-FB neurons' connectivity with their downstream targets in the FB (*Figure 34B, C*). For example, PFNa neurons that innervate glomerulus R5 synapse onto FC1B neurons in C6, and L5 neurons synapse onto FC1B neurons in C4, consistent with their anatomical projection pattern (top panel, *Figure 34C*). More nuanced morphological characteristics also affect connectivity. For example, PFNp_a neurons, which arborize in spatially restricted regions of the FB (*Figure 29*), primarily connect single PB glomeruli to single FB columns, while PFNa neurons, which have slightly wider FB arbors, connect single PB glomeruli to several adjacent FB columns (*Figure 34C*). Thus, PB-FB phase shifts structure columnar input to the FB network.

The above analysis views PB-FB projections as connecting individual glomeruli in the PB to discrete columns in the FB. However, as noted above (*Figures 29, 31 and 32*), FB columnar neurons form a type-specific number of columns and there is a large variability in how evenly each type tiles the FB. This raises the possibility that the directional tuning of FB neurons may vary continuously according to their medial-lateral position in the FB. To assess this possibility, we used connectivity to infer the directional tuning of FB neurons by taking circular means of the phases that FB neurons inherit from their presynaptic PB-FB inputs (*Lyu et al., 2020*; *Figure 34—figure supplement 1*). This analysis revealed a

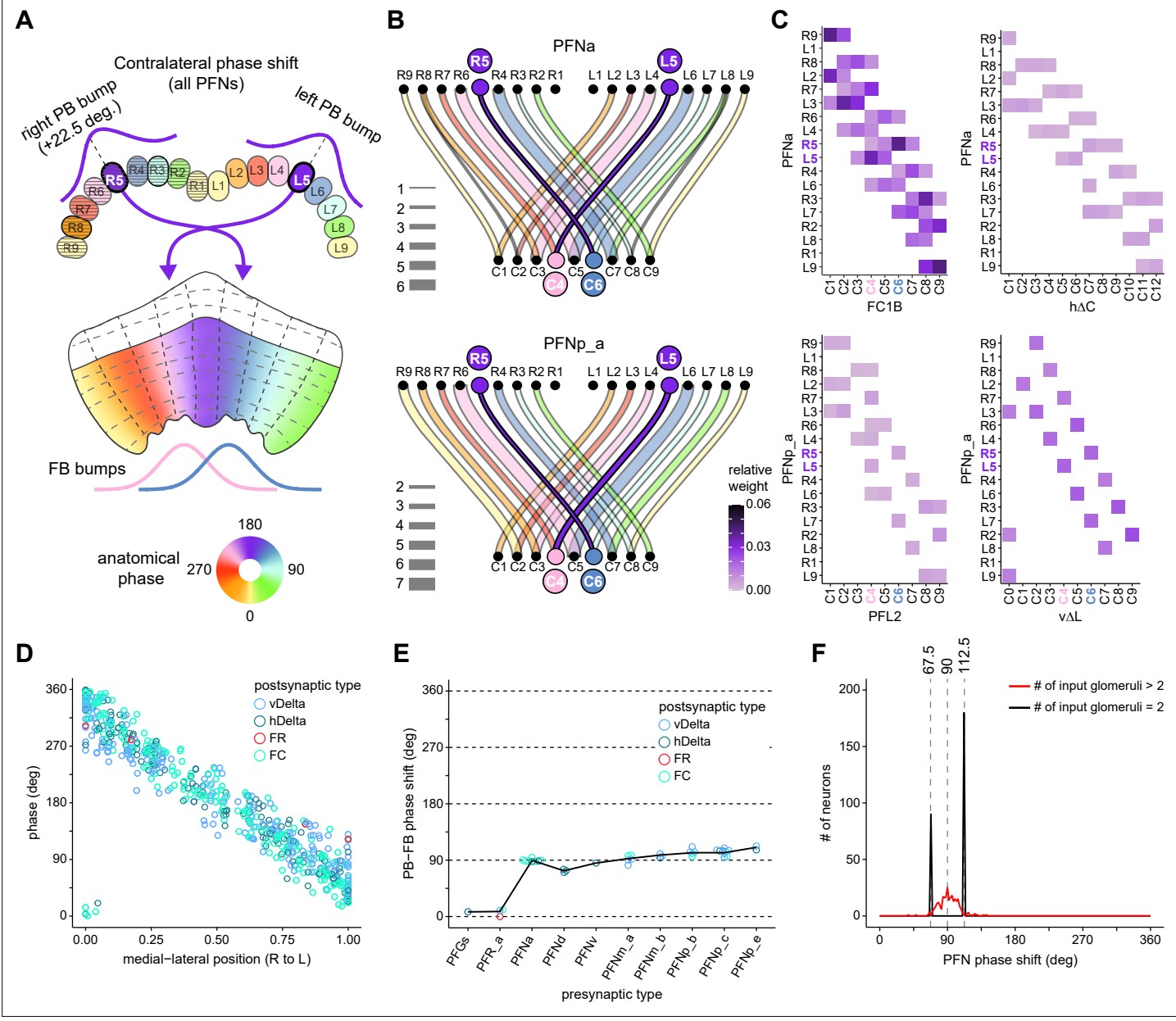

**Figure 34.** Protocerebral bridge to fan-shaped body (PB-FB) projection patterns determine FB neuron's phase shift and directional tuning. (**A**) Schematic of a PB-to-FB projection pattern showing the one-column contralateral phase shift employed by PFN types, as in *Figure 30B*. (**B**) Graphs showing the projection pattern from PB glomeruli to FB columns for all neurons in the PFNa (top panel) and PFNp_a (bottom panel) populations, as in *Figure 30B*. (**C**) Connectivity between PFNa (top panel) or PFNp_a (bottom panels) neurons and two of their downstream partners within the FB. Notice that PFN neurons that arborize in glomeruli R5 or L5 connect with distinct columns in the FB, consistent with their PB-FB phase shifts. (**D**) Scatter plot showing the estimated directional tuning of FB neurons as a function of their medial-lateral position. For every vΔ, hΔ, or FX neuron postsynaptic to a PB-FB type, directional tuning was estimated by assigning angles to PB-FB neurons according to the PB glomerulus they innervate and by taking a circular mean across all angles inherited by the postsynaptic FB neuron, weighted by connection strength (*Lyu et al., 2020*; *Figure 34—figure supplement 1*, and Materials and methods). Medial-lateral position was normalized from 0 (right border to FB) to 1 (left border of FB) to account for the varying width of the FB layers occupied by each postsynaptic type. (**E**) Anatomical phase shift for PB-FB neuron types. Each circle is an estimated phase shift from the presynaptic PB-FB type to one of its postsynaptic types (vΔ, hΔ, or FX). Phase shifts were estimated across all postsynaptic neurons of a type individually and the circular mean was taken as the type average (black line). Note that PFR_b, PFNp_a, and PFNp_d types were excluded from this analysis due to inconsistent downstream connectivity (*Figure 35*) or because they exclusively target hΔ types on both axonal and dendritic compartments (*Figure 37*), both of which complicated phase shift estimates (see Materials and methods). (**F**) Histograms of PFN phase shift magnitude across all postsynaptic FB neurons (vΔ, hΔ, or FX), colored according to whether the postsynaptic FB neurons sample from presynaptic PFN neurons

*Figure 34 continued on next page*

Figure 34 continued

from two glomeruli (black) or from presynaptic neurons from more than two glomeruli (red). For individual neurons to have a 90° phase shift, they must sample from presynaptic PB-FB neurons that innervate at least two PB glomeruli (see *Figure 34—figure supplement 1B*).

The online version of this article includes the following figure supplement(s) for figure 34:

**Figure supplement 1.** Estimating protocerebral bridge-fan-shaped body (PB-FB) phase shifts and directional tuning of FB neurons.

strong correlation between FB neurons' medial-lateral position and their estimated directional tuning (*Figure 34D*). Consistent with the notion that the FB inherits a sinusoidal activity bump from the PB, the medial and lateral borders are tuned to similar phases, as expected for a circular representation. In addition, FB neurons' directional tuning is quite evenly distributed, suggesting that, rather than forming discrete columns that are consistent across neuron types, FB neurons can take on a range of angles that is largely determined by their medial-lateral position and, therefore, the subset of PB-FB inputs that they sample from.

This approach (*Lyu et al., 2020*) also allowed us to estimate the magnitude of the PB-FB phase shift for PB-FB neuron types (*Figure 34E*). In agreement with our projection-based analysis (*Figure 30*), PFGs and PFR_a were found to have anatomical phase shifts close to 0°. In contrast, all PFN neuron types have an ~90° phase shift, with some variability across neuron type. To investigate the source of PFN phase shift variability, we analyzed how individual FB neurons sample from the left and right PB-FB populations (*Figure 34F* and *Figure 34—figure supplement 1*). Consistent with our angular assignments in the PB, FB neurons that sample from just two distinct angles – one from the left PFN population and one from the right PFN population – have phase shifts that are usually 67.5 or 112.5° (*Figure 34F*). This is because no left-right pair of glomeruli is separated by 90° (*Figure 34—figure supplement 1B*). Instead, neurons with a 90° phase shift sampled from at least three PB glomeruli. This effect explains some of the type-to-type variability in PFN phase shifts. For example, FB neurons that receive input from PFNa neurons have phase shifts very close to 90° since they mostly sample from at least three glomeruli (*Figure 34—figure supplement 1C*), likely due to PFNa neurons' wide FB arbors, as mentioned above. In contrast, neurons that receive input from PFNp types often sample from just two glomeruli that are 112.5° apart. This may be because PFNp types form spatially clustered arbors in the ventral FB, which favors a more restricted connectivity pattern, as mentioned above. Since this effect reflects the diverse ways in which FB neurons sample from the left and right PB populations, it is unlikely to be due to the precise angles assigned to PB glomeruli. We return to the potential functional role of PB-FB phase shifts in the context of vector navigation in 'Discussion.'

These PB-FB projection patterns raises the possibility that the bumps conveyed from the left and right PB could propagate independently through the FB network, a scenario we refer to as 'functional lateralization.' Does the FB network's connectivity support such a scenario? One signature of lateralization would be that PFN neurons from different sides of the PB might project to different neuron types in the same FB columns (*Figure 35A*). However, we found no systematic differences in left versus right PFN inputs to different downstream FB neuron types (*Figure 35B*). A second signature of lateralization might be PFN neurons from different sides of the PB projecting to distinct neurons within a downstream type, perhaps organized by demi-columns (*Figure 35C*). However, we found that the input contributions of left and right PB neurons were very similar for most downstream neurons (*Figure 35D*). Although there are some exceptions to this rule, these exceptions are almost all neurons with both weak and inconsistent connections at the population level (small dots to the top left of the plot in *Figure 35E* and an absence of large dots at the top right). Note that the PFNd neurons do preferentially and consistently make lateralized connections with other PFNd neurons. That is, within the FB, PFNd neurons from the left PB synapse onto PFNd neurons from the right PB, and those from the right PB onto those from the left PB, but neither population makes synapses with other PFNd neurons from the same side of the bridge (see arrow in *Figure 35E*). Thus, consistent with physiological reports from some PB-FB neuron types (*Shiozaki et al., 2020*), the bumps from the left and right PB are likely to be summed by each downstream FB neuron type, rather than multiple bumps that might then be processed independently within the FB. Note, however, that this does not rule out the possibility of multiple activity bumps arising from *different* PFN types, the navigational implications of which we explore in 'Discussion.' Note also that although there appears to be no lateralization in the context of FB bump dynamics, there is considerable asymmetry in how some FB neurons from the left and right halves of the structure project to the AB.

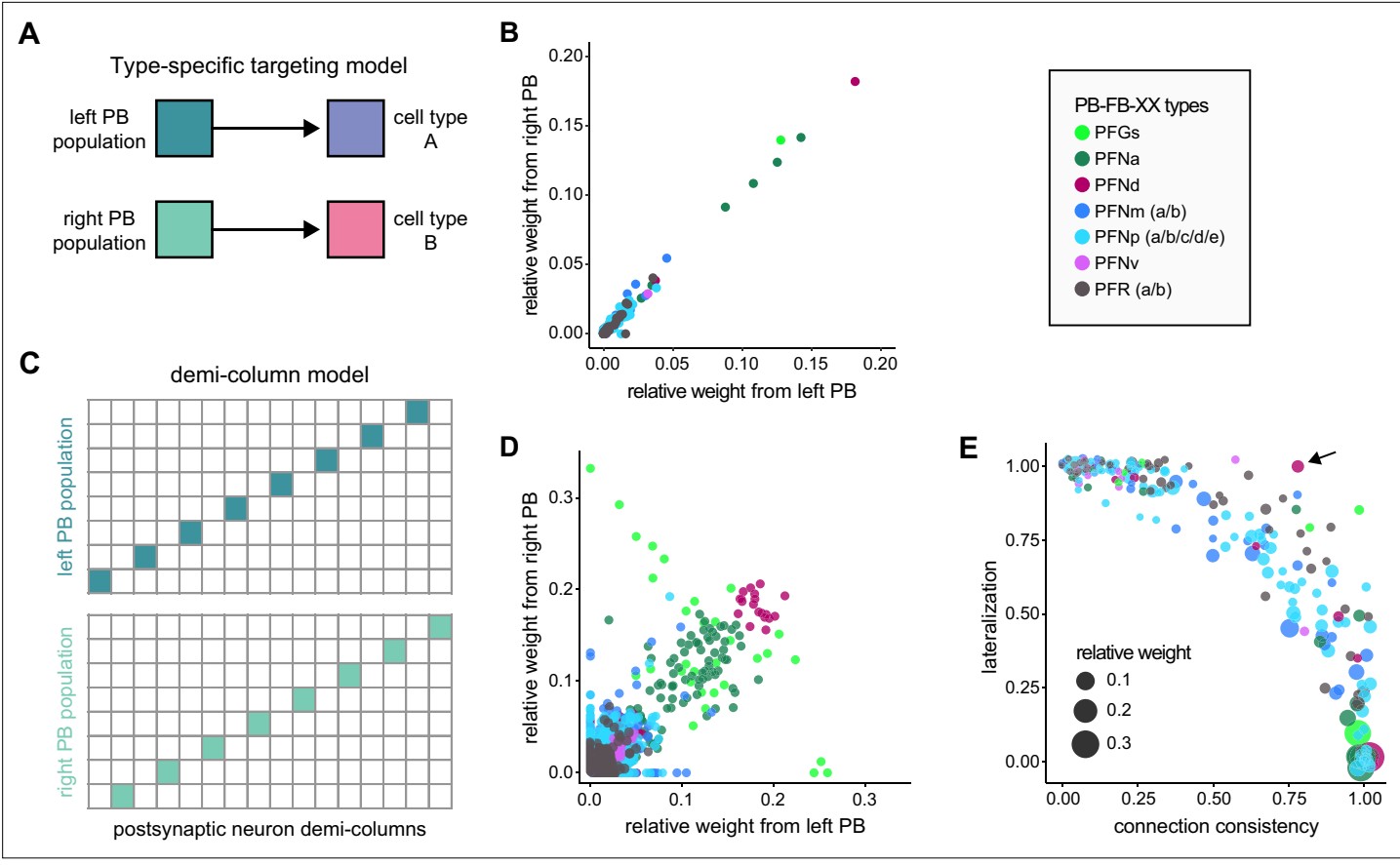

**Figure 35.** Right and left PB-FB-* populations target the same FB neuron types and neurons. (**A**) Schematic showing one potential mechanism – type-specific targeting by left and right PB-FB-* populations – by which activity from the left and right PB could propagate through separate FB channels. This model predicts that PB-FB-* neurons from the left and right PB should target distinct downstream neuron types in the FB. (**B**) Scatter plot showing the average input from left (x-axis) and right (y-axis) PB-FB-* neurons onto downstream neuron types. Each circle is a downstream neuron type, and circles are colored according to the upstream PB-FB-* type (see legend). If the model from (**A**) were true, some points should lie along the x and y axes, indicating specific input from the left or right PB populations. Instead, every downstream type receives approximately equal input from left and right PB populations, ruling out the model from (**A**). (**C**) Schematic showing a second potential mechanism – demi-column-specific targeting by left and right PB-FB-* populations – by which activity from left and right PB could propagate through separate FB channels. This model predicts that individual neurons in a downstream population should receive input from the left or the right PB population (high 'lateralization'), but not both. (**D**) Scatter plot showing the average input from left (x-axis) and right (y-axis) PB-FB-* neurons onto individual neurons in downstream populations. Each circle is a downstream neuron, and circles are colored according to the upstream PB-FB-* type (see legend). If the model from (**C**) were true, all neurons (circles) in a downstream population would lie along the x- or y-axis, indicating specific input from the left or right PB population. While some points do lie along the axes, most circles lie along the diagonal, suggesting roughly equal input from the left and right PB populations, similar to (**B**). (**E**) Scatter plot showing left/right lateralization (y-axis. the proportion of downstream neurons that receive input from the left or the right PB but not both) according to the connection consistency (x-axis. the proportion of neurons in a downstream neuron type targeted by a PB-FB-* neuron type). The size of circles indicates connection strength. Each circle is a downstream neuron type targeted by an upstream PB-FB-* type. The model from (**C**) predicts that points should lie in the upper-right portion of the plot, indicating a strong connection that is highly lateralized. instead, only weak and inconsistent connections show lateralization. With few exceptions (e.g., PFNd-to-PFNd), strong and consistent connections have downstream neurons that receive input from both left and right PB populations (low lateralization). This rules out the model from (**C**). Note that a small jitter was introduced so that overlapping points could be resolved. Arrow marks PFNd to PFNd connectivity, a connection that is fairly strong and lateralized.

## The AB: the only clearly asymmetrical structure in the fly brain

The AB, which sits at the base of the FB (*Figure 36A*), is now considered a core structure of the CX (*Wolff and Rubin, 2018*). As suggested by its name, the structure is notable for having distinctly different sizes on either side of the midline, with the right side typically being larger than the left (*Jenett et al., 2012*; *Pascual et al., 2004*; *Wolff and Rubin, 2018*). There is some evidence suggesting that flies with an AB that is roughly equal in size in both the right and left hemisphere have reduced long-term memories of shock-associated odors as compared to their more asymmetric conspecifics,

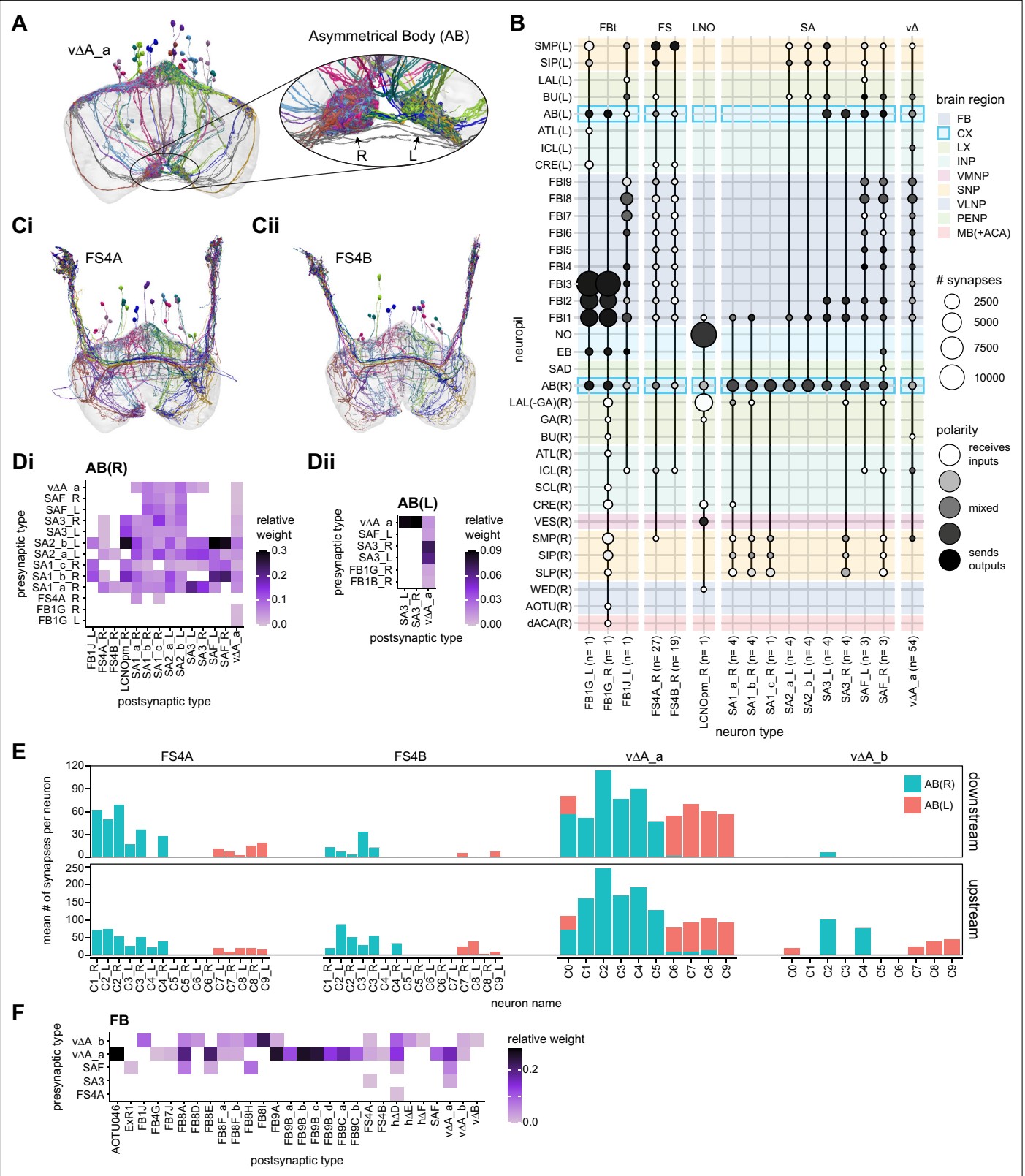

**Figure 36.** Overview of the asymmetric body (AB). (**A**) Morphological renderings of the vΔA_a neurons, which arborize in the fan-shaped body (FB) and in the AB. They are columnar in the FB, with columns 1–5 projecting into the right AB and columns 6–9 projecting into the left AB. (Inset at right) The right AB is noticeably larger than the left AB. (**B**) Region arborization plots for each neuron type that contains arbors in the AB. (**C**) Renderings of FS4A (**Ci**) and FS4B (**Cii**) neural populations. These neuron types are columnar and receive input in both the AB and the FB and output to the superior medial

*Figure 36 continued on next page*

*Figure 36 continued*

protocerebrum (SMP). (**D**) Type-to-type connectivity matrix for the right (**Di**) and left (**Dii**) AB. The smaller left AB has fewer types that make significant connections. (**E**) The mean number of downstream (top) and upstream (bottom) synapses in the right (cyan) or left (red) AB by their FB column of origin for the columnar FB-AB neurons. (**F**) Type-to-type connectivity matrix for the downstream partners of the columnar FB-AB neurons in the FB.

The online version of this article includes the following figure supplement(s) for figure 36:

**Figure supplement 1.** Additional asymmetrical body (AB) connectivity.

though their short-term memories are unaffected (*Pascual et al., 2004*). Neurons in the AB have also been associated with fructose feeding preference (*Musso et al., 2021*). The AB may therefore serve multiple functional roles. Indeed, the AB receives inputs from many different brain regions, including the SMP, SIP, ATL, and CRE, and sends its outputs primarily to the dorsal and ventral layers of the FB (*Figure 36B*). Pathways upstream from the AB inputs primarily originate in the LH, though clock neurons, the AL, vPNs, and other sources also appear (*Figure 36—figure supplement 1A*). Many of the AB neuron types that innervate the FB have been described previously (*Wolff and Rubin, 2018*), but we identified two new types that primarily get input in the AB: the FS4A and FS4B neurons (*Figure 36C*). FS4A and FS4B also receive input throughout the FB and output in the SMP/SIP/SLP.

The asymmetry of the two sides of the AB is clearly reflected in the connectivity of the neuron types that arborize there. The left AB primarily connects the vΔA_a neurons with the SA3 neurons, while the right AB connects many more types of AB neurons (*Figure 36D*). Another notable feature is that FB-AB columnar neurons that arborize in the right half and center of the FB send processes to the right AB, while neurons that arborize in the left FB send processes to the left AB (*Figure 36E*, *Figure 36—figure supplement 1B*). This is in stark contrast with the NO, whose left and right halves both receive PFN projections from all FB columns. The FB targets of AB-FB neurons are varied, including tangential, vΔ, and hΔ neurons (*Figure 36F*, *Figure 36—figure supplement 1C*). The role of the asymmetry introduced by the AB is not currently understood.

## Intra-FB columnar connectivity: a constrained 2D grid for recurrent computations

Unlike the EB network, whose columnar neurons form recurrent EB-PB loops, FB columnar neurons mainly receive input in the PB and provide output to the FB, suggesting a primarily feedforward pathway (*Figure 22*). Even PFNv neurons, which have presynaptic sites within the PB, do not receive much columnar input in the FB, making recurrent columnar loops unlikely. This suggests that the FB acts as a way station along a feedforward pathway, receiving a PB bump – modified by input from accessory structures like the NO – before initiating actions by transmitting commands to premotor centers. However, also in contrast to the EB, the FB is characterized by dense and highly specific intercolumnar and interlayer recurrent connections between different neuron types. This connection matrix likely strongly influences bump dynamics and enables recurrent network dynamics *within* the structure. We now focus on describing the most prominent motifs in this network.

As described in a previous section (*Figure 31*), the backbone of the FB's recurrent network is a 2D grid formed by the hΔ and vΔ neurons. More generally, all FB columnar types can be divided into two broad categories based on the morphology of their neurons (*Figure 37A*). The processes of 'vertical' neuron types are largely confined to a single column and include all PB-FB-*, FX, and vΔ types. 'Horizontal' hΔ neuron types, by contrast, have processes in two distinct columns. As depicted in *Figure 37A*, these vertical and horizontal neuron types connect to each other in several different ways. Importantly, however, the connectivity between different vertical and horizontal types closely respects some simple rules. We found that connections were typically either localized to the same column (*Figure 37Bi*), jumped half the width of the FB (*Figure 37Bii*), or did both (*Figure 37Biii*). In this last case, a presynaptic neuron in one FB column synapses onto postsynaptic hΔ neurons, some with primarily dendritic arbors in the column and some with primarily axonal arbors in the column. Because hΔ neurons are assigned to columns based on the location of their dendritic arbor, this connectivity motif produces two diagonal bands in the column-to-column connectivity matrix (bottom panel of *Figure 37Biii*).

It is as yet unknown whether these connections are excitatory or inhibitory, something that – much as is the case with the Δ7 neurons in the PB – would impact how activity is propagated across these

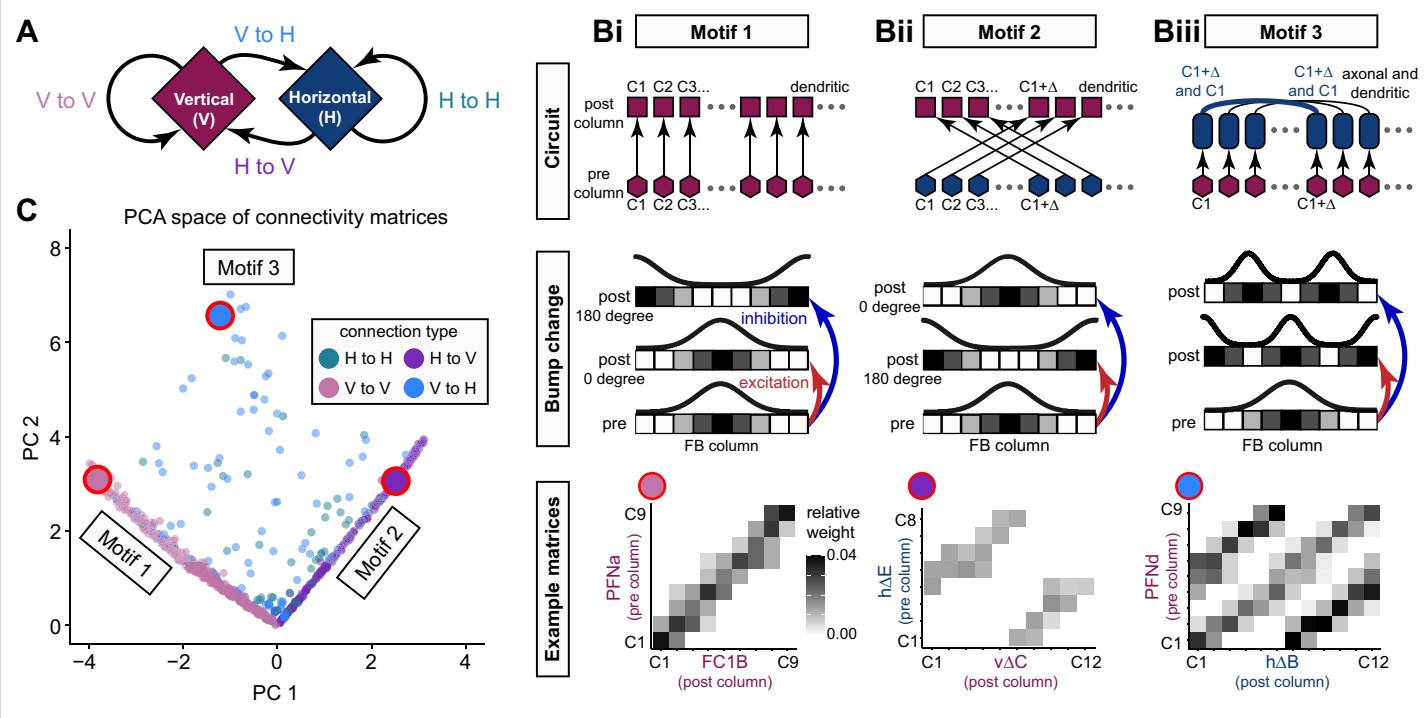

**Figure 37.** The intra-fan-shaped body (intra-FB) columnar network is built from a small number of circuit motifs. (**A**) FB columnar neurons can be divided into vertical and horizontal types. Throughout the figure, vertical types are marked in maroon and horizontal types are marked in dark blue. Note that hΔ neurons are named according to the column containing their dendritic arbor, which impacts the connectivity matrix structure, as shown in the examples in (**B**). Vertical and horizontal neurons give rise to four connection types. vertical-to-vertical (V to V), vertical-to-horizontal (V to H), horizontal-to-horizontal (H to H), and horizontal-to-vertical (H to V). (**B**) Three columnar-to-columnar connectivity motifs generated by three circuit motifs. Top panels show circuit motifs that generate the corresponding column-to-column connectivity matrix shown in the bottom panels. The middle panels show how excitatory or inhibitory connections would impact bump phase. In each circuit diagram, all presynaptic columns are marked with hexagons, while postsynaptic columns can be dendritic (squares), axonal (circles), or contain multiple hΔ neurons with either dendritic or axonal arbors (rounded rectangles). See legend in *Figure 37—figure supplement 1* for details. Circuit motifs are shown with ellipsis (…) to indicate variable column numbers, while connectivity matrices and bump change diagrams are shown with the nine-column pattern typical of most FB columnar neurons. (**Bi**) Motifs that generate a diagonal column-to-column connectivity matrix. Excitatory connections could pass the bump to a second layer while maintaining its phase, while inhibitory connections could shift the bump's position by 180°. (**Bii**) Motifs that generate a shifted column-to-column connectivity matrix. Excitatory connections would shift the bump by 180° while inhibitory connections would maintain its phase (bottom panel). (**Biii**) Motifs that could produce a column-to-column connectivity matrix with two diagonal bands. Excitation and inhibition could produce a double-bump pattern, as shown in the bottom panel. Alternatively, if the axonal compartment receives inhibitory input and the dendritic compartment receives excitatory input, a single bump would be preserved (not shown). (**C**) Scatter plot showing that FB connectivity matrices can be grouped into one of the three motifs. Each circle in the scatter plot marks the location of a single connectivity matrix in principal component space. Briefly, each column-to-column connectivity matrix was coerced into a nine-column scheme, binarized, and transformed into a vector. Principal component analysis (PCA) was performed using a matrix containing all connectivity vectors (n = 903 connectivity matrices), and each vector was projected onto the largest two PCs (PC1 and PC2). Circles are colored according to pre-to-post connection type (see legend). Note that the large majority of connectivity matrices correspond to motifs 1 and 2 (diagonal point clouds), whose orthogonality is preserved in PC space. Points lying off these diagonals largely correspond to motif 3. The three large circles outlined in red correspond to the connectivity matrices in the bottom panels of (**B**).

The online version of this article includes the following figure supplement(s) for figure 37:

**Figure supplement 1.** Detailed description of intra-fan-shaped body (intra-FB) columnar connectivity motifs.

**Figure supplement 2.** Principal component analysis of fan-shaped body (FB) columnar connectivity.

connections. For example, a column-matched excitatory connection (*Figure 37Bi*) would preserve the phase of the bump, but if that connection were inhibitory, it would shift the bump by half the width of the FB (roughly 180° in azimuth). Similarly, a connection across half the FB's width might shift the bump by 180°, if it happens to be excitatory, or keep it in place (0°) if it were inhibitory (*Figure 37ii*). We analyzed the complete set of connectivity matrices (see, e.g., the bottom row of *Figure 37Bi–iii*) between all vertical and horizontal types using principal components analysis (PCA) and found that the three broad intercolumnar motifs that we identified accounted for most of the variance across the

entire set (*Figure 37C* and *Figure 37—figure supplements 1 and 2*, see Materials and methods). Not knowing the neurotransmitters and receptors involved in these motifs, we cannot exactly say how the activity bump from the PB is modified by the FB network. However, the connectivity strongly suggests that azimuthal comparisons and transformations of activity bumps are a key function of much of the FB's recurrent circuitry. We describe the potential significance of these specific motifs for bump-driven navigational computations in 'Discussion.'

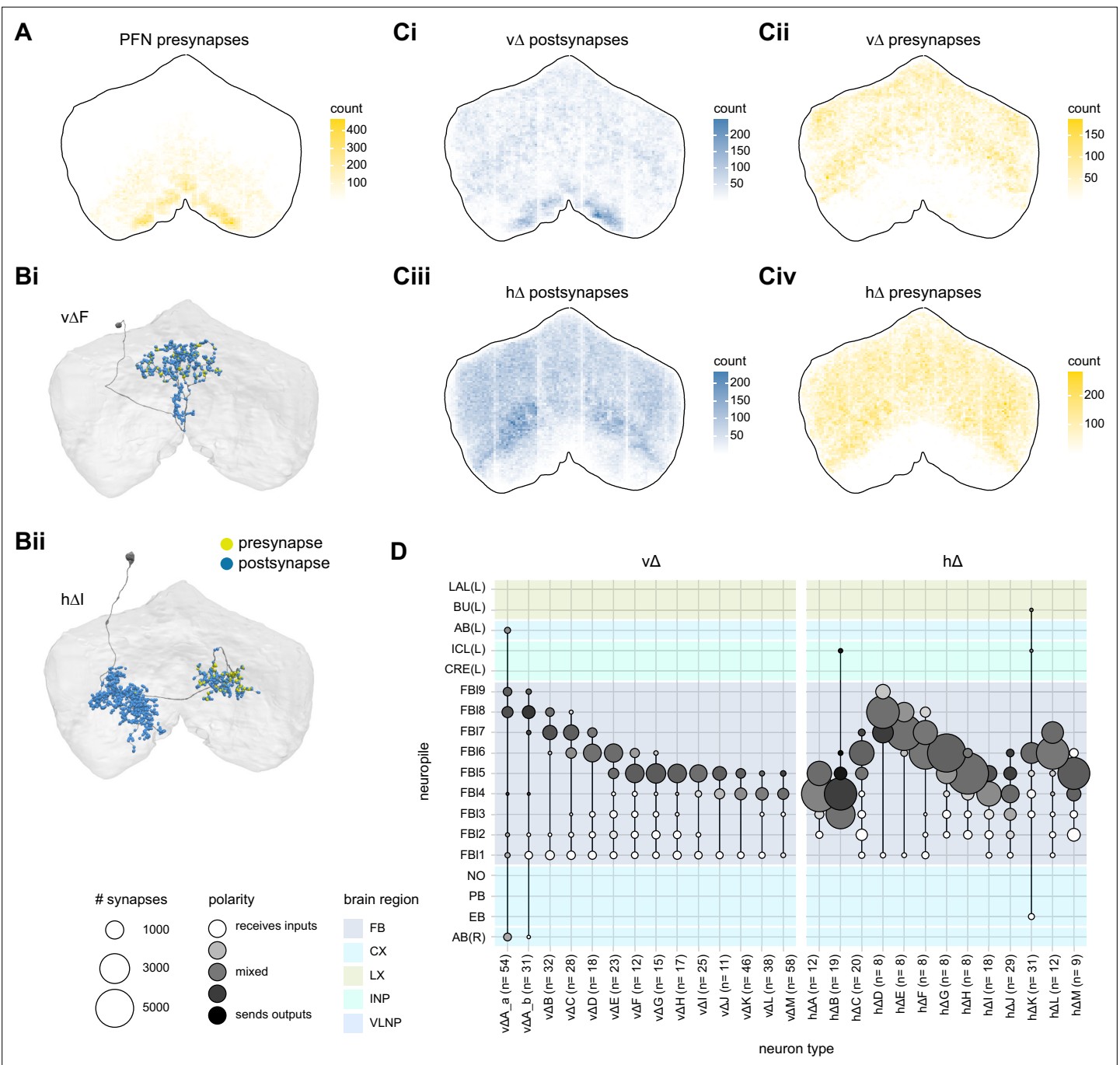

**Figure 38.** Δ neurons in the fan-shaped body (FB) preferentially take input in lower FB layers and output to upper FB layers. (**A**) All presynaptic sites for all the PFN neuron types. (**B**) Morphological renderings of vΔF (**Ai**) and hΔI (**Aii**) neurons. Presynaptic sites are shown in yellow while postsynaptic sites are shown in blue. Both types have postsynaptic sites throughout their arbors, but their presynaptic sites output in their upper layer FB arborizations. (**C**) All postsynaptic (left) and presynaptic (right) sites for all the vΔ (top) or hΔ (bottom) neurons. Postsynaptic sites are visible in the lower FB layers while presynaptic sites are restricted to the upper layers of the FB. (**D**) Region arborization plots for each vΔ and hΔ type.

We now turn to the other axis of information flow in the FB, the vertical axis. As noted in a previous section, the PFN neurons provide the FB with most of its columnar input. These neurons target the ventral layers of the FB (*Figure 38A*) – in fact, most PFN neurons target layers 1 and 2, while PFNa, PFNd, and PFNv types target layers 3 and 4 (*Figure 28—figure supplement 1*). The PFR and PFGs types are the exception to this rule, with presynaptic specializations up to layer 6 (*Figure 28—figure supplement 1*). When we examined the 2D grid of neuron types formed by the FB's interneurons, we found a discernable vertical direction to information flow within the structure. Specifically, most of the intra-FB columnar neurons transfer information from the ventral layers of the FB to the dorsal layers, as can be seen from the distribution of their postsynaptic and presynaptic specializations (*Figure 38B and C*). Both the hΔ and vΔ types receive most of their extra-FB columnar input in their ventral arbors and provide output in their dorsal arbors (as denoted by the dark circles being consistently higher than light circles in *Figure 38D*). The FB output neurons do not appear to selectively get input from the lower FB layers (*Figure 28*), suggesting that the dorsal flow of information in the FB is primarily used in intra-FB computations.

## Columnar phase shifts redux: the PFL neurons

As discussed above (*Figures 30 and 34*), the primary PB-FB columnar inputs to the FB enter the structure with type-specific anatomical phase shifts, which we believe to subserve specific vector computations required for navigation (see 'Discussion'). The previously identified output neurons, three types of PFL neurons (*Lin et al., 2013*), also display type-specific phase shifts (*Figure 39*, *Figure 39—figure supplement 1*). These PFL phase shifts position them well to modulate or control directed actions, an idea developed further in 'Discussion.'

The PFL2 neurons are distinguished from all other PB-FB columnar neurons based on their coverage of PB glomeruli. In contrast with the other columnar neurons, these neurons only receive inputs from the inner five PB glomeruli on each side (*Figure 39Ai–iii*). Thus, these neurons only inherit a single bump in the PB since they effectively sample from a 360° space that is split between the left and right halves of the PB (by contrast, most other PB types sample from 360° space in both the left and right PB). The PFL2 neurons are also distinguished by their four-column phase shift, giving them an ~180° phase shift in their FB innervation relative to their input PB glomerulus or glomeruli (since individual PFL2s sometimes arborize in two neighboring glomeruli) (*Figure 39Aii–iv*, *Video 11*). Notably, these neurons send projections to both sides of the LAL, a pattern of connectivity whose potential navigational function for forward locomotion we explore in a later section on output pathways of the CX, and revisit in 'Discussion.'

By contrast, almost all PFL1 neurons have a single column (~45°) ipsilateral phase shift and project to the contralateral LAL (*Figure 39Bi–iv*). The exceptions to this rule are, first, the innermost PFL1 neurons. These neurons, which originate in PB glomeruli L1 and R1, send their outputs to the LAL on the same side. Second, the PFL1 neurons in L7 and R7 are not phase shifted as expected, but project to C1 instead of C2 and C9 instead of C8 in the FB, respectively.

Finally, the PFL3 neurons display a two-column (~90°) ipsilateral phase shift and also project to the contralateral LAL (*Figure 39Ci–iv*, *Video 12*). Exceptions to this rule are described in the figure legend. Just as the 180° phase shift of the PFL2 neurons and their projection to both sides of the LAL suggest a role in directing forward locomotion, this 90° phase shift suggests a potential role for PFL3 neurons in directing turns towards an FB-specified goal, an idea that we develop in the section on CX output pathways and in 'Discussion.'

Overall, the PFL neurons, along with the other FB columnar outputs, appear to constitute dedicated circuits for a variety of bump-dependent navigational behaviors. How does the fly set directional goals based on internal state and context and then select the behavioral programs to achieve those goals? The large set of tangential inputs into the FB are obvious candidates to play a leading role in this process, which we turn to next.

## Tangential inputs to the FB network

In earlier sections, we described the ring and ExR neurons that bring tangential input to the EB. Most ring neurons bring information about directional sensory cues to the EB-PB compass network, tethering the fly's internal compass to its surroundings, while several of the ExR neurons likely provide

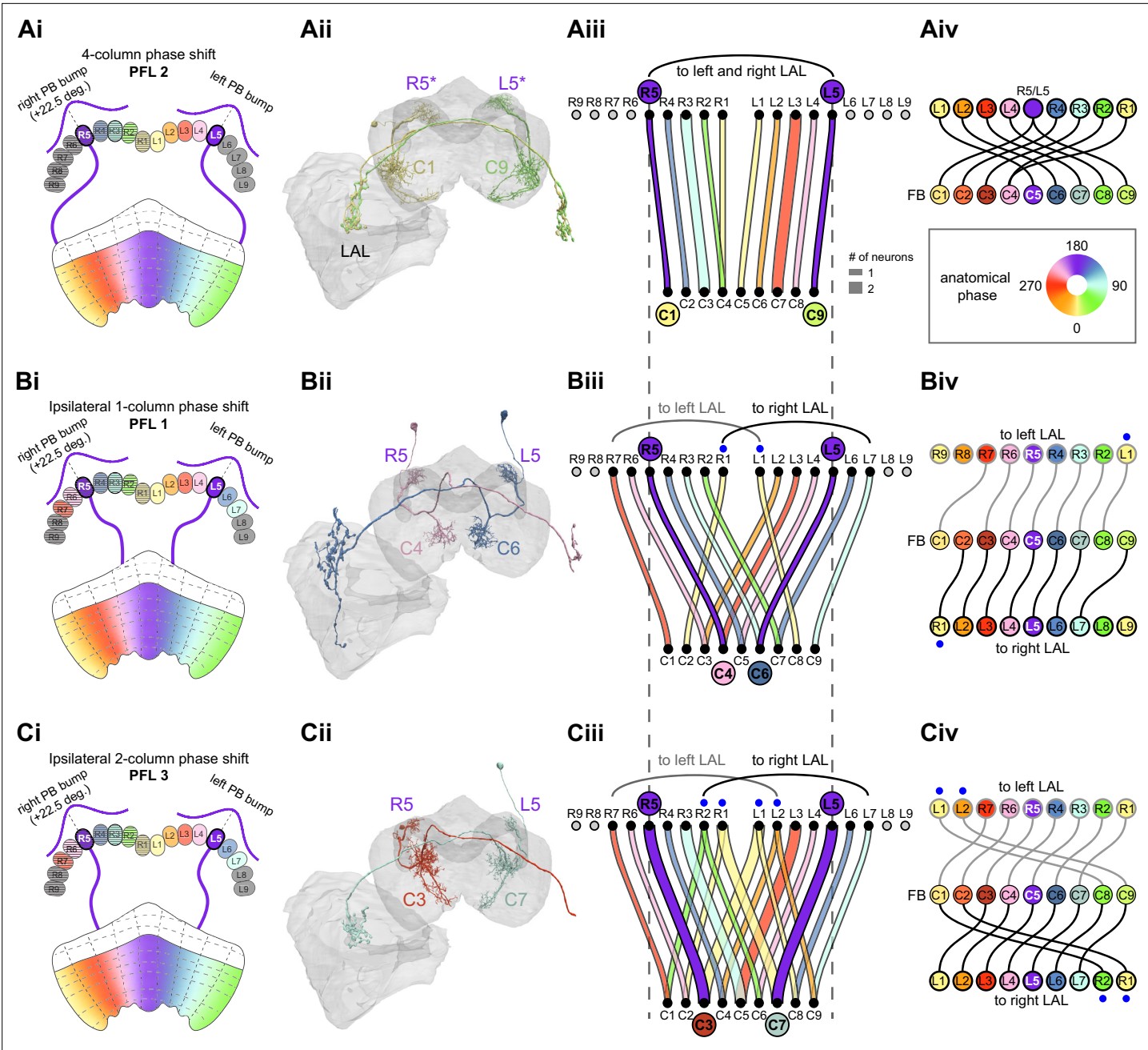

**Figure 39.** PFL neurons, a major fan-shaped body (FB) output, have type-specific phase shifts in protocerebral bridge-to-fan-shaped body (PB-to-FB) projections. (**A**) PFL2 neurons have a four-column (~180°) PB-FB phase shift and bilateral LAL projections. (**Ai**) Schematic of a PB-to-FB projection pattern with a four-column phase shift, as shown for the R5 and L5 glomeruli. PB glomeruli and FB columns are colored according to anatomical phase, which indicates matching bump locations. (**Aii**) Morphological renderings of PFL2 neurons innervating R5 and L5. Neurons are colored according to their FB column. Notice that R5 and L5 neurons end up at C1 and C9, respectively. R5/L5 have been given asterisks because individual PFL neurons can innervate multiple PB glomeruli (in this case, R4/R5 and L4/L5). (**Aiii**) Graph showing the projection pattern from PB glomeruli to FB columns for all neurons in the PFL2 population. R5 and L5 projections have been highlighted as in (**Ai**), and edges are colored according to PB glomerulus. Unlike all other PB-FB-* neurons, the PFL2 population should only inherit one bump in the PB since the neurons sample from an ~360° region of PB space, split between left and right halves (R5–R1 and L1–L5). (**Aiv**) Functional graph showing the mapping between phases in the PB (top row) and phases in the FB (bottom row). Circles are colored by anatomical phase (legend). With one exception (R1), every PB glomerulus connects to a FB column with an ~180° phase shift. (**B**) PFL1 neurons have a one-column (~45°) ipsilateral phase shift and project to the contralateral LAL. (**Bi**) Same as in (**Ai**), but schematizing the one-column ipsilateral phase shift of PFL1 neurons. (**Bii**) Same as in (**Aii**), but for two PFL1 neurons. Notice that the R5 neuron projects to C4, and the L5 neuron projects to C6. (**Biii**) Similar to (**Aiii**), but for PFL1. Black and gray arched lines indicate groups of glomeruli that project to the right or left LAL, respectively. R1 and L1 are marked with blue dots because they project to the ipsilateral LAL, unlike the other neurons in the population. (**Biv**) Similar

*Figure 39 continued on next page*

*Figure 39 continued*

to (**Aiv**), but for PFL1. Here, instead of dividing glomeruli by their left vs. right PB innervation (as in **Biii**), glomeruli are grouped by whether they project to the left LAL (top row, gray outlined circles) or the right LAL (bottom row, black outlined circles), and sorted by anatomical phase. With the exception of R7 and L7, each glomerulus projects to FB columns with a one-column (~45°) ipsilateral phase shift. (**C**) PFL3 neurons have a two-column (~90°) ipsilateral phase shift and project to the contralateral LAL. (**Ci**) Same as in (**Bi**), but schematizing the two-column ipsilateral phase shift of PFL3 neurons. (**Cii**) Same as in (**Bii**), but for two PFL3 neurons. Notice that the R5 neuron projects to C3, and the L5 neuron projects to C7. (**Ciii**) Same as in (**Biii**), but for PFL3. R1/R2 and L1/L2 are marked with blue dots because these glomeruli contain neurons that either project to the contralateral LAL (like most of the population) or project to the ipsilateral LAL (unlike most of the population). (**Civ**) Same as in (**Biv**), but for PFL3. Notice that every neuron that projects to the left LAL (top row) and right LAL (bottom row) samples FB columns with a two-column (~90°) phase shift.

The online version of this article includes the following figure supplement(s) for figure 39:

**Figure supplement 1.** Columnar structure of PFL types.

modulatory input. In contrast to the ring neurons, the FB's many tangential inputs are less well understood.

The FB has nine different layers of varying widths. Layers 4 and 5 are the widest, while layers 1 and 9 are the smallest (*Figure 40A*, *Video 13*). All FB tangential (FBt) neurons have presynaptic specializations in characteristic FB layers and most bring input from accessory structures of the CX, such as the CRE, SMP/SIP/SLP, and LAL, as well as from the NO (*Figure 40A*, *Figure 40—figure supplement 1A*). The FB4O neurons, for example, receive input in the CRE and SMP/SIP/SLP and send their outputs solely to layer 4 in the FB (*Figure 40Aii*).

Like the ExR neuron types, there is considerable variability in neuronal morphology across the different FB tangential neuron types (*Figure 40B*). While most types target one FB layer, some, such as FB1I, target multiple layers (*Figure 40Bi*). Separately, while most types have processes external to the CX, some, such as FB4Z, are intrinsic FB interneurons, with no external processes (*Figure 40Bii*). Notably, not all FB tangential neurons uniformly fill their layer. FB4Z, for example, sends out selective processes to target specific partners within layer 4 (*Figure 40—figure supplement 1B*). There is also considerable variability in where the FB tangential neurons arborize outside of the CX. For example, the FB tangential neurons in layers 3–5 preferentially send processes to the LAL and CRE, while the upper layers tend to target the SMP/SIP/SLP (*Figure 40C*, *Figure 40—figure supplement 2*). Finally, in stark contrast with EB ring neurons, most FB tangential neurons consist of only one or two neurons per side (*Figure 40D*). We found no clear evidence of side preference for neurons originating on the right or left within a given type. Neurons from both sides target PB-FB columnar neurons from both sides of the PB. The small

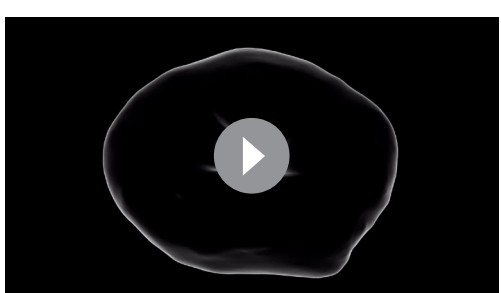

**Video 11.** PFL2 phase shift. Similar to *Videos 6–7*, this video highlights the 180° phase shift of the PFL2 population. The video begins by showing morphological renderings of an EPG neuron that contacts a PFL2 neuron in the PB. Notice that the PFL2 neuron, shown in blue, has processes in both R4 and R5. Next, the video shows the synaptic connection from the PFL2 neuron onto a DN in the LAL, and then briefly highlights an EPG to PFL2 connection in the left PB. At this point, notice that the PFL2 neurons innervating R5 and L5 (blue and gold), which share similar directional tuning, project to the lateral border of the FB, ~180° away from their PB regions. Lastly, the video shows individual morphological renderings of all PFL2 neuron before showing the population as whole. Related to Figure 39A.

https://elifesciences.org/articles/66039/figures#video11

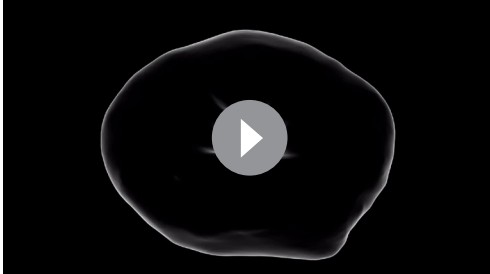

**Video 12.** PFL3 phase shift. Similar to *Video 11*, but showing the +/−90° phase shift of the PFL3 population. In the last portion of the video, pairs of PFL3 neurons from the left and right PB are shown. Each pair inherits a similar directional tuning in the PB but projects to distant regions of the PB, generating 90° phase shift. Related to Figure 39C.

https://elifesciences.org/articles/66039/figures#video12

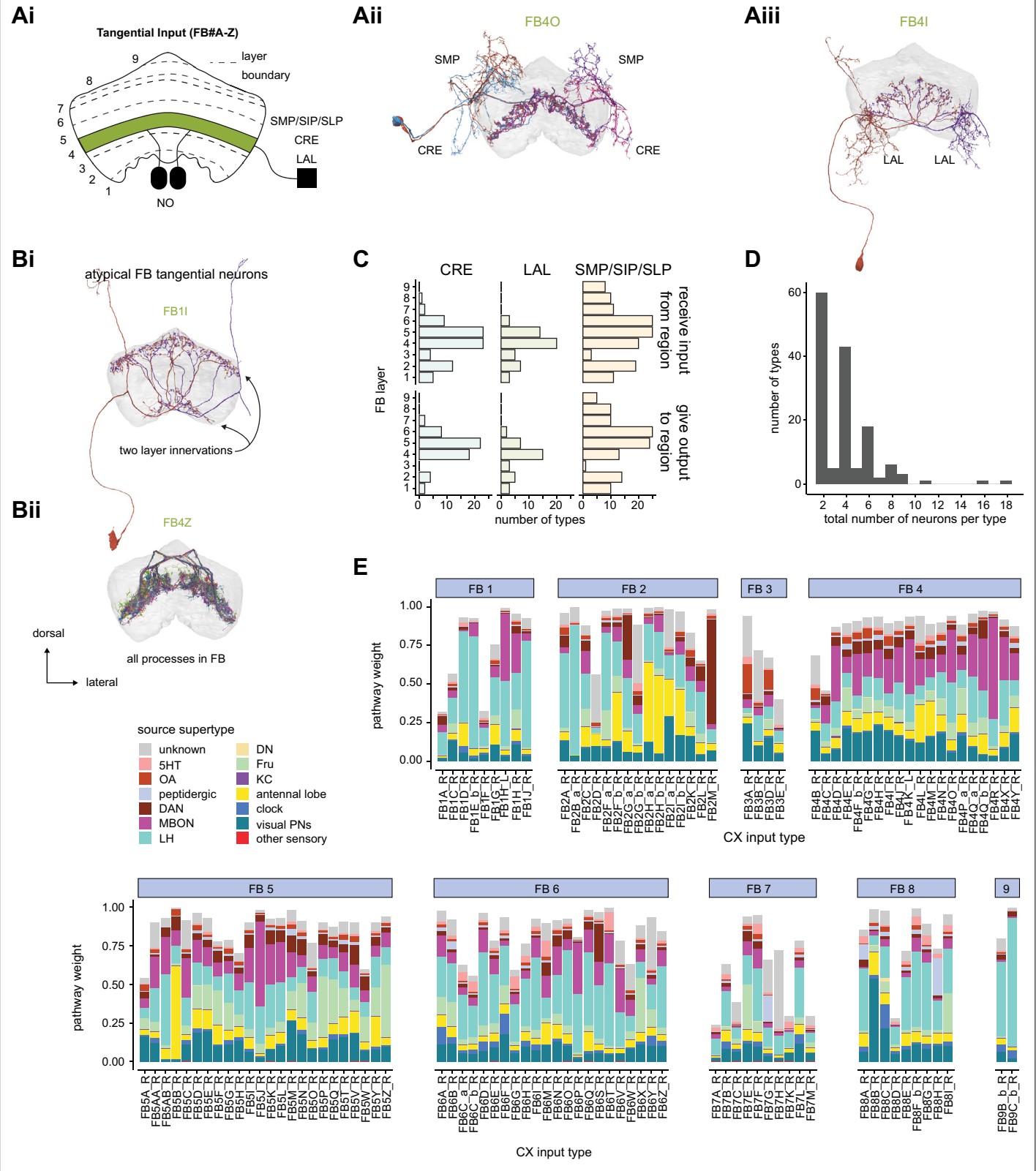

**Figure 40.** Fan-shaped body (FB) tangential overview. (**A**) Overview of FB tangential neurons. (**Ai**) FB tangential neurons output in single or multiple layers of the FB (e.g., in layer 4, shown in green) and may have mixed arbors in the noduli (NO), superior protocerebrum SMP/SIP/SLP, lateral accessory lobe (LAL), crepine (CRE), or other brain regions outside of the central complex. (**Aii**) A morphological rendering of FB4O neurons, which receive input in the SMP and CRE and output to layer 4. (**Aiii**) A morphological rendering of FB4I neurons, which receive input in the LAL and output to layer 4. (**B**) While

*Figure 40 continued on next page*

*Figure 40 continued*

most FB tangential neurons arborize in one FB layer and receive input external to the central complex (CX), there are exceptions. Some FB tangential neurons, for example, arborize in multiple FB layers. (**Bi**) shows a morphological rendering of one such type, the FB1I neurons. In contrast, some FB tangential neurons arborize exclusively within the FB. (**Bii**) shows a morphological rendering of one such type, the FB4Z neurons. (**C**) The number of FB tangential types that receive input from (top) or give output to (bottom) the CRE, SMP/SIP/SLP, or LAL. The FB layer refers to the layer where a given type has the most expansive processes. For this analysis, only the right neurons of the type are considered, and each right neuron of that type must have, on average, at least three synapses in the given region. (**D**) The number of neurons per FB tangential type. With a few rare exceptions, both the right and left FB contribute an equal number of neurons to each type. (**E**) Input pathway classifications for the FB tangential input neurons. Types are counted as inputs if they have at least 20 synapses of a given polarity outside of the CX and are the postsynaptic partner in at least one significant type to type connection outside of the CX. See *Appendix 1—figure 3* for an explanation of pathway weight.

The online version of this article includes the following figure supplement(s) for figure 40:

**Figure supplement 1.** Fan-shaped body (FB) arborizations by region.

**Figure supplement 2.** Fan-shaped body (FB) tangential synaptic sites that are outside of the central complex (CX).

number of neurons per type suggests that they convey specific, uniform information to their targeted layer(s).

The number of different FB tangential neurons that provide output to any given FB layer and the diversity of brain regions from which they draw their inputs suggests that every computation that the FB participates in is likely modulated by context. Precisely what these modulatory influences are is largely unknown. They could be purely sensory, for example, or they could convey the state of the animal. However, some hints may come from the upstream partners of the different FB tangential neurons (*Figure 40E*). For example, MB output neurons preferentially target layer 4 (which will be discussed in the next section). The LH, which receives direct olfactory, thermosensory, and hygrosensory input and multisensory inputs from the visual, mechanosensory, and gustatory systems as well (*Dolan et al., 2019*; *Schlegel et al., 2020*), is part of input pathways that project to most FB layers. vPNs, AL neurons, and the courtship-associated Fru neurons are also upstream of many FB tangential types. This diverse array of upstream partners could convey a range of contextual cues to layers throughout the FB.

Although little is known about most of the FB's tangential neurons, there are some exceptions. Most notably, dorsal FB tangential types are known to be involved in sleep-wake control (reviewed in *Artiushin and Sehgal, 2017*; *Donlea, 2017*; *Dubowy and Sehgal, 2017*; *Helfrich-Forster, 2018*), a topic we return to below (Figures 48–53). In addition, a recent study focused on the LH identified an FB tangential neuron type they called PV5k1, which, when optogenetically stimulated under closed-loop visual conditions, leads to a reduction in the fly's wingbeat frequency (*Dolan et al., 2019*). The neurons targeted by the GAL4 lines used in that study likely correspond to FB2H_a, FB2H_b, and/or FB2I_b, neuron types that target some hΔ neuron types and also the PFL2 and PFL3 neuron types, consistent with the direct influence on the fly's behavioral patterns (see later sections on PFL neurons). Another recent study found that FB tangential neurons that target layer 6 encode food choice (*Sareen et al., 2020*). Inhibiting these neurons made hungry flies more likely to eat bittersweet food with 500 mM sucrose instead of purely sweet food with 50 mM sucrose. The activity of these neurons was also shown to encode the food choice.

The ExFl1 neurons (*Liu et al., 2006*; *Weir et al., 2014*; *Young and Armstrong, 2010b*),

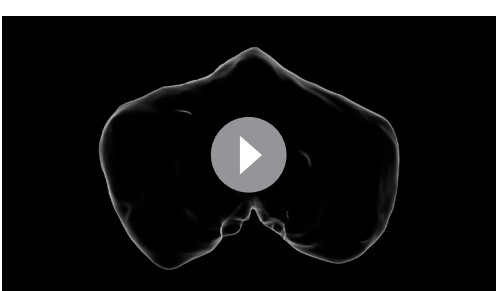

**Video 13.** Morphological renderings of the fan-shaped body (FB) tangential neurons. The FB tangential neurons are shown layer by layer. First, all of the FB tangential neurons that predominantly arborize in layer 1 are shown, then all of the FB tangential neurons that predominantly arborize in layer 2 are shown, and so on, all the way through layer 9. Each layer is assigned a unique color on a continuous scale that goes from yellow (layer 1) to green (layer 5) to blue (layer 9). As the neurons in each layer are displayed, they are rotated around the z-axis to allow all the processes in the 3D volume to be seen. Note that the FB tangential neurons in each layer tend to send their processes to a distinct brain region outside of the CX. After all the individual layers are shown, they are combined so that all the FB tangential neurons can be seen together.
https://elifesciences.org/articles/66039/figures#video13

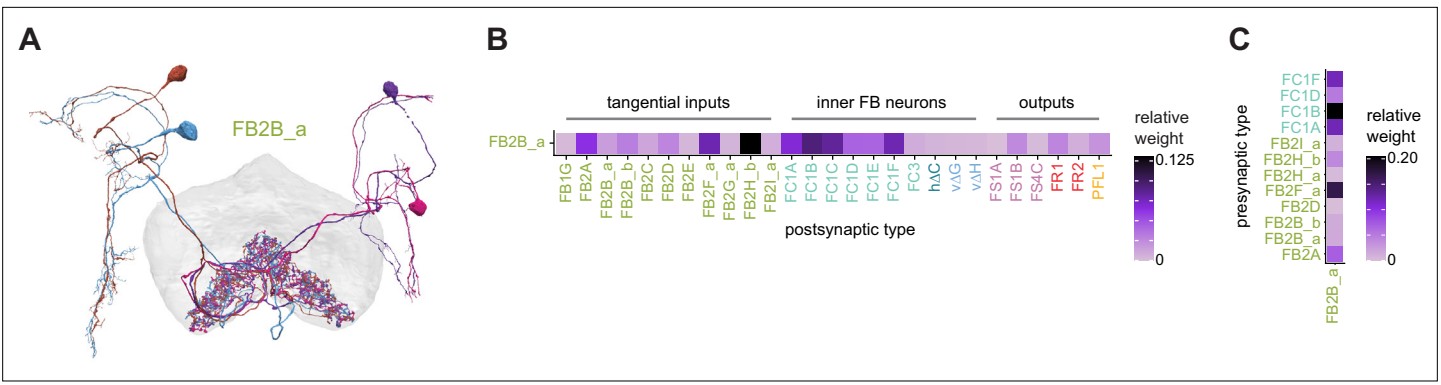

**Figure 41.** FB2B_a connectivity. (**A**) FB2B_a neurons, arborize in the SIP, the crepine (CRE), and fan-shaped body (FB) layer 2. (**B**) Postsynaptic FB2B_a partners in the FB. Partners include other FB tangential cells, FB neurons that are both pre- and postsynaptic in the FB, and FB outputs. (**C**) Presynaptic FB2B_a partners in the FB.

which are likely the FB2B_a and/or FB2B_b neurons (*Figure 41A*), have also been characterized. These neurons respond to progressive optic flow and are strongly modulated by whether or not the fly is flying (*Weir et al., 2014*), providing a potential indication the motor state of the animal. Information from the FB2B_a neurons is fed to other FB tangential neurons, FB interneuron types (including some hΔ and vΔ neurons), intermediate types (such as the FC neurons), and output neurons (such as the FR neurons, *Figure 41B*). The FB2B_a neurons also get input in the FB from the other layer 2 FB tangential neurons and from the columnar FC1 neurons (*Figure 41C*). Such a connectivity profile is typical of most FB tangential neuron classes, as shown in *Figures 42–44*.

Some FB tangential neuron types in FB layers 2 and 8 have been proposed to have a major role in visual learning (*Liu et al., 2006*). In such a situation, the tangential neurons could be ideally placed to provide information about positive or negative reinforcers, a function typically carried out in the fly brain by DANs. Indeed, several FB tangential neurons are known to be DANs (discussed further below), as indicated by the gray bars in the connectivity matrix in *Figure 42*. DANs are not the only neuromodulatory neurons amongst the FB tangential neurons. It is likely that several other FB tangential neurons may be modulatory and peptidergic, but we could only confirm one additional such neuron, the octopaminergic OA-VPM3 (*Figure 42*, first row in the connectivity matrix).

Overall, the FB tangential neurons primarily target the intra-FB network and rarely target the PFN neurons, the major source of columnar input to the FB (*Figure 42*). This intra-FB targeting by potential contextual and neuromodulatory signals further emphasizes the importance of the FB's interneurons and recurrent network in shaping circuit dynamics in the structure.

As with the ring neurons of the EB, the FB's tangential neurons synapse onto each other near their presynaptic specializations in the FB (*Figure 43*). However, in contrast to the ring neurons, there are far fewer neurons in each FB tangential type, and their subnetworks seem less tightly clustered, making it more difficult to detect hierarchies amongst the potential contextual inputs (squares marked in *Figure 43*). Like the ring neurons, all tangential neurons of the FB, including the DANs, also receive considerable intra-FB input near their presynaptic sites in the FB, (*Figure 44A*), much of it from their targets (*Figure 44B*).

We also examined the connectivity matrix within and across different FB tangential neuron types at the level of individual neurons (*Figure 45A*). Similar to the ring neurons, a subset of FB tangential types have reciprocal connections between the individual neurons within that type. For example, FB2I neurons interact strongly with each other, and these interactions are spread uniformly across columns within their layer of innervation (*Figure 45B*). If these interactions are inhibitory, as is the case with ring neurons, such connectivity may facilitate a competition between similar contextual inputs that are vying to influence navigational computations based on the columnar position of the bump. We revisit this possibility in 'Discussion.'

## Pathways from the mushroom body to the central complex

FB tangential neurons receive much of their input in central brain regions like the SMP/SIP/SLP (*Figures 28 and 40F*), which, in turn, receive inputs from many other parts of the brain, such as the

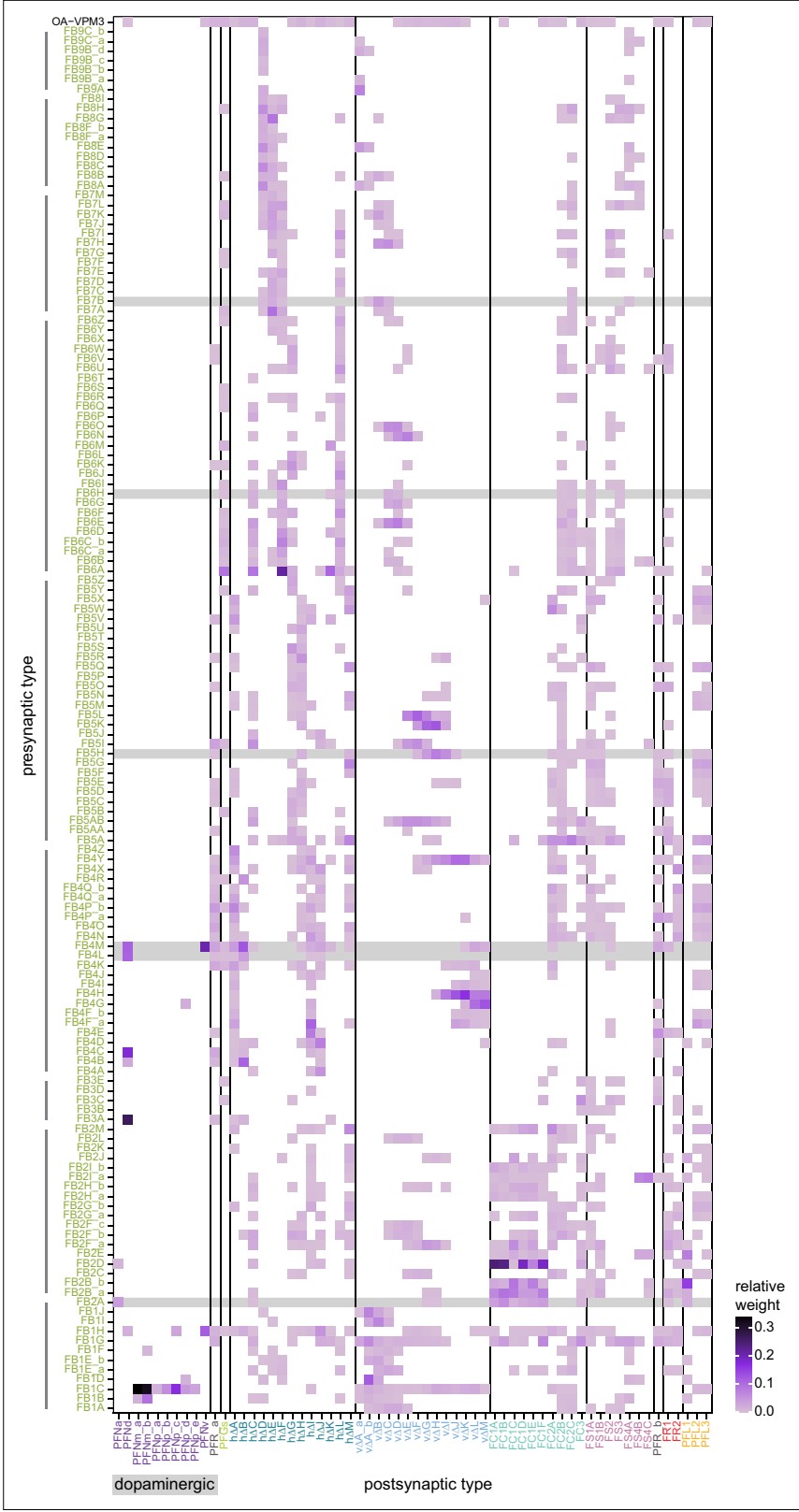

**Figure 42.** Fan-shaped body (FB) tangential postsynaptic partners in the FB. Type-to-type connectivity matrix for the FB tangential presynaptic partners in the FB. The connectivity of known dopaminergic neurons is highlighted in gray. Vertical lines adjacent to the y-axis mark groups of FB tangential neurons that primarily arborize in the same layer.

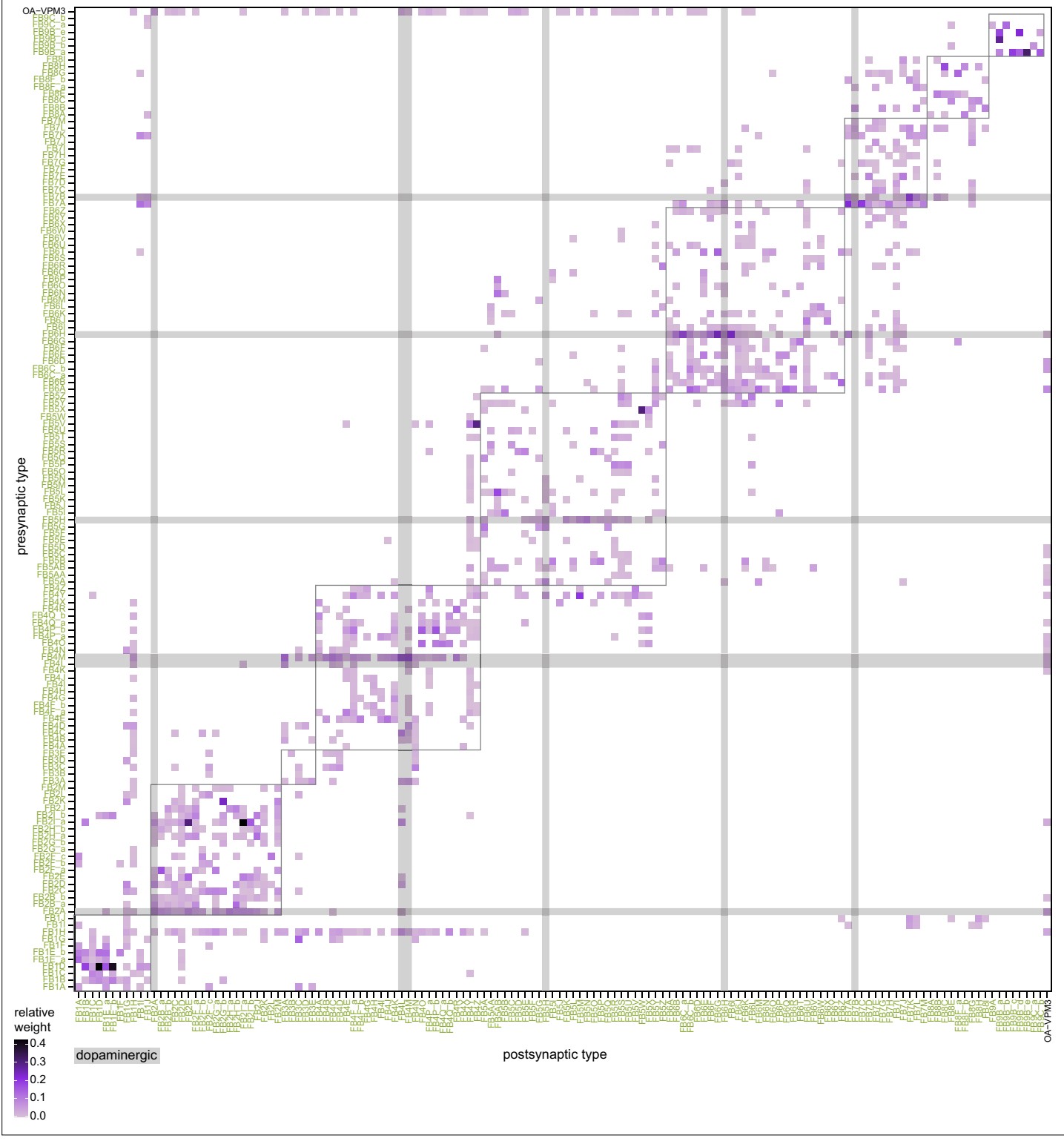

**Figure 43.** Fan-shaped body (FB) tangential to FB tangential connections in the FB. Type-to-type connectivity matrix between FB tangential types in the FB. The connectivity of known dopaminergic neurons is highlighted in gray. Boxes surround connections between FB tangential neurons that have their primary arborizations in the same FB layer.

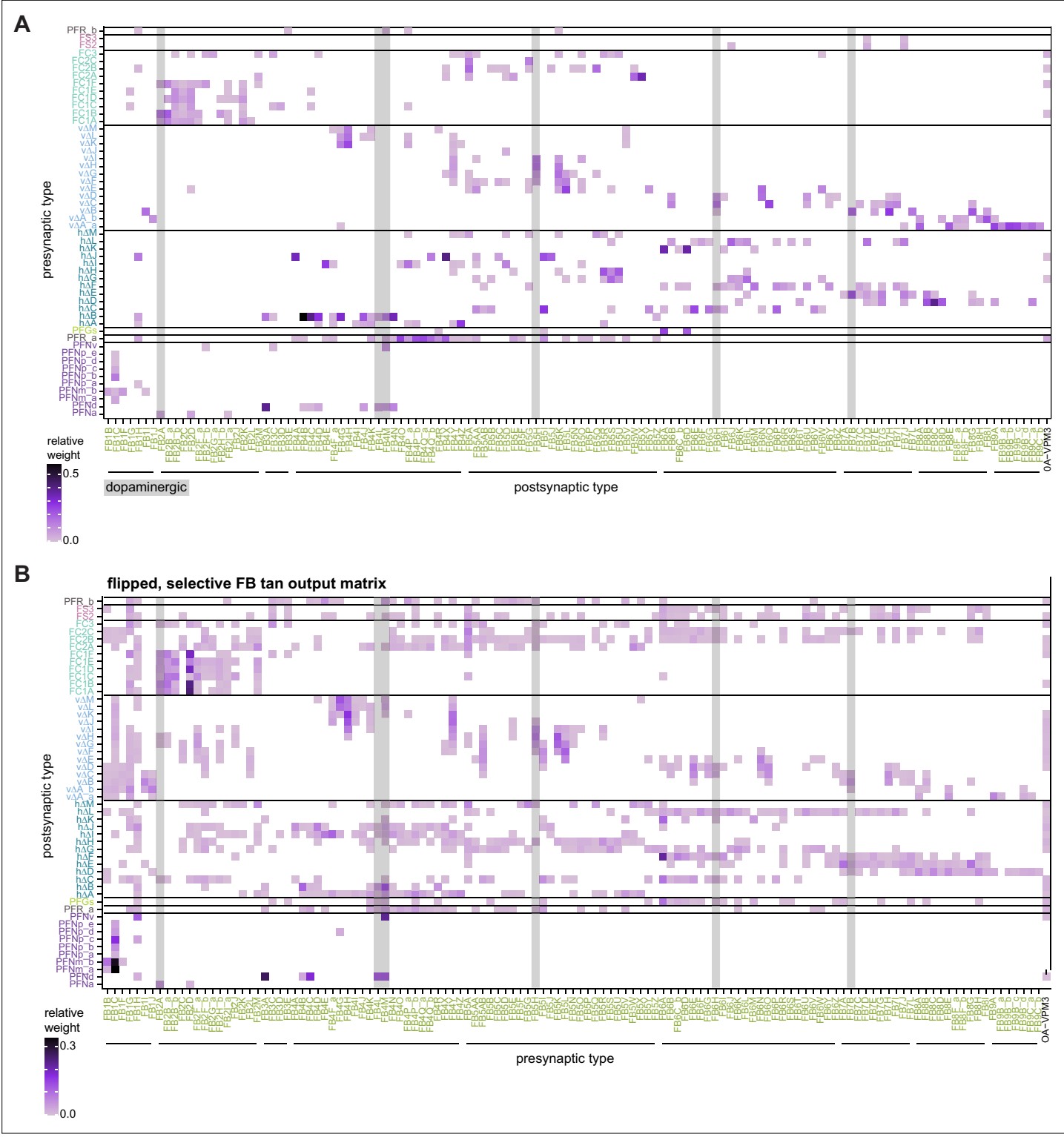

**Figure 44.** Fan-shaped body (FB) tangential postsynaptic partners in the FB. (**A**) Type-to-type connectivity matrix for the FB tangential postsynaptic partners in the FB. The connectivity of known dopaminergic neurons is highlighted in gray. Horizontal lines adjacent to the x-axis mark groups of FB tangential neurons that primarily arborize in the same layer. (**B**) Type-to-type connectivity matrix for the FB tangential presynaptic partners in the FB, as seen in *Figure 42A*, where the axes are now flipped and only the partners from (**A**) are plotted.

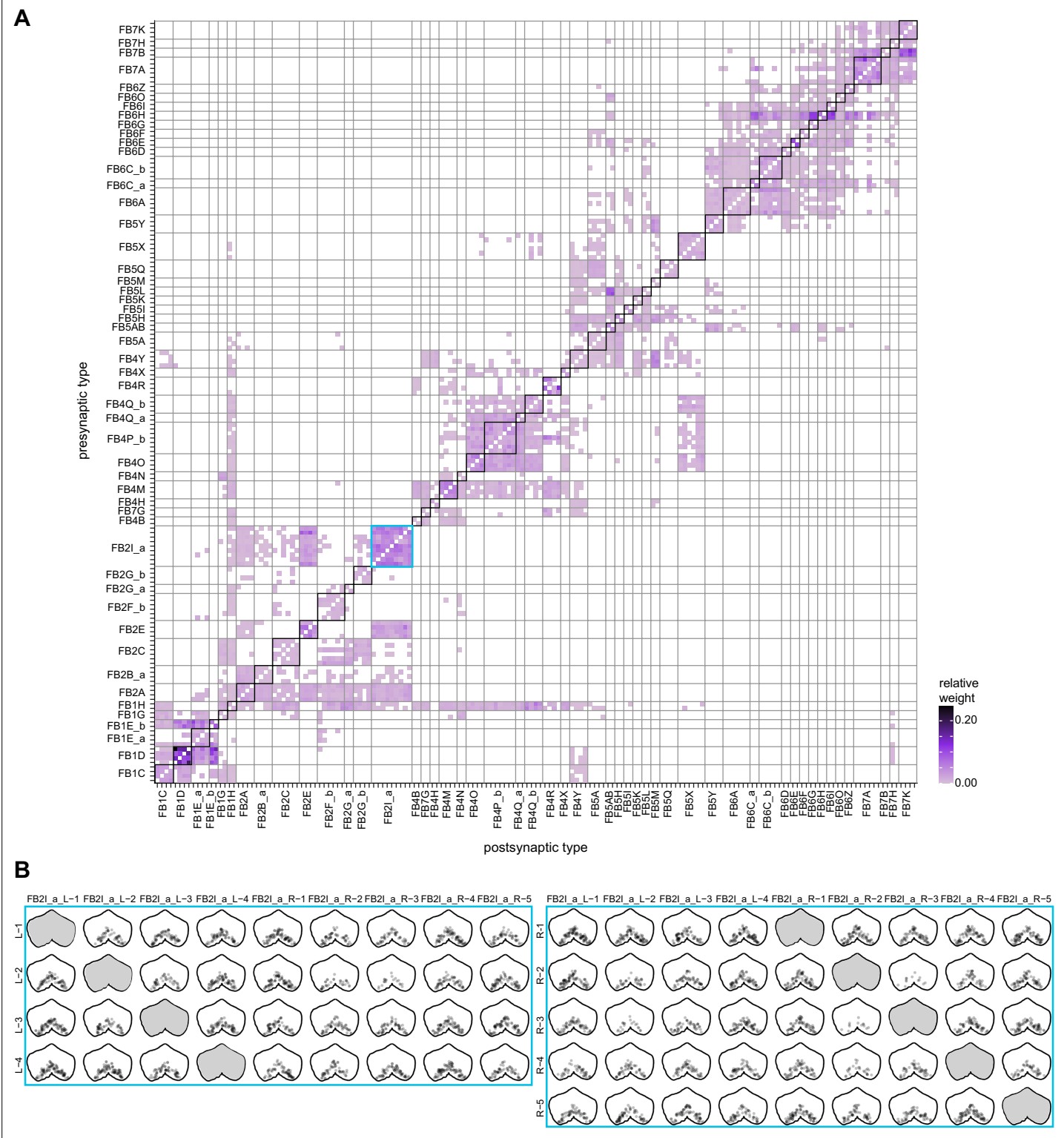

**Figure 45.** Several fan-shaped body (FB) tangential neuron types show all-to-all connections that resemble connectivity patterns within and between ER neuron types. (**A**) Neuron-to-neuron connectivity matrix for the FB tangential types. (**B**) Locations of synapses between individual FB2I neurons (box in blue in **A**). FB2I neurons synapse onto other neurons of the same type across all columns of the layers that they innervate.

MB, consistent with the idea that FB tangential neurons enable the modulation of CX-controlled behavior, perhaps according to context and internal state. Our knowledge of the precise nature of these signals has been limited by a lack of extensive characterization of inputs to these less structured brain regions. A notable exception is the set of inputs that arrive from the MB. The MB is thought to be a center for associative learning (*Cognigni et al., 2018*; *Modi et al., 2020*). It receives inputs from nearly all sensory modalities, is innervated by multiple types of DANs (*Li et al., 2020*), and provides an architecture to flexibly convert sensory information from thousands of Kenyon cells (KCs) into experience-dependent (and DAN-mediated) valence signals carried by the MB's output neurons, the MBONs. MBONs broadcast these valence signals to other areas of the brain, but how these signals drive the fly's behavioral responses is not fully understood. In this section, we analyze pathways connecting the MB and the CX. Communication between these structures may play an important role in sleep (*Dubowy and Sehgal, 2017*; *Sitaraman et al., 2015*), memory consolidation (*Berry et al., 2015*; *Dag et al., 2019*; *Donlea, 2019*), context-dependent feeding decisions (*Sareen et al., 2020*; *Scaplen et al., 2021*; *Scaplen et al., 2020*), and perhaps also the conversion of the MB's valence signals into goal-directed actions during navigation (*Collett and Collett, 2018*; *Sun et al., 2020*).

As described in a companion paper on the MB (*Li et al., 2020*), about half of the MBONs directly synapse onto FB tangential neurons that target the middle layers (4–6) of the FB (*Figure 46A*). Some MBONs contact only a single type of FB tangential neuron. For example, MBON09 connects exclusively to FB4R (*Figure 46Bi*). Many other MBONs contact multiple downstream targets in the FB. MBON04, for example, targets FB tangential neurons in both layers 1 and 6 (*Figure 46Bii*).

MBONs of different neurotransmitter types, both excitatory and inhibitory, often converge onto the same FB tangential neurons (lines of different colors in *Figure 46A*; see *Li et al., 2020* for how neurotransmitters for each of the MBONs were identified). For example, the glutamatergic (and likely inhibitory) MBON04 and the cholinergic (excitatory) MBON12 both contact FB4A (*Figure 46A and Biii*), and two MBONs that express different (likely) inhibitory neurotransmitters, MBON05 (glutamate) and MBON09 (GABA), converge onto FB4R, along with a third excitatory (cholinergic) MBON21 (*Figure 46A*).

FB neurons are not the only CX neurons targeted by direct projections from the MB. The glutamatergic MBON30 neuron type targets the LCNOp neurons (*Figure 46Biv*) that themselves feed the PFNp columnar neurons. The strength of these direct connections varies widely. Some, such as connections from MBON09 and MBON21 to FB4R, represent a significant fraction of their downstream target's input (longer bars in *Figure 46—figure supplement 1B*), while others are much weaker in the influence that they exert on even their most preferred downstream partners in the CX (short bars in *Figure 46—figure supplement 1B*). Note that, as explained in an early section of the article, our summary plots and connectivity matrices exclude connected pairs whose connection weights fall below our threshold (for significance criteria, see Materials and methods). Thus, *Figure 46A*, for example, shows slightly fewer connections than the equivalent figure in the companion paper on the MB connectome (*Li et al., 2020*). This dependence of connectivity on the threshold chosen is shown in *Figure 46—figure supplement 1A and C*.

In addition to these direct connections, MBONs also connect to both dorsal and ventral layers of the FB through intermediate neurons in regions like the SMP, SIP, and CRE (*Figure 47*). Several of these one-hop pathways feature a mix of convergence and divergence from different MBONs onto neurons in intermediate layers (complete set of paths in *Figure 47—figure supplement 1*). Note, once again, that not all MB-to-CX pathways involve FB tangential neurons. For example, MBONs 26 and 31 reach the CX, and even the FB, through the LAL and the LCNO and LNO neurons (*Figure 47E*).

In some cases, we could use what is known about the KC inputs to different MBON types (see Figure 15 in *Li et al., 2020*) to determine the type of sensory information in specific downstream FB tangential neurons (*Figure 47C and D*). For example, several MBONs conveying valence signals associated with visual information send divergent streams of information to an intermediate layer of neurons in diverse brain regions. These intermediate neurons, in turn, feed FB tangential neurons of different classes that go to both ventral and dorsal layers of the FB (*Figure 47C*). MBONs that receive thermosensory and hygrosensory information project to a largely different set of intermediate neurons, which project to their own, largely distinct set of FB tangential neurons (and LNO neurons) (*Figure 47D*).

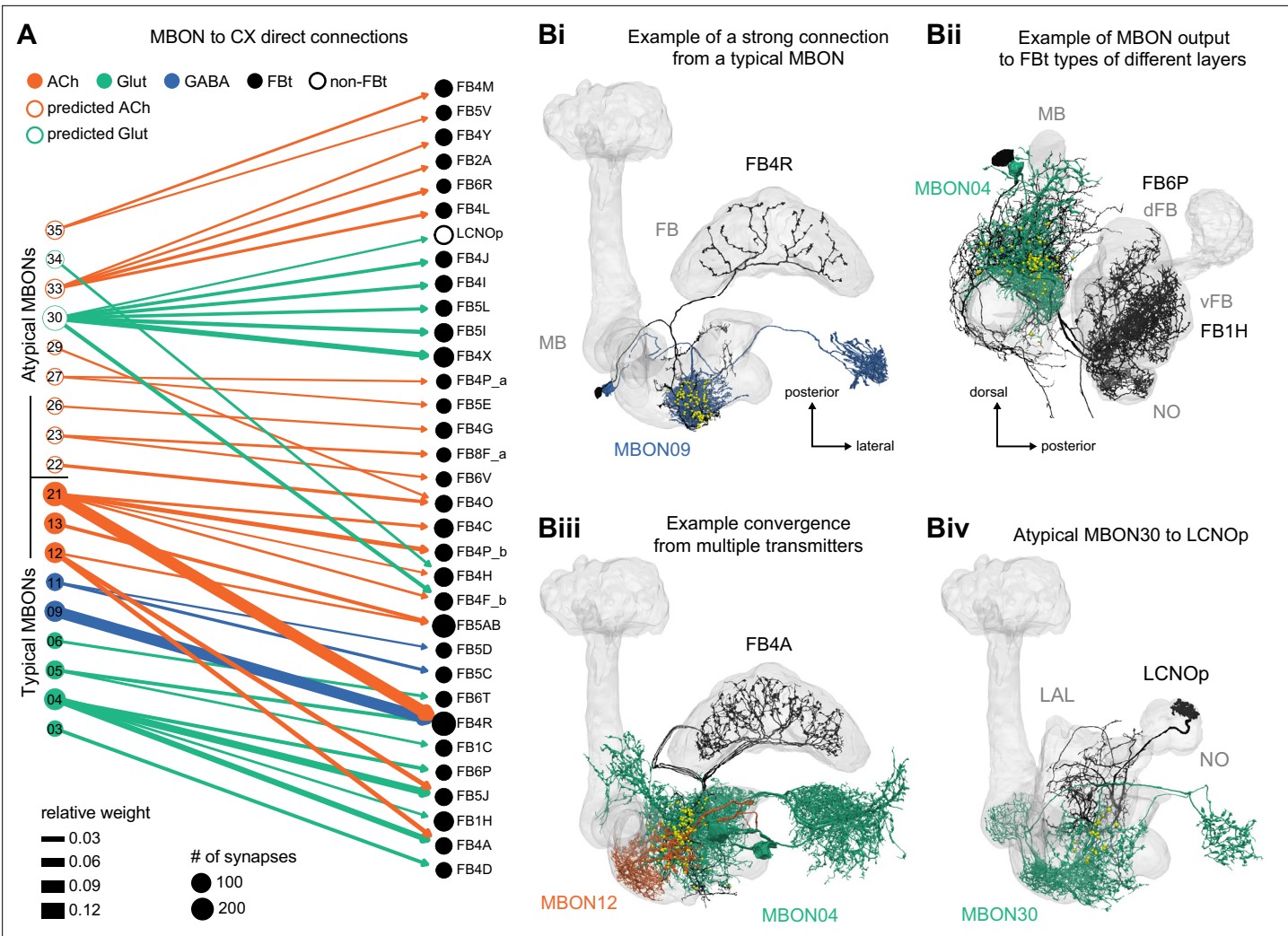

**Figure 46.** Direct connections from mushroom body output neurons (MBONs) to central complex (CX) neurons. (**A**) Network graph showing direct connections from MBONs to CX neuron types, all of which are fan-shaped body (FB) tangential cell types, with the exception of one connection involving LCNOp, a LAL-NO type. MBON nodes are colored according to their neurotransmitter identity as determined by RNA-seq (filled circles) (*Aso et al., 2019*) or as predicted by an artificial neural network trained on EM images of presynaptic boutons from MB neurons with known transmitter types (open circles) (*Eckstein et al., 2020*; *Li et al., 2020*). Both typical (01–23) and atypical (24–35) MBONs are included. The size of each node is proportional to the total number of outgoing (MBON types) or incoming (CX types) synapses, and the width of each edge is proportional to the connection's relative weight, averaged over all right hemisphere regions of interest (ROIs) outside of the CX. Graph includes all direct connections with at least 10 synapses (as in *Li et al., 2020*) and a relative weight greater than 0.01 (see Materials and methods and *Figure 46—figure supplement 1* for more details). (**B**) Morphological renderings illustrating several aspects of MBON-to-CX connectivity. (**Bi**) shows an example of a strong direct connection, from MBON09 to FB4R. (**Bii**) shows an example of an MBON (MBON04) that contacts multiple FB tangential neurons that innervate dorsal (FB6P) or ventral (FB1H) FB layers. (**Biii**) shows an example of two MBONs that release different neurotransmitters (MBON12, acetylcholine; MBON04, glutamate) but provide convergent output to the same target (FB4A neurons). (**Biv**) highlights the one direct connection that does not involve an FB tangential type, with atypical MBON30 synapsing onto LCNOp. Yellow circles mark synapse locations.

The online version of this article includes the following figure supplement(s) for figure 46:

**Figure supplement 1.** Connection threshold dependence of mushroom body output neuron (MBON) to central complex (CX) connectivity.

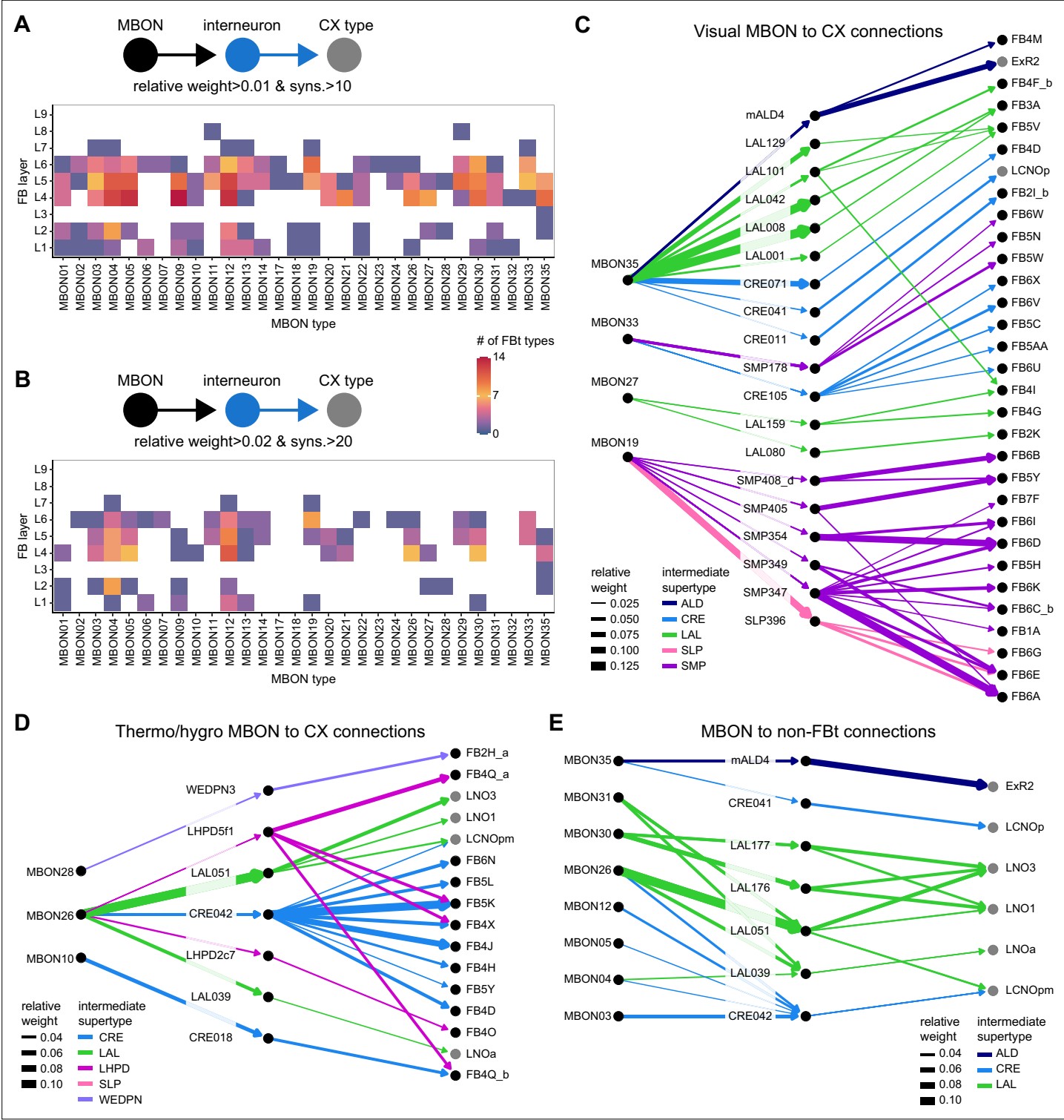

**Figure 47.** Indirect mushroom body output neuron (MBON) to central complex (CX) connections. (**A**) Plot showing the number of fan-shaped body (FB) tangential types, per layer, indirectly targeted by MBONs through one intermediate neuron. Only indirect pathways where each connection involved more than 10 synapses and a relative weight greater than 0.01 were considered. (**B**) Same as in (**A**), but for strong pathways, with greater than 20 synapses and relative weights greater than 0.02. (**C**) Network graph showing all strong (thresholds as in **B**), indirect connections from MBONs that receive at least 20% of their input from visual projection neurons (vPNs) to CX neuron types through one intermediate layer. Edges are colored by the intermediate neuron's supertype, which largely reflects the region of interest (ROI) that contains its arbors. Non-FB tangential targets have gray nodes. (**D**) Same as in (**C**), but for MBONs the receive at least 20% of their input from thermosensory or hygrosensory projection neurons. (**E**) Network graph

*Figure 47 continued on next page*

*Figure 47 continued*

showing all strong, indirect connections from MBONs to non-FB tangential CX neurons. Notice that, other than ExR2, the non-FB tangential targets belong to a LAL-NO type.

The online version of this article includes the following figure supplement(s) for figure 47:

**Figure supplement 1.** Indirect pathways from mushroom body output neurons (MBONs) to central complex (CX) neurons.

In 'Discussion,' we examine the potential role of the connectivity between the MB and FB in navigation, sleep, and memory consolidation.

## Sleep-wake circuits in the dorsal FB and their upstream and downstream connections

Previous studies have established a functional role for several types of dFB tangential neurons in tracking sleep need and controlling sleep-wake states (reviewed in *Artiushin and Sehgal, 2017*; *Donlea, 2017*; *Dubowy and Sehgal, 2017*; *Helfrich-Forster, 2018*). Specifically, a heterogeneous population of tangential neurons targeted by the R23E10 GAL4 line (*Figure 48A and B*; *Qian et al., 2017*) encodes sleep need through changes in both intrinsic excitability and spontaneous firing rates, and induces sleep when activated (*Donlea et al., 2014*; *Donlea et al., 2018*; *Donlea et al., 2011*; *Liu et al., 2016*; *Ni et al., 2019*; *Pimentel et al., 2016*; *Qian et al., 2017*). Counteracting these sleep-promoting populations are wake-promoting DANs (*Liu et al., 2012*; *Ueno et al., 2012*) that are able to inhibit neurons contained in R23E10 (*Pimentel et al., 2016*).

The specific neuron types composing these sleep- and wake-promoting populations remain mostly unknown. To address this, we matched individual neurons in R23E10 to their corresponding EM-defined neuron type by comparing their light- and EM-level morphologies (*Figure 48—figure supplements 2–7*, *Videos 14–15*, see Materials and methods). This analysis identified nine neuron types, each composed of 1–3 neurons per hemisphere, that are targeted by the R23E10 line (*Figure 48C and D*). These neurons occupy type-specific layers and sublayers of the FB, where most of their presynaptic specialization reside and innervate distinct regions of the SMP/SIP/SLP, where they form mixed arbors containing mainly postsynaptic specializations (*Figure 48B and D*). To identify the wake-promoting DANs of the dFB, we took a similar approach, which involved generating a split-GAL4 line, SS56699, that drives expression in three TH + neurons per hemisphere belonging to the PPL1 dopaminergic cluster and then matching these tangential neurons to their corresponding EM neuron types: FB5H, FB6H, and FB7B (*Figure 49*, *Video 16*).

How are the sleep-promoting neurons in R23E10 connected to the wake-promoting PPL1 DANs, and how does the network regulate sleep-wake states? To begin to address this question, we looked at the connectivity between these neuron types within the dFB, where their arbors overlap most strongly and where the majority of their presynaptic sites reside (*Figure 50*). Plotting neuron-to-neuron connectivity revealed a densely recurrent network, but with variable connection strengths, even across neurons of the same type (*Figure 50A*). Not surprisingly, given its more ventral arbors, the dopaminergic FB5H neuron type lacks direct connections with the types present in R23E10. *Figure 50B* shows a network graph of the connectivity between R23E10 neuron types and the two remaining DAN types. The network can be roughly divided into two clusters: one containing layer 6 neurons and the other layer 7 neurons. Each cluster contains both putative sleep-promoting R23E10 neuron types and a putative wake-promoting DAN type. Importantly, the two DAN types make reciprocal connections to nearly every 23E10 neuron in their layer. This simple motif resembles the classic 'flip-flop' circuit model of brain state regulation thought to underlie sleep-wake control in mammals (reviewed in *Saper et al., 2010*). In this model, wake-promoting and sleep-promoting neuron types have reciprocal inhibitory connections, ensuring that only one population is active at a time. While dopamine is known to inhibit R23E10 neurons (*Ni et al., 2019*; *Pimentel et al., 2016*), it is currently unknown whether R23E10 neurons inhibit DANs (*Figures 51–53*, *Figure 53—figure supplement 1*).

The ability of R23E10 neurons and dFB DANs to regulate sleep-wake states likely depends on their upstream and downstream connections, which remain largely unknown. We found that the dFB sleep-wake neuron types have the potential to influence a large number of neuron types both inside and outside the FB (*Figure 51*). Sleep-wake types target both columnar and tangential neuron types within the FB, and also have downstream targets in the SMP/SIP/SLP, consistent with the presence of

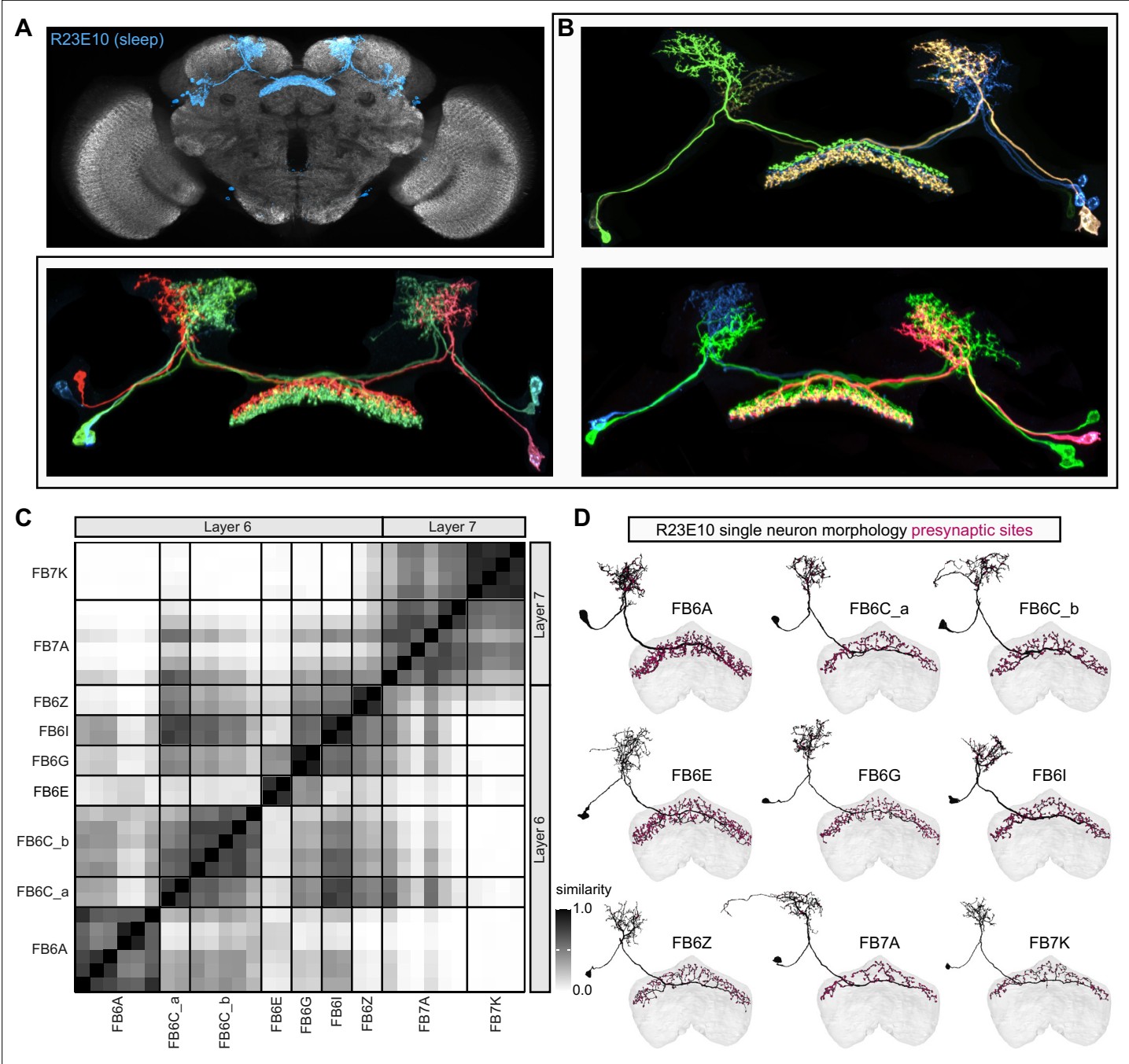

**Figure 48.** Identification of the sleep-promoting dorsal fan-shaped body (dFB) tangential neuron types in the R23E10 GAL4 line. (**A**) Front view of a 3D rendering of a confocal stack showing the R23E10 expression pattern (blue) along with immunohistochemical staining against nc82 (gray). The raw confocal stack was warped to a standard reference brain and rendered in 3D using VVDviewer, which facilitates direct comparison of the R23E10 expression pattern to the EM morphologies of candidate neuron cell types (see Materials and methods and *Figure 48—figure supplements 2–4*). (**B**) Stochastic labeling of subsets of R23E10 neurons made using the MCFO method (*Nern et al., 2015*). Note the differences in arbor morphology outside the FB and the different layers of arbors within the FB for the individual neurons in the line. (**C**) Matrix comparing the similarity in connectivity within the FB for the nine putative R23E10 neuron cell types (31 neurons total, see Materials and methods). (**D**) Single neuron morphological renderings from the EM dataset for each of the nine neuron types that were identified in the R23E10 line. Magenta circles mark presynaptic sites. Two anatomical features of R23E10 neurons —the lateral location of their soma and a fiber tract that enters the FB slightly medial to the lateral border— unambiguously identified 14 candidate tangential neuron types with processes in layers 6 and 7 whose general morphology matched that of the R23E10 pattern. Comparison of these neuron morphologies with candidate EM neuron types allowed us to exclude 5 of the 14 candidates based on the presence of arbors that lie well outside the R23E10 pattern (*Figure 48—figure supplements 2–6*).

*Figure 48 continued on next page*

**Figure supplement 1.** Region arborization plot of R23E10 and dopaminergic neuron cell types (FB6H and FB7B, see *Figure 49*).

**Figure supplement 2.** Summary of sleep-promoting and PPL1 dopaminergic neuron (DAN)-type identification.

**Figure supplement 4.** Cell types not found in R23E10, though they have similar morphology.

mixed arbors in these regions. Similarly, many upstream neuron types have the potential to influence the sleep-wake neurons (*Figure 52*), especially through inputs to their dendritic arbors in SMP/SIP/SLP (see early section on 'Assessing the relative importance of different synaptic inputs'). Within the FB, many of the neurons upstream of sleep-wake neuron types are also downstream of sleep-wake types, forming recurrent loops. For example, FB6A and FB6C_b are reciprocally connected with OA-VPM3, a wake-promoting octopaminergic neuron type with processes spanning many brain regions (*Ni et al., 2019*; *Seidner et al., 2015*). The physiological relevance of these candidate upstream and downstream connections remains to be determined, providing many targets for future physiological investigation.

Previous studies have suggested that a recurrent loop links sleep-wake circuits in the dFB with those in the EB (*Donlea et al., 2018*; *Liu et al., 2016*), but identifying the neurons and pathways involved has proven challenging. To address this, we constructed a network graph that contains the dFB types identified above, a few of their main partners, as well as previously reported CX sleep-wake neuron types: ER5 (*Liu et al., 2016*), ExR1 (i.e., 'helicon cells'; *Donlea et al., 2018*), and ExR3 (*Liu et al., 2019*). This analysis revealed that three neuron types – ExR1, ExR3, and hΔK – directly link sleep circuits in the EB with those in the dFB through distinct channels (*Figure 53*).

ExR1 and ExR3 form a subnetwork in the EB with extensive reciprocal connections involving ER5 and ER3d types. ER5 neurons are known to be involved in homeostatic sleep control (*Liu et al., 2016*; *Raccuglia et al., 2019*) and receive circadian input from the anterior-projecting DN1 pathway (*Figure 6*; *Guo et al., 2018*; *Lamaze et al., 2018*). ER5 neurons also synapse onto EL neurons, which target nearly all other ring neuron types in the EB (see *Figure 10F* for network graph). This may allow ER5 neurons to impact activity throughout the EB circuit. ER3d neurons innervate the inferior-posterior portion of the BU, a region known to contain glomeruli that lack ipsilateral or contralateral visual receptive fields (*Omoto et al., 2017*; *Shiozaki and Kazama, 2017*). In addition, ER3d neuron types are targeted by a driver line that labels neurons expressing the 5HT7 serotonin receptor and

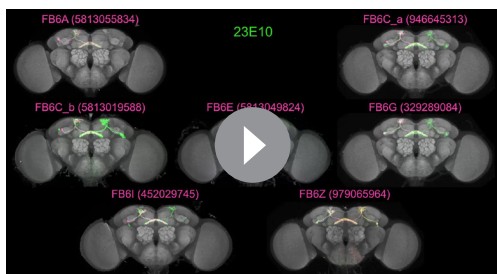

**Video 14.** 3D overlap of individual R23E10 neurons and corresponding EM neuron types from fan-shaped body (FB) layer 6. Video showing 3D morphological renderings comparing individual R23E10 cells from the right hemisphere (green), generated using the MCFO stochastic labeling technique (*Nern et al., 2015*), to single EM neuronal morphologies of FB tangential neurons arborizing in layer 6 (magenta).

https://elifesciences.org/articles/66039/figures#video14

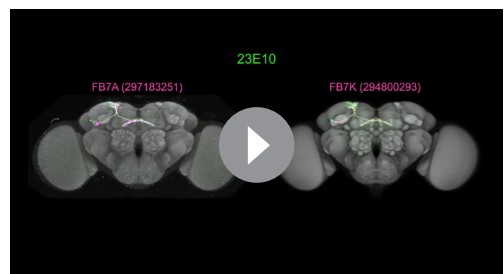

**Video 15.** 3D overlap of individual R23E10 neurons and corresponding EM neuron types from fan-shaped body (FB) layer 7. Video showing 3D morphological renderings comparing individual R23E10 cells from the right hemisphere (green), generated using the MCFO stochastic labeling technique (*Nern et al., 2015*), to single EM neuronal morphologies of FB tangential neurons arborizing in layer 7 (magenta).

https://elifesciences.org/articles/66039/figures#video15

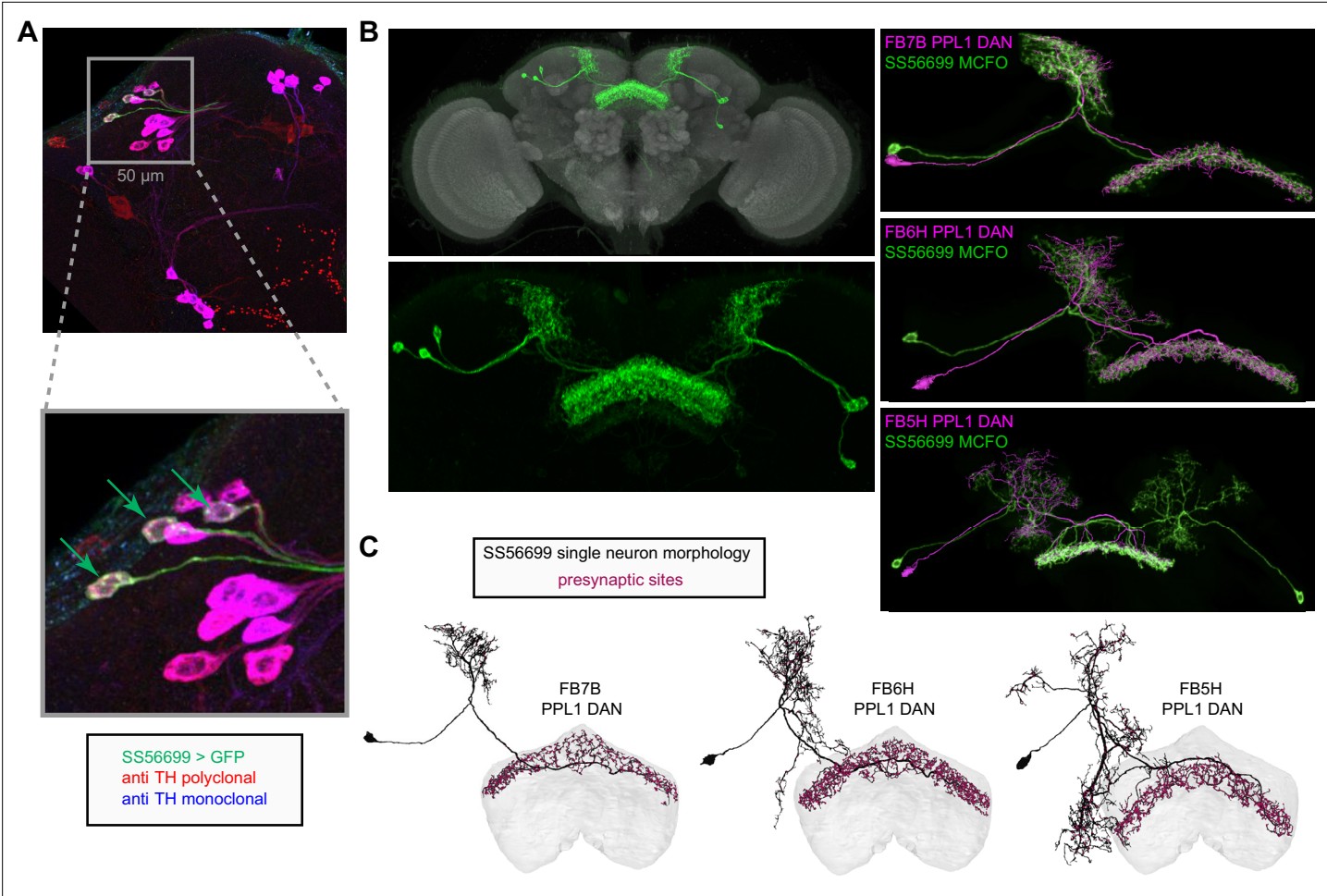

**Figure 49.** Identification of wake-promoting, PPL1 dopaminergic dorsal fan-shaped body (dFB) tangential neuron types. (**A**) Confocal micrographs showing a portion of the expression pattern of a split-GAL4 line, SS56699 (green), focused on the cell bodies of the three neurons expressed in each brain hemisphere of this line along with immunohistochemical staining against TH using a polyclonal (red) and monoclonal (blue) antibody. Inset shows a zoomed-in view of the three SS56699 soma in the right hemisphere, marked by green arrows, which are all TH+. This result was consistent across 12 hemispheres from six brains. (**B**) Expression pattern of the SS56699 line with nc82 reference staining (top) and zoomed-in view of the expression pattern alone (bottom). One of hemidriver parents of this line uses an 11kb genomic segment of the TH tyrosine hydroxylase (TH) gene (see *Aso et al., 2014a*) to drive its expression. Morphological renderings comparing the three putative dFB dopaminergic neuron types (magenta)—FB7B (top panel), FB6H (middle panel), and FB5H (bottom panel)—to individual neurons from SS56699, generated by MCFO stochastic labeling (green; *Nern et al., 2015*). (**C**) Single neuron morphological renderings from each of the three identified PPL1 dopaminergic neuron types: FB7B, FB6H, FB5H. Magenta circles mark presynaptic sites. See *Video 16* for 3D comparisons.

have been previously implicated in the serotonergic ExR3 sleep-wake circuit of the EB (*Figure 53—figure supplement 1*; *Liu et al., 2019*). Finally, hΔK neurons have purely dendritic inputs in the EB, where they receive input predominately from PEN_b neurons. This may be one pathway by which the head direction signal, whose amplitude correlates with the fly's locomotor activity (*Turner-Evans et al., 2017*), can be passed directly to dFB circuits.

ExR1, ExR3, and hΔK neurons all send projections to the dFB, where they connect with distinct clusters (indicated as gray regions in *Figure 53*). The largest cluster consists of a recurrent FB network that includes PFGs, a putative neuromodulatory columnar type (as discussed in a later section), FB6A neurons, which are contained in 23E10, as well as ExR3 and hΔK neurons. While this cluster is anchored by FB6A, it seems likely that even sleep-wake neuron types that lack direct connections with ExR1 and hΔK may be able to influence their activity since sleep-wake neuron types are recurrently connected, as described above.

These results identify a complex network that may allow for communication between sleep-wake circuits in the FB and those in the EB. However, the vast majority of the input and output to sleep-wake

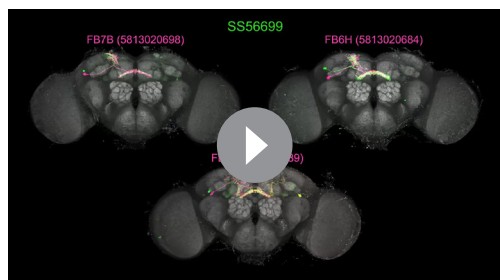

**Video 16.** 3D overlap of PPL1 dopaminergic neurons (DANs) in the dorsal fan-shaped body (dFB) (SS56699) and corresponding EM neuron types. Video showing 3D morphological renderings comparing individual PPL1 dFB tangential neurons contained in the SS56699 line from the right hemisphere (green), generated using the MCFO stochastic labeling technique *Nern et al., 2015* neurons (magenta).
https://elifesciences.org/articles/66039/figures#video16

neuron types involve additional pathways, most of which have not been functionally character-ized, as described above. For example, most of the output from dFB sleep-wake neuron types is onto other FB neurons, raising the possibility that inducing sleep may involve previously unrec-ognized neuron types and pathways (*Dag et al., 2019*; *Tomita et al., 2020*). For example, Lei, Keleman et al. (in preparation) have identified the neuron that corresponds to the sleep-promoting split-GAL4 line SS57264 described in *Dag et al., 2019* as FB2B_a. This neuron is not connected to the R23E10 neurons by the columnar neurons in a single step, so these sleep-promoting mecha-nisms are not obviously coordinated.

## Pathways that leave the CX: feedback, motor output, and more

In the preceding sections, we focused on char-acterizing the connectivity patterns of different structures within the CX. We now turn our attention to potential output pathways from the CX. The CX is thought to communicate with the motor centers through the PFL neurons in the LAL (*Hanesch et al., 1989*; *Heinze and Homberg, 2008*; *Namiki and Kanzaki, 2016*), where they contact DNs involved in the control of steering (*Rayshubskiy et al., 2020*). Here we systematically analyzed the

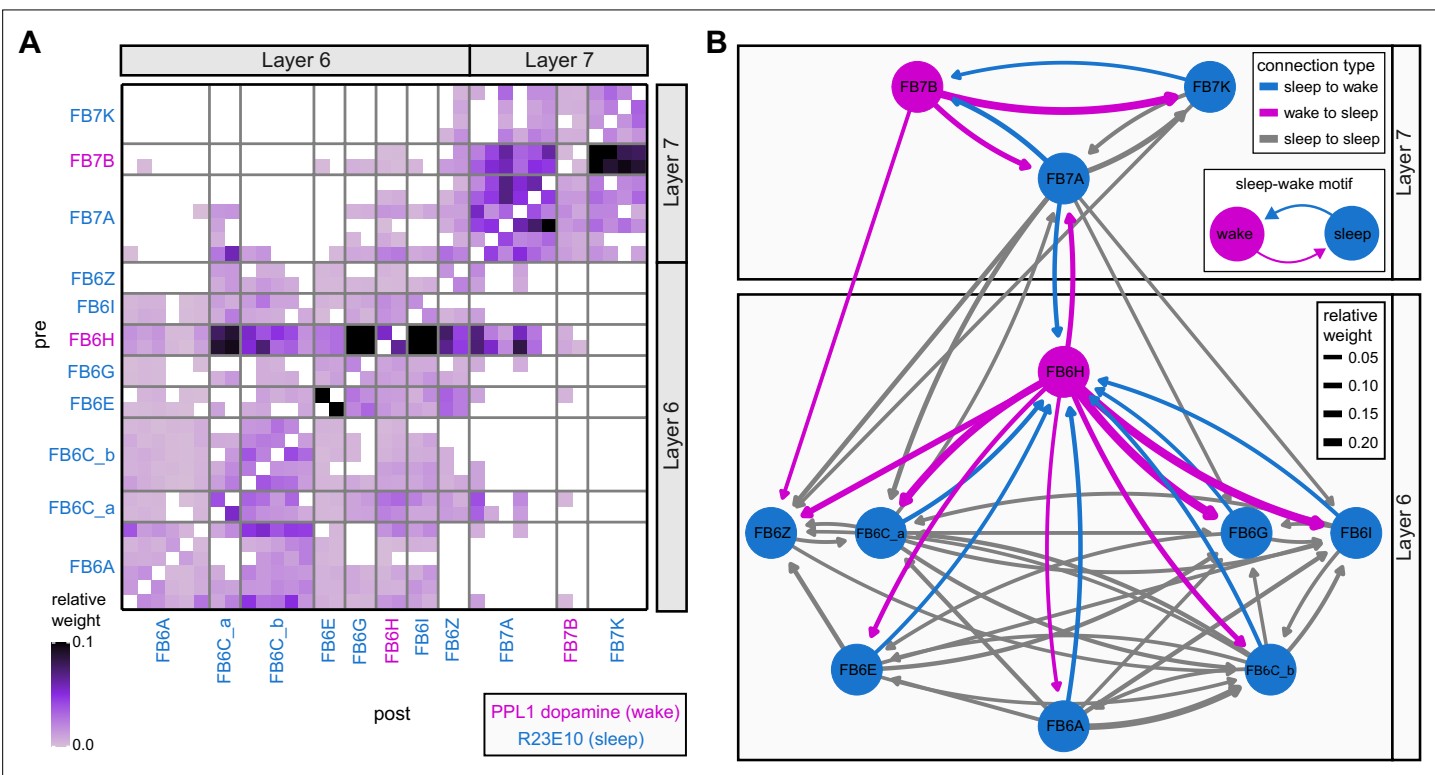

**Figure 50.** A potential sleep-wake flip-flop switch in the dorsal fan-shaped body (dFB). (**A**) Neuron-to-neuron connectivity matrix between R23E10 neurons (marked by blue text) and dopaminergic neurons (marked by magenta text). Note that most layer 6 neurons connect to other layer 6 neurons, and layer 7 neurons to other layer 7 neurons, but there are many fewer connections between layers, consistent with tangential neurons' layer-specific innervation patterns. (**B**) Network graph showing the intra-FB connections between the R23E10 and dopaminergic types. Arrow width is proportional to connection strength, and arrow color indicates connection type. Node color indicates whether the neuron type belongs to a putative wake-promoting type (magenta) or a putative sleep-promoting type (blue). Connections within a type have been omitted for clarity, but can be observed in (**A**).

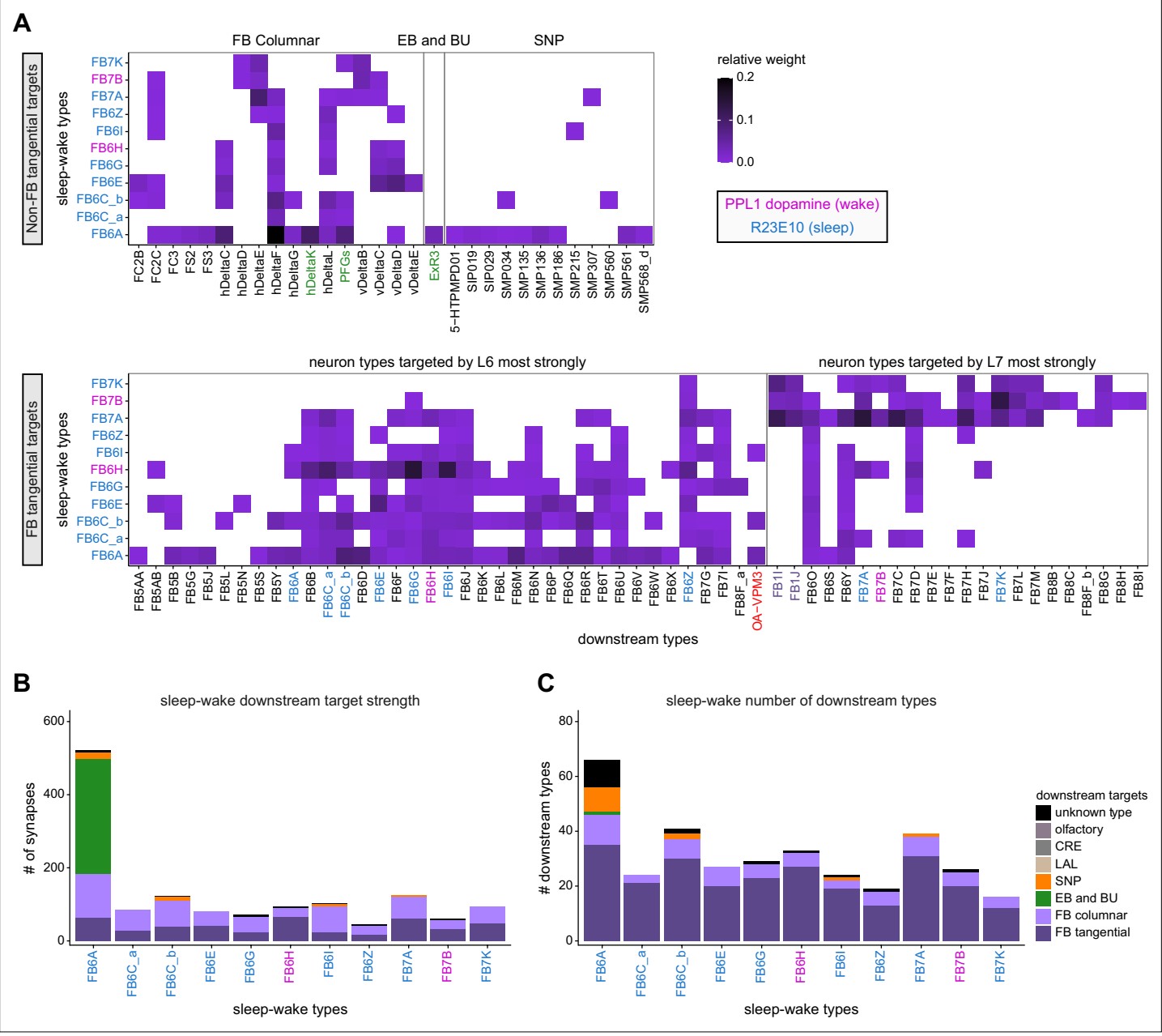

**Figure 51.** Downstream targets of dorsal fan-shaped body (dFB) sleep-wake neurons. (**A**) Type-to-type connectivity matrices showing the neuron types targeted by each of the sleep-wake neuron types. The downstream neurons are divided into groups, with non-FB tangential targets shown in the top panel and FB tangential targets shown in the bottom panel. Green text marks neurons involved in ellipsoid body-FB (EB-FB) sleep-wake circuit (see *Figure 53*). Connections with relative weights below 0.005 were excluded from this analysis. (**B**) Number of synaptic connections from each sleep-wake neuron type to other FB neuron types (tangential or columnar), neuron types with prominent arbors outside the FB (EB/bulb [BU], superior protocerebrum SMP/SIP/SLP, lateral accessory lobe [LAL], crepine [CRE], and olfactory), or unknown types (i.e., neurons that have not been assigned a type name). (**C**) Same as in (**B**) but plotting the number of downstream neuron types reached.

circuits downstream of CX neurons in all CX accessory structures and constructed a more complete picture of all CX outputs. We show that, besides being involved in a large number of recurrent loops back into the CX, CX output neurons are upstream of numerous circuit modules that span most of the brain's neuropils.

We began by examining the projection patterns of all core CX neuron types that contact neurons in other brain regions (*Figure 54A*). These neurons target a narrow column around the CX that extends through the BU, GA, CRE, LAL, SPS, WED, RUB, ROB, SMP, SLP, IPS, and SPS (*Figure 54A*

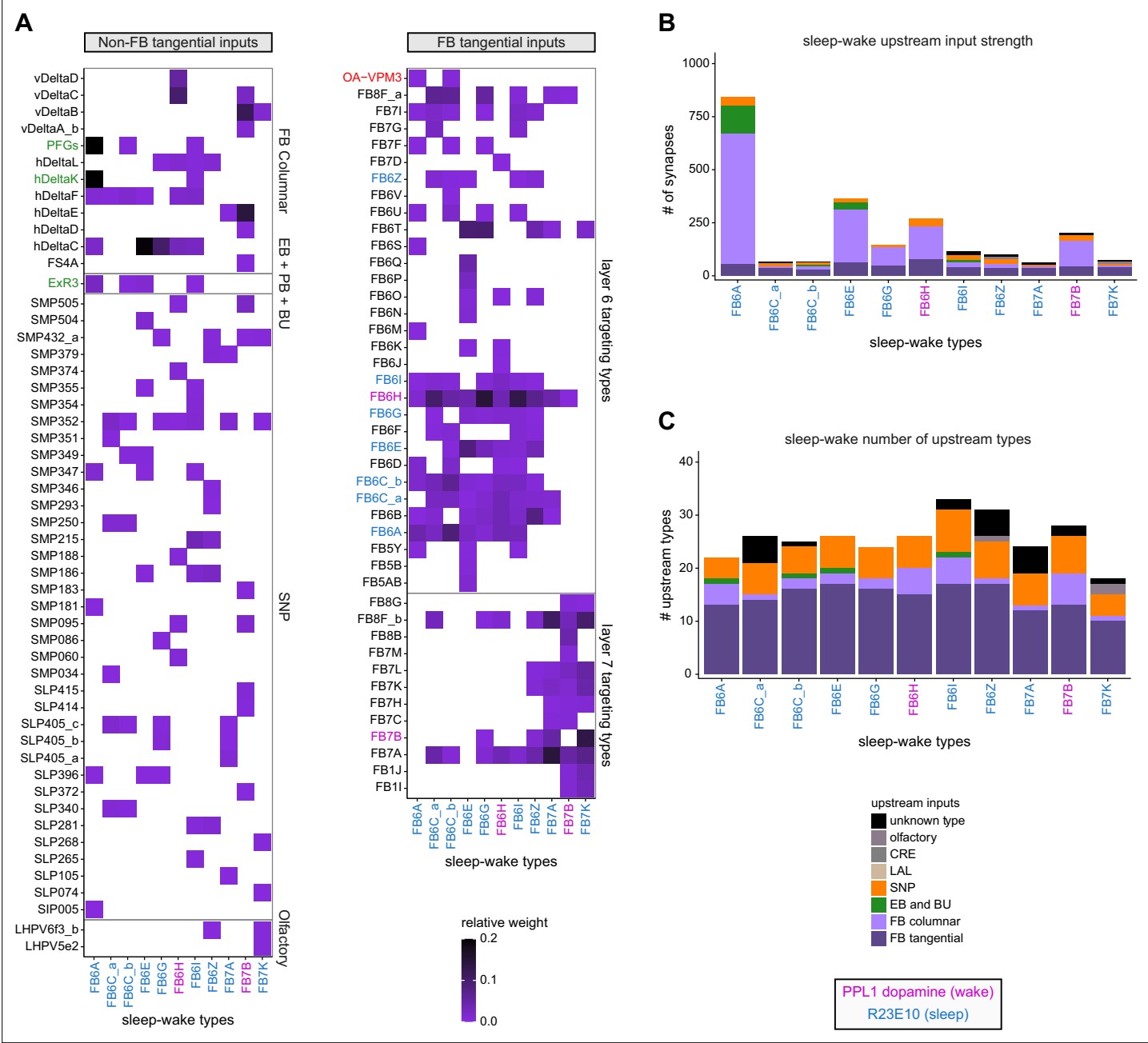

**Figure 52.** Inputs to dorsal fan-shaped body (dFB) sleep-wake neurons. (**A**) Type-to-type connectivity matrix showing the neuron types that target each of the sleep-wake neuron types. The upstream neurons are divided into groups, with non-FB tangential targets shown in the left panel and FB tangential targets shown in the right panel. Connections with relative weights below 0.005 were excluded from this analysis. (**B**) Number of synaptic connections to each sleep-wake neuron type from other FB neuron types (tangential or columnar), neuron types with prominent arbors outside the FB (ellipsoid body/bulb [EB/BU], superior protocerebrum SMP/SIP/SLP, lateral accessory lobe [LAL], crepine [CRE], and olfactory), or unknown types (see legend). (**C**) Same as in (**B**) but plotting the number of upstream neuron types.

*and B*), as partially described in previous studies (*Hanesch et al., 1989*; *Li et al., 2020*; *Lin et al., 2013*; *Rayshubskiy et al., 2020*; *Wolff et al., 2015*; *Wolff and Rubin, 2018*). The neural projections and synapse locations within these regions appear to largely segregate by type (see *Figure 54B*). Different types of neurons contribute very different numbers of synapses to their target areas, from tens of synapses for some FB tangential neurons to more than 10,000 synapses for PFL3 neurons (*Figure 54C*). Thus, information from the CX is broadcast widely across the brain, but likely through different, target-specific communication channels of various strengths.

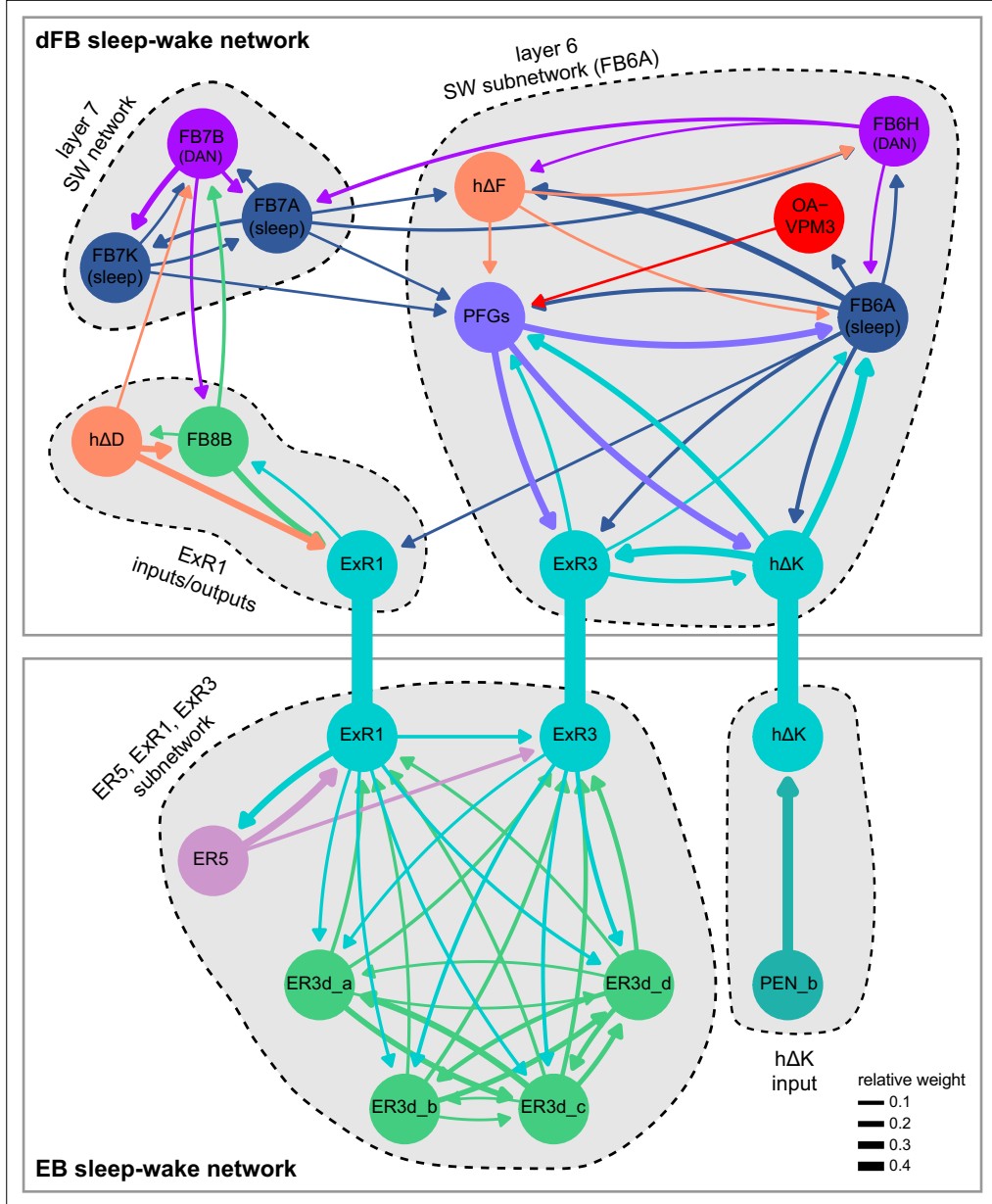

**Figure 53.** A direct pathway linking sleep-wake neurons in the dorsal fan-shaped body (dFB) and ellipsoid body (EB). Network graph showing the connections between ExR1, ExR3, and hΔK, along with some of their major upstream and downstream connections in the dFB (top panel) and EB (bottom panel). dFB types contained in the R23E10 line have "(sleep)" below their name, while the wake-promoting dopaminergic types have "(DAN)". Note that these circuits are embedded in the highly recurrent dFB and RB networks, whose many neuron types and connections have been omitted for clarity.

The online version of this article includes the following figure supplement(s) for figure 53:

**Figure supplement 1.** Ellipsoid body (EB) neuron types in 5HT7-GAL4.

To assess how far information spreads across all output pathways, we computed the number of neurons reached by each successive synaptic 'hop' along the individual pathways (*Figure 55A*). We found that within just two hops, that is, by the third network 'layer' (a term we use to denote neurons a certain number of synaptic steps away rather than in the conventional sense of a purely feedforward network), information from the CX reaches several thousand neuron types and over 10,000 individual neurons within the hemibrain volume (*Figure 55B*). By the fifth network layer, the reach of the CX extends to 80% of all neurons in the brain (similar to what is reported in Figure 20 in *Scheffer et al.,*

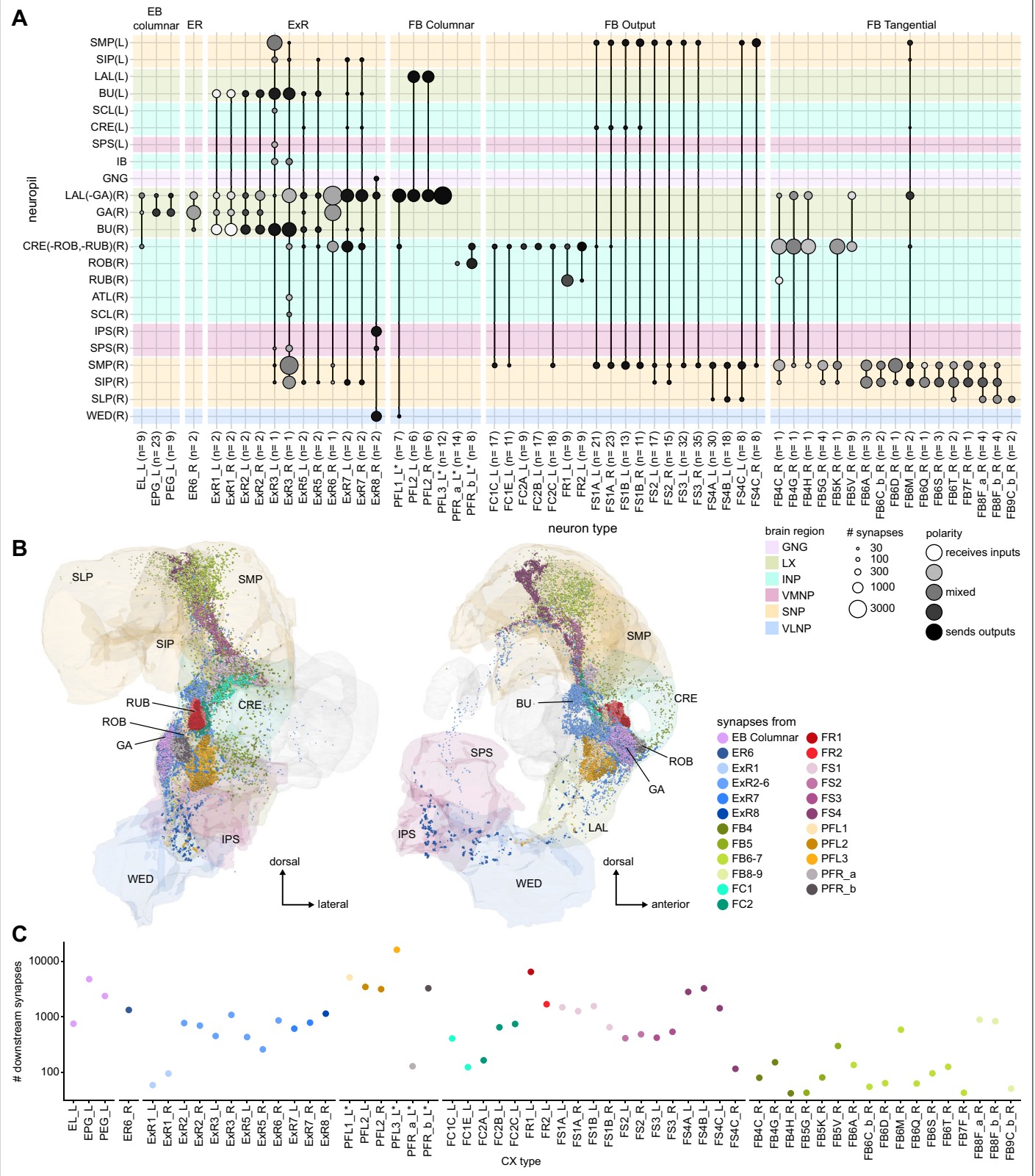

**Figure 54.** Central complex (CX) neurons with downstream synapses outside the CX. (**A**) Neuropil innervation plot of all CX types having downstream connections outside the central complex. Only CX neuron types that have a significant number of presynaptic terminals in other brain regions are shown. The CX is excluded to highlight the connections in non-CX neuropils. The CX innervation of the same neurons can be found in *Figure 10* (ellipsoid body [EB] columnar, ER neurons, extrinsic ring [ExR] neurons), *Figure 28—figure supplement 1* (FB columnar), and *Figure 40—figure*

*Figure 54 continued on next page*

Figure 54 continued

**supplement 1** (FB tangential neurons). (**B**) Presynaptic site locations outside of the CX in the right hemibrain for the neuron types shown in (**A**). Left: frontal view. Right: side view. The locations are overlaid on an anatomical rendering of the relevant neuropils. Dot colors indicate the neuron type. The color code for neuropils is identical in (**A**) and (**B**). (**C**) The total number of synapses across all significant type-to-type connections outside of the CX in the right hemibrain for all neurons of the types shown in (**A** and **B**).

*2020*). Additionally, information from each network layer feeds back into previous layers (*Figure 55A*, lower loops). Recurrent loops constitute a large percentage of connections within each layer (loops in *Figure 55A*, also see *Figure 55B*).

Although many of these downstream neurons, especially those reached in later layers, arborize in regions that have been poorly characterized, we could, in some cases, classify target neurons. We labeled these targets either by their type or by their main neuropil (*Figure 55C and D*), following the same classification scheme as detailed in Appendix 1—table 6 in *Scheffer et al., 2020*. We discovered that within two hops information from the CX reached areas as far as the LH, and neuron types as diverse as MBONs, VPNs, DNs, and a variety of neuromodulatory and peptidergic neurons (*Figure 55D*). We describe these different targets in more detail in subsequent sections.

Different pathways out of the CX show varying degrees of divergence (*Figure 55E*). Divergence here refers to the number of new neuron types reached at each network layer. For example, pathways that start at EL, ER6, ExR1, FB6Q, or FB6T neurons reach very few neuron types overall, suggesting that the information in these pathways does not spread widely. By contrast, pathways that start at PFL3 and FS1A neurons reach many different neuron types within a few hops, suggesting that information from these types is widely shared across the brain (*Figure 55E*). The number of types reached partially reflects the difference in synapse numbers (compare *Figure 55E* to *Figure 54C*). Besides the PFL neurons, the FS, FC, and FBt neurons, as well as the ExR7 and ExR8 neurons, also reach a large number of neurons.

From this point on, we use pathway weights (*Appendix 1—figure 3*; also see Materials and methods) to quantify the influence exerted by a given CX output neuron onto a different neuron. Briefly, pathway weights quantify the relative aggregate influence of one neuron onto another neuron through all the pathways that link them.

## Recurrent connectivity within lateralized neuron populations in CX accessory structures

Not all connections in CX accessory structures feed into feedforward output circuits. In fact, we found that output synapses in four structures –the GA, ROB, RUB, and BU – are mostly to other CX neurons, forming recurrent CX-to-CX pathways (*Figure 56A*). Recurrent pathways via the BU – an important input hub to the EB – are formed exclusively by ExR neurons (*Figure 56B*, see earlier section on the ExR neurons). These recurrent pathways likely serve to modulate inputs to the EB. In contrast, pathways through the GA, ROB, and RUB are composed of EB and FB columnar neurons (*Figure 56B*). We will focus on these three structures below.

The GA, ROB, and RUB are paired across the midline and are selectively innervated by either the left or right PB-projecting population of columnar neurons (*Figure 56C*). The GA appears to primarily house recurrent connections between CX neurons, creating CX-to-CX recurrent loops that may allow EB columnar neurons to form connectivity patterns independent of their EB arborizations (*Figure 56D*). For example, all EPG and PEG neurons arborize in the GA. Depending on which EB wedge the neurons innervate, they target either the dorsal or ventral GA compartment (*Figure 56E*; this compartment specificity was previously described in *Wolff et al., 2015*). In their respective compartments, EPG neurons then synapse onto PEG neurons, resulting in a checkerboard-like connectivity pattern (*Figure 56D*). As a result, each set of columnar neurons innervating one GA subcompartment comprises four tiles separated by 90° increments. The functional significance of this motif is currently unknown. A few non-columnar types also participate in recurrent motifs between the GA and the EB. ER6, one of the few ring neuron types that does not connect to EPG neurons in the EB (*Figure 11A*, *Figure 56Fii*), is recurrently connected to both EPG and PEG neurons in the GA (*Figure 56D*), and to PEG neurons in the EB (*Figure 56F*). ExR6 also outputs to the EPG neurons in the GA (*Figure 56—figure supplement 1A*), and ExR6 and EPG neurons are recurrently connected in the EB (*Figure 14—figure supplement 3*). Finally, it is worth noting that EPG neurons also receive

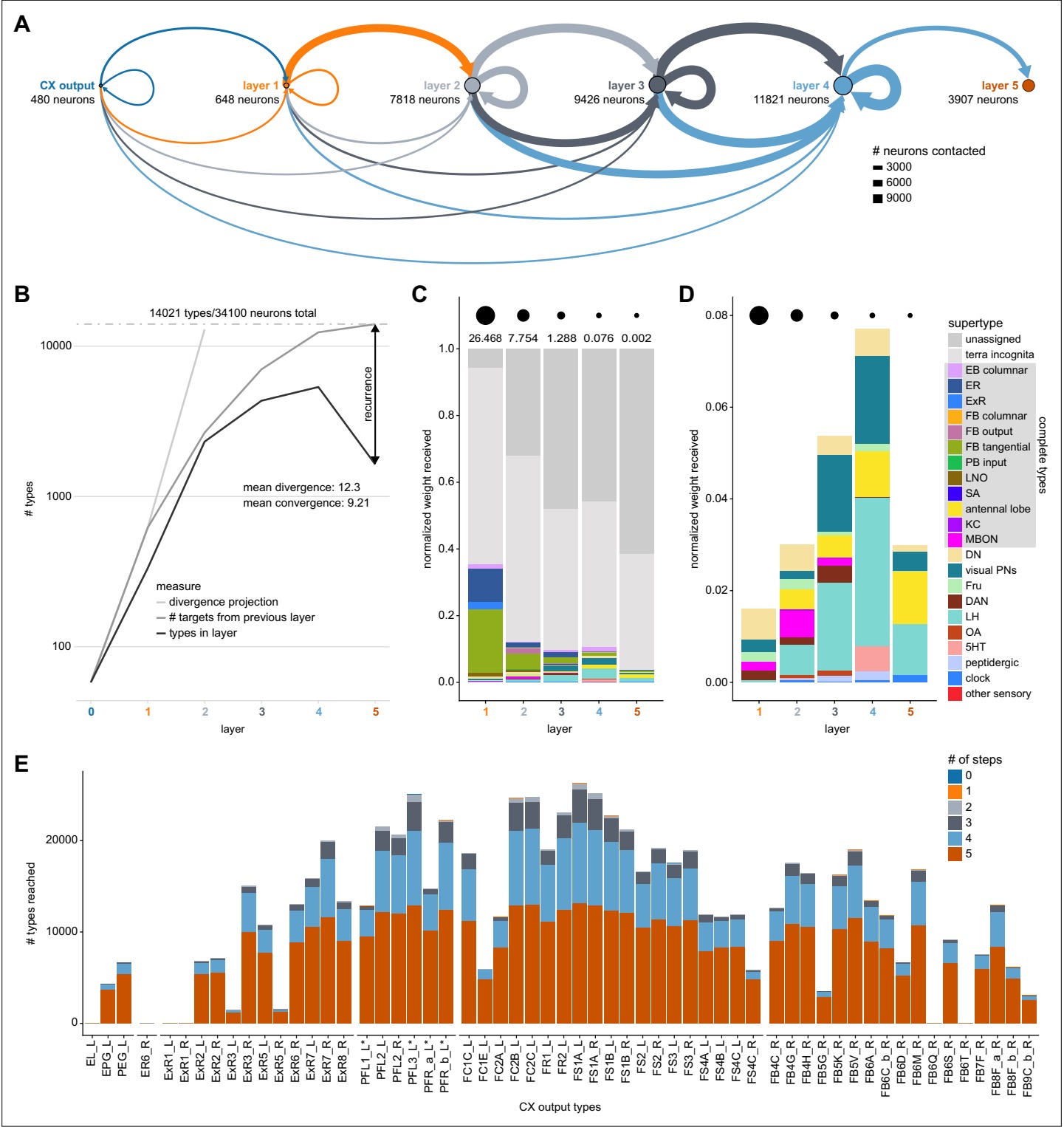

**Figure 55.** Divergence of output networks. (**A**) Diagram of the number of neurons contacted while walking five steps downstream from central complex (CX) neurons that arborize outside the CX. Size of the circles represents the number of new neurons in each layer. The layer a neuron is assigned to corresponds to the length of the shortest path from the CX to that neuron. The thickness of the connecting lines indicates the number of neurons reached in the same layer (loops), in the next layer (top arc) or in previous layers (bottom arcs). Color of the connector indicates the layer of origin. (**B**) The number of types per layer (black), the total number of targets of the previous layer to any layer (dark gray), and the projected number of types from the mean divergence of the previous layer (light gray). The difference between the total number of targets (dark gray) and the number extrapolated from the divergence (light gray) reveals the level of convergence of the output pathways. The difference between the number of types per layer

*Figure 55 continued on next page*

*Figure 55 continued*

(black) and the total number of targets (dark gray) corresponds to connections that are not simple feedforward connections and reveals the amount of recurrence of the output pathways. On average, each type connects 12.3 other types (divergence of 12.3) and is contacted by 9.21 other types (convergence of 9.21). Note that the total here exceeds the number of neurons in the database as they include simulated pathways on the side of the brain not present in the volume (see Materials and methods). Of the 34,100 neurons reached, 21,363 (out of 26,190 for the entire dataset) are in the hemibrain dataset, and 12,737 are mirror symmetric neurons from existing neurons inferred from symmetric connections. (**C**) Relative type composition of the different layers weighted by the pathway weight (see Materials and methods, *Appendix 1—figure 3*) they receive from the CX. Circles on the top row represent the total pathway weight received in every layer. The total pathway weight decreases as the layer gets farther away from the CX, as is expected with the metric used, which multiplies relative weights across a pathway then sums pathways ending on the same neuron. Reflecting the composition of the database, the majority of neurons reached either belong to poorly studied neuropils ('terra incognita') or have no name in the database ('unidentified'). Note that in the first layer, most identified targets are CX types. (**D**) Same as (**C**), but zoomed-in onto known types excluding CX types. Types with a gray background in the legend are those for which most existing neurons of that type are present in the database. The fraction of known targets increases to reach a maximum in the fourth layer. (**E**) Number of types reached outside of the CX for every CX neuron innervating outside of the CX in different downstream layers. Note that very small numbers are not visible on this scale.

strong input from a LAL neuron of unknown function in the GA (*Figure 56—figure supplement 1A*). In summary, the GA mostly hosts recurrent connectivity motifs between EPG, PEG, and a defined subset of ER and ExR neurons.

Some columnar neuron types, such as the EL and PFGs neurons, do not target the GA proper. Instead, they target an undefined region surrounding the GA termed the GA surround (GAs). In the GAs, EL neurons appear to primarily form all-to-all connections to other EL neurons from the same hemisphere (*Figure 56D and Fi*). Curiously, neither the EL neurons nor the PFGs neurons make many synapses in the GAs. We therefore looked in the EM images for evidence of other signaling mechanisms. We found that the synapses formed by both PFGs and EL neurons are elongated-bar (E-bar rather than T-bar) synapses (*Shaw and Meinertzhagen, 1986*; *Takemura et al., 2017a*) and that they contain dense core vesicles (DCVs) (*Figure 56—figure supplement 1B–D*). DCVs have been associated with neuropeptide or neuromodulator release (*Burgoyne and Morgan, 2003*; *Hammarlund et al., 2008*; *Nassel and Winther, 2010*), suggesting that synaptic activity in the GA may be broadly regulated in ways that have not yet been explored.

The round body (ROB) and rubus (RUB) are innervated by the FB columnar PFR and FR neurons, respectively (*Figure 56C*). Both PFR types arborize in the ROB. There, PFR_a receives input only from PFR_b, while PFR_b neurons make all-to-all within-type connections as well as a number of output connections (*Figure 56—figure supplement 2A and B*). The most significant connection in the ROB appears to be from PFR_b to the only non-CX neuron targeting a large extent of the ROB, LAL002, which in turn connects to several other non-CX neurons in the LAL and CRE (*Figure 56—figure supplement 2B and C*). Similar to PFR neurons in the ROB, only one of the FR neurons, FR1, forms strong within-type all-to-all connections in the RUB (*Figure 56—figure supplement 3A*). However, unlike in the ROB, both FR1 and FR2 make connections to non-CX partners in the RUB (*Figure 56—figure supplement 3B*). Notably, the sets of downstream partners as well as the output pathway circuits of FR1 and FR2 are largely distinct (*Figure 56—figure supplement 3B and C*). Thus, while the ROB and RUB contain recurrent connections between CX columnar neurons like the GA does, they differ from the GA in that they can also be considered CX output structures. Some of their outputs will be described in later sections.

## CX feedback pathways through other regions

CX output neurons that project to other regions that are less tightly linked to the CX – like the LAL, CRE, or SMP – also participate in feedback pathways. Indeed, pathways that start with a CX output neuron and reach another CX neuron directly or indirectly outside of the CX account for a large fraction of the total output pathway weights of most CX output neurons (*Figure 57A*). For example, 75% of the outputs of the ExR7 neurons feed pathways that reenter the CX, mostly through ER neurons and FBt neurons. In a structure associated with motor control, such feedback could enable a broad class of neurons, including all the columnar neurons, to be notified of an upcoming CX-initiated action (see 'Discussion').

Not all CX output neurons contribute equally to feedback pathways. Feedback constitutes only 25% of total pathway weights out of the PFL neurons, and the ExR8 neuron type contributes virtually nothing to pathways that feedback into the CX. This corresponds to differences between output

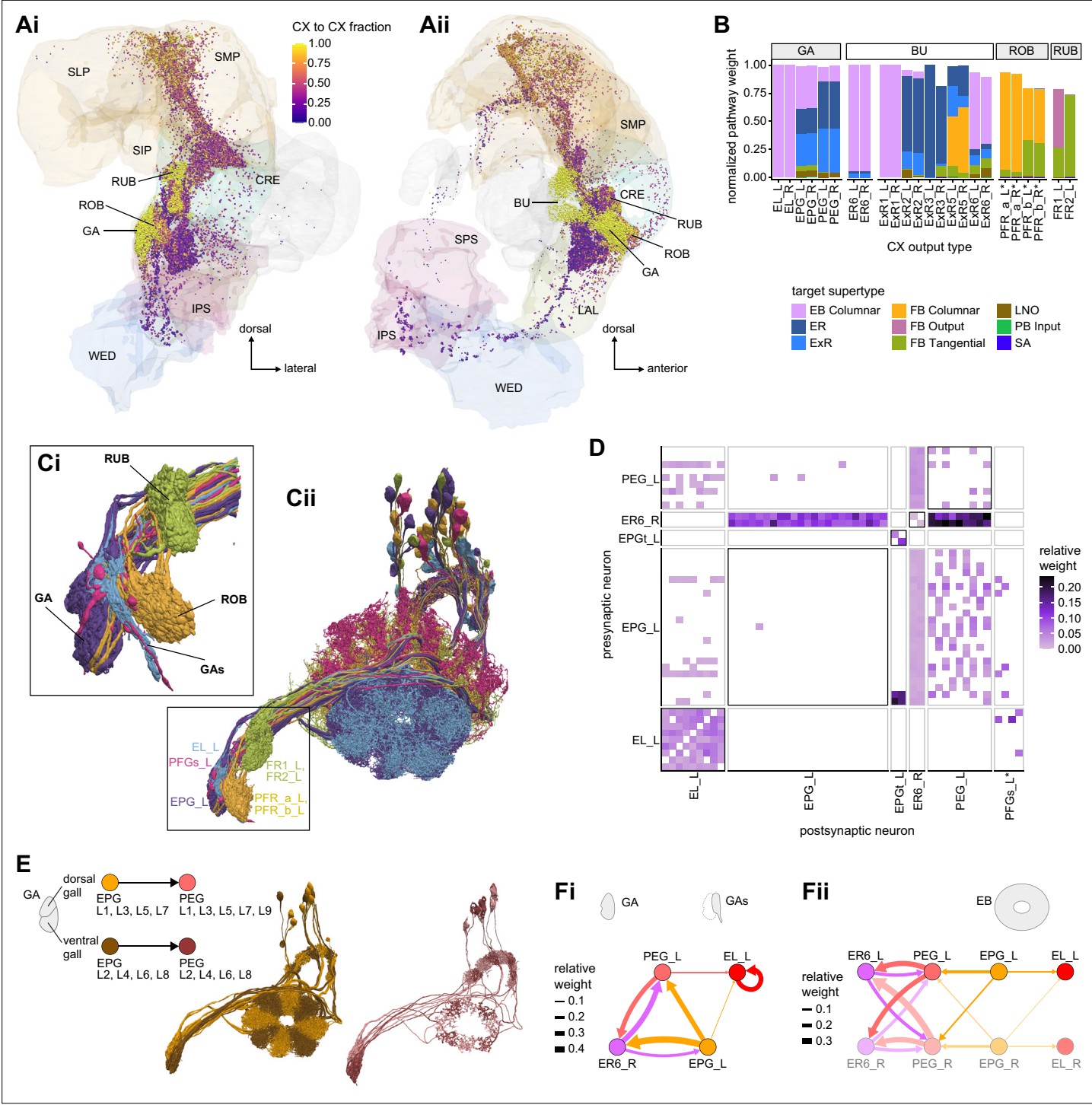

**Figure 56.** Central complex (CX) to CX connections in the gall (GA), bulb (BU), round body (ROB) and rubus (RUB). (**A**) Downstream synapses of potential CX output neurons, colored by the fraction of their target pathways that contribute to pathways coming back to the CX (see Materials and methods). The GA, BU, ROB, and RUB contribute most of their outputs to the CX and the lateral accessory lobe (LAL) almost none, while the upper neuropiles are mixed. (**Ai**) Frontal view; (**Aii**) side view. (**B**) Pathway weights of all pathways that start in the GA, BU, ROB, and RUB and end on another CX neuron. The weights are normalized for each type of origin. If the normalized pathway weight is 1, it corresponds to a neuron for which all output pathways come back to the CX. Connections are separated by the supertypes that these recurrent pathways reach. EB neurons mostly reach other EB neurons, whereas FB neurons mostly reach other FB neurons. (**C**) Morphological renderings of seven selected neuron types that innervate the GA, gall surround (GAs), ROB, and RUB. (**Ci**) Closeup of the four structures. (**Cii**) Illustration of the full morphology of the seven neuron types, showing the left population only. (**D**) Connectivity matrix of neurons that arborize in the right GA and GAs region. All connections outside of the CX regions (ellipsoid

*Figure 56 continued on next page*

*Figure 56 continued*

body [EB], protocerebral bridge [PB], fan-shaped body [FB], noduli [NO]) were considered because the GA region of interest does not capture the GAs. PFGs neurons were included in the analysis, but did not make any significant connections. (**E**) Illustration of selective connectivity between EPG neurons to PEG neurons from odd and even wedges of the EB in the dorsal and ventral GA, respectively. Left: schematic. Middle: rendering of EPG cells targeting the right GA with those from odd wedges colored in orange and those from even wedges colored in brown. Right: rendering of PEG neurons shown analogously as EPG cells (maroon: even-numbered PB glomeruli; pink: odd-numbered PB glomeruli). (**F**) Connectivity graphs on the level of neuron types showing any connections with at least 0.05 relative weight. (**Fi**) Connectivity in the GA and GAs. (**Fii**) Connectivity in the EB between the same neuron types as in (**Fi**), including the neurons from the other hemisphere.

The online version of this article includes the following figure supplement(s) for figure 56:

**Figure supplement 1.** Gall (GA) and gall surround (GAs).

**Figure supplement 2.** Round body (ROB).

**Figure supplement 3.** FR connectivity in the rubus (RUB).

neuropils. Most synapses made by CX output neurons in the LAL, WED, and PS are to 'true' outputs pathways that leave the CX, whereas the situation is more mixed in regions like the CRE and SMP (***Figures 56A and 57A***).

Unsurprisingly, the major input types of the CX – ER neurons and FBt neurons – are the main recipients of synapses from feedback pathways (***Figure 57A***, ***Figure 57—figure supplement 1***). Some columnar types are also targeted by feedback pathways, and these pathways usually involve columnar-to-columnar recurrence, similar to the motifs we described in the GA, ROB, and RUB (***Figure 57B***, ***Figure 57—figure supplement 1B***). Moreover, only weak feedback pathways connect EB and FB neurons. EB columnar neurons and ExR neurons mostly talk between themselves (***Figure 56***, see also ***Figure 15***), whereas FB neurons mostly reach FB tangential neurons (***Figures 56B and 57A***). EB-FB and FB-EB pathways *outside* the CX are largely absent and weak when present because EB neurons and FB neurons innervate nonoverlapping areas outside of the CX (***Figure 54B***) and therefore form few direct connections between each other in these accessory regions. However, there are a few notable exceptions. For example, the PFL types participate in output pathways that feed back into the EB through ER and ExR neurons (***Figure 57Ci and Cii***). Additionally, PFL2 and PFL3 neurons reach several LAL-NO neuron types (***Figure 57Ci and Ciii***). Since the PFL2 and PFL3 neurons are hypothesized to carry motor commands (see 'Discussion'), these pathways from the PFL neurons to the EB and NO could be used to bring self-motion information back into the CX in the form of efference copies.

## Network motifs involving CX neurons with external projections

Above we described how CX-neuron-to-CX-neuron connections outside of the CX are most commonly made between neurons originating in the same core CX structure (EB or FB). We next asked if these same neuron types also connect inside the CX, and if so, what types of motifs best captured their connectivity patterns inside and outside the CX (***Figure 58***). Following the same line of analysis that we used for analyzing ExR connectivity (***Figure 15***), we distinguished three possible motifs (***Figure 58A***): 'canonical feedback,' 'parallel connections,' and 'linked targets.' The 'canonical feedback' motif corresponds to cases in which CX output neurons make synapses outside of the CX onto other CX neurons, which, in turn, project back to a core CX structure in which they contact the original output neurons. For example, PFL1 neurons provide convergent input to FB2B_b neurons, which then feed back onto PFL1 neurons in the FB (***Figure 58B***). Similar feedback motifs have frequently been found to be involved in inhibitory gain control and gating (***Womelsdorf et al., 2014***). In our example, if FB2B_b were to be inhibitory, it would potentially allow PFL1 neurons to regulate the timing and magnitude of their output. The fact that many ER neurons are known to be inhibitory makes gain control through feedback loops a potential function of those motifs in the EB. The second motif involves 'parallel connections' from one type onto another in multiple structures. For example, neurons of the tangential type FB6T synapse onto neurons from the ipsilateral FB6E type in both the SMP/SIP and the FB (***Figure 58C***). The function of such a motif is not clear, but it is likely to depend on the extent to which electrical activity in these neurons is compartmentalized. The third and final motif involves neurons projecting to 'linked targets.' These recurrent subnetworks are composed of multiple neuron types that connect to each other in the CX while sharing a common input outside the CX. A prominent example of such a case is the FB8F_a neuron type that projects to a set of FB6 neurons interconnected

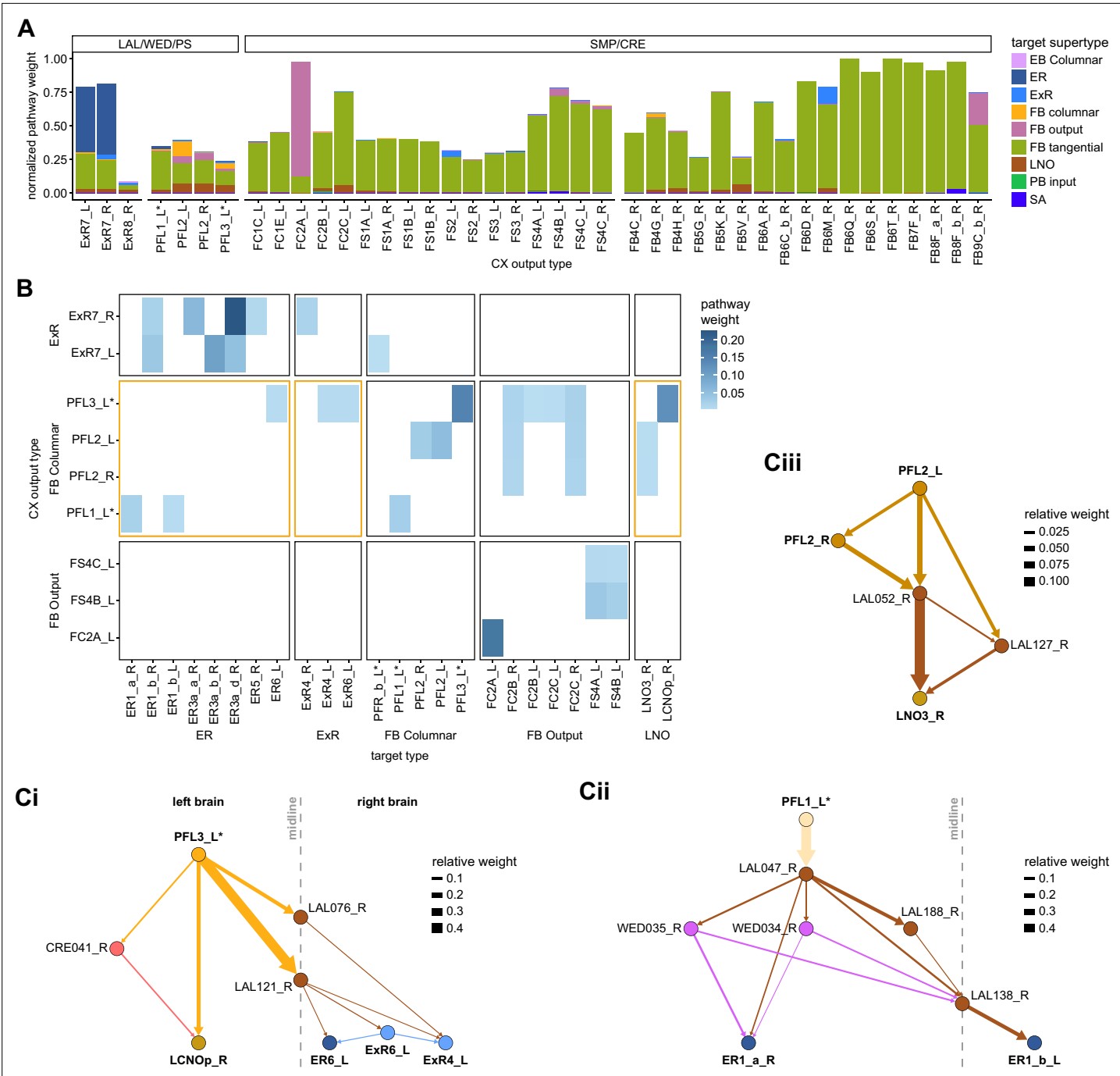

**Figure 57.** Central complex (CX) to CX connections in other regions. (**A**) Pathway weights (see *Appendix 1—figure 3*) of pathways that end on another CX neuron for neurons innervating the neuropils not described in *Figure 56*. Lateral accessory lobe (LAL), wedge (WED), posterior slope (PS), crepine (CRE), and superior medial protocerebrum (SMP). The weights are normalized for each type of origin (see Materials and methods). The connections are separated by the supertypes that these recurrent pathways reach. (**B**) Type-to-type pathway connectivity matrix of those same neuron, excluding the fan-shaped body (FB) tangential neurons. Connections from the FB to ellipsoid body (EB) and noduli (NO) neurons are highlighted. (**C**) Pathways from PFL neurons to EB and NO neurons. (**Ci**) PFL3 neurons connect ipsilaterally to LCNOp and contralaterally, through midline crossing LAL interneurons, to ER6, ExR6, and ExR4 neurons. (**Cii**) PFL1 neurons, through a multilayered network, reach ER1_a neurons ipsilaterally and ER1_b neurons contralaterally. Note that LAL138 is the WL-L neuron described in the section about mechanosensory inputs. (**Ciii**) PFL2 neurons reach LNO3 in two steps.

The online version of this article includes the following figure supplement(s) for figure 57:

**Figure supplement 1.** All central complex (CX-to-CX) connections.

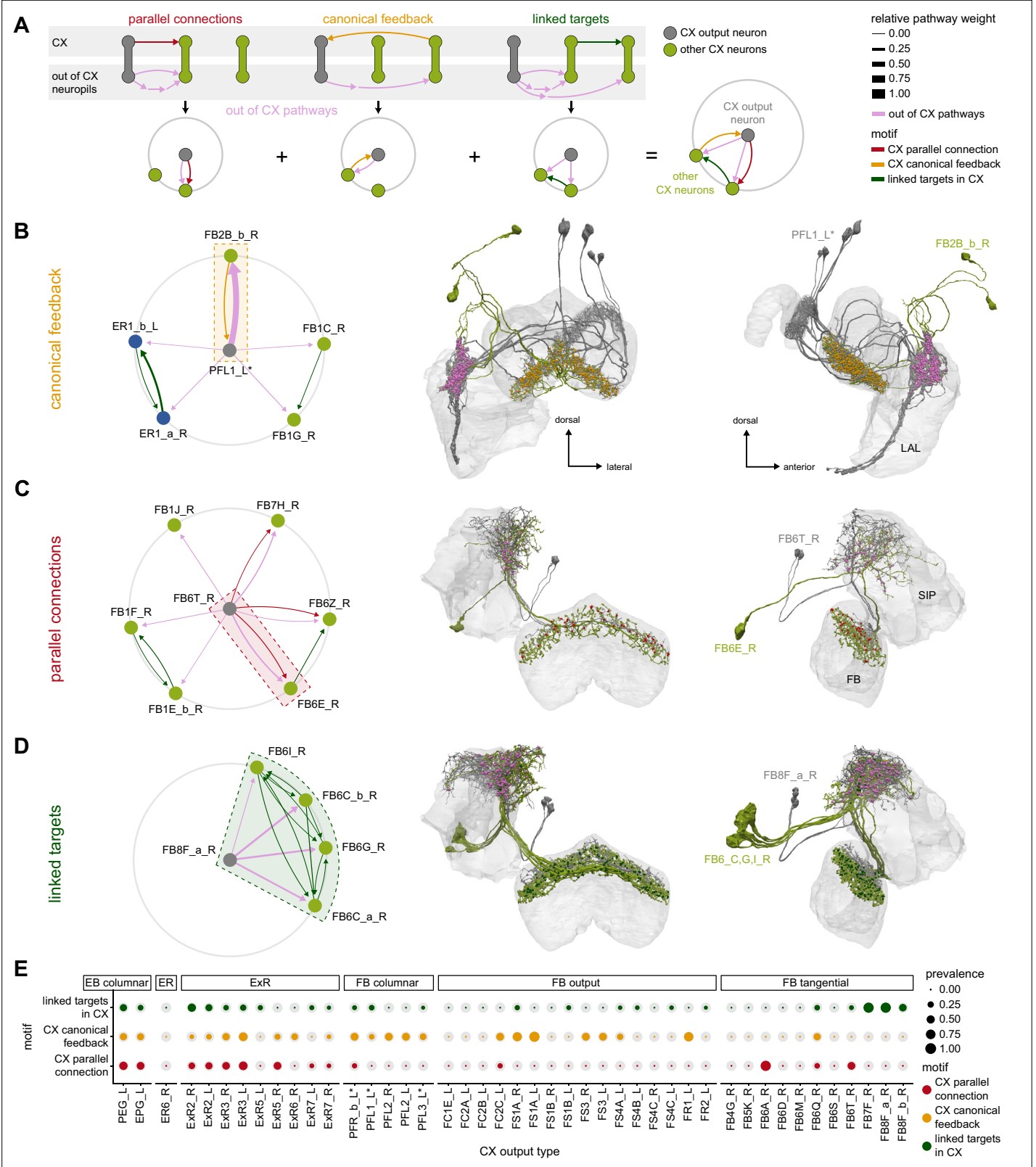

**Figure 58.** Central complex (CX-to-CX) motifs. (**A**) Schematic of the three motifs considered and their equivalent representation in a compact circular network plot. The CX output type of interest is in gray and at the center of the circular diagram. It reaches other CX neurons (green) through pathways that can be constituted of multiple steps (pink). Motifs are formed by the relation between those pathways outside of the CX and the connections formed inside of the CX by the same CX neurons. 'Canonical feedback' corresponds to the target of the pathway contacting the source type in the CX

*Figure 58 continued on next page*

Figure 58 continued

(yellow). 'Parallel connections' occur when the source neurons also contact the pathway target neuron inside the CX (red). 'Linked targets' are neurons connected in the CX that are targets of the same neuron outside of the CX (green). The equivalent circular plot is provided below each motif, and their combination in a single polar plot is on the right. (**B–D**) Example motifs. Left: circular motif graph showing the motif in the context of all the CX-to-CX motifs that the type of origin is implicated in. Right: frontal and lateral morphological renderings. (**B**) FB2B_b forms a canonical feedback loop with PFL1 neurons. PFL1 contacts FB2B_b in the LAL while FB2B_b contacts PFL1 in the fan-shaped body (FB). (**C**) Parallel connections. FB6T contacts FB6E both in the SIP/SMP and FB. (**D**) Linked targets. FB8F_a contacts four FB6 neurons who are themselves interconnected in the FB. (**E**) Prevalence of the three motifs for all the potential CX output types. The colored circles represent the prevalence of each specific motif. The gray circles represent the total number of all the motifs of the same type that could form given that type's partners outside of the CX. Ellipsoid body (EB) columnar, ExR2, and ExR3 neurons form a large proportion of all the possible motifs, reflecting the high level of recurrence in EB circuits.

in the FB (*Figure 58D*). This linked target motif could allow a CX output neuron to regulate the activity of a group of CX neurons as a whole.

Neurons that participate in any one of three motifs are usually part of a larger network containing several motifs (*Figure 58B–D*, left panels). However, not all CX output neurons participate equally in all three motifs (*Figure 58E*), and this may reflect different functional roles. In the EB, EPG, PEG, and ExR2 and ExR3 neurons participate in all three motifs. This is likely a consequence of the high degree of recurrence between EB columnar, ring, and some ExR neurons. Such recurrence could help sustain ring attractor dynamics in the EB-PB network. In the FB, the columnar neurons making the strongest contributions to downstream networks outside the CX (PFL, FS, and PFR_b neurons) almost exclusively form canonical-feedback motifs. These neuron types constitute the main channels by which the CX communicates with the rest of the brain and likely modulates the fly's actions; feedback inhibition – if indeed it is inhibition – could enable greater temporal precision and faster switching between different actions while also controlling the amplitude of these outputs. Finally, the linked-target motif is predominant in the dorsal layers of the FB. This motif may be involved in controlling dedicated modules associated with behavioral state and sleep (see 'Discussion').

## CX projections to brain areas outside the CX

Pathways that leave the CX and send information on to other brain regions are necessary for the CX to exert its influence on the fly's behavior. For example, during sleep, the CX is thought to trigger the consolidation of courtship memories by driving DANs that project to the MB (*Dag et al., 2019*). The CX also influences the fly's wakefulness and activity levels based on internal states, such as circadian rhythm (*Liang et al., 2019*) and the need for certain nutrients (*Sareen et al., 2020*). To this end, the CX is well known to play an important role in the initiation and direction of movement (*Bender et al., 2010*; *Guo and Ritzmann, 2013*; *Harley and Ritzmann, 2010*; *Kathman et al., 2014*; *Krause et al., 2019*; *Martin et al., 2015*; *Poeck et al., 2008*; *Rayshubskiy et al., 2020*; *Strauss, 2002*; *Strauss and Heisenberg, 1993*; *Triphan et al., 2010*; *Triphan et al., 2012*). Besides directly regulating motor pathways, the CX may also exert its influence by modulating the gain and tuning of early pathways.

Different CX neurons make vastly different contributions to downstream networks (*Figure 59A*), reflecting their unequal number of synapses (*Figure 54C*), number of downstream targets (*Figure 55E*), and contribution to CX-to-CX loops (*Figure 57A*, *Figure 57—figure supplement 1A*). The PFL, PFR_b, and FS4 neurons are the strongest contributors to pathways external to the CX, followed by ExR8, FR, and FS1-3 neurons. FC neurons, ExR7, ExR3, and a handful of FB tangential neurons make weak contributions. All other CX neurons innervating structures outside of the CX that we considered are purely involved in CX-to-CX loops. For the sections that follow, note that our efforts to characterize networks downstream of the CX were limited by our inability to identify neurons with projections outside the hemibrain volume, and also by the limited characterization of neurons in less structured brain regions like the CRE, WED, or SMP. Indeed, for all CX output types, most of the downstream targets were in such less studied neuropils, limiting our ability to extract functional insight (*Figure 59A*).

The majority of neurons in networks downstream of the CX receive only a very small contribution from the CX (*Figure 59B*). We focused on analyzing the neuron types that receive more than 0.5% of their inputs from pathways originating in the CX. Plotting the weights of pathways from CX neurons onto strong downstream targets (see Materials and methods) and clustering them by their inputs revealed that these targets are largely segregated. The majority of external targets receive significant input from only one CX type (*Figure 59C*). This segregation of targets stems from the anatomical

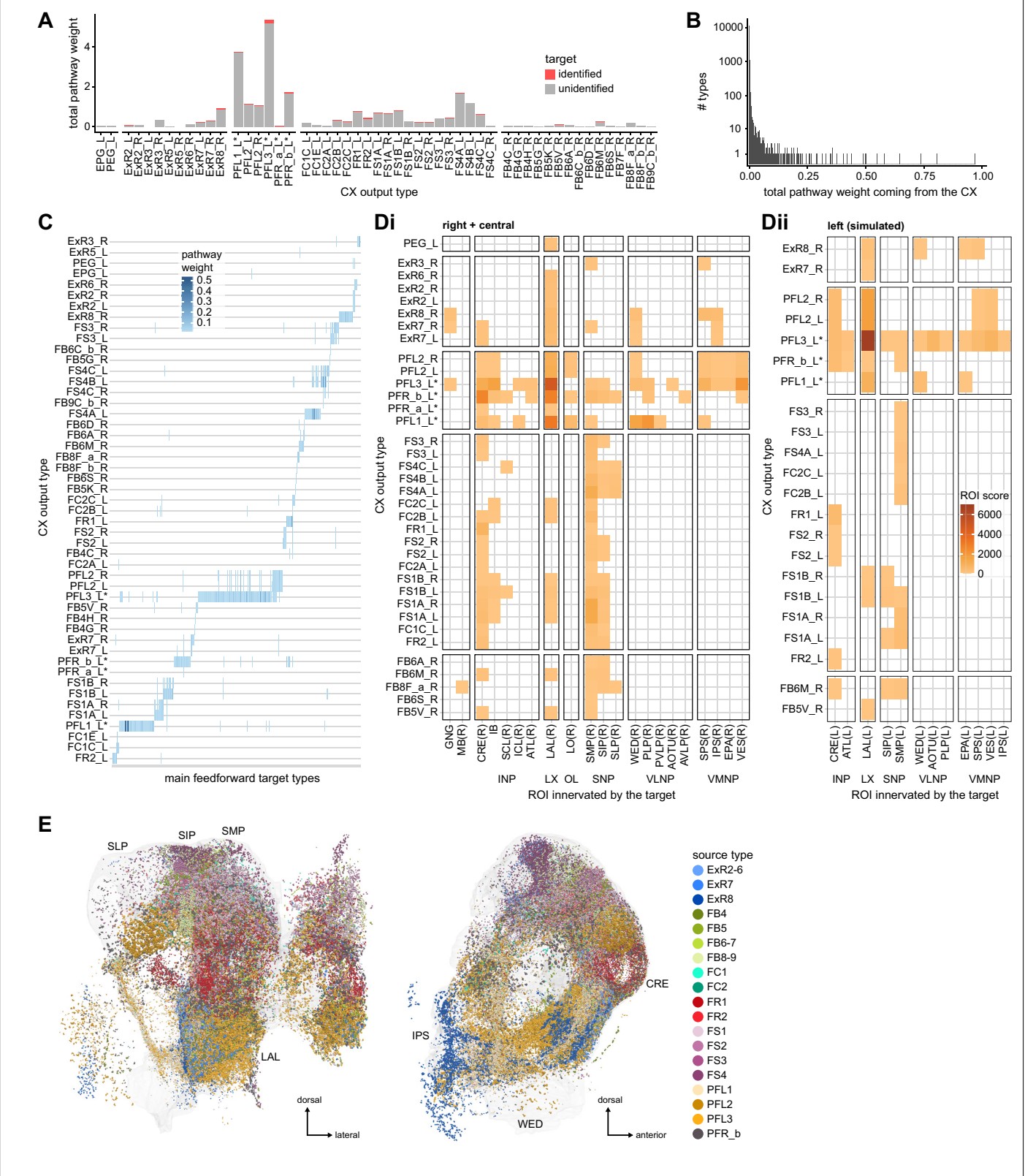

**Figure 59.** Feedforward output networks. (**A**) Total pathway weights contributed by the different central complex (CX) output neurons (summed over all neurons in the graph, see Materials and methods and *Appendix 1—figure 3*). Color indicates if the receiving types are identified or not (unidentified means that they are part of poorly studied neuropils or unknown). Compare to *Figure 54C*. some types (e.g., FR and FS neurons) reach a large number of neurons but have much lower pathway weights than other prominent types (for example, PFL neurons). This discrepancy arises because these types

*Figure 59 continued on next page*

*Figure 59 continued*

only make modest contributions to their targets. (**B**) Histogram of total pathway weight received from the CX for every downstream type. Most neurons receive very weak inputs from the CX. Note the log scale, without which the handful of neuron types receiving strong CX contributions would be invisible. (**C**) Connectivity matrix from the CX to every type in the downstream network graph receiving more than 0.5% total pathway weight (filtered for individual weights > 0.5%). The CX output types are ordered according to the similarity of their output vectors (see Materials and methods). Most important targets are influenced mainly by a single CX type. PFL3 neurons contact more types than any other CX type. Note that for convenience of display the names of the targets are not displayed on the x-axis. (**D**) Downstream neuropil innervation of the targets of CX output neurons, starting on the right side of the brain. The innervations are weighted by the pathway weight they receive. The region of interest (ROI) score is the sum of pathway weights for all the target types innervating the ROI times their number of downstream synapses in the same ROI. (**Di**) Measured innervations (right and central neuropils). (**Dii**) Simulated innervations (left neuropils) from the known symmetric types. This is necessarily an underestimate of the extent to which the CX pathways reach the contralateral side of the brain. (**E**) Downstream synapses of targets receiving more than 0.5% of pathway weight from the CX, colored by the CX type contributing the most to their inputs.

The online version of this article includes the following figure supplement(s) for figure 59:

**Figure supplement 1.** Clustering at different depths.

**Figure supplement 2.** Modularity of output networks.

**Figure supplement 3.** Same as *Figures 54B and 59E* for PFL, FS, and FC neurons alone.

**Figure supplement 4.** Same as *Figures 54B and 59E* for FR, PFR, and ExR neurons alone.

**Figure supplement 5.** Neuron-to-neuron output connectivity of the main columnar output neurons.

---

segregation of the CX neurons synapses (*Figure 54B*) and is maintained several synapses downstream (*Figure 59—figure supplement 1*). We found that this occurs because each CX output type targets a distinct 'module,' a set of neurons much more connected to each other than to neurons of other modules (*Figure 59—figure supplement 2*, *Videos 17–22*). It is therefore likely that CX output channels differ in the information that they carry, with each channel serving the distinct functional needs of its downstream circuits. This segregation raises the intriguing possibility that different output neuron types may control distinct sets of behaviors (see 'Discussion').

When mapping the CX downstream networks at the scale of the brain region (*Figure 59D*), this segregation is less apparent. Even if each type targets a distinct set of regions, many CX output networks cover similar regions. The CRE, the LAL, the SMP, and SIP are reached by most FB output neurons, whereas both the PFL and ExR7/8 neurons target the ventral neuropils. When plotting the synapses of those downstream targets, the finer-scale segregation by downstream target type is partially visible as subclusters within brain regions (*Figure 59E*, *Figure 59—figure supplements 3 and 4*). The emerging picture is one where every CX output neuron type targets a relatively strongly interconnected subnetwork that is only weakly linked to the target subnetworks of other CX output neurons (*Figure 59—figure supplement 2*). Finally, columnar organization is lost at the output stage. All neurons of each columnar type converge onto the same neurons (*Figure 59—figure supplement 5*). This suggests that heading or head direction information may be lost in downstream partners and that the output

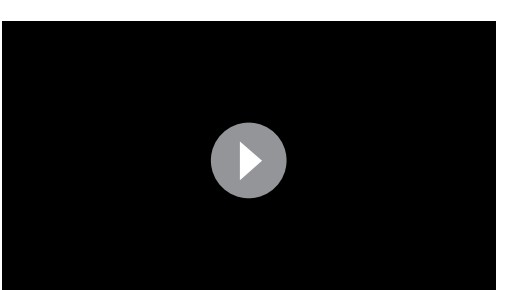

**Video 17.** PFL1 targets. Morphological renderings of the neurons in each output module. The video follows the Figure 74 diagram from top to bottom. For each CX output neuron, the subpart of the diagram corresponding to that neuron is shown on the right, the renderings on the left. The CX neuron is shown first, column by column if necessary, then the main target neurons in its output networks are shown in rapid succession before being shown together. When the CX neuron projects to several output modules, those are shown sequentially, with different colors.

https://elifesciences.org/articles/66039/figures#video17

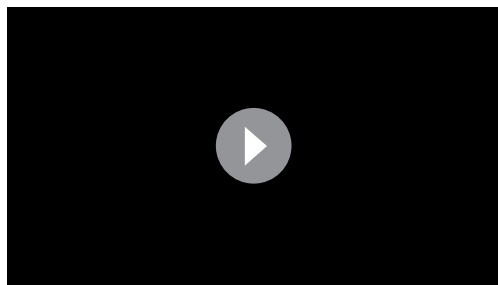

**Video 18.** PFL2 and PFL3 targets.

https://elifesciences.org/articles/66039/figures#video18

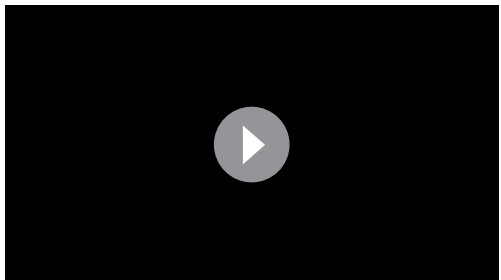

**Video 19.** FS3 and FS4 targets.
https://elifesciences.org/articles/66039/figures#video19

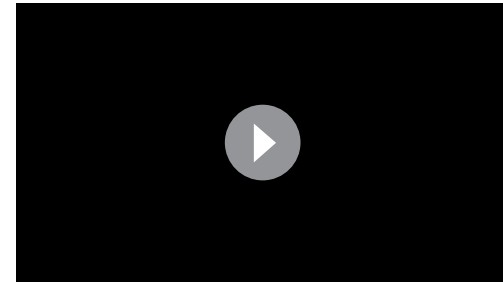

**Video 21.** FC1 and FC2 targets.
https://elifesciences.org/articles/66039/figures#video21

modules could act as simple functional units (see 'Discussion').

## Known neuron types reached by CX output pathways

Although the majority of pathways that exit the CX lead to poorly characterized neurons, we did find many well-identified targets (*Figure 60A and B*). These targets include neuromodulatory or peptidergic neurons, MBONs, vPNs, and DNs, among others. Of these, some of the most prominent are MBONs receiving input from FR, FS, PFR_b, and FB8F_a pathways, vPN neurons receiving input from ExR8 and PFL3 pathways, and the DNs targeted by PFL2 and PFL3 pathways (*Figure 60C*). We discuss these pathways in more detail below.

## CRE and SMP connections to MBONs and DANs

As discussed in an earlier section, the MB is a highly conserved center for associative learning and memory. These functions are, in part, mediated by interactions between MBONs and DANs. In the previous section (*Figures 46 and 47*), we focused on direct and indirect inputs from MBONs to the CX. We now turn our attention to information flow in the reverse direction. Such interactions could enable the CX to trigger the consolidation of courtship memories by driving DANs that project to the MB (*Dag et al., 2019*). More generally, these connections could play a role in modulating the learned behaviors that the MBONs are thought to drive (*Aso et al., 2014b*).

The FR, FS1, FS2, and PFR_b neuron types, and one type of FB tangential neuron type (FB8F_a), all send outputs to a weakly connected subnetwork of PPL DAN and MBON neuron types. This network is mostly located in the CRE and SMP (*Figure 61A–B*, *Video 20*). In that network, two MBON types receive direct input from one CX neuron type each: MBON30, from FR1 neurons (*Figure 61D*, also discussed in *Li et al., 2020*), and MBON27, from FS1B neurons. MBON27 and MBON30 are both 'atypical MBONs,' which receive some of their synaptic inputs outside the MB. This is to be expected as no CX neurons project directly to the MB. Within this network, single PPL and MBON neurons form small local subcircuits, interconnected by uncharacterized CRE and SMP neurons (*Figure 61B–E*). Interestingly, most of the prominent dopaminergic targets of the CX are PPL neurons innervating the MB. Only one PPL neuron in that set does not innervate the MB (PPL107). Neurons from the PAM cluster receive much weaker contributions (*Figure 61A*). PPL neurons are thought to carry punishment signals

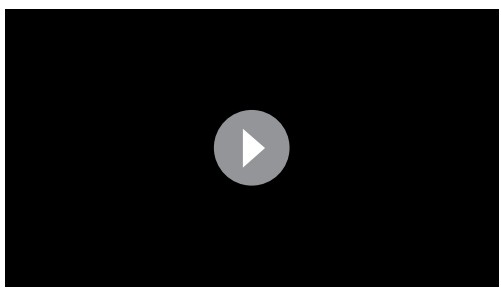

**Video 20.** FS1, FS2, FR, and PFR targets.
https://elifesciences.org/articles/66039/figures#video20

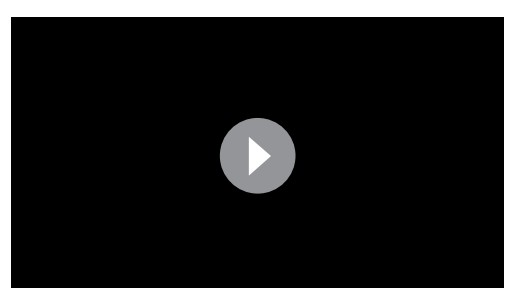

**Video 22.** Exr7 and ExR8 targets.
https://elifesciences.org/articles/66039/figures#video22

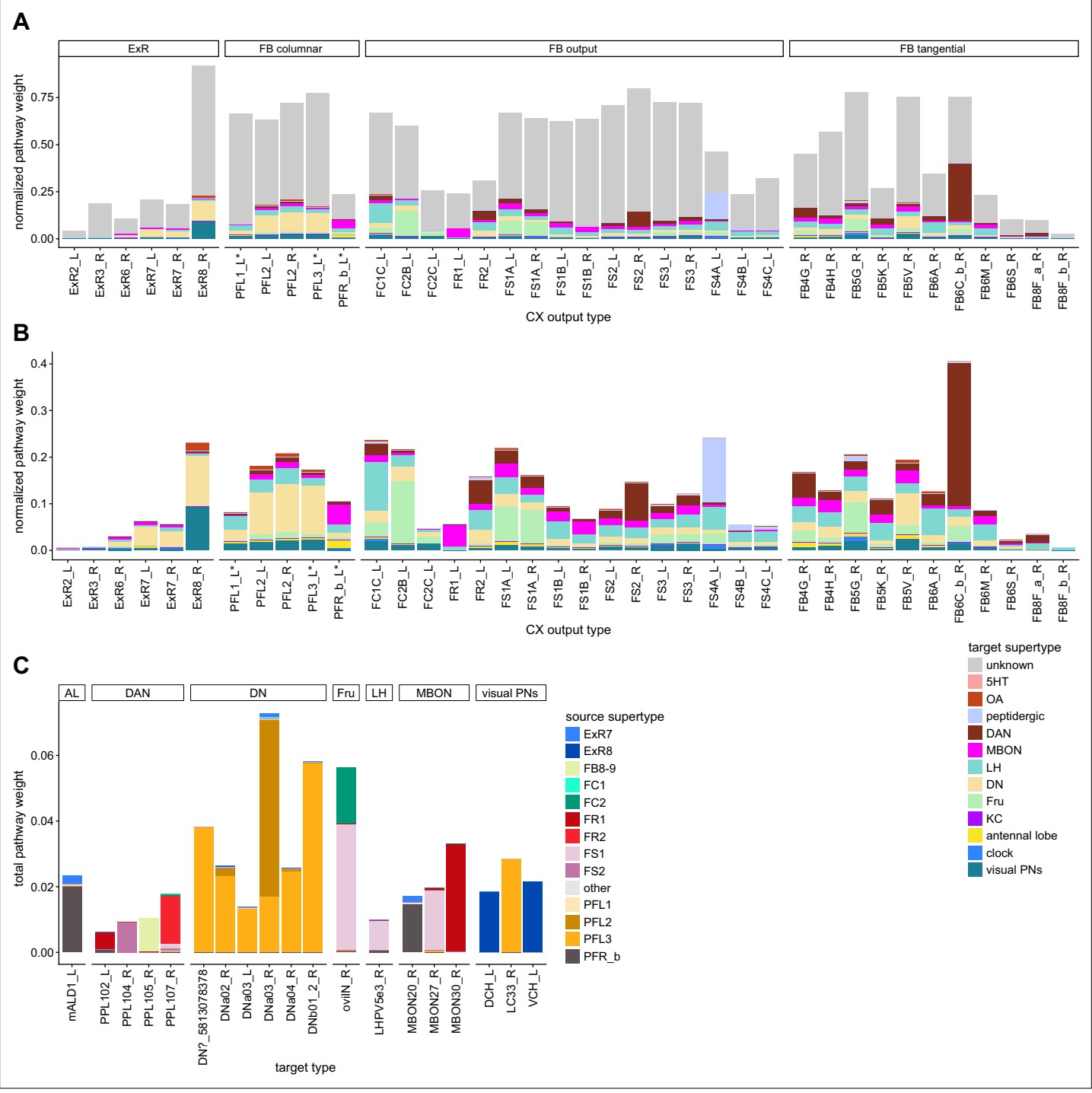

**Figure 60.** Connections to identified types. (**A**) Normalized pathway weights (see Methods *Appendix 1—figure 3* and Materials and methods) of pathways that end on various known types. Only central complex (CX) neurons that contribute at least 0.5% of their outputs to feedforward networks are included. 'Unknowns' correspond to pathways that never reach a known type. Note that some of the known groups are still very broadly defined. For example, the group labeled 'LH' contains a lot of functionally diverse neurons with branches in the lateral horn (one of which is the WPN neuron mentioned in the mechanosensory inputs section). (**B**) Same as (**A**), but zoomed in on the known types. (**C**) Pathway weights from the CX received by the most prominent known targets (getting at least 0.5% of their inputs from CX pathways), colored by the CX types of origin.

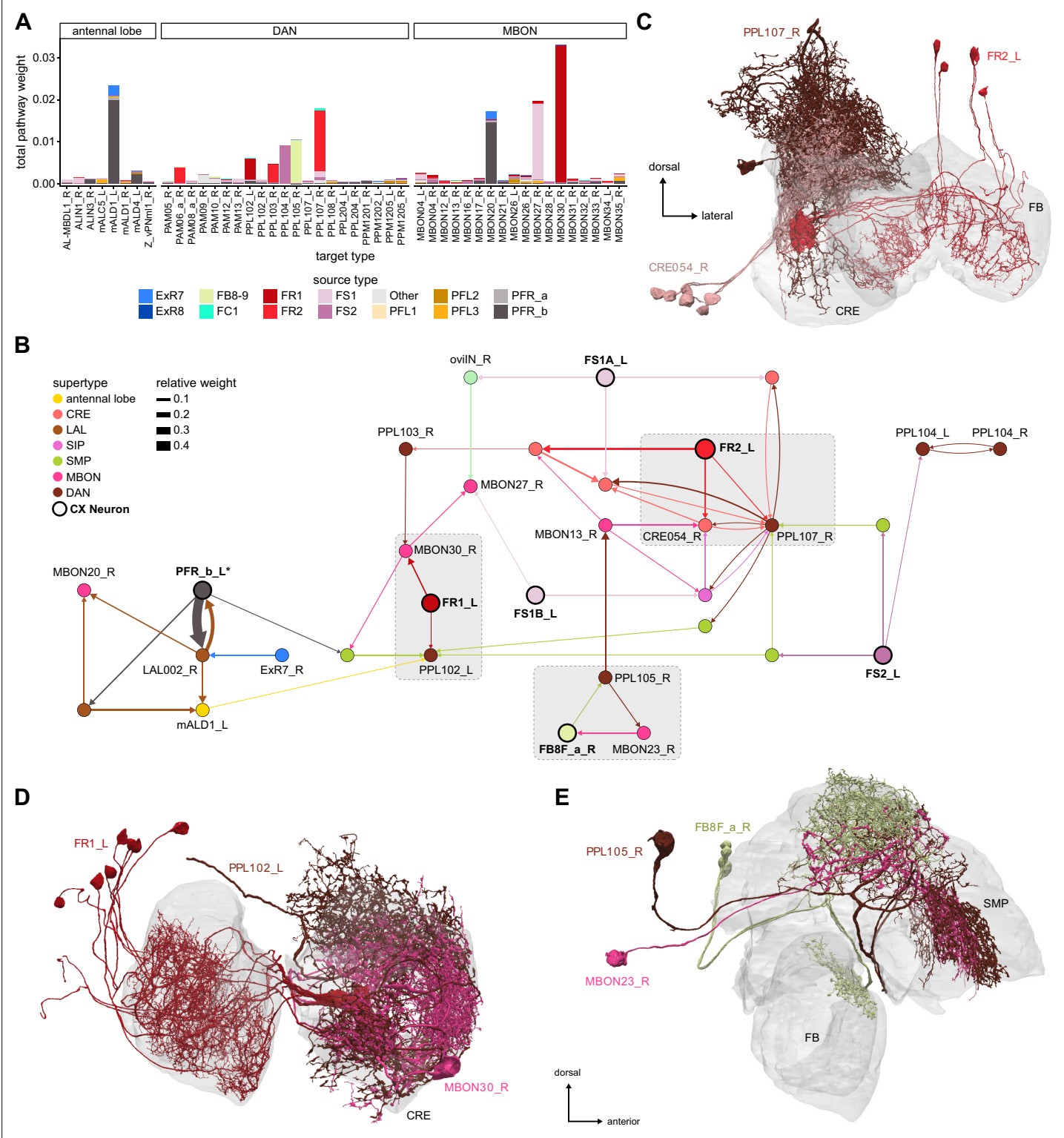

**Figure 61.** Connections to mushroom body output neurons (MBONs), dopaminergic, and antennal lobe neurons. (**A**) Pathway weights (see *Appendix 1—figure 3*) from the central complex (CX) to MBONs, dopaminergic (DANs), and antennal lobe neurons for which the total pathway weight is greater than 0.05%, colored by the CX type of origin. (**B**) Network diagram showing the interconnections between those pathways. Grayed areas correspond to the morphological renderings in (**C–E**). CX neurons have bold labels and circles. (**C**) FR2 to PPL107 and CRE054, morphological rendering. (**D**) FR1 to PPL102 and MBON30, morphological rendering. (**E**) FB8F_a/MBON23/PPL105 loop, morphological rendering.

The online version of this article includes the following figure supplement(s) for figure 61:

**Figure supplement 1.** Main circuits converging onto oviIN and MBON27.

in the MB (*Aso and Rubin, 2016*), in contrast with the reward-associated PAM cluster. It would therefore appear that the CX could modulate learning in the MB by preferentially targeting punishment signaling neurons. Since this modulation is coming from columnar neurons of the FB, it is possible that the CX modulates punishment signals in an orientation-dependent manner.

Overall, it is unclear what distinguishes the MBON neurons targeted by the CX. The MBONs in this subnetwork are not among the strongest MBON inputs to the CX (*Figures 46 and 47*). There is therefore no strong recurrent loop between the CX and the MB. One interesting exception is MBON30, which is weakly connected to some LAL-NO neurons (*Figures 46 and 47*). MBON27, on the other hand, is known to receive inputs from visual KCs in the MB and projects to DNs in the LAL (*Li et al., 2020*). Finally, MBON20 neurons, which are reached by PFR_b through a somewhat separate network (*Figure 61B*), are themselves strongly linked to two DNs, DNp42 and DNb05 (*Li et al., 2020*). DNp42 is required for innate aversive olfactory behavior (*Huoviala et al., 2020*). Besides the preferential targeting of punishment associated DANs, this is another indication that the CX to MB link may modulate aversion.

Two other neurons are associated with this CX to MBON subnetwork: mALD1, a giant AL neuron of unknown function (*Figure 62—figure supplement 1C*) and the large inhibitory oviposition interneuron oviIN (*Figure 61B*, *Figure 61—figure supplement 1*; *Wang et al., 2020*). These large neurons lie at the intersection of MB and CX outputs, where they can be flexibly modulated by both; this may be ideal for their potential role in triggering or modulating entire behavioral programs like oviposition.

## Interactions with visual projection pathways

Few of the vPNs, outputs from the optic lobes to the central brain (*Mu et al., 2012*; *Panser et al., 2016*; *Wu et al., 2016*), reach the CX. Instead, many vPNs interact fairly directly with motor pathways (*Namiki et al., 2018*). Nevertheless, we found links between CX outputs and a few vPNs, suggesting that the CX may selectively modulate specific, direct visuomotor pathways.

Four types of CX output neurons (PFL1, PFL3, PFR_b, and ExR8 neurons) interact with visual pathways (*Figure 62A*, *Figure 62—figure supplement 1*, *Videos 17–18*). Pathways originating from the PFL1, PFL3, and PFR_b neurons all target the output areas of a subset of vPNs. The PFL3 and PFL1 neurons reach three lobula columnar (LC) neurons (*Figure 62B–G*), whereas PFR_b indirectly contacts a range of lobula and medulla neurons (*Figure 62—figure supplement 1*). In all these cases, connections are axo-axonal (*Figure 62C, E and G*), meaning that these CX pathways likely regulate the output of the vPNs. Moreover, these connections are often reciprocal, which indicates that the vPNs also regulate the CX output pathways that target them. Finally, we also found that despite being reciprocally connected the CX output pathways only share a small fraction of their outputs with the vPNs they target (*Figure 62—figure supplement 2*). In short, FB output pathways and a select set of direct visuomotor pathways have the potential to influence each other's outputs. These reciprocal connections may allow the direct visuomotor pathways and indirect CX-mediated pathways to compete for control of the fly's actions.

Most of the vPNs targeted by PFL1, PFL3, and PFR_b neurons are columnar in the optic lobes and project to well-defined optic glomeruli. These kinds of neurons are generally thought to convey information about specific visual features (*Wu et al., 2016*), but the details of which neurons convey which features are still incomplete. Of the neurons found here, only one type, LC10, has been investigated for its function. LC10 neurons, which interact with PFL3 pathways in the AOTU (*Figure 62D and E*), have been shown to be essential to the small object tracking system used by males to follow females during courtship (*Ribeiro et al., 2018*). The interaction between PFL3 and LC10 pathways could therefore prioritize changes of direction driven by the small object tracking system versus those driven by the CX. The other LC type in contact with PFL3, LC33 (*Figure 62B and C*), was first described in the hemibrain dataset (*Scheffer et al., 2020*). Interestingly, in both cases, PFL3 pathways are only connected with a subset of the LC neurons of a given type, suggesting that the visuomotor pathways influenced by (and influencing) the CX output pathways have specificities beyond the feature specificity conferred by individual vPN types. The LC type in contact with PFL1 pathways, LC27 (*Figure 62F and G*), was also first described in the hemibrain dataset (*Scheffer et al., 2020*). Given the predominance of the PLP and PVLP in PFL1 outputs (*Figure 59Di*), and the fact that the optic glomeruli are positioned at the PVLP and PVLP/PLP boundary, it is possible that PFL1 output pathways interact with other LC neurons that are not presently identified or that are only partially traced in the volume.

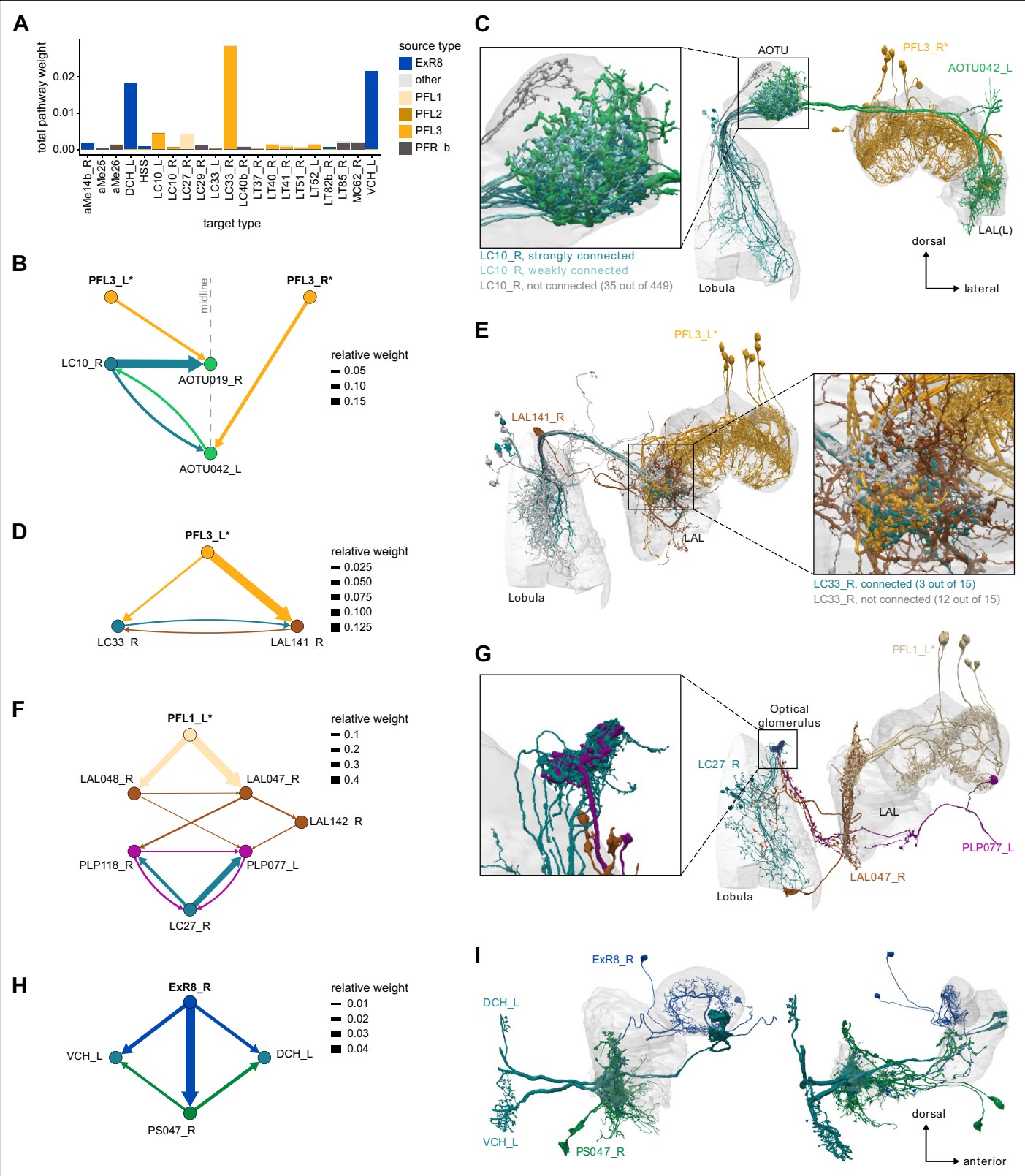

**Figure 62.** Connections to visual projection neurons (vPNs). (**A**) Total pathway weights (see Materials and methods and *Appendix 1—figure 3*) from the central complex (CX) to vPNs for which the total pathway weight is greater than 0.05%, colored by the CX type of origin. (**B, C**) PFL3 neurons interact with LC10 neurons through anterior optic tubercle-lateral accessory lobe (AOTU-LAL) neurons. (**B**) Network diagram showing how PFL3 neurons interact with LC10 neurons both ipsilaterally and contralaterally through AOTU-LAL neurons. The midline is denoted by the vertical dotted line. (**C**)

*Figure 62 continued on next page*

*Figure 62 continued*

Morphological rendering showing that PFL3 interacts with LC10 along a dorsoventral axis in the AOTU, corresponding to the anteroposterior axis in the lobula. Connections are stronger on the ventral side of the AOTU, corresponding to LC10 neurons innervating the posterior part of the lobula. (**D, E**) PFL3 neurons interact with a subpopulation of LC33 neurons. (**D**) Network diagram. PFL3 neurons synapse onto LC33 in the LAL both directly and through one of its strong targets (LAL141). (**E**) Morphological rendering showing which subset of LC33 neurons is associated with PFL3 neurons. (**F, G**) PFL1 neurons interact with LC27 neurons. (**F**) Network diagram showing that the connection is through two layers constituted by LAL and PLP neurons, respectively. (**G**) Morphological rendering showing that the PLP neurons downstream of PFL1 specifically target the LC27 glomerulus. (**H, I**) ExR8 neurons contact ventral centrifugal horizontal (VCH) and dorsal centrifugal horizontal (DCH), centrifugal neurons of the horizontal fiber system. (**H**) Network diagram showing that ExR8 reaches CH neurons both directly and indirectly through a PS neuron. (**I**) Morphological rendering. PS047 innervation closely follows ExR8 innervation.

The online version of this article includes the following figure supplement(s) for figure 62:

**Figure supplement 1.** PFR_b-to-visual projection neurons (PNs) connections.

**Figure supplement 2.** PFL3 and LC33 neuron-to-neuron connectivity.

In contrast with those axo-axonal connections made between FB output pathways and visual columnar neurons, the ExR8 output pathway contacts 'centrifugal' (CH) visual neurons, that project from the central brain to the optic lobes. Specifically, ExR8 reaches the dorsal centrifugal horizontal (DCH) and ventral centrifugal horizontal (VCH) neurons in the PS (*Figure 62H1*, *Video 22*). CH neurons, whose response properties have been characterized in the blowfly (*Hausen, 1976*), are part of the horizontal-motion-sensing network of the lobula, where they are both pre- and postsynaptic. However, these neurons receive the majority of their inputs in the central brain. In blowflies, the VCH and DCH neurons are both nonspiking and inhibitory, and they confer their motion sensitivity to at least one of their downstream targets, the small-object-motion-sensitive FD1 neuron (*Egelhaaf et al., 1993*). All these neurons respond to motion both in the ipsilateral (front-to-back) and contralateral (back-to-front) direction. The FD1 neuron, in particular, has been hypothesized to suppress motion responses during saccades (*Hennig et al., 2011*). The ExR8 to CH projection is therefore one place where the CX can directly influence sensory processing (in this case, visual motion processing). The ExR8 have not yet been functionally characterized, but are unusual for an ExR neuron type in that they seem like 'true' CX output neurons (*Figure 57A*). In addition to its EB inputs, the ExR8 neuron type receives input in the ipsilateral NO1 from PEN_b neurons (*Figure 25B*). ExR8 could therefore relay information about angular velocity from the fly's movements to circuits that detect visual motion and potentially suppress the visual responses that are the consequences of the animal's own movements.

## CX outputs to the motor system

As mentioned above, the CX is known to play a role in determining the fly's movements. Movements are controlled through DNs, which carry motor commands from the central brain to the VNC (*Hsu and Bhandawat, 2016*; *Namiki et al., 2018*). Light microscopy has enabled the identification of a few LAL-projecting CX neuron types that may link the CX to DNs (*Hanesch et al., 1989*; *Heinze and Homberg, 2009*; *Lin et al., 2013*; *Wolff et al., 2015*). A complete connectome should, in principle, allow us to identify all CX to DN neural pathways. However, the limited volume of the hemibrain has permitted at most one-third of these neurons to be identified (*Scheffer et al., 2020*). We used the identified DNs to focus our analysis of CX output neurons, while noting that a larger number of CX output neurons that project to other brain areas and to as-yet-unassigned neural segments might well make contributions to DNs that were difficult to identify in the hemibrain volume.

The main source of CX input to DNs comes from PFL2 and PFL3 neurons, with additional smaller contributions from ExR7, ExR8, and FR2 neurons (*Figure 63A*). Multiple DN types receive strong PFL2/PFL3 inputs. Most prominent among these are DNa02, DNa03, DNa04, and DNb01 neuron types, as well as a putative DNg neuron (5813078378). DNa02 is known to be involved in turning in walking flies (*Rayshubskiy et al., 2020*). DNa02, DNa03, DNa04, and DNb01 share inputs to varying degrees, suggesting that they could be part of an interacting premotor network. Since head direction- and heading-related information present in the FB is lost at the first synaptic relay in the LAL (*Figure 59—figure supplement 5*), the only obvious simple way asymmetries can arise downstream of the PFL neurons are through uneven innervation on the right or left side of the brain. All PFL2 neurons project to identical neurons on both sides of the brain, whereas left PFL3 neurons only contacts neurons in the right LAL, and vice versa. Consequently, the DN network is influenced symmetrically

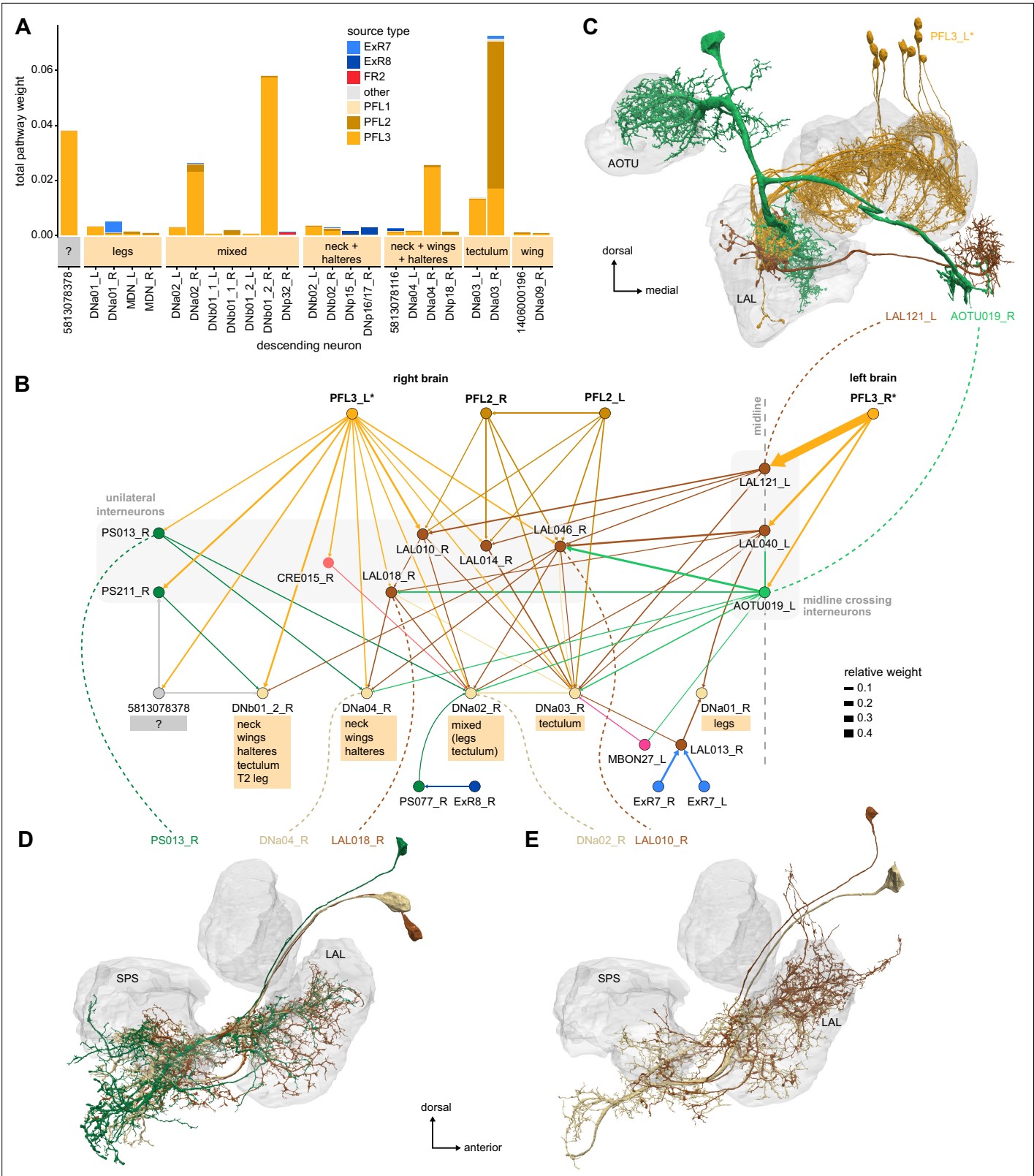

**Figure 63.** Connections to descending neurons. (**A**) Pathway weight (see *Appendix 1—figure 3* and Materials and methods) from the central complex (CX) to descending neurons (DNs) for which the total pathway weight is greater than 0.05%, colored by CX type of origin and separated by their putative ventral nerve cord (VNC) innervation. (**B**) Network diagram of the main CX to DN connections, restricted to DNs on the right side of the brain. DN connections primarily come from PFL2_R and PFL3 neurons, with smaller contributions made by ExR7 and ExR8 neurons. PFL2 neurons reach DNs

*Figure 63 continued on next page*

*Figure 63 continued*

through ipsilateral networks while PFL3 neurons also reach them via lateral accessory lobe (LAL) interneurons crossing the midline. Much of the circuit is shared between PFL2 and PFL3 pathways. VNC innervations for each DN type are indicated below, highlighting their diversity. (**C**) Morphological rendering of PFL3_R neurons with two of its midline crossing targets, AOTU019_R and LAL121_R. (**D**) Morphological rendering of PS013_R, DNa04_R and LAL018_R. Note how LAL018 innervations follow those from DNa04. (**E**) Morphological rendering of DNa02_R and LAL010_R.

The online version of this article includes the following figure supplement(s) for figure 63:

**Figure supplement 1.** Other connections to descending neurons (DNs).

by PFL2 neurons and asymmetrically by PFL3 neurons. This is consistent with the hypothesis that PFL2 neurons control forward walking and/or fixation in flight, and that PFL3 neurons control turning. In such a scheme, turns in walking flies would be controlled by asymmetric modifications of the fly's gait (*DeAngelis et al., 2019*; *Strauss and Heisenberg, 1990*) and turns in flight by asymmetric changes in wing kinematics (*Muijres et al., 2015*), both modulated by PFL3 neurons. No other CX output neurons appear to be in a position to execute such a function (also see 'Discussion'). It is interesting to note that the DN neurons in question innervate a variety of leg and wing neuropils in the VNC (*Namiki et al., 2018*), suggesting that control of different actuators is somewhat integrated or coordinated in the LAL. PFL2 and PFL3 neurons also reach the moonwalker neuron MDN (*Bidaye et al., 2014*; *Feng et al., 2020*), which has a bilateral innervation pattern and is known to drive backward walking. This connection is almost exclusively contralateral (*Figure 63—figure supplement 1A and B*), a suggestion that the MDN could also be involved in asymmetric behaviors.

PFL2 and PFL3 connections to their downstream DN targets are both direct and indirect through a LAL interneuron network (*Figure 63B–E*, *Video 18*). The LAL targets of PFL2 neurons on a given side contribute mostly to ipsilateral networks, while PFL3 neurons tend to also reach contralateral DNs through strong connections to midline crossing LAL neurons. Such neurons have been shown to function in flip-flop circuits mediated by inhibition in the silkworm moth brain (*Iwano et al., 2010*) and could participate in modulating left-right asymmetries in the activity of the output network (see

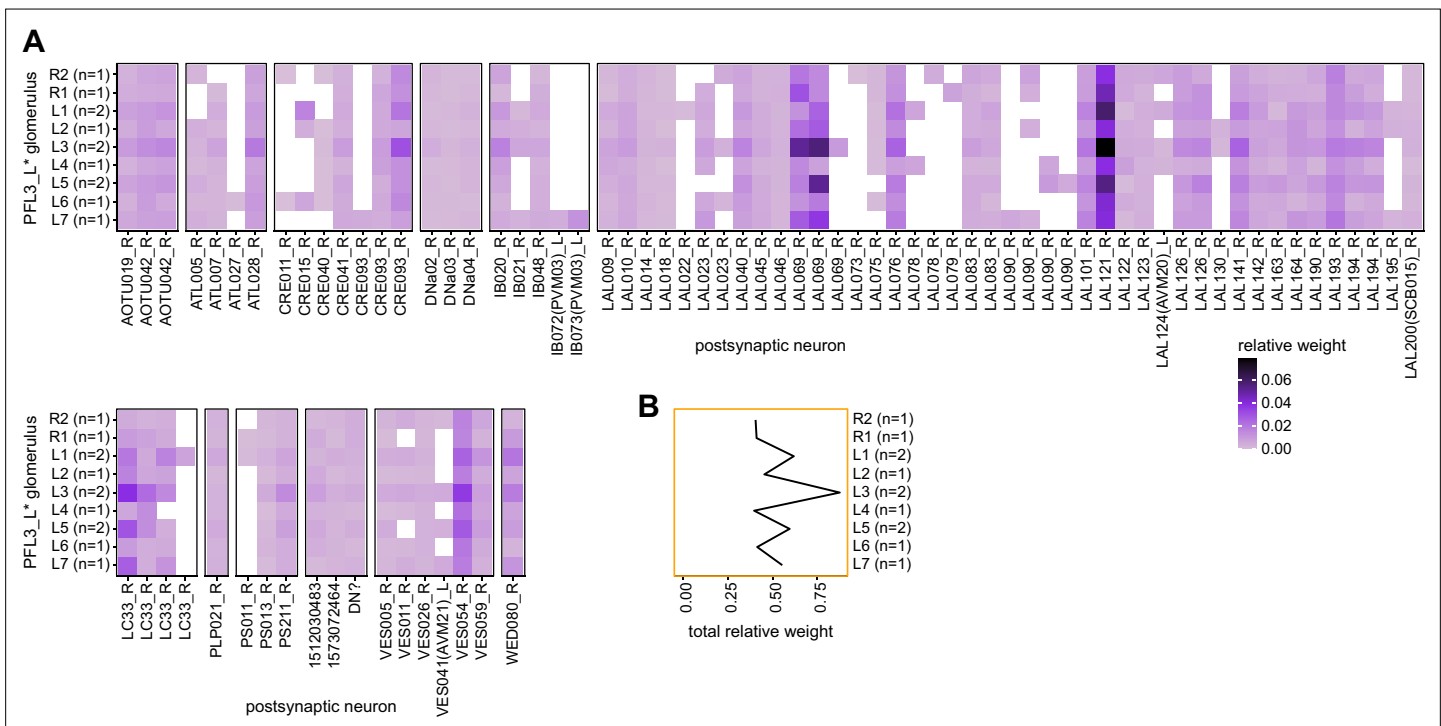

**Figure 64.** PFL3 outputs distribution. (**A**) Connectivity matrix between PFL3_L and its direct downstream partners outside of the central complex (CX), on the right side of the brain. PFL3 neurons are binned by protocerebral bridge (PB) glomerulus (numbers of neurons per glomerulus are indicated in parenthesis on the y-axis). (**B**) Sum of the relative weights across glomeruli. Connections are strongest for glomerulus L3.

'Discussion'). More generally, the heavily recurrent LAL networks could participate in the integration and coordination of activity between the different DNs.

ExR7 also indirectly reach some DNs in this network (*Figure 63B*). Additionally, we observed that ExR8 neurons target, through PS neurons (*Figure 63—figure supplement 1C–E*), a different set of DNs thought to innervate neck and haltere neuropils (*Namiki et al., 2018*). This connection, taken together with the ExR8 connection to the CH neurons (*Figure 62H and I*), suggests that ExR8 could play a role in controlling head movements and their interplay with optic flow signals.

Finally, FR2 reaches DNp32 through a SMP interneuron (*Figure 63—figure supplement 1F*). It is likely that a lot of similar connections in the dorsoposterior part of the brain are missed because they lie outside the hemibrain volume.

## Asymmetries in the distribution of columnar outputs

As mentioned previously, the columnar identity of the PFL neurons is lost in their targets in the LAL (*Figure 59—figure supplement 5*). However, we did notice that PFL neurons that innervate certain PB glomeruli consistently provide stronger inputs to their downstream targets in the LAL than PFL neurons coming from other glomeruli. *Figure 64* shows that the outputs of the left PFL3 neurons are strongest for the individual PFL neurons that innervate PB glomerulus L3. This observation can partly be explained by the difference in the number of neurons that innervate each glomerulus (*Figure 24—figure supplement 1*). By symmetry, R3 innervating PFL neurons would similarly be expected to contribute the strongest output on the right side.

Assuming that PFL3 neurons drive turns, and that the turn amplitude depends on the differential activity between the targets of the PFL3 neurons in the left and right LAL, this asymmetry could have consequences on the behavior of the animal in the absence of any stimulus and FB influences (unlikely though this may be). L3 and R3 are ~180° apart; when the bump is in L3, it is also near R7 and vice versa. Therefore, if the EPG bump of activity was in L3 or R3 and passed to the PFL3 neurons, the asymmetries in PFL3 output would then generate a turn. Only when the PFL3 outputs are symmetric, in R5/L5, would no turn be generated (see 'Discussion'). Input to the PFL3 neurons in the FB could confer flexibility on such a scheme (*Rayshubskiy et al., 2020*). Interestingly, the inhomogeneities we observed are strongest for the LAL interneurons targeted by the PFL3 neurons rather than their direct connections to the DNs (*Figure 64A*). This may mean that the LAL interneuron network plays a crucial role in shaping any rotational or translational signals generated by the CX.

The FS4A and B neuron types also show an interesting asymmetry (*Figure 65A* shows it for FS4A), where neurons innervating the leftmost part of the FB make the strongest contribution to their post-synaptic targets. This asymmetry is likely related to the fact that these neurons innervate the AB. The innervation pattern in the FB is denser on the side that does not go to the AB. FS4A prominently targets neurons projecting to the flange (FLA) (*Figure 65B*). The FLA, also known as the dorsal tritocerebrum, belongs to the SEZ and is thought to control food intake (*Hartenstein et al., 2018*; *Rajashekhar and Singh, 1994*), suggesting that FS4A neuron may be involved in directing feeding behaviors. What role this asymmetry could play in feeding behaviors is unclear. It is however reminiscent of the recent report that neurons linking the AB to the dorsal layers of the FB (likely vΔA_a) play a crucial role in coordinating feeding behaviors to fructose sensing (*Musso et al., 2021*).

## Known unknowns downstream of the CX

Many of the neurons and circuits that receive projections from CX output neurons have never been investigated experimentally. Beyond brain regions that are directly downstream of CX output neurons (e.g., the LAL for PFL neuron types, the CRE for FC, FR, and PFR neuron types, the SMP/SIP for the FS neuron types), a few other, more distant neuropils reappear regularly in these downstream circuits. The IB and ATL are targeted by prominent neurons in the PFL3, FS3, FC2C, and PFR_b downstream networks. The lateral part of the PLP is part of the PFL1, FC1E, and FR1 networks. As mentioned before, LC neuron output circuits likely also innervate some parts of the lateral PLP (*Scheffer et al., 2020*). The ExR8 and a FS1B networks reach ventral brain regions (SPS/IPS down to the GNG). Deciphering the organization and function of those areas will be necessary to obtain a complete picture of the many ways in which the CX likely shapes the fly's behavior.

Our analysis of the 'true' outputs of the CX reveals that PFL2 and PFL3 neuron types connect to DNs, enabling flexible, context- and goal-dependent control of the fly's orientation and locomotion.

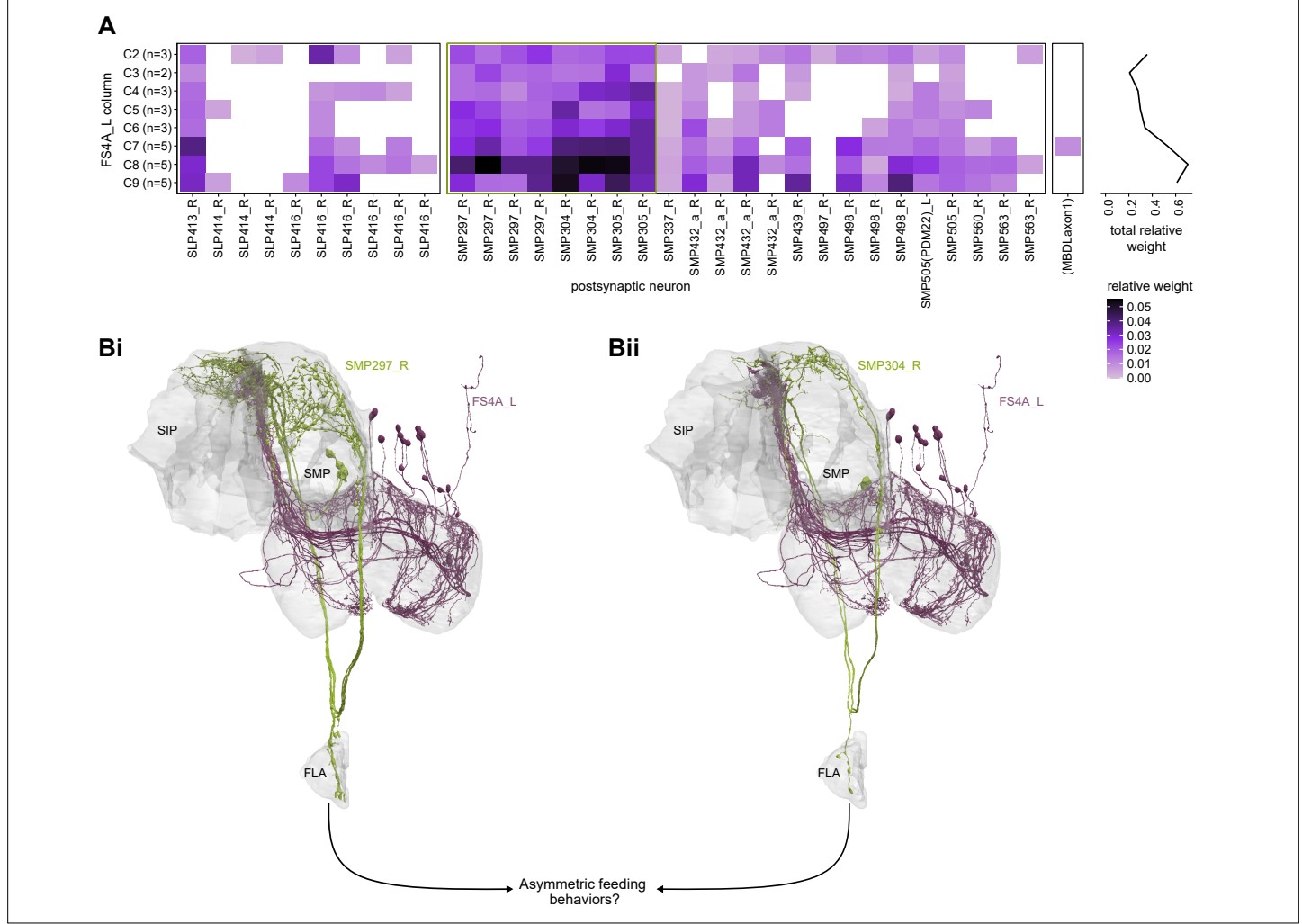

**Figure 65.** Asymmetric connection to the flange (FLA). (**A**) Connectivity matrix between FS4A_L and its direct downstream partners outside of the central complex (CX), on the right side of the brain. FS4A neurons are binned by fan-shaped body (FB) column (numbers of neurons per column are indicated in parenthesis on the y-axis). Right: sum of the relative weights across columns. Connections are biased towards columns C7–C9. (**B**) Morphological rendering of two of the strongest direct targets of FS4A, SMP297, and SMP304 (also circled in green in the connectivity matrix in **A**). All strong direct targets of FS4A project to the FLA, potentially participating in the control of feeding behaviors. FS4A neurons are columnar and project unilaterally in the superior medial protocerebrum (SMP), raising the possibility that they control feeding behaviors in a directed fashion. Besides, the asymmetry in FB innervation also suggests that this behavior could have a default directionality corresponding to the border columns of the FB.

There are also indications of interactions between CX-related output circuits and more direct visual output pathways, suggesting that these different sensorimotor pathways may differentially modulate each other during specific behavioral contexts. The more dorsal output networks of the FB, which act through other columnar neurons (FC, FS, FR, and PFR_b), are likely involved in modulating and directing behaviors that depend on the animal being correctly oriented in its surroundings. Those behaviors could be innate, like oviposition or feeding, or learned and influenced through the CX to MBON network. A prominent feature of nearly every layer of these output circuits is strong feedback to the CX, indicating that information related to the fly's intended actions is relayed back to the core CX structures. All these output circuits are also densely recurrent, a feature that should enable fine-tuning and coordination during action selection. Despite the incompleteness of our analyses, it is clear that information from the CX influences a wide variety of targets across the brain, from neurons that arborize in sensory systems to DANs and DNs.

## Discussion

Recent physiological and anatomical studies at the light and EM level have highlighted strong links between circuit structure and function in the adult fly central brain. These links have proven to be valuable both for generating hypotheses and for experimentally testing them (*Green et al., 2017*; *Klapoetke et al., 2017*; *Morimoto et al., 2020*; *Turner-Evans et al., 2017*; *Turner-Evans et al., 2020*). This recent history gives us reason to expect that connectomics (*Eichler et al., 2017*; *Ohyama et al., 2015*; *Schlegel et al., 2020*; *Takemura et al., 2017a*; *Takemura et al., 2017b*; *White et al., 1986*; *Zheng et al., 2018*) will continue to accelerate studies of circuit function (*Bentley et al., 2016*; *Deutsch et al., 2020*; *Eschbach and Zlatic, 2020*; *Gordus et al., 2015*; *Jovanic et al., 2016*; *Morimoto et al., 2020*; *Ohyama et al., 2015*; *Schretter et al., 2020*; *Tastekin et al., 2018*; *Turner-Evans et al., 2020*). CX circuits, in particular, are thought to be involved a wide variety of flexible, context-dependent behaviors (*Honkanen et al., 2019*; *Turner-Evans and Jayaraman, 2016*). In the 'Results' sections, we have provided a detailed description of CX neuron types and circuits, with a particular focus on extracting and examining network motifs from the perspective of what we currently know about CX circuit function and CX-mediated behavior. We found many repeating motifs, raising the possibility that an understanding of the computational roles of some of these may generalize to others. Some of these motifs match those that have been proposed previously to implement ring attractors for head direction computation (*Hulse and Jayaraman, 2019*; *Turner-Evans et al., 2020*). Others seem suitable for gain control in multiple structures. And still others seem to be ideal for vector computations that would be required for robust navigational behaviors. We found that information from the CX's output neuron types is broadcast through fairly segregated pathways that are distributed across the brain, not just to premotor centers but to sensory regions and, importantly, back into the CX itself. In the sections that follow, we discuss some functional implications of these motifs and of other results from our analyses. We derived these functional implications not just from our connectomic analyses and the historical precedent of structure predicting function in many different neural circuits, but also on published physiological and behavioral studies. Testing the hypotheses that we outline below will require a long series of functional experiments, but the connectome provides an invaluable guide for the design and prioritization of such experiments.

### EM circuit reconstruction: how complete is complete enough?

The value of EM-level connectomes in understanding the function of neural circuits in small and large brains is widely appreciated (*Abbott et al., 2020*; *Litwin-Kumar and Turaga, 2019*; *Schlegel et al., 2017*). Although recent technical advances have made it possible to acquire larger EM volumes (*Scheffer et al., 2020*; *Zheng et al., 2018*) and improvements in machine learning have enabled high-throughput reconstruction of larger neural circuits (*Dorkenwald et al., 2020*; *Januszewski et al., 2018*), the step from acquiring a volume to obtaining a complete connectome still requires considerable human proofreading and tracing effort (*Scheffer et al., 2020*). As part of our analysis of the CX connectome, we found that although increased proofreading led to an expected increase in the number of synaptic connections between neurons, it did not necessarily lead to significant changes in the relative weight of connections between different neuron types (*Figures 3 and 4*). While it is important to note that we made comparisons between the hemibrain connectome at fairly advanced stages of proofreading in the CX, our results do suggest that it may be possible to obtain an accurate picture of neural circuit connectivity from incomplete reconstructions. It may be useful for future large-scale connectomics efforts to incorporate similar validation steps of smaller sample volumes into reconstruction pipelines to determine appropriate trade-offs between accuracy and cost of proofreading.

### Connectivity and neural processing beyond the typical synapse

Although we provide a detailed description of the CX's hundreds of neuron types, recurrent networks, and pathways, there is still more information that could be extracted from the CX connectome. The CX is innervated by a large number of modulatory and peptidergic neurons (*Kahsai et al., 2012*; *Kahsai et al., 2010*; *Kahsai and Winther, 2011*), many unidentified and almost all of unknown function. These neurons likely significantly modulate the function of recurrent networks in ways that few studies address (*Bargmann and Marder, 2013*). Knowing their identities – whether by matching LM images of known neuron types to their EM counterparts in the hemibrain (*Bogovic, 2020*; *Jody*

*et al., 2020*; *Otsuna et al., 2018*) or by advances in machine learning-based identification of neuro-modulator/neuropeptide and receptor types (*Eckstein et al., 2020*) – would help guide circuit studies into context- and internal state-dependent processing in the CX.

A large number of CX neuron types that make T-bar and E-bar synapses (*Shaw and Meinertz-hagen, 1986*; *Takemura et al., 2017a*) in CX structures also send projections to other structures in which they make no such synaptic connections. We investigated these projections in more detail and consistently found DCVs (*Burgoyne and Morgan, 2003*; *Hammarlund et al., 2008*; *Nassel and Winther, 2010*) in these otherwise nearly synapse-free processes (*Figure 56—figure supplement 1B and C*). Although the involvement of some of these neuron types, for example, PFGs neurons, in sleep-wake circuits suggests a plausible scenario for state-dependent modulation of CX circuits, such explanations are not easily available in all cases.

It is important to note that our use of relative weights to assess synaptic strength was informed by observed correlations between synapse counts and the area of synaptic contact in larval *Drosophila* (*Barnes et al., 2020*), and the dependence of synaptic strength on synaptic surface area, at least in the mammalian neocortex (*Holler-Rickauer et al., 2019*). We expect relative weights to provide only an approximate measure of true functional strength. Further, synapses across the *Drosophila* brain undergo structural changes depending on the time of day, sleep, activity, and the animal's specific experiences (*Bushey et al., 2011*; *Kremer et al., 2010*; *Pyza and Meinertzhagen, 1999*); properly accounting for the impact of such factors on connectivity patterns would require comparisons across multiple connectomes. Also, as previously discussed, the hemibrain connectome does not capture glial networks or gap junctions. Despite all these limitations, the identification of chemical synapses between CX neurons and examining their relative weight based on synapse counts allowed us to extract network motifs that make strong predictions about function. We discuss these insights in the following sections.

## What the CX's network motifs tell us about its navigational computations

Many flexible, goal-driven behaviors unfold over longer durations than fast reflexive responses and are robust to the temporary loss of sensory cues directly associated with the goal. Desert ants, for example, use path integration to return to their nests after long foraging trips in relatively featureless landscapes (*Wehner, 2020*), and mammals use working memory to perform delayed match-to-sample tasks (*Romo et al., 1999*). For such behaviors, brains are believed to rely on intermediate representations and neural dynamics that persist or update even in the absence of direct sensory inputs. Such persistent representations have long been believed to be generated, updated, and maintained by recurrent attractor networks (*Brody et al., 2003*; *Durstewitz et al., 2000*). These more abstract intermediate representations also enable disparate sensory and self-motion cues of different modalities to be registered to a shared reference frame. A path integrating ant, for example, may use such a representation to register cues from polarized light, visual optic flow and proprioception (but see *Pfeffer and Wittlinger, 2016*) and all Diptera likely need to register visual and haltere input, as flesh flies do (*Kathman and Fox, 2019*). Ultimately, information in these reference frames must still be dynamically converted to a body-centered reference frame for situation-appropriate action. Decades of experimental work in a variety of species have led theorists to propose gain fields for the implementation of such coordinate transformations (*Andersen et al., 1993*; *Pouget and Sejnowski, 1997*; *Pouget and Snyder, 2000*; *Salinas and Abbott, 2001*; *Zipser and Andersen, 1988*), but the predicted neural circuit connectivity has not been directly identified. In addition, for an animal to learn from experience, any past associations of the current context with good or bad outcomes must be recalled and used to modify neural dynamics at the level of such intermediate representations, raising computational questions that have been explored in the field of reinforcement learning (*Sutton and Barto, 2018*). The repertoire of flexible navigational behaviors that insects display suggests that their small brains may solve many of these computational challenges. Further, insect circuits may have evolved solutions to these problems that resemble those proposed by theorists to account for neural response properties in mammalian circuits (*Hulse and Jayaraman, 2019*).

Flies in particular use short-term memory to orient towards the last-known positions of attractive visual beacons that have disappeared (*Neuser et al., 2008*). They learn about their body size and use that information when attempting to cross gaps (*Krause et al., 2019*). They learn to avoid heat

punishment by using visual patterns around them to orient to safety (*Liu et al., 2006*). Although they are not central place foragers like bees and ants, they are capable of returning to a spot of food even when exploring their surroundings in darkness (*Brockmann et al., 2018*; *Corfas et al., 2019*; *Kim and Dickinson, 2017a*), and of remembering visual landmarks to navigate to safe spots in an otherwise hostile open space (*Haberkern et al., 2019*; *Ofstad et al., 2011*). The CX is thought to be essential for many of these behaviors. In the sections that follow, we will discuss how the patterns of connectivity revealed by the CX connectome may enable the neural dynamics, coordinate transformations, and learning-induced changes in action selection associated with meeting the computational challenges of some of these behaviors.

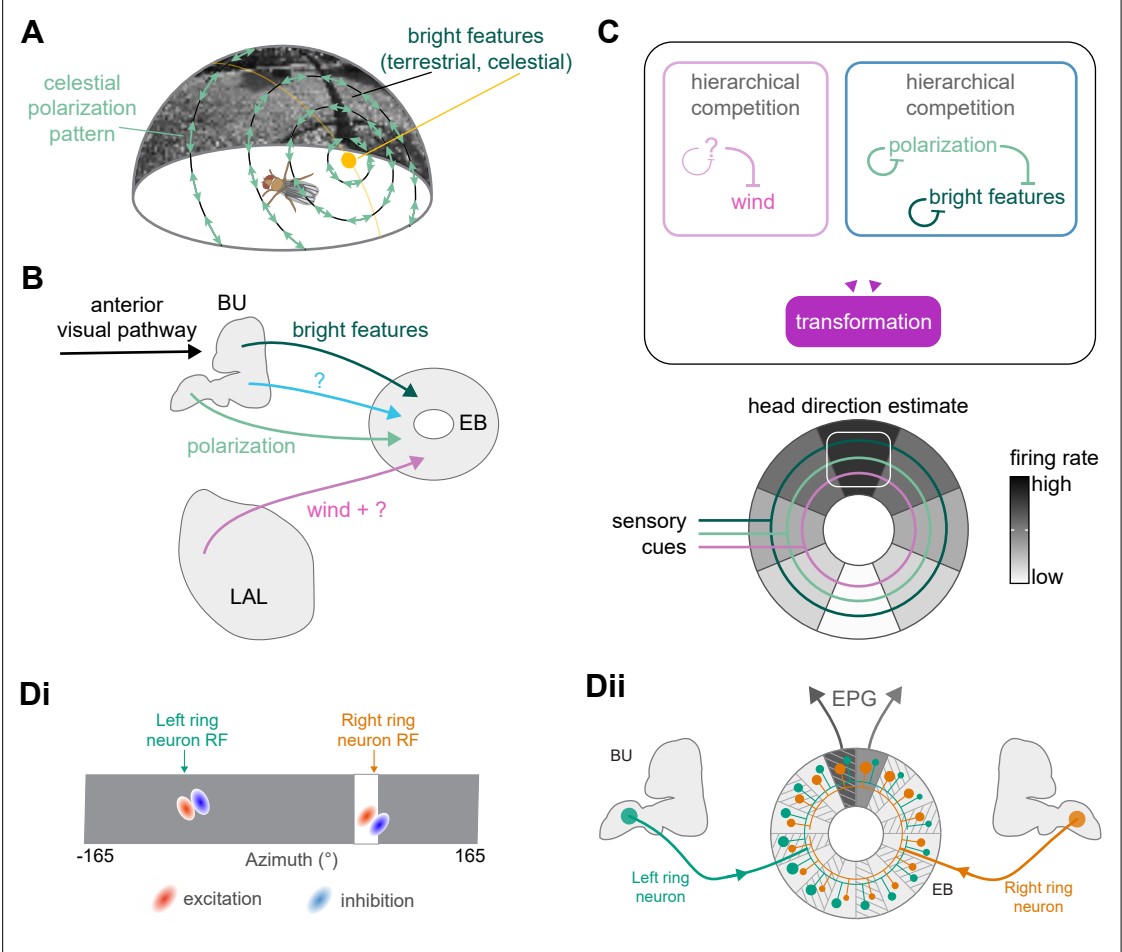

**Figure 66.** Mapping multisensory cues to a flexible head direction representation. (**A**) Illustration of different types of visual cues found in a natural setting that can inform the fly about its orientation. The sun represents a prominent bright landmark but also creates a polarization pattern that covers the full sky. In addition, terrestrial features create a visual scene that can be mapped onto the head direction representation (*Fisher et al., 2019*; *Kim et al., 2019*). (**B**) Ring neurons bring sensory information to the central complex (CX), where they provide input to the fly's head direction system. Sensory pathways have been described for mechanosensory information about wind direction (*Okubo et al., 2020*), celestial visual cues related to the polarization pattern of the sky or visual features (*Seelig and Jayaraman, 2013*). (**C**) Hypothetical competition and transformation that could occur through interactions between ring neuron types conveying distinct directional information. Due to hierarchical competition, one sensory cue, for example, polarization pattern, could dominate at the expense of other, less reliable cues. The transformation from sensory information represented by ring neurons to the head direction estimate allows for complementary directional cues to be combined. (**D**) Schematic of ring neurons that respond to local features in a visual scene (**Di**). Plasticity between these ring neurons and EPG neurons (**Dii**) ensures that the compass reliably tethers to the visual scene.

## Generating a stable representation of head direction in dynamic, multisensory environments

Head direction representations enable an animal to flexibly rely on a variety of different cues, including self-motion, to orient. Work in *Drosophila* and other insect species has established that the CX builds a stable head direction representation using information from ring neurons, which convey directional sensory cues, such as polarized light, visual landmarks, and wind direction (*Figure 66A and B*, *Table 5*; *Heinze and Homberg, 2007*; *Homberg et al., 2011*; *Okubo et al., 2020*; *Seelig and Jayaraman, 2015*; *Varga and Ritzmann, 2016*). In *Drosophila*, visual head direction information reaches the CX via the anterior visual pathway, which appears to convey different visual information in separate, parallel 'channels' (*Figures 66A, B and 6–8*; *Omoto et al., 2017*; *Seelig and Jayaraman, 2013*; *Shiozaki and Kazama, 2017*; *Sun et al., 2017*; *Timaeus et al., 2020*). Some 'channels' of this pathway have been characterized functionally, while the function and sensory tuning of other groups of neurons remains elusive. For example, most of the ring neurons (and their inputs) in the superior BU are spatiotemporally tuned to visual features with some degree of orientation preference (*Seelig and Jayaraman, 2013*; *Sun et al., 2017*) and the pathway through the anterior BU appears to be dedicated to polarization signals (*Hardcastle et al., 2020*) (see below). In contrast, little is known about the role of ring neurons that get their inputs in the inferior BU in informing the head direction representation (*Omoto et al., 2017*; *Shiozaki and Kazama, 2017*). Wind stimuli reach the compass circuitry through a separate input pathway via the LAL (*Okubo et al., 2020*), and it is unknown whether other sensory modalities are conveyed through this route.

Our connectivity-based analysis suggests that there are 22 ring neuron types, 18 of which receive inputs via the anterior visual pathway. In contrast, an anatomical and developmental characterization of ring neurons found only 11 distinct morphological types (*Omoto et al., 2018*). Notably, our connectome-based typing likely represents a subdivision of the previously suggested types rather than a drastic reorganization. Given that past neurophysiological studies have only tested tuning to a relatively small number of sensory stimuli, it remains to be seen how many functionally distinct input types exist.

The connectome reveals mechanisms by which sensory stimuli are integrated to inform the fly's head direction estimate. Our findings suggest that different cues exert differing levels of influence on the EPG neurons that carry the head direction representation (*Figure 66C*). A prioritization of certain sensory cues is reflected in the relative locations of synaptic input from different sensory streams onto the EPG dendrites in the EB (*Figure 12E*), in the relative weight of those inputs (*Figure 11A*), in the feedback that some ring neuron pathways receive from the EPG neurons (*Figure 13—figure supplement 1B*), and in the relative weight of across-type inhibition from some ring neuron types onto others (*Figure 13A and C*). The implicit hierarchy of ring neuron inputs to the fly compass indicates that the EPG head direction representation preferentially tethers to environmental references that are likely to indicate a global direction. Bright visual landmarks, for example, may originate from celestial bodies such as the sun, but they could also be generated by local terrestrial objects (e.g., gaps in a forest canopy). By contrast, a polarization pattern in the sky, if available, represents a reliable global reference, which might explain the observed circuit motifs that suggest the preferential use of polarization cues to update the fly's head direction representation (*Figure 66C*). However, the relatively high connection strength between ER4m and EPG neurons may also arise from this fly not being exposed to polarized light stimuli (see Materials and methods). Such deprivation could have prevented these connections from being subjected to the synaptic depression that other visual pathways may have experienced (*Fisher et al., 2019*; *Kim et al., 2019*) (but note that there is no evidence yet for long-term structural changes at any of these synapses).

Particularly when navigating over long distances, skylight cues allow the head direction representation to be tethered to global landmarks such as the sun and to the polarized light patterns of the sky (*Heinze and Reppert, 2011*). Indeed, polarized light e-vector information has long been thought to be important for the determination of sky compass-based head direction in many insects. A dorsal band of the insect eye called the dorsal rim area is structurally specialized for the detection of polarized light e-vectors in the sky (*Labhart, 1999*). Despite their comparatively small dorsal rim area (*Fortini and Rubin, 1991*; *Wada, 1974*; *Wernet et al., 2003*), flies can also use polarized light cues to determine their heading (*Hardcastle et al., 2020*; *Mathejczyk and Wernet, 2019*; *Warren et al., 2018*; *Weir and Dickinson, 2012*; *Wernet et al., 2012*). Sensory information about the celestial

polarization pattern reaches the *Drosophila* CX via a dedicated pathway to the ER4m neurons (*Hardcastle et al., 2020*; *Weir et al., 2016*, *Figures 6–8*). Although only five ER4m neurons from each hemisphere show strong tuning to e-vector orientation, this tuning collectively covers a large part of the 180° range of possible e-vector orientations (*Hardcastle et al., 2020*; *Weir et al., 2016*). However, in contrast to the position of the sun, the 180° symmetric polarized light patterns do not immediately provide the ability to distinguish a specific direction from one directly opposite to it.

The CX connectome suggests that the fly's compass may have evolved a solution to this problem. For the polarization-tuned ER4m neurons, we observed that synapse numbers to EPG neurons varied smoothly along the circumference of the EB, but with mirror-symmetric profiles for ER4m neurons from the left and right hemisphere, respectively. If synapse counts correlated with synaptic strength, this would result in stronger connections from ER4m neurons of the left hemisphere for EB wedges on the right half of the EB and stronger connections from the right hemisphere to the left half of the EB (*Figures 11D and 67A*). This structure was even more clearly revealed when we analyzed the pairwise correlation of EPG neurons according to their ER4m inputs (*Figure 11—figure supplement 1*, *Figure 67B*): all EPG neurons on the right side of the EB were positively correlated with each other, while being anticorrelated with those on the left side, and the inverse pattern was observed for the left EPG population. Given that polarized light has a 180° symmetry (*Hardcastle et al., 2020*; *Weir et al., 2016*), this connectivity pattern may allow the fly to generate a complete, 360° head direction representation from polarized light input (*Figure 67B*).

One possible mechanism by which this could be achieved hinges on the geometry of the fly's polarization sensors in the dorsal rim area and how it interacts with the natural polarization pattern of the sky. The receptive fields of the fly's polarization sensors in the left and right eye face the contralateral celestial hemisphere and tile a small strip along the rostral-caudal axis of the fly (*Figure 67C and D*; *Heinze, 2014*). Along this strip tuning to e-vectors varies continuously and covers nearly the full 180° range of possible e-vector orientations. Given the naturalistic celestial polarization pattern schematized in *Figure 67C*, the geometry of the slightly curved receptive field 'strip' might act as a rough 'matched filter', such that neurons in the dorsal rim area on the side of the sun (facing the contralateral sky) are systematically more strongly activated than those on the side facing away from the sun (*Figure 67D*). The all-to-all inhibition between left and right ring neurons in the EB (*Figure 13A*) may then systematically select either the left or the right ring neurons to tether the head direction depending on which direction the fly is facing relative to the current position of the sun, thus disambiguating the 180° mirror symmetry in the polarization signal.

In locusts, TL-neurons, homologs of the fly's ring neurons, and PB neurons have been shown to exhibit matched-filter-like tuning to the full-sky polarization patterns generated by the sun (*Bech et al., 2014*; *Zittrell et al., 2020*). A corollary of these studies is that individual TL neurons in the locust have receptive fields that span large parts of the sky. Indeed, the area of the sky that is sampled by photoreceptors in the dorsal rim area is significantly larger in locusts compared to flies (*Heinze, 2014*), and it is plausible that further sensory processing along the anterior visual pathway towards the CX differs between species as well, in which case different insects might employ different strategies for disambiguating polarized light stimuli.

The mechanism described above would not require that the sun be directly visible, but it might still be beneficial to have ring neurons that have multimodal tuning to both polarized light and sun-like stimuli. Such cells have been described in other insects (*el Jundi et al., 2015*; *Heinze and Reppert, 2011*; *Pegel et al., 2018*; *Pfeiffer et al., 2005*). While this has yet to be demonstrated experimentally, multimodal ring neurons tuned to both visual features and polarized light e-vector orientation may also exist in flies. Hardcastle and colleagues report polarization-tuned neurons in the superior bulb (*Hardcastle et al., 2020*), where tuning to bright features has also been observed (*Omoto et al., 2017*; *Seelig and Jayaraman, 2013*; *Sun et al., 2017*).

Besides visual cues, mechanosensory wind stimuli can drive the fly's head direction system in the EB (*Okubo et al., 2020*; *Figures 9 and 66B*). Information about wind direction reaches the EB via ring neurons that arborize in the LAL. Wind tuning has been demonstrated in both ER3a and ER1 neurons, although only ER1 neurons were able to update the head direction estimate (*Okubo et al., 2020*). Analysis of the connectome suggests that both the ER1 and ER3a neuron populations consist of multiple types with distinct inputs. We found that only ER1_b and ER3a_b neurons got strong inputs from cells that we believe to be the wind-sensitive LAL138 (WL-L) and WPN neurons (*Figure 9*; *Okubo*

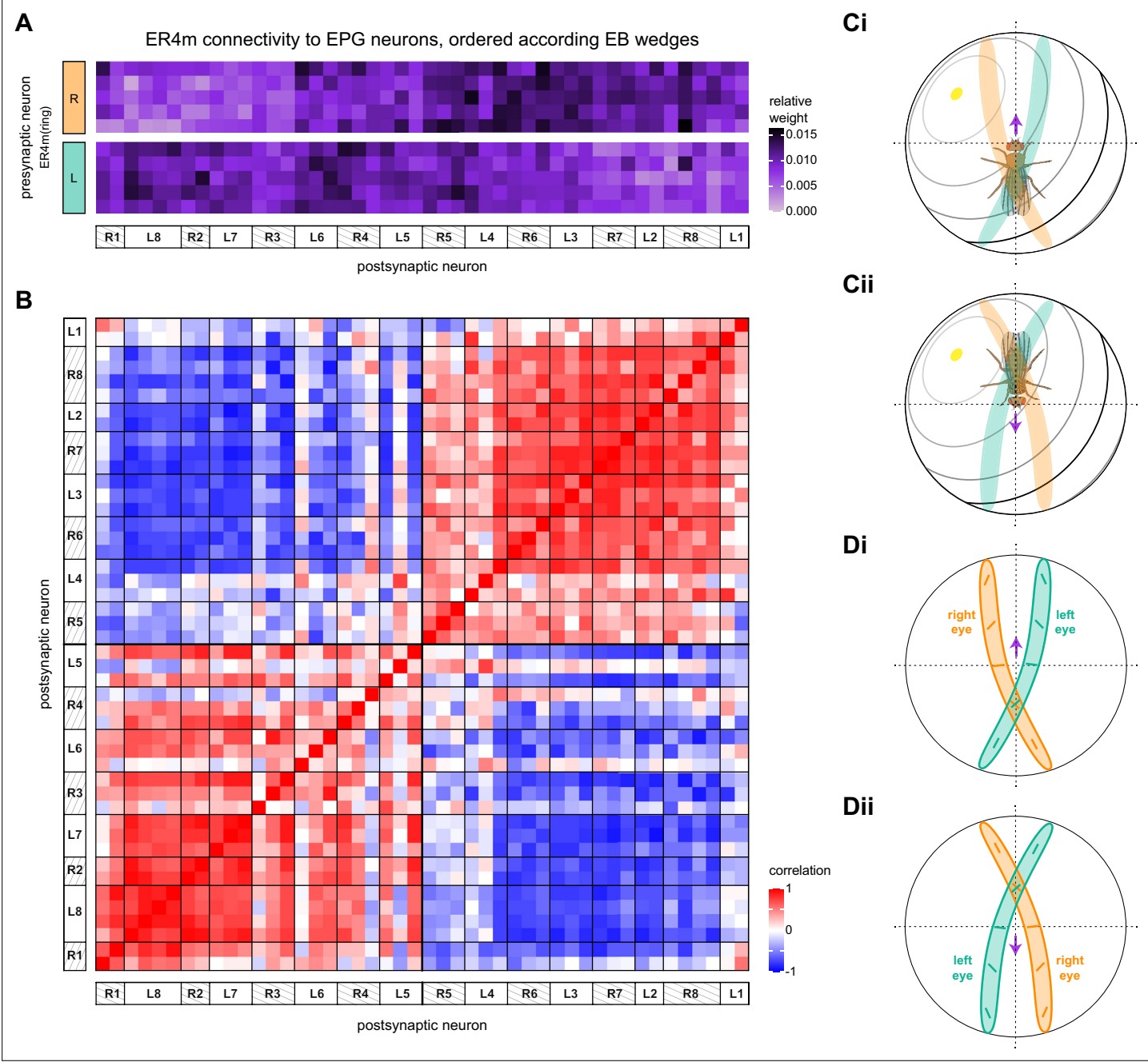

**Figure 67.** Disambiguating directional information from polarized light sensors. (**A**) Connectivity matrix of ER4m inputs to EPG neurons in the ellipsoid body (EB). EPG neurons are sorted according to the EB wedge they innervate. See also *Figure 11D*. (**B**) Pairwise Pearson's correlation measured between individual EPG neurons according to the pattern of their ER4m neuron inputs. See *Figure 11—figure supplement 1* for details. (**C, D**) Under most conditions, the two eyes, and thus the left and right polarization-sensitive dorsal rim areas, are expected to receive different input. (**C**) Schematic of the polarization pattern and the sun position of the sky in relation to the fly's eyes depending on the fly's orientation. Receptive fields of the polarization sensitive dorsal rim area for the left (green) and right (orange) eye are overlaid. (**D**) Receptive fields of the left and right dorsal rim area now shown with an indication of the orientation of the e-vector direction that different parts are sensitive to. (**Ci, Di**) The sun is located to the left of the fly. (**Cii, Dii**) The sun is located to the right of the fly.

*et al., 2020*; *Suver et al., 2019*). The connectivity of these two ring neuron types onto EPG neurons, with strong connections from ER1_b but no connections from ER3a_b neurons, is consistent with the observation that ER1 but not ER3a neurons can drive the head direction representation (*Figure 11A*). It is also noteworthy that in the EB, ER1_b neurons deviate from the within-type all-to-all inhibition

motif that all other ring neurons show in the EB (*Figures 13A and 66C*). A possible reason is that an accurate mapping from ER1 neuron activity to a head direction representation requires pooling information from multiple ring neurons at once (*Okubo et al., 2020*). Our analysis also suggests that ER1_b input to the EPG neurons is suppressed by ER1_a neurons, but it is presently unknown whether ER1_a neurons also encode wind direction or whether these neurons are tuned to a different stimulus (*Figure 66C*). ER1_a and ER1_b inputs in the LAL are distinct and unfortunately little is known about the inputs ER1_a receives.

Whether over short or long distances, olfactory cues are strong indicators of good food sources. Flies are known to fly upwind when they encounter an appetitive odor (*Budick and Dickinson, 2006*), a strategy also employed by other insects navigating to an odor source (*Carde and Willis, 2008*). A robust navigational strategy would allow an insect to maintain the same heading using other cues even if the wind were to transiently die down. Based on the proximity of different ring neuron inputs to the putative spike initiation sites of EPG neurons, the head direction representation is likely to be strongly tethered to wind direction by input from ER1_b neurons (and perhaps also ER1_a neurons, although their function is currently unknown) (*Okubo et al., 2020*). If visual cues are flexibly mapped onto head direction representation using this wind direction input as a reference (*Fisher et al., 2019*; *Kim et al., 2019*), the EPG compass could allow the fly to preserve its heading using those cues even in the absence of wind.

The relative importance of synapse location in determining the cues to which the EPG compass tethers will only be clear with in-depth investigations of EPG neuron biophysics. More broadly, future studies of ring neuron and EPG interactions should provide an implementation-level understanding of a variety of computations related to dynamic multisensory integration (*Pouget et al., 2002*) and the resolution of conflicts between cues of different reliability (*Deneve et al., 2001*; *Hoinville and Wehner, 2018*; *Wystrach et al., 2015*).

## All-to-all inhibition for noise reduction in sensory inputs to the compass network

Recent studies have proposed an important role for fast-timescale, short-term plasticity of synaptic connections between ring and EPG neurons in enabling the EPG compass to quickly adapt to different sensory settings (*Figure 66D*; *Fisher et al., 2019*; *Kim et al., 2019*). The connectome suggests that the ring neuron network may also preselect more salient cues for the compass through all-to-all inhibitory connectivity within each type. The precise impact of all-to-all inhibition on the ring neuron network's preprocessing of localizing cues that are used to generate the head direction representation will depend on the timescale of the inhibitory conductance (*Ermentrout, 1992*), which is as yet unknown. If the inhibitory conductance is fast, all-to-all inhibition would create winner-take-all dynamics in which a few ring neurons receiving the strongest inputs effectively shut down all other ring neurons. In sensory settings characterized by a single dominant sensory cue, such as the sun or polarized light e-vector orientation in a desert landscape during the day (*Coyne et al., 1982*; *Weir and Dickinson, 2012*), only a handful of ring neurons with appropriately tuned receptive fields (*Fisher et al., 2019*; *Omoto et al., 2017*; *Seelig and Jayaraman, 2013*; *Shiozaki and Kazama, 2017*; *Sun et al., 2017*) would be active within each type for any particular head direction. Fast all-to-all inhibition in this setting would enhance the activity of the most dominant ring neuron within each type and would minimize the impact of noise from the others, which might otherwise disrupt the stability of the EPG compass.

However, stable and unique heading representations are also generated within scenes with multiple strong cues, such as within a forest or when walking on the branches of a tree, as long as the two-dimensional arrangement of cues allows for a unique determination of heading (*Kim et al., 2019*). In the presence of multiple salient cues, we would expect multiple ring neurons to respond with comparable strength for any given heading of the fly and several to respond weakly to any additional visual cues ('clutter') in the scene. Although fast all-to-all inhibition in this scene would still filter out these weaker responses, it could allow multiple, strongly responsive ring neurons to remain active for each heading. A slower inhibitory conductance would, in this situation, induce oscillatory spiking dynamics between these multiple 'winners' (*Ermentrout, 1992*), a situation that has been referred to as 'winner-less competition' (*Rabinovich et al., 2001*), and that has been suggested to be useful for sequential memory (*Seliger et al., 2003*).

## State-dependent modulation of ring neurons

Ring neuron responses are not determined purely by sensory cues. These neurons appear to be modulated by state, maintain a baseline level of activity, and may be biophysically configured to support oscillatory population activity linked to sleep need (*Raccuglia et al., 2019*). The many additional inputs that many ring neuron types receive in the BU (*Figure 8*) provide clues as to how the activity of these neurons might be modulated by the fly's behavior and its internal state. The visually tuned superior BU ring neurons primarily receive input from a large interhemispheric AOTU neuron (AOTU046), which may mediate dynamic stimulus selection through delayed contralateral inhibition (*Sun et al., 2017*). The same group of ring neurons also shows changes in activity with the fly's behavioral state (flight versus walking) (*Seelig and Jayaraman, 2013*) and indeed many of these neurons receive input from the dopaminergic ExR2 neuron that has been linked to changes in the fly's motor activity (*Kong et al., 2010*; *Liang et al., 2019*; *Tao et al., 2020*). A different set of ring neurons that receive their inputs in the inferior BU receives strong inputs from two ExR neurons – ExR1 and ExR3 – that have been linked to the control of sleep (*Figures 8 and 15*; *Donlea et al., 2018*; *Liu et al., 2019*), and may gate sensory stimuli according to the fly's behavioral state (*Donlea et al., 2018*).

## A ring attractor network with all the trimmings

The fly's head direction representation tethers to directional sensory cues conveyed by ring neurons, but is also updated by self-motion cues (*Green et al., 2017*; *Green and Maimon, 2018*; *Hulse and Jayaraman, 2019*; *Seelig and Jayaraman, 2015*; *Turner-Evans et al., 2017*; *Turner-Evans et al., 2020*) and is maintained across periods of immobility (*Seelig and Jayaraman, 2015*). Strong experimental and theoretical evidence suggests that the representation is maintained by a ring attractor network (*Kim et al., 2017c*), which includes at least some of the recurrently connected columnar neurons that link the EB and the PB: the EPG, PEN_a, PEN_b, and PEG neuron types (*Green et al., 2017*; *Turner-Evans et al., 2017*; *Turner-Evans et al., 2020*). The patterns of connectivity between individual neurons of these types are consistent around the entire EB and across the length of the PB. Similarly, the broad connectivity patterns of individual neurons within these types to tangential neurons – Δ7 neurons in the PB and different types of ring neurons in the EB – are similar across these structures. Notably, however, two distinct classes of neurons (EPGt neurons and the P6-8P9 neurons, *Figures 18 and 21*, respectively) only innervate the edges of the network; both innervate the outer glomeruli of the PB, and the EPGt neurons also innervate the corresponding wedges in the EB. These types may help to stitch together what might otherwise be a discontinuity in the ring attractor network. Indeed, the EPGt neurons in the left and right PB arborize in wedges in the EB that lie directly in between the wedges occupied by the EPG neurons on either side of the potential discontinuity (in PB glomeruli 1 and 8). The EPGt neurons may therefore represent angular positions halfway between the edge angles, bridging the gap. We note, however, that the Δ7 neurons and the P6-8P9 neurons that output in these outer glomeruli each receive unique input from different sets of EPG neurons, making it hard to assign a clear corresponding EB angle to glomerulus 9 (*Figure 68Ai*).

The hemibrain connectome further allowed us to identify several neuron types and connectivity motifs that are likely involved in the network's function, but whose roles await experimental investigation. Many of these additional types are tangential neurons. Some of them appear to provide additional sources of inhibition, potentially regulating overall network activity (*Franconville et al., 2018*; *Turner-Evans et al., 2020*). The ER6 neurons, for example, receive input from the EPG and PEG neurons in the GA and send outputs to the PEG, EL, and PEN_b neurons in the EB (*Figure 10*, *Figure 13—figure supplement 1*, *Figure 56*), potentially modulating the EPG-to-PEG-to-PEN_b--to-EPG feedback loop. Furthermore, many of the ExR neurons make connections to and receive input from EB columnar neurons (*Figure 14*). The PB receives neuromodulatory input from the dopaminergic LPsP neurons and the octopaminergic P1-9 neurons (*Figure 23*). The LPsP neurons may enable changes in synaptic strength in the PB. Such plasticity in the PB has been suggested to allow flies to calibrate their directed movements to their body size (*Krause et al., 2019*). In sensory brain regions, octopaminergic neurons are known to modulate neuronal conductances based on the fly's behavioral state (*Strother et al., 2018*; *Suver et al., 2012*), and it is an open question whether the P1-9 neurons play a similar role in the PB. Notably, the Δ7 neurons connect recurrently to each other in the PB, but the function of this recurrence is unknown. One possibility is that recurrent Δ7 connections may increase the stability or robustness of the ring attractor network (*Pisokas et al., 2020*).

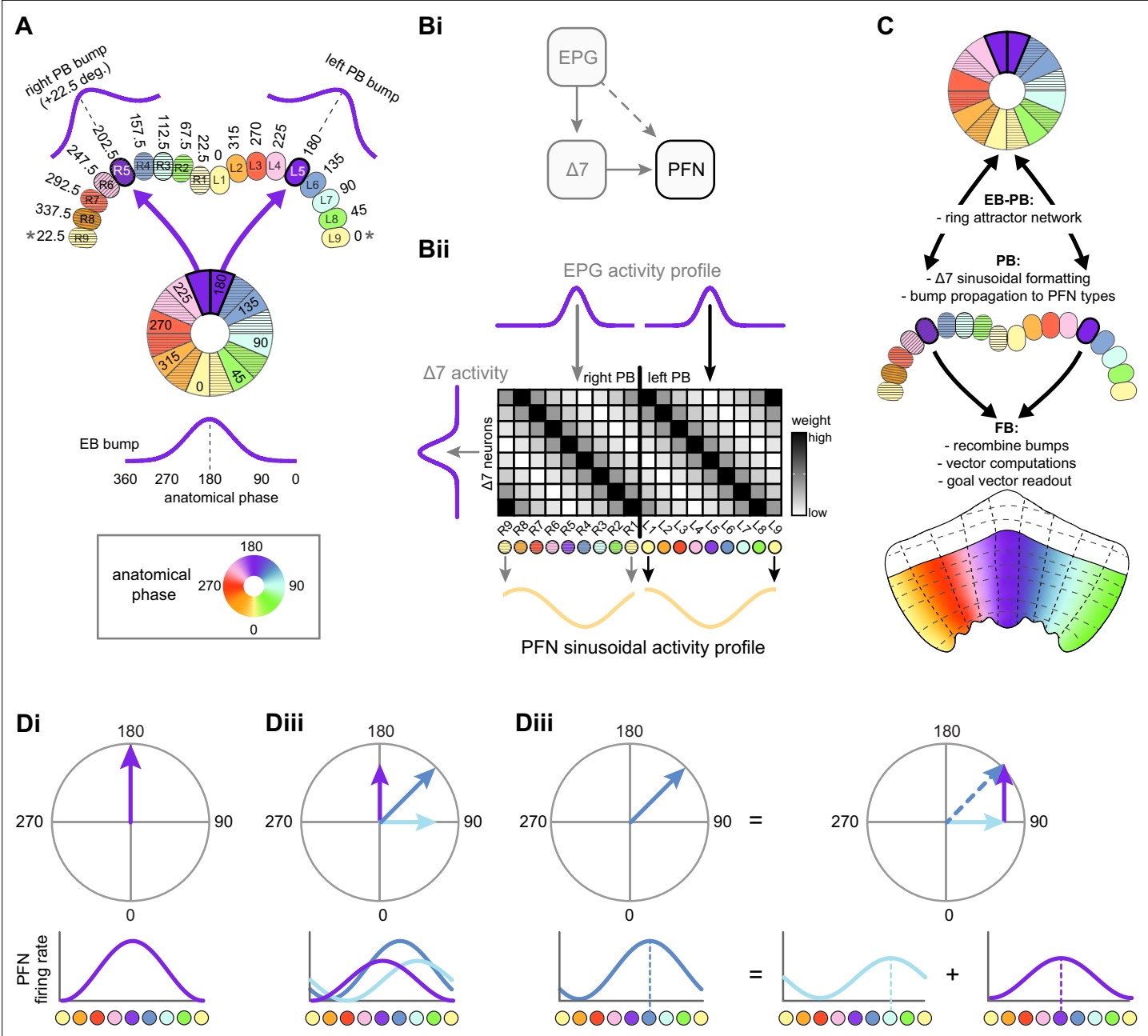

**Figure 68.** Conveying and transforming the head direction representation from the ellipsoid body (EB) to the fan-shaped body (FB). (**A**) Schematic showing how a bump of activity gets conveyed from the EB to the left and right PB. EB wedges and PB glomeruli are colored by their anatomical phase (i.e., directional tuning). Based on data from *Figure 16*. (**B**) Δ7 neurons in the PB transform any EPG activity profile into a sinusoidal activity profile that gets inherited by FB columnar neurons. PFN neurons receive the EB bump directly from EPG neurons (dashed arrow) as well as through Δ7 neurons (solid arrow) (**Bi**). Schematic showing how the connectivity of Δ7 neurons might transform any EPG activity profile into a sinusoidal activity profile (illustrated for PFN neurons; yellow curve) (**Bii**). Based on data from *Figure 20*. (**C**) Overview of bump propagation through the central complex (CX) along with the major computations carried out in each region. (**D**) Phasor representation of a sinusoidal activity profile. Any sinusoidal activity bump can be represented as a vector whose angle encodes bump phase and whose magnitude encodes bump amplitude. Schematic of hypothetical sinusoidal activity bump (purple line), centered at 0°, encoded by a population of neurons that function as a sinusoidal array, and phase representation (purple arrow) of the same activity (**Di**). Vector addition can easily be implemented by sinusoidal arrays carrying different activity bumps (purple, blue, light blue) (**Dii**). Two vectors (purple and light blue) can be summed to generate a new vector (blue) (**Diii**). Based on data from *Figure 20*.

Two additional classes of columnar neurons also contact the ring attractor network: EL and IbSpsP. The connectivity pattern of EL neurons in the EB is remarkably similar to that of the EPG neurons (*Figure 11*), but their function is unknown. In the PB, the IbSpsP neurons bring input into specific glomeruli from regions associated with premotor functions, potentially allowing them to exert an influence on the dynamics of the bump in the PB.

## Bumps on the move: duplication and sinusoidal reformatting

The ring attractor network described above generates a single activity bump in the EB that encodes the fly's head direction (*Green et al., 2017*; *Heinze, 2017*; *Seelig and Jayaraman, 2015*; *Turner-Evans et al., 2017*). The connectome allowed us to follow this activity bump through the CX as it gets duplicated, reformatted, recombined, and, finally, read out. In the process, we discovered network motifs that seem ideally suited for performing vector computations. These motifs place constraints on the network's computational capacity and inspire conceptual models for how the network might function. We begin by describing how the activity bump is forwarded from the EB to the PB, where it is duplicated and reformatted into a sinusoidal profile. In subsequent sections, we consider how the FB network may recombine these bumps to perform vector computations in support of goal-directed behavior.

The EPG population divides the EB into 16 'wedges,' suggesting that the fly's head direction system samples angular space at 22.5° intervals (*Figure 68A*; *Hanesch et al., 1989*; *Lin et al., 2013*; *Seelig and Jayaraman, 2015*; *Wolff et al., 2015*). Importantly, this does not mean that the system cannot resolve head directions at resolutions higher than 22.5° since the differential activation of columnar neurons with distinct directional tunings can effectively represent any arbitrary angle within the 360° around the fly. From the EB, EPG neurons convey the HD bump to both the left and right PB, generating two bumps that are sampled at approximately 45° intervals (*Figure 68A*). Due to the EPG projection pattern (*Wolff et al., 2015*), there is a 22.5° shift in the directional tuning between EPG neurons in left and right PB, as recently confirmed by physiological recordings (*Lyu et al., 2020*). Importantly, the bumps in the left and right PB still encode the same head direction, but do so using sets of neurons whose sampling of angular space is shifted by 22.5°.

Within the PB, FB columnar neurons inherit a head direction bump directly from EPG neurons and indirectly through Δ7 neurons (*Figure 68Bi*; *Franconville et al., 2018*; *Green et al., 2017*; *Turner-Evans et al., 2017*; *Turner-Evans et al., 2020*). The Δ7 populations appear ideally suited to reformat the EB bump into a sinusoidal profile, regardless of its original shape (*Figure 20*). Individual Δ7 neurons provide output to 2–3 PB glomeruli spaced ~180° apart (i.e., separated by seven glomeruli). Between these axonal compartments are dendritic segments whose EPG input weight is well fit by a sinusoid, suggesting that individual Δ7 neurons should have a sinusoidal tuning curve. Assuming the Δ7 population uniformly samples angular space (e.g., with a 45° sampling interval), this would manifest as two sinusoidal bumps across the PB, one in the left PB and one in the right PB (*Figure 68Bii*). Furthermore, recurrent Δ7 connections may enforce a sinusoidal activity pattern on the Δ7 population itself, which could improve the ability of Δ7 neurons to reformat the activity bump into a sinusoidal profile before passing it on to PB-FB types. The Δ7 population provides input to ~10 types of PB-FB neurons, effectively duplicating the activity bump in the process. As discussed below (*Figure 69*), this duplication may allow the FB network to recombine bumps in a way that implements a compact vector calculator (*Figure 68C*).

Why might the Δ7 population reformat the activity bump into a sinusoidal shape? Perhaps because sinusoids are a particularly suitable representation for vector-based computations (*Touretzky et al., 1993*), since the sum of any two sinusoids of equal frequency is also a sinusoid. One way of schematizing this process is to use phasor diagrams (*Figure 68D*). Viewed in this way, the sinusoidal activity bumps become vectors whose magnitude reflects bump amplitude and whose angular position indicates bump phase, with each phase mapping to an allocentric direction. Adding vectors is equivalent to adding sinusoidal activity profiles.

## Path integration: an example of a canonical vector-based computation

Path integration is a canonical vector-based navigation strategy used by a diverse array of both flying and walking animals (reviewed in *Collett, 2019*; *Heinze et al., 2018*; *Wehner, 2020*), potentially including *Drosophila* (*Brockmann et al., 2018*; *Corfas et al., 2019*; *Dethier, 1957*; *Kim and*

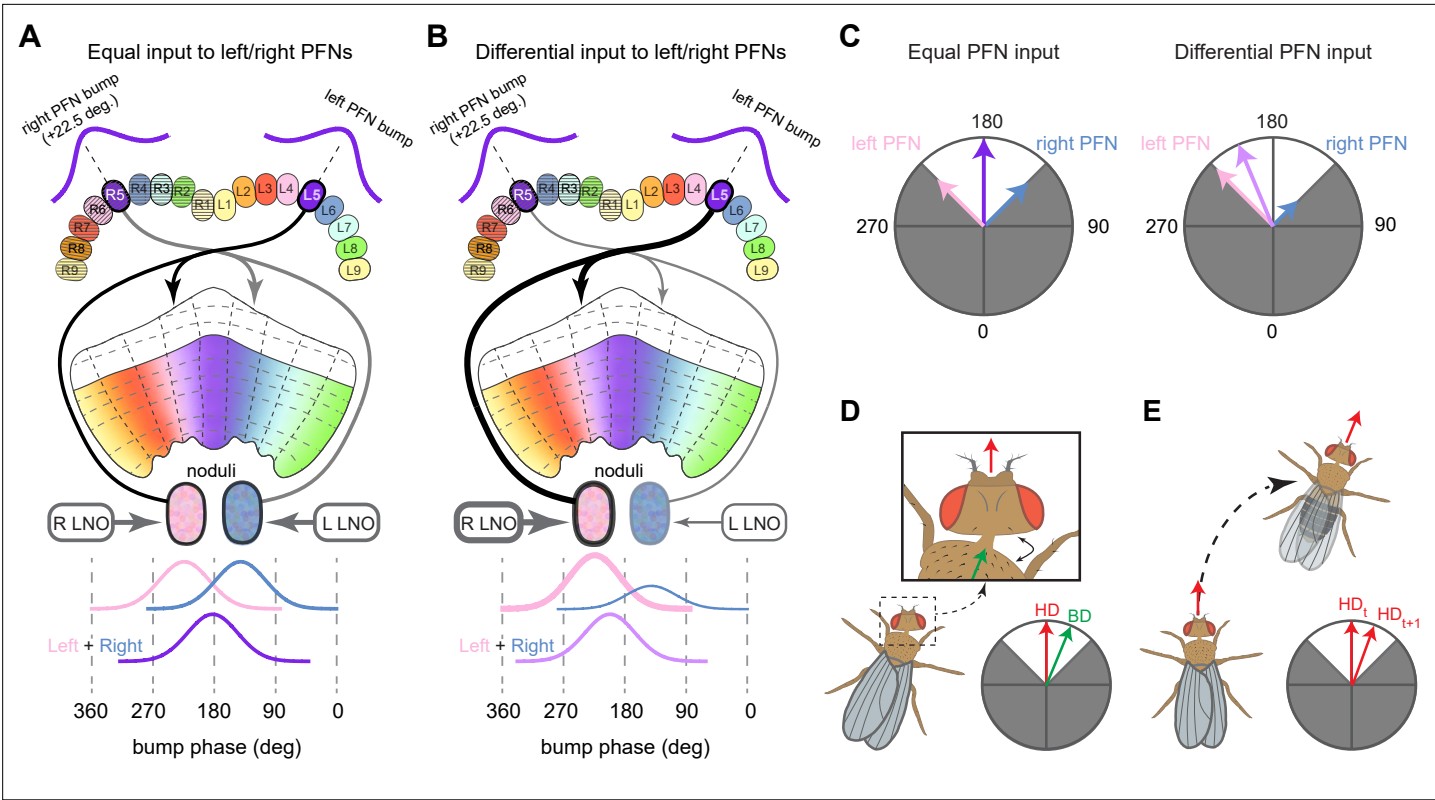

**Figure 69.** Conceptual model showing that PFN phase shifts, when combined with differential noduli (NO) input, could produce ±45° bump shifts between the protocerebral bridge (PB) and fan-shaped body (FB). (**A**) Schematic of a PB-to-FB projection pattern with a one-column contralateral phase shift. PB glomeruli and FB columns are colored according to anatomical phase (from –180° to 180°), which indicates matching bump locations. PFN neurons innervating the right PB project to the left NO, where they receive input from LAL-NO neurons carrying self-motion information from the left lateral accessory lobe (LAL). Similarly, PFN neurons innervating the left PB project to the right NO and receive self-motion inputs from LAL-NO neurons innervating the right LAL. As shown in the bottom panel, when these two LAL inputs are equal, the spatially offset bumps from the left and right PB sum to generate a new bump located halfway between the two. (**B**) Same as in (**A**), but with differential NO input. In this case, as shown in the bottom panel, the R LAL neurons increase the bump amplitude of the left PB population (pink bump), and the L LAL neurons decrease the bump amplitude of the right PB population (blue bump). The sum of these two bumps will end up closer to –90°, the location of the left PB bump, due to the difference in bump amplitudes. (**C**) Phasor diagram interpretations of the scenarios from (**A**) and (**B**). In the left panel, with equal NO input, the left PFN and right PFN bumps sum to produce the purple vector located at 0. In the right panel, with differential NO input, the left PFN bump becomes bigger than the right PFN bump (as in **B**), and therefore, their sum is closer to the left PFN bump. This effectively shifts the bump by a phase that depends on the difference in amplitude between the left and right PFN populations. Importantly, the PFN neurons' one-column ipsilateral phase shift limits such bump shifts to ±45° from the PB bump. Phases shifts outside this area (marked in gray) cannot be produced. (**D**) Illustration of a head direction to body direction coordinate transformation. (**E**) Illustration of a forward model of head direction. Figure describes conceptual models based on data from **Figures 30 and 34**.

*Dickinson, 2017a*; *Murata et al., 2017*). In its most basic form, 2D path integration requires that an animal keep track of its direction and distance relative to a stored goal location, such as a food source or nest, often without the use of external landmarks. The direction and distance to the goal location is thought to be computed through the integration of self-motion signals and stored as a 'home vector.' To return to the goal location, animals are thought to generate appropriate motor commands by comparing their current heading to the stored home vector. While many insects are thought to generate and use visual snapshots of their surroundings to guide return trips (*Collett et al., 2013*; *Collett and Zeil, 2018*; *Freas et al., 2019*), and while such visual homing may involve the MB (*Buehlmann et al., 2020*; *Collett and Collett, 2018*; *Kamhi et al., 2020*; *Sun et al., 2020*), we will focus here on how a home vector might be constructed and read out in the CX using only a stable head direction signal, a situation that can arise in featureless landscapes or in darkness. Although there is as yet no definitive evidence that the CX is used for path integration, in the next few subsections, we show how a network built from FB-inspired circuit motifs could compute a translational velocity (TV) vector in an allocentric reference frame whose integration would yield a home vector. We note

that the framework for vector computations that we describe below is likely to be useful for a much broader array of behaviors involving oriented action selection.

## The potential for vector computations in the FB

PFN neurons serve as the major columnar input to the FB network. The ±45° phase shift that is characteristic of all PFN neuron types (*Figures 30 and 34*) implies that activity bumps from the left and right PB would end up ~90° apart in the FB (*Figure 69A*). The amplitude of these activity bumps is likely to be strongly influenced by the different inputs that PFN neurons receive through their lateralized projections in the NO (*Figure 25*), setting up the possibility of bump-based vector computations in the FB. The PEN_a neurons, which are conjunctively tuned to head direction and angular velocity, perform a similar computation in the EB by providing phase-shifted input to the EPG neurons, thereby updating the position of the EPG bump when the fly turns (*Green et al., 2017*; *Turner-Evans et al., 2017*; *Turner-Evans et al., 2020*). Inside the FB, vector computations fed by phase-shifted PFN bumps whose amplitudes are controlled by different conjunctive signals could ultimately drive PFL neuron types to generate appropriate motor commands (*Figures 39 and 63*), an algorithmic idea (*Hartmann and Wehner, 1995*; *Wittmann and Schwegler, 1995*) for which an FB implementation was first proposed in *Stone et al., 2017*.

Although the intra-FB columnar network is highly recurrent, much of it is built from a limited number of circuit motifs (*Figure 37*). These motifs serve as the backbone of a 2D grid in which activity bumps are constrained to either maintain their phase (i.e., maintain their column) while moving across layers, or shift phase by 180° (i.e., shift by half the width of the FB). While some pathways directly connect PFN neurons to output pathways, such as those involving PFL neurons, many more pathways run through this 2D grid (*Figure 33*). Thus, the network has depth, providing multiple layers with which to process activity bumps. In addition, a large number of tangential neuron types selectively innervate different layers of the FB, suggesting that the FB's vector computations are influenced by context and internal state. In the sections that follow, we draw from published experimental and theoretical work to explore the navigational implications of PB-FB neuron phase shifts. Importantly, for the purposes of discussion, we assume that the magnitude of PFN phase shift is precisely 90°, a simplifying assumption about symmetry in the circuit that ignores the type-to-type variability in estimated phase shifts across PFN types (*Figures 30 and 34*), the functional significance of which remains unknown. Similarly, we assume all LNO types to be excitatory, but the proposed conceptual models could be built from inhibitory LNO types as well. Finally, while the columnar structure of the various vΔ and hΔ types shows considerable variability, we assume these neurons can either maintain the phase of an FB bump or shift it by 180°.

## Potential function of PFN phase shifts: forward models and coordinate transformations

Despite the PFN phase shifts, the two 90°-separated activity bumps arising from a single PFN type cannot propagate independently through the FB network because nearly all single neurons and neuron types that are postsynaptic to PFN neurons sample from left and right populations equally (*Figure 35*). Instead, each postsynaptic FB type likely sums the bumps from the left and right PFN populations, indicating that each postsynaptic type represents a single, summed bump that will propagate through the FB network.

Much like the PEN neurons, the PFN neurons innervating the left PB project to the right NO and neurons innervating the right PB project to the left NO, where they receive input from various LNO types (*Figure 25*; *Lin et al., 2013*; *Wolff et al., 2015*). One potential function that this differential NO input to the left and right PFN populations could serve is to produce a new, transformed directional representation that could take on angles ±45° around the fly's instantaneous head direction. For example, as shown in *Figure 69B*, a strong excitatory input to the right nodulus would increase the bump amplitude of the left PFN population relative to the right PFN population. In turn, the summation of these two bumps by a postsynaptic neuron type in the FB would result in a new bump that lies closer to that of the left PFN population.

*Figure 69C* shows phasor diagrams of this process. Critically, because these vectors can only take on positive values (firing rates above 0), such differential input could only shift the resulting vector's phase by ±45° around the fly's instantaneous head direction. What might this transformed

directional representation encode? The answer likely depends on the nature of the input that PFN neurons receive from LNO types (*Figure 25*; *Wolff and Rubin, 2018*). Recent work in *Drosophila* has shown that some PFN neurons show differential activity in the NO that reflects the fly's turning behavior during flight (*Shiozaki et al., 2020*), but the nature of the rotational velocity signal remains to be determined. We briefly outline two hypothetical scenarios that differ in the specific information carried by PFN neurons.

Directly wiring sensors to actuators in different ways can, in principle, allow a simple agent to display a variety of behaviors (*Braitenberg, 1984*). But flies, like most animals, have to deal with an additional complication. Some of their sense organs are on body parts that are different from those that enable them to move. PFN phase shifts could enable coordinate transformations, such as converting the allocentric head direction representation into an allocentric body direction representation (*Figure 69D*; *Andersen and Cui, 2009*; *Andersen et al., 1993*; *Batista, 2002*; *Bicanski and Burgess, 2020*). Flies make head movements that change their head-body angles by as much as 30° during both flight (*Duistermars et al., 2012*) and walking (*Fujiwara et al., 2017*; *Geurten et al., 2014*). In this scenario, LNO neurons that provide input to PFN neurons arborizing in the left and right NO would encode how much the head is rotated to one or the other azimuthal direction of the body's axis, perhaps derived from proprioceptive information (*Paulk and Gilbert, 2006*; *Preuss and Hengstenberg, 1992*). When properly calibrated, such differential input could allow the PFN phase shift (which, at 45°, is sufficient to encode the entire range of head-body angles) to rotate the head direction vector by an angle equal to the head-body angle. This would effectively transform the fly's head direction vector into an allocentric body direction vector. To do so, the network could use gain fields (*Andersen et al., 1993*; *Pouget and Sejnowski, 1997*; *Pouget and Snyder, 2000*; *Salinas and Abbott, 2001*; *Zipser and Andersen, 1988*), with an intermediate layer composed of PFN neurons whose head direction tuning curves are gain-modulated by shared input related to the head-body angle. The neuron types downstream of PFN neurons would complete the coordinate transformation through their structured sampling of PFN neurons with distinct directional tunings. Coordinate transformations such as these may be useful when combining allocentric directional representations with body-centered velocity estimates to estimate direction and distance. We return to this idea in more detail in subsequent sections on vector computations related to path integration.

A second scenario is shown in *Figure 69E*. if we assume that an LNO type carries a motor efference copy of the fly's rotational velocity, as has been shown to exist in the fly visual system (*Fujiwara et al., 2017*; *Kim et al., 2017b*; *Kim et al., 2015*), then bump shifts driven by differential input to the left and right PFN populations could function as a forward model (*Webb, 2004*) that encodes a prediction of the fly's future head direction or body direction. Why might a forward model of head or body direction be useful? Intracellular recordings from neurons in the ring attractor network have revealed that PEN activity is tuned to the fly's rotational velocity, but that this activity lags behavior by ~100 ms (*Turner-Evans et al., 2017*). Similar lags may result from propagation delays in neural processing, either along sensory pathways into the EB or in passing the bump from EB to FB. In situations where flies might rely on the CX to direct their movements, especially in time-critical scenarios, such delays in updating the compass could be costly. One way to overcome this is for the PFN network to compute the fly's approximate future head or body direction so that navigational decisions can effectively be made in real time, a strategy that dragonflies have been shown to use during rapid prey capture maneuvers (*Mischiati et al., 2015*). A forward model could also allow the fly to distinguish changes in its head or body direction associated with voluntary movements from those induced by external disturbances, such as changes in wind direction (*Currier et al., 2020*). More generally, matching the predicted head and body direction with the actual direction could enable the fly to fine-tune its movements to produce intended motor outputs (*Krause et al., 2019*). As described below, PFL neuron types have anatomical phase shifts that appear well-suited to perform such comparisons (Figure 73).

The PFN neuron types that the computations hypothesized above might involve are as yet unclear. However, our analysis of PFN inputs from LNO types does allow us to draw some inferences about the sort of self-motion information that different PFN neuron types might carry. For example, considering that PFL2 neurons provide selective feedback to the LNO3 neuron type and considering that the PFL3 neurons feed the LCNOp neurons (*Figure 57C*), we would hypothesize that the former may provide its downstream PFN neurons – the PFNv neurons (*Figure 27*) – with efference information related to translational (and potentially forward) movement and that the latter my provide its PFN targets

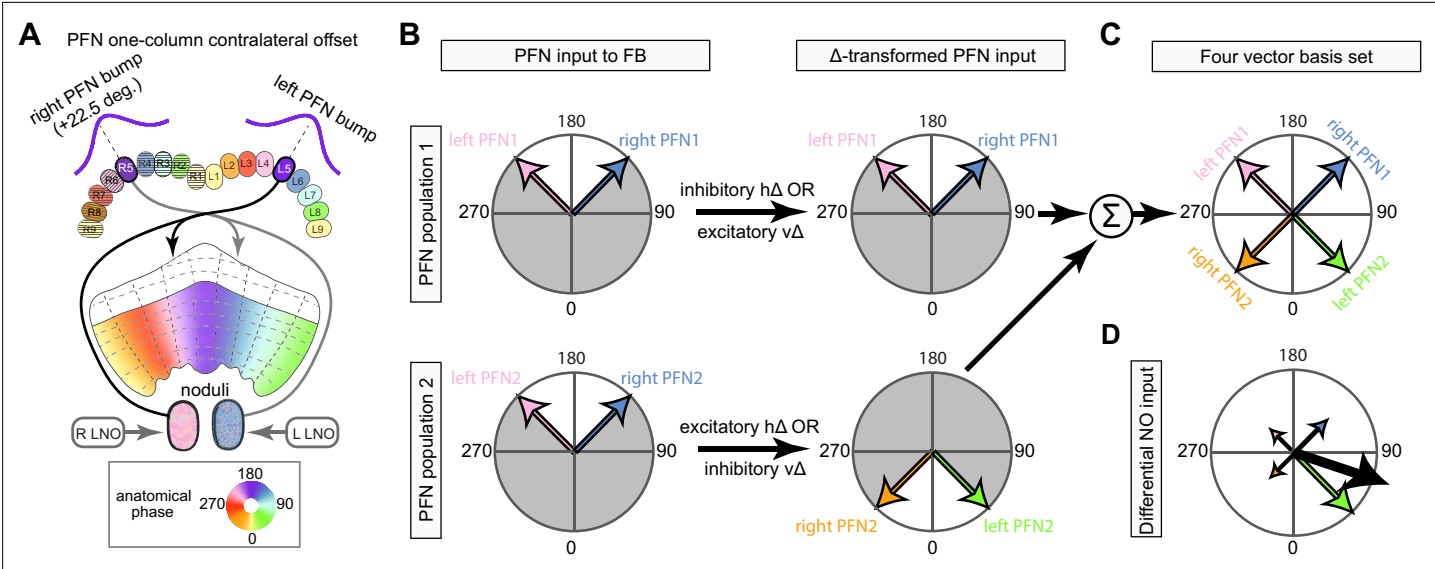

**Figure 70.** Conceptual model showing how two PFN populations, when combined with differential noduli input, could form a four-vector basis set whose summation could produce any vector. (**A**) Schematic of a protocerebral bridge-to-fan-shaped body (PB-to-FB) projection pattern with a one-column contralateral phase shift. (**B**) Phasor diagram interpretation showing how hΔ and vΔ motifs could be used to transform PFN bumps. For the first PFN population (top panels), an inhibitory hΔ or an excitatory vΔ could pass the PFN1 bump along while maintaining is phase. In contrast, for the second PFN population (bottom panels), an excitatory hΔ or and inhibitory vΔ could shift the PFN2 bump by 180°. For both PFN population 1 and 2, differential noduli (NO) input could shift their summed FB bump location by ±45°, but not outside this region (marked by gray portions of each circle). (**C**) Phasor diagram of a hypothetical downstream neuron that sums the input from the PFN1 and PFN2 populations. In this case, the downstream neuron's population activity would be the sum of four independent bumps, located 90° apart from one another and whose amplitudes are modulated by independent noduli input. These four bumps could act as a basis set for computing an arbitrary vector, thus freeing the resulting bump from the ±45° range. (**D**) Phasor diagram showing how modulating the amplitude of the left and right PFN1 and PFN2 populations could be used to compute an arbitrary vector (shown in black). Figure describes a conceptual model based partially on data from *Figure 30*, *Figure 31*, *Figure 33*, *Figure 34*, and *Figure 37*.

The online version of this article includes the following figure supplement(s) for figure 70:

**Figure supplement 1.** Dynamic updating of the four-vector basis set.

**Figure supplement 2.** The fan-shaped body (FB) network has the necessary connectivity and depth to form a basis set: bump propagation using simulated activity through actual FB connectivity.

– several PFNp subclasses (*Figure 27*) – with efference information related to rotational movements. In addition, many LNO types are downstream of pathways from vPNs (*Figure 5*), consistent with the use of optic flow-based self-motion signals (*Stone et al., 2017*). Furthermore, given that LNO types are the target of multiple input pathways, these neurons could carry combinations of sensory and motor signals to encode self-motion. Physiological recordings in behaving flies would be needed to test such hypotheses.

## Intra-FB circuit motifs for vector computation

Phase shifts of a single PFN neuron type could enable the generation of vectors that are within ±45° of the fly's instantaneous head direction. Intriguingly, the FB network appears to be wired to expand this range to allow for computations with vectors of arbitrary angles.

As shown in *Figure 70*, the vΔ and hΔ neuron types could, in principle, allow for vector computations across arbitrary azimuthal angles. In this example, we consider two hypothetical PFN populations (PFN1 and PFN2), both with 45° contralateral phase shifts, as shown in the phasor diagrams in *Figure 70B*. On their own, these PFN populations are limited to directional representations spanning 45° around the head direction signal that they inherit. However, an excitatory hΔ (or an inhibitory vΔ) would invert the PFN2 vectors, shifting them by 180°. Thus, if a postsynaptic neuron type were to sum the input from the noninverted PFN1 neuron population and an inverted PFN2 neuron population, it could form a representation of any arbitrary vector over the full 360° range, even though the PFN1 and PFN2 populations are individually range-limited (*Figure 70B and C*). A similar inversion

could happen at the level of the PB if one PFN population were to receive excitatory Δ7 input while the other received inhibitory Δ7 input, which is likely how the 180° separation of PEN_a and PEN_b population bumps is generated and maintained in the PB (*Green et al., 2017*; *Turner-Evans et al., 2017*). Together, the noninverted PFN1 and inverted PFN2 neuron populations form a basis set of four basis vectors, all separated by 90° (*Figure 70C*). As mentioned previously, the requirement for PFN neurons to have a positive firing rate prevents any single PFN population from forming a basis set on its own; instead, forming a basis set requires four independent bumps located at 90° intervals. When this situation is achieved, independent NO input could alter the relative amplitudes of bumps carried by each of the four PFN populations (inverted and noninverted PFN1 and PFN2 populations), enabling their sum to encode a vector with any angle (*Figures 70C and 25*) and could thus represent such independent vectors. Importantly, during navigation, the orientation of this set of four vectors would be dynamically updated with the head direction representation, such that any computations derived from these vectors would be independent of the fly's current head direction (*Figure 70—figure supplement 1*).

Could the intra-FB network support the construction of arbitrary vectors, which requires two layers beyond PFN input? The type-to-type network graph in *Figure 33A* suggests that there are many pathways within the FB's 2D grid that could potentially implement a four-vector basis set. We chose two arbitrary PFN types, PFNd, and PFNp_c, and used their connectivity with two downstream vΔ and hΔ neuron types – vΔK and hΔA – to illustrate how this might work (*Figure 70—figure supplement 2*). We also show how a downstream neuron type – PFL3 in this case – could sum the input from the vΔ and hΔ to represent arbitrary vectors determined by independent NO inputs to the left and right PFNd and PFNp_c populations (*Figure 70—figure supplement 2C*). Physiological investigations will be required to establish which of the FB's many pathways implement such computations, and whether or not the large number of these pathways is an indication of vector computations in different behavioral contexts.

## Connectome-driven assessment of models of path integration

Having established that the FB network of *Drosophila* could, in principle, compute arbitrary vectors, we now explore the potential utility of PB-FB phase shifts and intra-FB connectivity motifs for path integration. A variety of models have been proposed for path integration (*Benhamou, 1997*; *Benhamou et al., 1990*; *Benhamou and Séguinot, 1995*; *Bernardet et al., 2008*; *Cheung, 2014*; *Gallistel, 1990*; *Goldschmidt et al., 2017*; *Haferlach et al., 2007*; *Hartmann and Wehner, 1995*; *Issa and Zhang, 2012*; *Jander, 1957*; *Kim and Hallam, 2000*; *Kim and Lee, 2011*; *Maurer, 1998*; *Merkle et al., 2006*; *Mittelstaedt, 1983*; *Mittelstaedt and Mittelstaedt, 1973*; *Müller and Wehner, 1988*; *Stone et al., 2017*; *Vickerstaff and Di Paolo, 2005*; *Wittmann and Schwegler, 1995*). These models have several differences, including whether the home vector is stored in an allocentric reference frame or an egocentric reference frame, and whether it is stored using a 'static vectorial basis' or a 'dynamic vectorial basis' (for details, see *Heinze et al., 2018*; *Vickerstaff and Cheung, 2010*). Here we focus on models that store the home vector in an allocentric reference frame using a static vectorial basis (*Hartmann and Wehner, 1995*; *Stone et al., 2017*), which has been shown to have several theoretical advantages (*Cheung and Vickerstaff, 2010*; *Vickerstaff and Cheung, 2010*) and whose implementation is directly suggested by the FB's network architecture. Path integration models can be further divided into two groups according to whether the home vector is stored and read out as independent components or as a single vector.

An example of the first type of path integration model, which stores the home vector as two independent components, was recently put forward by *Stone et al., 2017*. This work combined anatomical and functional data from bees, including physiological recordings of optic flow-sensitive LNO neurons and EM data, to build an anatomically inspired model of path integration based on the projection and innervation patterns of CX neurons, but without access to their synaptic connectivity. The model utilized PB-FB phase shifts to read out a home vector and, importantly, could also account for holonomic motion, which occurs when animals move in directions that are not aligned with their head/body axis, an issue we return to below. At its core, this model and those derived from it (*Le Moel et al., 2019*) function by modulating the amplitude of left and right PFN bumps according to the insect's motion in the leftward or rightward direction, respectively. Integration of the left and right PFN activities can then store a home vector as two independent components. During readout, a

population of PFL neurons is assumed to compare the insect's current head direction to that of directions 45° to the left and to the right to decide which direction is closer to the implicitly stored home vector. This 45° 'functional offset' (phase-shift) was derived from physiological recordings, demonstrating that some LNO neurons function as optic flow sensors whose optimal expansion points are 45° to the left and right of the bee (*Stone et al., 2017*), a feature we return to below. While conceptually elegant, one major feature of this model is inconsistent with the anatomy and connectivity of the homologous neurons in the *Drosophila* connectome. In particular, the model requires that the left and right PFN bumps independently propagate to right and left PFL populations, respectively. This operation is unlikely to be supported by the FB columnar network since every neuron and neuron type postsynaptic to PFN neurons receives input from both the left and right populations (*Figure 35*).

In the next two sections, we use the additional anatomical and connectivity information provided by the CX connectome to propose two conceptual models for computing an allocentric TV vector whose integration could be stored as a single home vector. The first model builds on the work of *Stone et al., 2017* and uses PFN offsets to simplify home vector computation. The second model is more relevant to walking insects and incorporates a head-to-body coordinate transformation to compute the fly's TV vector. In both cases, the key computation performed by the FB network is a coordinate transformation that ensures that egocentric velocity signals and allocentric directional representations are directionally aligned.

## Computing an allocentric TV vector using head-centered optic flow sensors during flight

Flying insects are thought to perform visual odometry by relying on optic flow sensors to estimate their velocity relative to the ground (*Leitch et al., 2020*; *Srinivasan, 2014*; *Srinivasan, 2015*), consistent with leg-based cues being of little use and motor signals being unreliable in the face of external perturbations, like gusts of wind. In addition, during flight many insects make banked turns involving body rolls that are accompanied by gaze-stabilizing head rotations that keep the head near the horizontal plane (*Kim et al., 2017b*; *Muijres et al., 2015*). Importantly, flight trajectories often contain a significant sideslip component as well, during which the insect's TV is in a direction that is different from that of its head-body axis (*Braun et al., 2012*).

The FB's recurrent circuitry described above could use self-motion information to compute a flying insect's allocentric TV vector. One potential model is shown in *Figure 71*. This model is composed of two PFN neuron types that receive independent input from two hypothetical LNO neuron types,

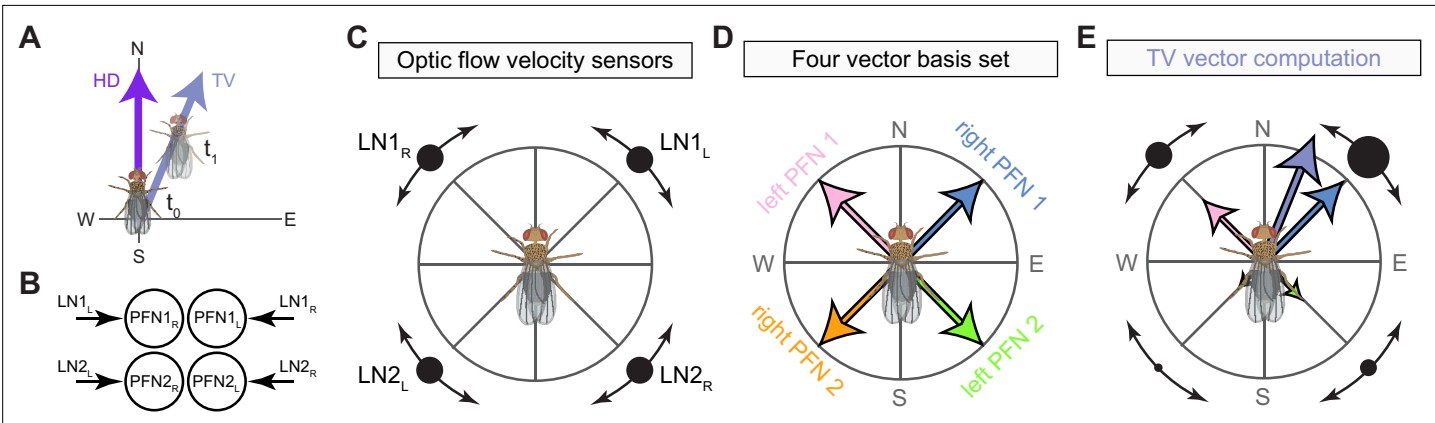

**Figure 71.** A conceptual model that computes an allocentric translational velocity vector using head-centered optic flow sensors during flight. (**A**) Illustration of a fly whose head and body direction are pointed north and whose translational velocity vector is 22.5° east of north. (**B**) Schematic of noduli circuitry, showing that the left and right PFN1 and PFN2 populations receive input from right and left LN1 and LN2 neuron types, respectively. (**C**) LN1 and LN2 neuron types are those described by *Stone et al., 2017*. They function as optic flow-based velocity sensors with preferred expansions points spaced at 45° intervals around the fly's head. (**D**) Schematic of a four-vector basis set. Importantly, note that each PFN vector points in the same direction as its upstream LN neuron's preferred optic flow direction. (**E**) Schematic showing how the four-vector basis set, whose vectors are amplitude-modulated by the LN velocity sensors, can compute the fly's translational velocity vector. In this case, because the fly is moving just east of north, LN1_L is driven most strongly, which increases the amplitude of right PFN1 population (blue vector). When properly calibrate, summing the amplitude-modulated PFN vectors compute the fly's translational velocity vector. Figure describes a conceptual model based on previous work (*Stone et al., 2017*).

LN1 and LN2 (*Figure 71B*), for which there are multiple candidates (*Figure 25*). It exploits the FB network's ability to form a set of four basis vectors to compute a single TV vector. To do so, it employs the optic flow sensors described by *Stone et al., 2017* – with their preferred expansion points spaced at 45° intervals around the fly's head – to modulate the amplitudes of the four basis vectors such that their sum encodes an instantaneous allocentric TV vector (*Figure 71C–E*). Importantly, this model relies on the fact that the basis vectors and optic flow sensors are directionally aligned (*Figure 71C and D*). That is, at every moment in time, each bump in the basis set has its amplitude modulated by a velocity input that senses movement in the same direction as encoded by the bump. Much like the model of *Stone et al., 2017*, this model can account for holonomic motion (i.e., an animal's movements in directions not limited to its heading and head direction). Another feature of this model is that it should be insensitive to head movements in the yaw plane since the optic flow sensors and FB basis vectors are both in head-centered coordinates. A recent study found that, similar to the optic flow sensors described above, some PFN neuron types and their LNO inputs are preferentially tuned to air flow oriented ~45° to left or right of the fly's head (*Currier et al., 2020*), providing for a second potential velocity estimate whose tuning is aligned to PFN basis vector. The next conceptual model explores how this might work in walking insects, when the velocity sensors may be in a body-centered reference frame while the directional representation is in a head-centered allocentric reference frame.

## Computing an allocentric TV vector using body-centered velocity estimates during walking

Could the FB network compute an instantaneous TV vector in cases where its velocity and directional estimates are in different reference frames? The model shown in *Figure 72* explores such a scenario using a hypothetical example of a walking fly whose velocity estimates are computed using cues that operate in an egocentric, body-centered reference frame. These velocity estimates could be derived from motor efference copies or proprioceptive cues, and we assume the existence of estimates for both forward (parallel to the body axis) and sideslip (perpendicular to the body axis) velocity. Computing a TV vector in this scenario is more complicated than in the previous model because the direction of the head and that of the body are not necessarily aligned, which requires a head-to-body coordinate transformation. As shown in *Figure 72B–D*, this model uses the head-body angle to compute the total TV vector as the sum of two components, which represent the distance traveled parallel and perpendicular to the fly's body axis. To compute the parallel and perpendicular components of the TV vector, the model uses two basis sets that receive NO input related to the head-body angle as well as either a forward or sideslip velocity signal.

A circuit for computing the component of the TV vector parallel to the fly's body axis is shown in *Figure 72B*, which involves two calculations that occur in parallel. The circuit recruits two independent PFN populations, one to encode movement in the forward direction, and the other for movement in the backward direction. A velocity signal increases the amplitude of the two PFN vectors that point in the direction the fly is moving (for the example situation in *Figure 72B*, in the forward direction), resulting in a vector whose amplitude encodes velocity and whose direction is either the fly's head direction (*Figure 72B*), or its reverse head direction, which would be captured by the PFN2 population in *Figure 72B*. At the same time, an input related to the head-body angle transforms the head-centered vector into a body-centered vector, as described above (*Figure 69*). The result is a single vector encoding the component of the fly's movement parallel to the body's axis in either the forward or backward direction. As mentioned above, this sort of computation could employ gain fields (*Andersen et al., 1993*; *Pouget and Sejnowski, 1997*; *Salinas and Abbott, 2001*), but with the transformed representation (i.e., body direction) being scaled by the fly's velocity in the process.

The component of the TV vector that is perpendicular to the fly's body axis could be computed using the same circuitry as above, but with right/left sideslip velocity signals (*Figure 72C*) instead of forward/reverse velocity signals. As shown in *Figure 72—figure supplement 1*, such a circuit would work regardless of whether the fly is sideslipping right or left or whether its head is to the right or left of the body axis. The output of these two circuits could then be summed to compute a single vector that encodes the fly's instantaneous TV in an allocentric reference frame (*Figure 72D*).

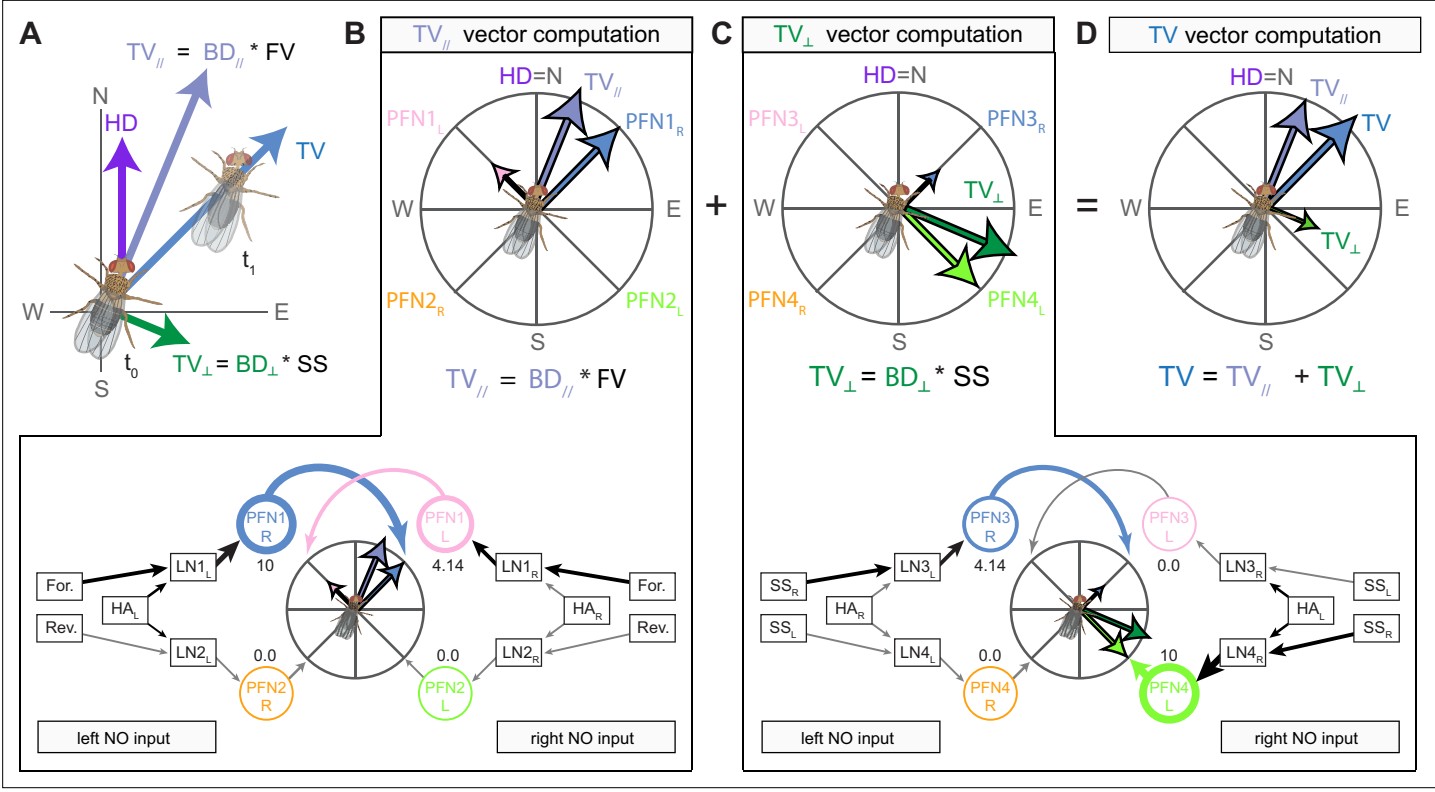

**Figure 72.** A conceptual model that computes an allocentric translational velocity (TV) vector using body-centered velocity estimates during walking. (**A**) Illustration of a walking fly whose head direction, body direction (BD), and TV direction are all different. The fly's head is pointing north, its body 22.5° east of north, and it's walking northeast (that is, 45° east of north). The fly's TV vector can be computed by summing the component of its movement parallel to its body axis (TV$_{//}$) with the component of its movement perpendicular to its body axis (TV$_\perp$). Circuits for computing these quantities are shown in (**B**) and (**C**), respectively. (**B**) Circuit for computing the component of the TV vector parallel to the fly's body axis. As shown in the bottom panel, this circuit uses a four-vector basis set whose PFN vector amplitudes are modulated by lateral accessory lobe-noduli (LAL-NO) inputs that encode whether the head is left (HA$_L$) or right (HA$_R$) of the fly's body axis as well as either a forward (For.) or reverse (Rev.) velocity signal. The firing rate of each PFN population is noted below each PFN node. Arrow width is proportional to firing rate. Gray arrows indicate neurons that are silent. Note that head angle input alone is insufficient to bring the LN neurons to threshold, but it can boost PFN firing when combined with a velocity input. In this case, LN2$_L$ remains silent despite receiving a head angle input from HA$_L$, and LN1$_L$ is strongly driven by both the forward velocity signal and HA$_L$. LN1$_R$, meanwhile, is moderately driven by the forward velocity signal alone. This conditional effect of the head angle input could be achieved in other ways, but the core conceptual model would remain the same. In all cases, the circuit would require proper calibration for the vector summation to accurately compute the fly's TV$_{//}$ vector. (**C**) Circuit for computing the component of the TV vector perpendicular to the fly's body axis. The circuit shown in the bottom panel operates like that described in (**B**), but the forward and reverse velocity signals have been replaced by left (SS$_L$) and right (SS$_R$) sideslip velocity signals. As in (**B**), a head angle input alone is insufficient to bring LN neurons to threshold. Note that these circuits function regardless of which direction the fly's head is facing and which direction the fly is moving, as detailed for four other examples in *Figure 72—figure supplement 1*. (**D**) Phasor diagram showing how summing the output from the circuits in (**B**) and (**C**) yields an exact TV vector whose integration would compute the path integration vector. Figure describes a conceptual model.

The online version of this article includes the following figure supplement(s) for figure 72:

**Figure supplement 1.** The circuit for computing TV$_\perp$ operates independent of the fly's head-body angle and which direction the sideslip component is towards.

## Summary: TV computation

The conceptual models described above – one for flight and the other for walking – could, in principle, compute an allocentric TV vector whose integration would yield an exact home vector. To accomplish this, the models use coordinate transformations to ensure that allocentric vectors are directionally aligned with the egocentric velocity estimates that control their amplitudes. While these particular models highlight the general utility of such transformations, the FB circuitry could, in principle, accommodate many similar models. In addition, it is possible that animals structure their movements during outbound paths to simplify the computation of the TV vector. For example, if an animal were to only move forward during outbound paths, then circuit components dedicated to encoding backward

motions would not be needed by the path integration circuit, a feature explicitly used by the model of *Stone et al., 2017*. Similarly, it is possible that an exact solution is not always required to perform path integration. For example, if a model generates errors that tend to cancel out during the integration process, the home vector can still effectively guide behavior. The local search behavior of foraging *Drosophila*, for example, involves relatively short loops that may not require a precise accounting of the goal location (*Brockmann et al., 2018*; *Haberkern et al., 2019*; *Kim and Dickinson, 2017a*). In most situations, the fly should also be able to use local sensory cues in addition to path integration during such search behaviors. Finally, it is possible that egocentric velocity signals could come prealigned to FB bumps, assuming that the LAL could implement the trigonometric functions needed to scale velocity signals according to, for example, head-body angle. Taken together, these models highlight the connectome's ability to inspire novel, implementation-level hypotheses about network computation. They also provide a framework for generating many similar models, with specific implementations that likely depend on cell type, species, and behavioral need. Ultimately, evaluating models like these necessarily requires physiological recordings from animals in specific behavioral contexts. Indeed, two contemporaneous studies have discovered direct physiological evidence that FB circuits compute the fly's TV and have independently proposed theoretical models that are conceptually similar to those described above (*Lu et al., 2020b*; *Lyu et al., 2020*). Yet, it is currently unclear if the output of this computation encodes the fly's TV vector or just the phase of this vector. Similarly, how the type-to-type variability in PFN phase shift magnitude (*Figure 34*) affects these computations requires future study. Finally, it is also possible that tangential neurons carrying feedback or self-motion signals (*Weir and Dickinson, 2015*; *Weir et al., 2014*) could scale the magnitude of these vectors within the FB network itself.

## Potential mechanisms for TV integration and home vector storage

Theoretical work has suggested several potential ways to integrate TV vectors. First, TV could be integrated and stored using two separate circuits: a ring attractor that encodes the angle of the home vector and a line attractor that encodes the length of the home vector (*Hartmann and Wehner, 1995*). Rather than keeping track of the distance traveled in each allocentric direction, this network would shift the columnar location of an activity bump to encode the phase of the home vector. Solutions like these seem unlikely to be implemented by the FB since they require FB-centered attractors with circuit motifs for shifting the bump, similar to those found in the EB-PB attractor, which we see no evidence for in the FB network. Second, the model of *Stone et al., 2017* employed 18 neurons per FB column and used structured recurrent connections between them to integrate and store a two-component home vector (*Stone et al., 2017*). The hemibrain connectome provides little evidence for such structured recurrent PFN connections, especially in the NO, where some PFN types show all-to-all connectivity. Finally, the conceptual models described above allow for the computation of a single TV vector (*Figures 71 and 72*), suggesting that the FB network could simply integrate the corresponding activity bump and store the resulting home vector directly. In doing so, this integration processes would function by keeping track of the distance traveled in each allocentric direction. How might the network integrate the TV vector and store the resulting home vector?

Integration and storage could occur through several complementary mechanisms operating at different scales, from changes in synaptic strength or the excitability of individual neurons to persistent activity (*Major and Tank, 2004*) at the single neuron (*Yoshida and Hasselmo, 2009*) or network level (*Aksay et al., 2007*). In addition, integration and storage mechanisms may vary across species and environmental context depending on the animal's needs. For example, desert ants can maintain multi-day memories of multiple goal vectors to food sites and remember home vectors over 1–2 days (*Wehner et al., 2004*). This sort of long-term maintenance would favor stable storage mechanisms, such as changes in synaptic strength. In contrast, *Drosophila* performing relatively brief local searches close to a food source may rely on short-term mechanisms that could involve persistent neural activity. The connectome alone does little to constrain the space of possible storage mechanisms, but it can provide information regarding the likely site of storage and inspire a few conceptual models for how the vector could be stored.

Several considerations narrow the potential site of home vector storage in *Drosophila*. In the framework of the above conceptual models, the home vector would be stored downstream of the four-vector basis sets used to compute the TV vector. The PFL2 and PFL3 neuron types are well positioned

to read out the home vector by comparing it to the fly's instantaneous head direction (see section below), suggesting that the home vector is perhaps stored by neuron types that provide inputs to the PFL neurons. The PFL neuron types could also store the home vector themselves. Some insects are likely to maintain more than one goal vector (*Dacke and Srinivasan, 2008*; *Mangan and Webb, 2012*), but the PFL neurons could store these different goal vectors through input-synapse-specific presynaptic (*Goldschmidt et al., 2017*) or postsynaptic plasticity. In addition to direct PFN input, PFL2/3 neurons receive shared input from a handful of hΔ types, several FC2 types, one or two vΔ types, and many FB tangential neuron types, each of which could also potentially store a home vector.

Several potential storage mechanisms seem plausible. Many hΔ neuron types have within-type recurrent connections, forming small loops that connect pairs of hΔ neurons that encode directions 180° apart. If the biophysical properties of these neurons allowed for graded, persistent activity, and hΔ neurons have inhibitory connections, each column-pair could encode the direction traveled along one dimension. Alternatively, while the FC neurons providing input to PFL2/3 neurons largely lack within-type recurrent connections, they could maintain a vector in working memory through graded changes in their excitability or activity. Finally, the FB's tangential neurons could potentially store home vectors through column-specific plasticity, as is known to occur between ring neurons and EPG neurons. In general, some recurrent architectures may allow for the storage of home vectors (*Wittmann and Schwegler, 1995*), but an FB ring attractor, if it were to exist, would likely not allow for home vector storage since these networks have the undesirable property of forming a single activity peak at the expense of the activity in distance columns that would be needed to fully encode the home vector.

Overall, although the connectome can do no more than rule out some circuit implementations of how the home vector might be stored, it should prove useful in prioritizing a search for the likely neural targets for such a function. It is important to note that the entire circuitry described above must function in different modes depending on the animal's behavioral needs – integrating direction and distance traveled to update the home vector when the fly is searching, but switching to reading out the home vector when the fly is attempting to return to a previously visited spot. The likeliest candidates for such behavioral mode switching are the FB's tangential neurons.

## Reading out the home vector

Once formed, how might an insect 'read out' the home vector to return to its goal location? In our formulation, the home vector points *from* the nest *to* the insect's current location. Returning home, then, requires that an insect move in a direction *opposite* to the home vector. To accommodate the other behaviors and computations that these circuits are likely to be involved in, we refer to the home vector as the 'stored vector,' which is read out to orient the insect along a 'goal vector' (*Figure 73*, bottom panel). However, unlike an ant or bee, the fly is not a central place forager. Thus, 'goal' in this context refers only to a spot that the fly is likely to return to during a local search, such as a food source (*Brockmann et al., 2018*; *Kim and Dickinson, 2017a*). PFL neurons are generally regarded as the major columnar output of the FB network (*el Jundi et al., 2015*; *Hanesch et al., 1989*; *Heinze et al., 2013*; *Heinze and Homberg, 2008*; *Homberg, 1985*; *Lin et al., 2013*; *Skutt-Kakaria et al., 2019*; *Wolff et al., 2015*). Their PB-FB offsets strongly implicate them in reading out stored vectors in ways first proposed by theoretical work (*Hartmann and Wehner, 1995*; *Wittmann and Schwegler, 1995*) and then, at the implementation level, by *Stone et al., 2017*. In particular, PFL neurons may use their PB-FB phase shifts to compare the fly's instantaneous head direction, which they receive in the PB, to that of the stored vector, which they may receive in the FB, to generate appropriate motor commands to guide the fly to its goal. In doing so, they effectively generate egocentric motor commands based on allocentric directional variables. Interestingly, each of the three PFL types have characteristic phase shifts that strongly predict their involvement in generating distinct motor commands (*Figure 73*).

PFL2 neurons may use their 180° phase shift and bilateral LAL projections to increase the fly's forward velocity when its heading is directly away from the stored vector, which in our formulation is towards the goal location (*Figure 73A*, bottom panel). Unlike the other PFL types, PFL2 neurons receive only a single bump as input in the PB (*Figure 39*). This suggests that the population cannot make left versus right activity comparisons. In agreement with this, individual PFL2 neurons make bilateral projections to the left and right LAL. Because of their 180° phase shifts, the PFL2 population activity will be largest when the fly is heading directly towards its goal location. The above

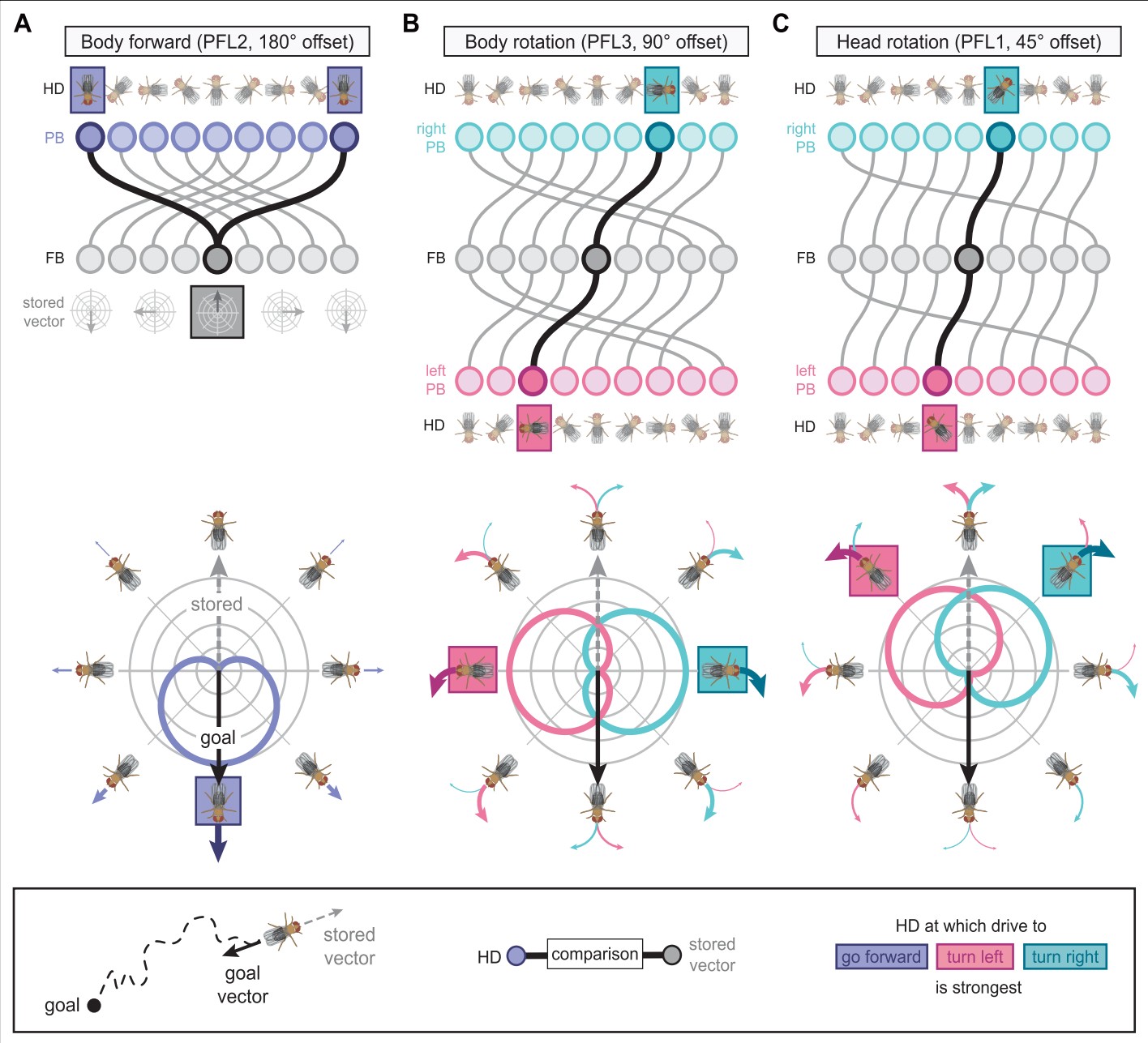

**Figure 73.** PFL neurons could generate egocentric motor commands by comparing the fly's allocentric head direction to an allocentric vector stored in the fan-shaped body (FB). (**A**) PFL2 neurons could use their 180° protocerebral bridge (PB)-FB phase shift to generate a forward velocity signal that is largest when the fly is oriented towards the 'goal vector,' which in our formulation is away from the 'stored vector' (see bottom panel and 'Discussion'). PFL2 neurons sample a single bump in the PB and individual PFL2 neurons project to both the left and right lateral accessory lobe (LAL), consistent with a bilateral velocity signal like forward walking. Top panel shows a schematic of the PFL2 180° phase shift between PB glomeruli (top row) and FB columns (bottom row). In this example, the stored vector points due north. To return to the goal location, PFL neurons compare the fly's instantaneous head direction to the stored vector. The 180° phase shift ensures that PFL2 output will be largest when the fly is oriented towards the goal direction (and opposite the stored vector). (**B**) Similar to (**A**), but for the PFL3 neuron type and its 90° PB-FB phase shift. Unlike PFL2 neurons, PFL3 and PFL1 neurons (**C**) sample head direction bumps from the left and right PB, and individual neurons project to either the left or right LAL, consistent with motor commands with a left/right asymmetry, such as turning. In the case of PFL3 neurons, the 90° phase shift ensures that the left PFL3 population will be most active when the fly is 90° to the right of the goal direction. Similarly, the right PFL3 population will be most active when the fly is 90° to the left of the goal direction. If we assume that the right PFL3 neurons generate right turns and left PFL3 neurons generate left turns, then the motor command would act to align the fly's heading with that of the goal direction. (**C**) Same as in (**B**), but for PFL1 neurons and their 45° PB-FB phase shift. Note that in all cases, the PB-FB phase shifts are an idealized version of those from *Figure 39*. The actual PFL phase shifts are not as stereotyped, since the phase shifts are continuous in anatomical space, unlike the discrete mapping schematized here.

*Figure 73 continued on next page*

*Figure 73 continued*

The online version of this article includes the following figure supplement(s) for figure 73:

**Figure supplement 1.** Numerosity and systematic asymmetries in synapse counts across columns may set up a potential 'default goal vector' through the PFL neurons.

characteristic suggests that PFL2 neurons are ideally suited to generate a motor command related to forward velocity.

PFL3 neurons may use their 90° phase shifts and lateralized LAL projections to orient the fly towards the goal. As shown in *Figure 73B*, their 90° offset predicts that the left and right PFL3 populations will have their maximum activity when the fly is 90° to the right or left of the goal direction, respectively. If the left PFL3 population generates left turns and the right PFL3 populations generated right turns, then the orienting behavior of the fly will have two equilibrium points: a stable equilibrium that occurs when the fly is oriented towards the goal direction and an unstable equilibrium when the fly is oriented in the opposite direction. This sort of read out would ensure that flies orient directly towards the goal location. It is additionally possible that across-column inhomogeneities in the EPG-> PFL synaptic profile (*Figure 73—figure supplement 1*) and in the PFL-> LAL network (*Figure 64*) may provide the fly with a 'default goal' in the absence of any FB input, similar to a hypothesis recently advanced in an independent study (*Rayshubskiy et al., 2020*). The 45° offset of PFL1 neurons may serve a related function, although they target distinct downstream neurons compared to PFL2/3 (*Figure 57C*). One possibility is that the PFL2/3 neurons affect body orientation while the PFL1 population controls a separate variable, such as sideslip or head-body angle. Ultimately, it is also important to remember that brain regions like the LAL and CRE house complex recurrent networks with inter-hemispheric pathways that are likely to be inhibitory (see Output sections) (*Iwano et al., 2010*). These networks are likely to play a major role in the transformation of PFL population activity into motor commands for the fly, something that our hypotheses do not incorporate.

## Summary: vector computations in the FB

The discussion above supports the notion that the FB network has the computational capacity to compute, store, and read out vectors in support of goal-directed navigational behaviors. While we have focused on path integration as a canonical vector-based computation, *Drosophila* are known to perform several other behaviors that may rely on the formation of goal vectors, including: local search, a path-integration-based foraging strategy (*Corrales-Carvajal et al., 2016*; *Dethier, 1957*; *Kim and Dickinson, 2017a*); menotaxis, where a constant heading is maintained relative to an arbitrary goal direction to generate straight trajectories that support long-distance dispersal (*Giraldo et al., 2018*; *Green et al., 2019*; *Haberkern et al., 2019*; *Leitch et al., 2020*); place learning, which requires associating visual cues with the presence of a cool spot in an otherwise hot 2D environment (*Melnattur et al., 2020*; *Ofstad et al., 2011*); and the detour paradigm, where flies orient towards directions associated with attractive landmarks even after they have disappeared (*Neuser et al., 2008*). In addition, ethologically based studies in behaving insects have established a range of vector-based behaviors, from long-distance migrations that require a time-compensated sun compass (*Heinze and Reppert, 2011*; *Perez et al., 1997*) to the waggle dance that bees use to communicate the distance and direction of a food source (*Frisch, 1967*). The ability of some insects to store multiple goal vectors and the fact that different insect species may use vector computations to support distinct behaviors has important implications for FB circuits. The FB may have evolved as a general vector calculator that can be co-opted, whether by evolution or in support of distinct behaviors, to support vector-based navigation strategies generally. In support of this idea, FB circuits, neuron types, and motifs are highly conserved across insects (*Strausfeld, 2012*), including PB-FB phase shifts (*Sayre et al., 2021*). Additionally, the ability of some insects to store multiple goal vectors requires mechanisms for switching between them, a function perhaps mediated by the large class of FB tangential neurons that could convey context and state information to the columnar networks involved in vector operations.

## Beyond navigation: the CX as a multifunctional network for context-based action selection

While we have focused much of our discussion on column-specific computations supporting vector navigation, the CX also receives input from over 150 distinct tangential neuron types. In the sections below, we briefly highlight these neurons' role in sensorimotor processing, memory-guided decision-making, circadian rhythms, sleep-wake control, and nutrient homeostasis. Together, these findings suggest that the CX operates as a multifunctional network supporting state- and context-dependent action selection for high-level behavioral control.

### Sensorimotor processing

Consistent with the CX's involvement in navigation, several studies have implicated FB tangential neurons in sensorimotor processing. For example, ExFl1 neurons (*Homberg, 1994*; *Liu et al., 2006*; *Weir et al., 2014*; *Young and Armstrong, 2010b*), which are likely FB2B_a and/or FB2B_b neurons (*Figure 41A*), are strongly modulated by whether or not the fly is flying and are tuned to progressive optic flow (*Weir et al., 2014*), providing a potential indication of the fly's current sensory and motor state. Similar activity patterns may be expressed by several other FB types as well (*Weir and Dickinson, 2015*). In addition, a recent study focused on the LH identified an FB tangential neuron type called PV5k1 (FB2H_a, FB2H_b, and/or FB2I_b) whose activation during closed-loop visual conditions leads to a reduction in the fly's wingbeat frequency (*Dolan et al., 2019*). Sensorimotor signals like these are well positioned to influence CX-driven motor commands based on the fly's immediate sensory environment and ongoing motor state.

### Memory-guided decision-making

Flexible behavior also requires animals to respond to their immediate sensory surroundings by evaluating past associations regarding the valence and novelty of available sensory cues. To investigate this, we focused on tracing pathways between the MB – the fly's main learning and memory center – and the CX (*Figures 46 and 47*). In agreement with results from a companion manuscript focusing on the MB (*Li et al., 2020*) and trans-Tango-based circuit mapping (*Scaplen et al., 2021*), we found extensive pathways leading from MBONs to FB tangential neurons. In the context of navigation, the MB is considered a potential source of visual snapshot memory, which may allow insects to base their navigation decisions on remembered panoramic views (*Collett and Collett, 2018*; *Sun et al., 2020*). Consistent with this general notion, some FB tangential neuron types in FB layers 2 and 8 have been proposed to play a major role in visual learning (*Liu et al., 2006*). In addition, recent studies have implicated MB-to-CX pathways in behaviors other than navigation. For example, MB-to-CX circuits may be important for experience-dependent alcohol preference (*Ojelade et al., 2019*; *Scaplen et al., 2021*; *Scaplen et al., 2020*). In addition, MB-to-CX circuits are involved in consolidating courtship experience into long-term memory (*Dag et al., 2019*). The sheer number of connections between MBONs and FB tangential neurons suggests that this prominent pathway is involved in many behaviors that make use of valence and novelty signals extracted from past associations that the fly has made with its current sensory surroundings.

### Circadian influence on the CX

Animals also select their actions based on latent environmental variables, such as the time of day, which are predictive of environmental conditions like temperature and humidity. Flies are most active around dawn and dusk, and show consolidated periods of inactivity throughout the night and during a daytime siesta (*Dubowy and Sehgal, 2017*). This daily rhythm is imposed by outputs from the circadian network and functions to restrict behavior to appropriate times of day. Previous studies have identified a population of anterior-projection DN1 clock neurons that convey circadian information through TuBu neurons to EB ring neurons (*Figures 6C and 7D*; *Guo et al., 2018*; *Lamaze et al., 2018*). Thus, CX circuits are likely to receive circadian information that could be used to select behaviors according to time of day (*Liang et al., 2019*). Whether circadian pathways target other regions of the CX requires further investigation. In addition to receiving circadian inputs that could affect rest-activity rhythms, considerable evidence suggests that CX circuits are involved in tracking internal states, such as sleep need and nutritive state, which we turn to next.

## Sleep-wake control

While its functions remain largely unknown, sleep is associated with a variety of processes in *Drosophila*, including synaptic homeostasis (*Bushey et al., 2011*), memory formation and consolidation (*Berry et al., 2015*; *Dag et al., 2019*; *Donlea et al., 2011*), changes in gene expression (*Cirelli et al., 2005*; *Zimmerman et al., 2006*), and several metabolic processes (*Kempf et al., 2019*; *Vaccaro et al., 2020*). Sleep in flies is behaviorally defined as a reversible state of immobility that is homeostatically regulated and associated with an increased arousal threshold (*Hendricks et al., 2000*; *Shaw et al., 2000*). It is marked by drastic changes in brain-wide activity patterns (*Nitz et al., 2002*; *Tainton-Heap et al., 2020*; *Yap et al., 2017*). The neural circuits involved in tracking sleep need and inducing sleep are thought to partially reside in the CX. In particular, a heterogeneous population of FB tangential neurons labeled by the R23E10 GAL4 line induces sleep when activated and tracks sleep need through changes in baseline firing rate and intrinsic excitability (*Donlea et al., 2011*; *Pimentel et al., 2016*) (but see also *Tainton-Heap et al., 2020*). Similarly, ER5 ring neurons track sleep need, and reciprocal connections between the EB and dFB are hypothesized to form a core circuit for homeostatic control of sleep (*Donlea et al., 2018*; *Liu et al., 2016*). Counteracting these sleep-promoting neurons are wake-promoting DANs in the dorsal FB that are thought to promote wakefulness by inhibiting R23E10 neurons (*Ni et al., 2019*; *Pimentel et al., 2016*). Our connectomic analysis revealed extensive reciprocal connections between putative sleep- and wake-promoting populations within the dFB, which could function as a 'flip-flop' switch to ensure that only one population is active at a time (*Saper et al., 2010*). In addition, we identified a large number of previously undescribed pathways leading to and from sleep-wake neuron types whose potential involvement in sleep-wake control requires future investigation (*Figures 51–55*), including reciprocal pathways connecting neurons in the EB with those in the dorsal FB (*Figure 53*; *Donlea et al., 2018*; *Liu et al., 2019*).

Several limitations of the hemibrain dataset are notable in the context of sleep. Neurons that show structural changes as a function of the fly's sleep-wake history (*Bushey et al., 2011*), such as ER5 (*Liu et al., 2016*), could have sleep state-dependent connections different from those described here; similarly, at present, the hemibrain connectome does not include reconstructed glia, which are also known to be involved in sleep-wake control (*Blum et al., 2021*; *Sengupta et al., 2019*); lastly, the hemibrain dataset cannot resolve the presence of gap junctions, which may also be important for sleep-wake control (*Troup et al., 2018*).

## Nutrient homeostasis

Recent studies have suggested that the CX is involved in internal state-based action selection beyond sleep-wake control. Within the EB, a population of ring neurons allows flies to assess the nutritive value of sugars, independent of their taste (*Dus et al., 2013*; *Park et al., 2016*). Similarly, tangential neuron types in the dorsal FB have been implicated in feeding decisions based on the nutritive value of foods, and they may incorporate past experience into these computations (*Sareen et al., 2020*). And vΔA_a columnar neurons, which innervate the AB and dFB, show oscillatory dynamics that depend on hemolymph glucose levels, and altering vΔA_a activity levels affects fructose preference (*Musso et al., 2021*). Together, these studies implicate CX circuits in nutrient homeostasis, a process important for successful foraging based on the fly's metabolic needs (*Corrales-Carvajal et al., 2016*).

## Circuit motifs for high-level behavioral control and action selection

The need for high-level behavior selection may explain the potential interactions of circuits related to navigation, feeding, circadian rhythms, and sleep. Hungry flies, for example, are known to forgo sleep in favor of foraging (*Keene et al., 2010*). Similarly, both sleep and feeding are known to be under circadian control (*Dubowy and Sehgal, 2017*; *Murphy et al., 2016*; *Xu et al., 2008*), biasing their occurrence to appropriate times of day. Based on these considerations and the experimental evidence summarized above, it seems likely that the CX operates as a multifunctional network that can be dynamically reconfigured (*Marder, 2012*) to support a variety of goal-directed behaviors based on immediate sensorimotor variables, learned associations, time of day, sleep need, nutritive state, and other as-yet-unknown inputs. Such a view of the CX is consistent with the variety of neuromodulator and peptides released by FB neurons (*Kahsai et al., 2012*; *Kahsai et al., 2010*; *Kahsai and Winther, 2011*).

Our connectomic analysis identified circuit elements and motifs that may support appropriate action selection. Most notably, many tangential neuron types, including EB ring neurons, form dense recurrent connections, both within neurons of a type and across distinct neuron types. For example, the FB's tangential neurons in layer 6 that have been implicated in sleep-wake control are highly recurrently connected. It is possible that some of these neurons or other neurons in their layer are involved in decision-making related to feeding (*Musso et al., 2021*; *Sareen et al., 2020*). If so, inhibitory interactions between these different tangential neurons may – akin to the interactions of ring neurons for sensory control of the fly's compass – enable the fly to select appropriate actions based on internal need. Related to this, recent studies have reported oscillatory activity in ER5 ring neurons related to sleep-wake control (*Raccuglia et al., 2019*; *Yap et al., 2017*), but how the highly recurrent networks in the EB and FB might support such oscillations remains to be determined. One possibility is that all-to-all inhibition between ring neurons in the EB could, with the appropriate inhibitory conductances, induce such patterns of activity. A different issue raised by the highly recurrent architecture of sleep-wake networks concerns how activity may propagate in these networks. Artificial stimulation of neurons within such potentially self-regulating networks may trigger downstream activity that is never seen in more naturalistic situations, confounding the interpretation of experimental results. Testing such ideas will require a finer-resolution analysis of the role that these neurons play in the action selection process.

How might the CX's columnar architecture support these distinct behaviors? Links between the CX's role in sleep and navigation have begun to be explored both experimentally (*Donlea et al., 2018*; *Liang et al., 2019*) and computationally (*Valle et al., 2020*), but the CX connectome suggests that the number of pathways and neuron types that connect circuit elements known to be involved in these functions may have been underestimated. For example, the dFB tangential neurons involved in sleep-wake control contact many columnar neuron types (*Figure 51*). Although we believe this columnar structure – and the FB's 2D grid more generally – to be convenient for vector computations, why this columnar structure may be needed for sleep-wake control or for feeding- and satiety-related computations remains mysterious. One possibility is that head direction or traveling direction signals may be used as proxies for tracking the quality or quantity of the fly's waking experience, perhaps to estimate sleep and/or nutritional need. Alternatively, the FB's navigational signals may be inherently activity-promoting since they likely drive premotor neurons in the FB. If so, these navigational signals may require suppression to establish a sleep state or to enable a hungry fly to stop on a patch of nutritive food. Another possibility, suggested previously (*Donlea et al., 2018*), is that tangential neurons may gate incoming sensory information, which could promote sleep or perhaps encourage a hungry fly to continue a feeding bout by ignoring distractors. Ultimately, if the columnar neurons are the main output of the CX, as seems likely, the FB's tangential neurons must impact behavior through them.

Considering that the highest layers of the FB are associated with modulating the fly's activity depending on sleep state and satiety levels, the connectivity pattern within the FB suggests that information about the fly's current navigational state may enter the FB ventrally, that additional processing may happen in the middle layers, which receive considerable input from the MB, and that this processing may then determine the fly's next actions (or inaction) in the dorsal layers. An additional possibility suggested by the flow of bump information from ventral to dorsal layers of the FB, and by the diffusion of columns in the dorsal layers, is that the specificity of actions is organized along the vertical axis of the FB, with oriented actions modulated and signaled by output neurons originating in the middle layers (see next section) and the fly's overall state of activity modulated in directionally nonspecific ways by the highest layers.

Compared to the vector computation models suggested by the CX's columnar structure, deriving connectome-inspired insights into the function of the CX's action selection networks proved more challenging. One reason for this is that most FB tangential neurons receive input from CX-associated regions whose function remains poorly understood, like the SMP/SIP/SLP, making it hard to assign specific circuit functions to these neurons based on their inputs alone. In contrast, the vector computation models relied on a considerable amount of prior experimental data that, when mapped to the connectome, provided physiological hooks for generating novel hypotheses regarding circuit function. In addition, FB tangential types often have extensive reciprocal connections to other tangential types, which, given the absence of functional data, is hard to interpret. Once some of these functions are better understood, it may be possible to derive internal state hierarchies, like those we identified

for directional sensory cues carried by EB ring neurons, which could suggest how the CX prioritizes different internal states. However, many of these internal states involve variables that evolve over time, such as nutritive state or sleep need, suggesting the underlying CX networks may undergo considerable plasticity that may not be apparent in connectome-level connectivity. The dynamic interaction of different internal state variables is likely also governed by neuropeptidergic signals that bathe the CX (*Kahsai et al., 2012*; *Kahsai and Winther, 2011*), but that our analysis did not capture. Finally, given our limited understanding of the variety of behavior the CX may support, understanding how internal state cues may factor into these behaviors is hard to predict at present. To better understand which behaviors the CX may be involved in, we used the connectome to identify output pathways, a topic we turn to next.

## Directing and modulating movement based on the fly's current state

The outputs of the CX likely modulate the fly's actions in a variety of different behavioral contexts, including voluntary take-offs, negotiating uncertain terrain, feeding, oviposition, and fighting. The structure of the FB, in particular, suggests that it could modify the head direction signal to orient the fly with respect to behaviorally specific 'goal' directions. Such goals could be a source of food or safety or, for female flies, a good site for oviposition.

The FB's columnar output types (PFL, PFR, FR, FC, and FS neurons) feed relatively independent output subnetworks, which may support, through unknown mechanisms, the maintenance of independent goal locations associated with different behaviors. Alternatively, these subnetworks could control independent sets of behaviors (*Figure 74*, which is partly based on *Figure 74—figure supplement 1*, proposes a speculative set of modules for this; see also *Videos 17–22*). If true, each subnetwork may carry the potential for the execution of actions towards independent goal locations, each specific for a given behavior and carried by a specific FB columnar type (or set of types). For example, some subnetworks could control behaviors related to goals in front of the animal, such as feeding or gap crossing (*Poeck et al., 2008*; *Triphan et al., 2016*; *Triphan et al., 2010*). Some CX outputs contact a limited number of MBON-associated networks (*Figure 61*). These connections may allow the CX to modulate some behavioral responses to specific sensory contexts that have been associated with negative or positive valence through the MB. The fact that an oviposition neuron (oviIN) is associated with these MBON networks (*Figure 61*) could mean that CX networks influence spatial decision-making during oviposition, which is known to be informed by several external factors (*Yang et al., 2008*; *Zhang et al., 2020*). In contrast, how and why CX signals from the columnar FR1 neurons should directly influence MB neurons themselves (in the case of the FR1 neurons, the MBON30 neurons) is less clear. The variety of different interactions between the MB and CX suggests that investigations of memory-guided orientation and navigation may benefit from a study of both regions acting in concert. Consistent with such an idea, the atypical MBON, MBON27, targets the DNa03 neuron type (*Li et al., 2020*), which is also targeted by PFL3 neurons.

Another axis along which CX-mediated behaviors can be subdivided is the scale of orientation control. From the body to the head and legs, proboscis, abdomen, or antenna, all body parts have an orientation relative to the environment. Each of these body parts could benefit from coordinated but independent control and could be individually targeted by CX outputs. The CX could, in the context oviposition, direct abdomen bending for egg laying in a manner that incorporates the fly's internal sense of its body size, posture, and orientation relative to its surroundings. Hints for how the CX exerts such directional control may be found in the morphology of its outputs.

Output neurons with bilateral innervation patterns in premotor regions such as the LAL and CRE are likely to modulate symmetric actions (e.g., forward walking), while those with unilateral innervations in such regions likely control asymmetric actions (e.g., turning). Examples of the former include the PFL2 and FS1-3 neurons, while PFL1, PFL3, PFR, FR, FS4, and FC neurons all show unilateral innervations of premotor regions.

These different output signals could also vary in how directly they control flies' behavior. CX outputs could themselves direct the animal's movements and/or orientation towards a desired location or away from one associated with danger. The PFL2 and PFL3 neurons provide the most direct link from the CX to the motor center, known as the VNC (*Figure 63*). As the major output channel of the CX, they are prime candidates to guide orientation and/or movements to a CX-specified goal. However, these actions would need to be coordinated with movements of body parts that alter the

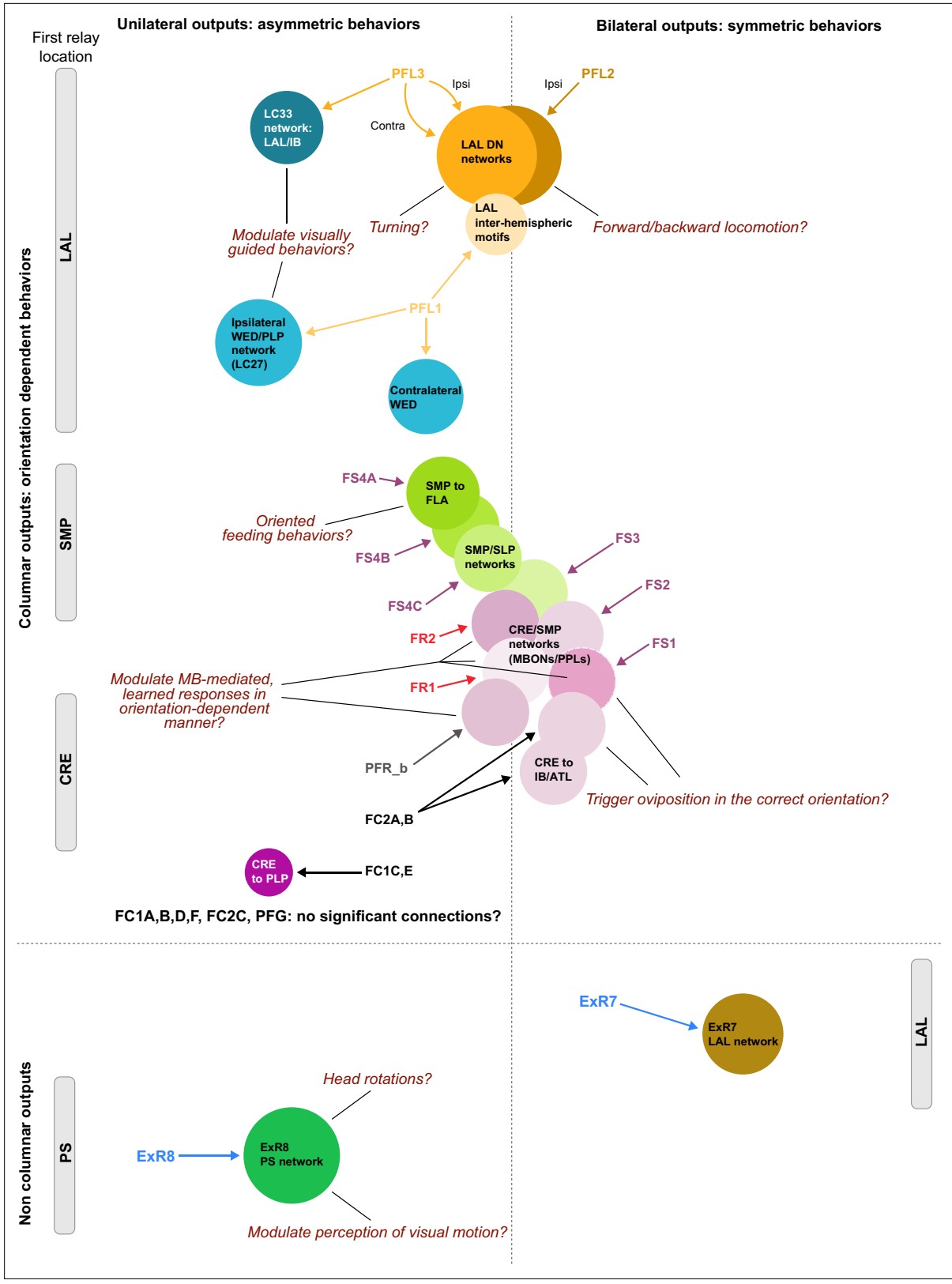

**Figure 74.** Summary of output networks. Schematic representation of the contributions of CX output neurons to various subnetworks and their potential functions. Outputs are divided between unilateral (like PFL3) versus bilateral (like PFL2), as those are likely to control different types of behavior (asymmetric vs symmetric), and between columnar and non-columnar, likely distinguishing between orientation-dependent and orientation-independent action selection.

*Figure 74 continued on next page*

Figure 74 continued

The online version of this article includes the following figure supplement(s) for figure 74:

**Figure supplement 1.** PFL1 subnetworks, rationale behind *Figure 74*.

sensed orientation, most notably head movement. The ExR8 neuron is a candidate to carry out some of those corrections, through connections both to DNs and to the visual system (*Figure 63—figure supplement 1*, *Figure 62*). The remainder of the CX's outputs act more indirectly, and may modulate and gate actions controlled by other brain regions rather than directly controlling them. This is well exemplified by the multiple points of convergence between visual pathways and CX output pathways (*Figure 62*). We remain entirely in the dark concerning some CX output neuron types (PFL1, ExR7, and most of the FC neurons) and much of their downstream circuitry. Further characterization of these underexplored brain regions, a more complete connectome, and genetically targeted imaging and perturbation experiments will help to identify the function of these pathways.

## Navigation with small networks and with numerical variation in columnar neurons

The remarkable behavioral repertoire of insects is still more remarkable when considering their small brains. The CX connectome suggests that part of the secret behind this wide-ranging repertoire lies in having evolved architectures that are precisely configured for sophisticated behavior, but – physiological and behavioral genetics studies suggest – with weights that are plastic to allow these behaviors to flexibly adapt to context and situational demand. It is likely that the impressive computational power of their brains may also derive from an underexplored aspect of their neurons: their capacity for arbor-specific local computations, possibly even subthreshold computations in which synaptic release does not require spiking, and molecular computations through signal transduction cascades (*Thornquist et al., 2020*). These issues will require further experiments, but the connectivity we observe in the EB, for example, hints at a rich potential for insights into subcellular computation in the CX.

Regardless of the true computational capacity of single neurons, it is remarkable that the fly can navigate with a head direction system of 16 directionally tuned columns (in the EB) and just a few thousand neurons performing vector computation (in the FB). In principle, such small networks should be exquisitely sensitive to any variations in the number of neurons encoding each direction. However, the CX connectome revealed a striking difference in the number of columnar neurons that innervate each of the 18 PB glomeruli (although these differences are mirror-symmetric). Although several studies have investigated the developmental origins of columnar CX neurons (*Pereanu et al., 2011*; *Sullivan et al., 2019*; *Walsh and Doe, 2017*; *Yang et al., 2013*; *Young and Armstrong, 2010a*), we do not know of any that have noted or focused on this systematic mirror symmetry. There are indications of numerical variations in some columnar neurons, such as the EPG, PEN, and PFL neurons, in the FAFB volume as well (*Rayshubskiy et al., 2020*; *Turner-Evans et al., 2020*), and more complete EM reconstructions of that volume (*Dorkenwald et al., 2020*; *Li et al., 2019*) should be able to clarify whether these variations exactly match what is seen in the hemibrain volume.

The functional consequences of the systematic variation of neuron numbers across columns are entirely unknown. It is possible that this variation builds redundancy into a critical navigational system or that the increased numbers of neurons in specific glomeruli ensure a preferred 'default' location for the bump to occupy within the more central columns of CX structures, and perhaps also a 'default' heading for the fly to adopt (*Figures 64 and 73*), an idea that is similar to a suggestion advanced independently in *Rayshubskiy et al., 2020*. A different possibility is that such variation is not stereotyped across flies, but is specific to individuals, and that this may account for locomotor biases across the population (*Ayroles et al., 2015*; *Buchanan et al., 2015*; *Skutt-Kakaria et al., 2019*). Functional experiments with specific perturbations of neuron numbers in different columns may be necessary to further investigate this issue. Regardless of their functional role, how such mirror symmetric numerical variation is achieved may be an intriguing question for future studies of CX development. We do not know if asymmetries and mirror symmetries in columnar neuron numbers are also present in the CX of other insects, but parallel efforts in connectomics (*Sayre et al., 2021*) should soon make this clear. Some Diptera, including *Drosophila*, have a closed EB, in contrast with most other insects, whose CBLs have an open, FB-like (CBU-like) structure (*Strausfeld, 2012*). The nonuniform distribution of the

EPG neurons at the base of the EB (see *Figure 18*), EPGt innervation at that location, and systematic modifications to neuron number across the columnar neuron types may represent an evolutionary adjustment to the closing of this structure.

## The CX as a tractable deep recurrent neural network

Technical advances over the past several decades have enabled increasingly large-scale recordings of neural activity from the central brains of a wide range of animals (*Jun et al., 2017*; *Lu et al., 2020a*; *Stringer et al., 2019*; *Vanwalleghem et al., 2018*). These recordings have, in turn, enabled high-throughput studies of neural response properties that have focused on relating patterns of neuronal activity to sensory, behavioral, and internal state variables. However, the biophysical and circuit mechanisms underlying these response properties have been more challenging to access. Similarly, dramatic progress in the field of machine learning has enabled the creation of sophisticated artificial agents that can solve a variety of different cognitive tasks, including flexible navigation (*Banino et al., 2018*; *Cueva and Wei, 2018*). Some units in these deep networks develop response properties broadly similar to those observed in real brains.

Insights into how such artificial neural networks generate the representations observed in their units – something that could, in principle, guide mechanistic hypotheses for the function of natural neural networks – have been slower to come (but see *Cueva et al., 2019*; *Uria et al., 2020* for progress in uncovering the architectural basis of navigational responses in these networks). In this era of deep learning, a broader question concerns the level of understanding that is appropriate or even possible for the function of large and complex neural networks (*Gao and Ganguli, 2015*; *Hasson et al., 2020*; *Lillicrap and Kording, 2019*; *Richards et al., 2019*; *Saxe et al., 2020*; *Yamins and DiCarlo, 2016*). What seems achievable is an understanding of learning rules and objective functions that can, in principle, generate networks with realistic population responses for specific cognitive tasks. The conservation of the CX's structure across arthropods (*Honkanen et al., 2019*; *Strausfeld, 2012*; *Turner-Evans and Jayaraman, 2016*) perhaps highlights the extent to which the region has, in practice, been shaped by such rules over evolutionary timescales in the service of flexible behavior. But what of an understanding of the actual network implementation itself? Some have argued against the necessity or desirability of such a level of understanding (*Hasson et al., 2020*; *Richards et al., 2019*). The fly's relatively brief history in systems neuroscience provides an increasingly compelling counterargument and may eventually offer a roadmap for implementation-level understanding that could scale to much larger brains and more complex cognitive functions.

The fly displays a wide repertoire of flexible behaviors, and some of its recurrent neural circuits show dynamics that have been linked to associative learning and navigation across animals. Its 100,000-neuron brain circuits may appear complex, but they also feature modularity, type-specific connectivity and topography that is genetically prespecified and has been refined over its evolutionary history. Some of these features apply to much larger brains as well (*Hodge et al., 2019*; *Maruoka et al., 2017*; *Saunders et al., 2018*; *Strange et al., 2014*; *Tasic et al., 2018*), although there is likely greater flexibility in the wiring of mammalian circuits and greater heterogeneity within cell types in the mammalian brain (*Cembrowski and Menon, 2018*; *Cembrowski and Spruston, 2019*). It is possible that developmentally driven organizational features of natural brains may actually make them more tractable than artificial neural networks for an understanding of their function (*Zador, 2019*). *Figure 75A*, for example, shows the connectivity of a small fraction of the fly CX's many neuron types arranged by layers. Taking a single-neuron-resolution view of this subnetwork shows just how densely recurrent it is (*Scheffer, 2020*), even at a small scale (*Figure 75B*). Indeed, if the types and connectivity of these neurons were unknown, extracting network structure from population responses would be a challenge. However, sorting the neurons into types – in this case, inhibitory types – makes the logic of the network clearer (*Figure 75C*; also see *Figure 75—figure supplement 1* for the part of the CX that we believe to be structured for navigational computations). Combining this circuit connectivity with physiological studies has enabled not only the generation of hypotheses for the computations that may be carried out by subnetworks at each layer, but, increasingly, tests of these hypotheses (*Fisher et al., 2019*; *Green et al., 2017*; *Kim et al., 2019*; *Kim et al., 2017c*; *Lu et al., 2020b*; *Lyu et al., 2020*; *Turner-Evans et al., 2017*). As a result, it is possible to establish circuit-level mechanisms underlying the generation of different response properties. Importantly, fly circuit connectivity is not always structured (*Caron et al., 2013*), many synaptic connections are plastic, and information from

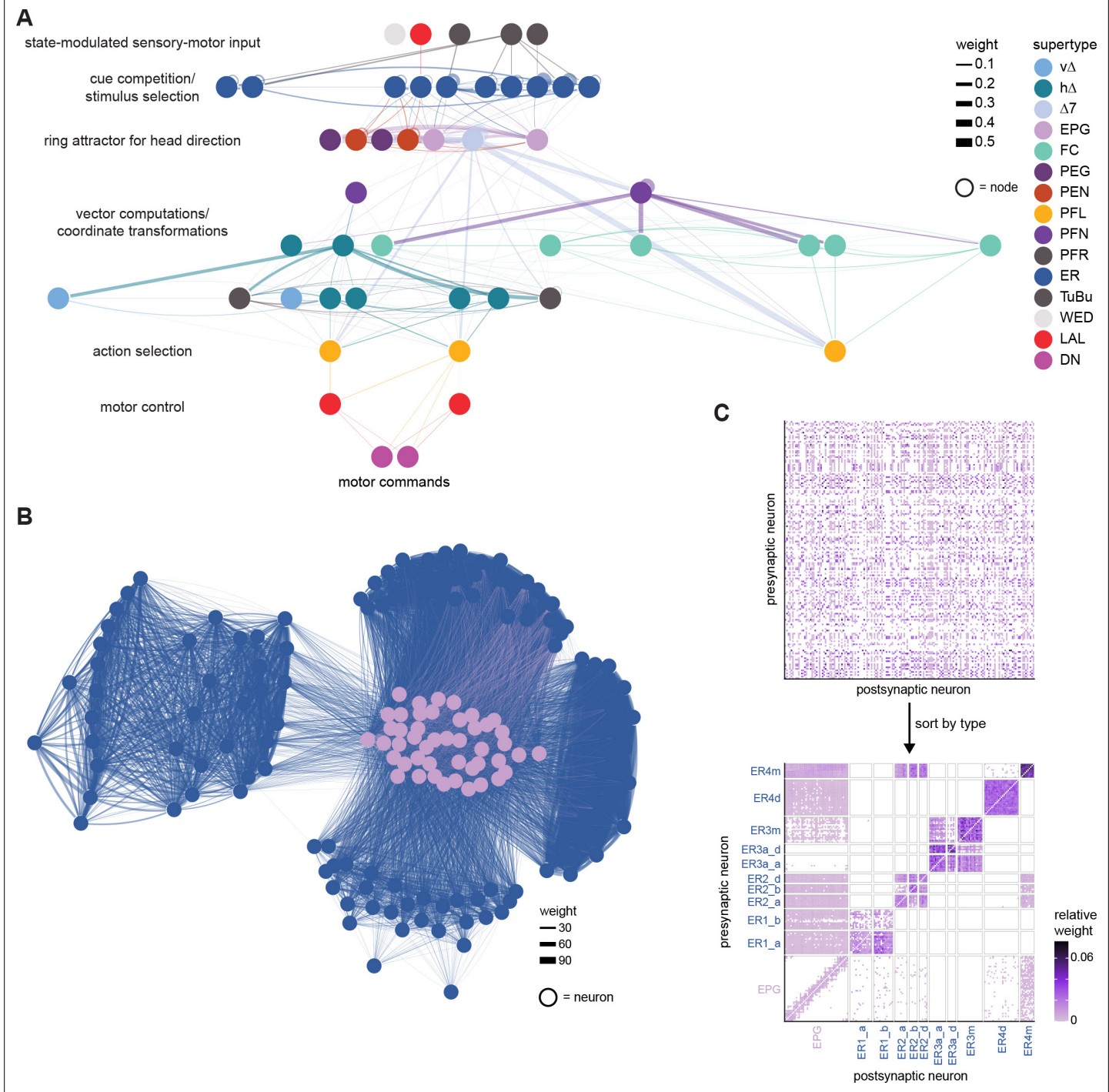

**Figure 75.** The central complex (CX) seen as a deep recurrent neural network for navigation. (**A**) A layered representation of the connectivity of a selection of neuron types in the CX, with a bias towards those involved in navigation. Layers have been labeled by their putative computational roles in a navigational context. (**B**) The connectivity of ER4m, ER3a_a,d, ER3m, ER4d, ER2_a,b,d, ER1_a,b, and EPG neurons is densely recurrent. However, different neuron types have specific roles in circuit function. The ER types plotted here are also the types plotted in layer 2 (cue competition/stimulus selection) in (**A**). (**C**) If neurons in (**B**) were unsorted, the structure in their connectivity would be difficult to recognize (left). When properly sorted by types, the structure in the network connectivity becomes clear (right). The neuron names were randomly shuffled to generate the unsorted plot at left.

The online version of this article includes the following figure supplement(s) for figure 75:

**Figure supplement 1.** The structure in the fan-shaped body (FB) connectivity becomes clear when neurons are sorted by type.

one part of the network often flows to all other parts of it (as we detail in sections of this article). Nevertheless, the developmentally prespecified organization of these networks makes them experimentally tractable. Although the computational capacity of the morphologically and biophysically complex neurons in these networks has likely been vastly underestimated, the connectome thus raises the prospects for at least a circuit-level understanding of how the fly's CX generates many of this small animal's flexible behaviors.

# Materials and methods

Details of all methods used in preparing the hemibrain connectome have been described previously (*Scheffer et al., 2020*). The connectome was reconstructed from the brain of a 5-day-old female fly of wild-type Canton S strain G1 x w^1118, raised on a 12 hr day/night cycle. Its brain was dissected 1.5 hr after lights-on.

## Nomenclature

Many neurons exhibit a unique morphology that could be matched to previous light-based descriptions. Neurons with practically indistinguishable shapes but with different connectivity patterns were split into connectivity subtypes within a morphology type (eg., the subclasses of PFN neuron types). For this connectivity-driven subclassification, we used a cell type clustering tool called CBLAST (*Scheffer et al., 2020*) in an iterative process, using neuron morphology as a template, and regrouping neurons after more careful examination of neuron projection patterns and their connections. CBLAST usually generated clusters that were consistent with the morphological groupings of the neurons, but often suggested new subgroupings. This clustering served as a naming guide after manual inspection (see *Tables 2 and 3* for all neuron types, with numbers for each type).

All CX neurons have been given two names, a short name that we use throughout the article, and a longer name that captures the overall morphology of the neuron. Both long and short names for the vast majority of PB, NO, and AB neurons are published (*Wolff and Rubin, 2018*) and remain largely unchanged, even though slight changes were made to certain formats in the FB nomenclature (*Scheffer et al., 2020*). For example, abbreviated names published in *Wolff and Rubin, 2018*, used a one-letter rather than a two-letter code for CX neuropils and non-CX neuropils included lower case letters (e.g., Sps, Ib). There are several cases in which these published names are identical to either abbreviated names in other brain regions or in other systems. These redundant names have consequently been changed, as follows. LN was changed to LNO, GLN was changed to GLNO, and EB ring neurons were changed from R1-R6 to ER1-ER6 to distinguish them from the names of fly photoreceptors. Details for the PB/NO/AB nomenclature scheme can be found in *Wolff and Rubin, 2018*. Below, we describe in broad terms the approach that was taken to name previously undescribed CX neurons, a more detailed account of which is published in *Scheffer et al., 2020*.

For long names, the same general rules used to name PB, NO, and AB neurons were followed for FB nomenclature, as follows:

1. A maximum of three neuropils, the two with the greatest input (except for tangential neurons, which are named for the neuropils with second and third highest input, see *Scheffer et al., 2020* for details), followed by one output, constitute the backbone of the name. The input neuropil with a larger arbor (and greater number of synapses) is listed first. Connectivity data was used to confirm the relative proportions of synapses in input neuropils. Neurons that project to only two neuropils are named by the input followed by output neuropil.
2. The standard two-letter abbreviations are used for CX neuropils (PB, FB, EB, NO, AB), whereas the three-letter abbreviations described in *Ito et al., 2014* are used for non-CX neuropils.
3. The highest synaptic input for many FB neurons is restricted to a relatively small group of neuropils, leading to a great deal of redundancy in anatomical names. Neurons that innervate the same set of neuropils are therefore distinguished from one another by the underscore symbol followed by a number (_1, _2, etc.).
4. Numbers that follow 'FB' indicate the FB layer arborized by the neuron. Confinement to the dorsal or ventral portion of an FB layer is indicated by a 'd' or 'v.' A hyphen between FB layers numbers indicates the arbor is more or less continuous across the noted layers. Numbers and letters that follow 'NO' identify either the nodulus or nodulus subcompartment. In an effort to

keep the anatomical names as short as possible, only the subcompartments are included for neurons that arborize in NO3, abbreviated NOa (anterior), NOm (medial), and NOp (posterior). Whereas most FB tangential neurons target a single FB layer, the FB columnar neuron arbors often span more than one layer, either as continuous arbors or as distinct arbors in different layers and brain hemispheres. The width of these arbors varies with neuron type, and therefore, column number is fluid and neuron type-dependent. In other words, there is no fixed number of columns in the FB as is shown for PB glomeruli, although most fall neatly into nine columns, as discussed in 'Results.' The placement of arbors, both within specific layers and across columns, is a defining morphological feature and is therefore included in columnar neuron names, as described in the following rule.

5. There are two basic blueprints for columnar neuron architecture: one in which input arbors are vertically arranged within a single column, indicated by the prefix 'v' in the short name and D0 in the anatomical name, and a second in which the arbors are horizontally distributed across the width of the FB, indicated by the prefix 'h' in the short name and D# in the anatomical name (see below; further details in *Scheffer et al., 2020*). The input arbor(s) for the horizontal class of neurons are vertically arranged within a column, as with the vertical class, and is/are separated by a neuron type-specific number of columns. The number of columns between these two arbors is referred to as Δ ('delta') and is indicated in the neuron name by 'Δ' followed by the number of columns that separate the input from the output arbors (details described in *Scheffer et al., 2020*). The following sample neuron illustrates the nomenclature scheme described above. FB2,3,4,5D5FB4,5v (hΔI) has a predominantly input arbor in layers 2,3,4, and 5 that extends through one column, its output arbor 'skips' five columns (D5) and extends vertically, within a column, through layers 4 and the ventral half of layer 5.

For short names, a detailed account of the naming scheme for the FB neurons has already been described (*Scheffer et al., 2020*). We briefly use the FB 'tangential' neurons as examples to illustrate the scheme. We found a total of 574 FB tangential neurons. Each neuron has widely extended processes spreading across a layer of the FB. The tangential processes typically extend throughout a single FB layer, but a few neuron types have multilayered arborizations. Their morphological classifications are based on the FB layers they innervate as well as their arborization patterns outside the CX and locations of their cell bodies. *Table 3* shows the number of neuron types for each FB layer and their cell counts. The numbers in the type name indicate the layer in which they arborize (i.e., FB1, FB2, …), followed by uppercase letters classifying their morphological types. When connectivity subtypes were identified, lowercase letters are used after underscores, for example, FB1E_a and FB1E_b. Methods *Appendix 1—figure 2* shows an example of identified connectivity types within a morphology type. We judged that these neurons were nearly indistinguishable based on morphology, so they were all given the same type name, FB2F. However, since CBLAST suggested these neurons should be clustered into three groups, FB2F was split into three connectivity types, FB2F_a, FB2F_b, and FB2F_c, with manual confirmation of their actual connections. For each morphological or connectivity type, there are always at least two neurons because we found pairs of neurons in the right and left side of the brain (even though the left side is only partially captured by the reconstructed hemibrain volume). We also identified several intrinsic columnar neurons in the FB, including hΔ and vΔ types, and columnar PNs, for example, FR (FB-RUB), FC (FB-CRE), and FS (FB-SMP). In addition, in the FB, we identified the EL (EB-LAL) neuron type.

## Software and code availability

We accessed the version 1.1 of the neuPrint database through its R API, neuprintr (http://natverse.org/neuprintr/; *Bates et al., 2019*). Unless otherwise stated, analysis was done in R. All analysis code is available at https://.com/jayaraman-lab/CX-connectome-analysis, (copy archived at swh:1:rev:5c7a-57256c15ea763e4cf0a7cf686ba0d3342d80, *Franconville, 2022a*) and the functions developed for the analysis are available as an R package at https://github.com/jayaraman-lab/neuprintrExtra, *Franconville, 2022b*.

Videos were produced using the neuVid system of Python scripts that work with Blender (*Hubbard, 2020*).

## Diameter of main neurites

We extracted main neurites of all CX neurons using the hemibrainr package (https://github.com/natverse/hemibrainr; *Bates, 2020*). The main neurite is defined as the process linking the soma to the primary branchpoint of the neuron. We then computed the median width of that fragment for each neuron.

## Validation methods

We compared the version of the dataset used in this article (v1.1) to an older version (v0.9) to test how much the connectivity motifs depend on tracing completeness. We selected neuron instances containing at least 200 upstream (respectively, downstream) synapses in the EB in the v1.1 dataset and quantified how their inputs changed (respectively, outputs) from the older version. We only considered connections between cell body identities (bodyids) present in both datasets and compared neuron to neuron connection weights.

We also compared different PB glomeruli that were traced to different levels of completeness. To compare inputs (respectively, outputs) of neurons of the same type across glomeruli, we chose types with at least one instance that had at least 20 upstream (respectively, downstream) synapses in each of the glomeruli that were considered. Further, at least 80% of that neuron instance's total synapses in those glomeruli were required to be upstream (respectively, downstream). The comparisons were made at the type-to-type level, between mirror symmetric glomeruli. When compiling type-to-type connection weights for a single glomerulus, we only considered instances innervating the glomerulus in question (e.g., only the EPG_L3 instances in glomerulus L3) even though the strict 'type' definition covers all glomeruli.

To compare the third column of the FB to the rest of the FB, we selected neurons with at least 200 upstream (respectively, downstream) synapses in the FB and at least 80% of those in FB column 3. Note that this restricts the analysis to columnar types. We compared type-to-type connections between these selected neurons and all other instances of neurons of the same type. To do so, we modified type definitions by dividing types between instances in column 3 and other instances.

## Quantifying and normalizing connectivity between individual neurons

Throughout, as per neuPrint convention, the number of synapses between two neurons is the number of postsynaptic sites on the postsynaptic neurons (*Scheffer et al., 2020*). Note that if polyadic synapses are present, as is common, the number of presynaptic sites will not equal the number of postsynaptic sites. In all analyses in which we quantify connections, we filter out all cases with strictly less than three synapses between two neurons. This significantly lightens processing load and gets rid of spurious connections.

Depending on the analysis, we used several different types of measures to quantify connection strength. As a default, we used *relative weight*, which is the relative input weight from some neuron $a$ to some neuron $b$ within a region of interest (ROI). Thus, to compute the relative weight of a connection from neuron $a$ to neuron $b$ in a region X, we counted the synapses from $a$ to $b$ in region X and divided this number by the total number of inputs that $b$ received in X. We feel that this metric is the best approximation of a functionally relevant measure. We also occasionally used a second connectivity measure, *output contribution*. Output contribution is the relative output weight from neuron $a$ to neuron $b$, computed by dividing the number of synapses from neuron $a$ to neuron $b$ in region X by the total number of outputs that $a$ sends in region X. While this measure is unlikely to be functionally meaningful, it is useful for comparing the output composition of neurons (e.g., *Figure 26B*, bar graphs and validation).

## Quantification of connectivity between neuron types

Each neuron type can consist of many neurons. If these neurons are connected to neurons of another type, the individual weights between partners may vary considerably. We therefore devised two criteria to judge if the connections between two types of neurons were significant in a given ROI. For our first criteria, we calculated the normalized total connection strength from neuron type A to neuron type B in the chosen ROI. $W_{DA}$. $W = \sum_{1}^{n} \sum_{1}^{m} w_{ij}$ is the total weight from type A, with n individual neurons, $a_1$ to $a_n$, to type B, with m neurons, $b_1$ to $b_m$, in the chosen ROI. $w_{ij}$ is the weight of the connection from $a_i$ to $b_j$ in the chosen ROI. $D_A$ is the total number of synapses downstream of type A neurons (to

all neuron types) in the chosen ROI. If the normalized total connection strength was greater than 0.8, we considered the connection significant. This criterion is meant to retain cases in which a neuron type sends a small process to a neuropil where it makes exclusive contact with another type.

For our second criterion, we calculated the relative weight contributed by neurons of type A to each type B neuron in a given ROI. $W_{rel_j} = \sum_1^n \frac{w_{ij}}{U_{b_j}}$, where $U_{b_j}$ is the total number of upstream synapses of neuron $b_j$ in the ROI. We then considered the sample $W_{rel_1} \ldots W_{rel_m}$. In the case that m=1 (i.e., if type B consisted of a single instance), the connection was considered significant if $W_{rel_1}$ was above a predetermined threshold (in practice 0.01). Otherwise, we used R's t-test with p=0.05 to determine if the connection was significant. For significant connections, we used the mean relative weight $\sum_1^m W_{rel_{jm}}$ to quantify the connection from type A $\lambda$ B.

## Deciding which neurons innervate a given ROI

A given type is considered to innervate an ROI if it satisfies two criteria.

1. Half of the instances of the type make synapses in the ROI.
2. The type makes at least one significant type to type connection in the ROI.

For example, we use this method to select neuron types to include in the connectivity graph in *Figure 10F*.

In neuropil innervation plots, only those regions where significant type-to-type connections exist are shown.

## Neuron renders

The 3D morphological renderings of neurons presented in the figures were generated using the visualization tools of NeuTu (*Zhao et al., 2018*).

## Connectivity graphs

Graphs were laid out manually (e.g., *Figure 6C* and *Figure 62*), by specifying the layer of each type and using the Sugiyama layout in igraph (e.g., *Figure 61—figure supplement 1*), or by using the stress majorization layout of the graphlayouts package (*Figure 17*, *Figure 59—figure supplement 2*, *Figures 60 and 9–22*).

## Community detection

Community detection in *Figure 59—figure supplement 2* is done using the label propagation algorithm implemented in the igraph package.

## Morphology analysis

The electrotonic distance of a synapse from the putative spike initiation zone was computed utilizing tools from the R packages nat and igraph. We started by generating a graph representation of a neuron. Using nat, we read in a neuron's skeleton and transformed this object into a directed graph for which the weights of the edges are the Euclidean distance between the vertices and the edges were directed away from the soma. Assuming the skeleton nodes are placed in such a way to capture the curvature of the neuron's arbors, the Euclidean distance between two nodes is a decent approximation of the arbor distance between these two points. Hence, the neuron graph contains sufficient information to calculate the distance along a neuron's arbor between any two points. These neuron graphs also contained information about the width of the arbor at each vertex. Each synapse location was projected to the closest (in Euclidean distance) graph node location. For more info, see http://natverse.org/nat/articles/neurons-as-graph.html.

Once we had the graph representation, we identified a root point that approximates the putative spike initiation zone. Extrapolating from studies in *Drosophila* PNs, we assumed the spike initiation zone for EPG neurons to be near the roots of the neuron's arbors in the EB (*Gouwens and Wilson, 2009*). To determine the root point, we intersected the EPG skeleton with the EB ROI. This process often 'fractured' the skeleton, generating several disconnected subgraphs. This was a problem because we rely on the graph representation of the neuron to calculate distances along the arbors. To 'heal' these fractures, we took each connected subgraph and repeatedly added the respective parent node from the original full graph back to the subgraph until all subgraphs were connected to each

other again. We then pruned down any nodes that were added during the 'healing' but were unnecessary to keep this graph connected. This ensures that the parent node of the graph, which will be the root point, is the point closest to the EB for which there exists a path between this point and each of the neuron's synapse locations in the EB. Note that, for our purposes, any point between this putative spike initiation zone and where the synapses of interest occur will be a sufficient point for the root point as the ordering of the synapses will be consistent relative to any of these points. This analysis is aimed at comparing the electrotonic distance between different synapses and the spike initiation zone for a single neuron. Hence, any stretch of arbor that occurs between the spike initiation zone and all the synapses of interest will necessarily be included in the calculation of electrotonic distance for all synapses and thus will not affect our comparison. The synapses of interest are determined to be those that occur within the EB ROI.

With the root point identified, we calculated the electrotonic distance between each synapse and this point using functions from the igraph package. Since the width of the arbor changes, we calculated the length constant $\lambda$ for each edge separately (prescribing the width of the edge to be the width at the terminating vertex of the edge). The electrotonic distance of that edge then is the length of the edge normalized by the edge's length constant. To determine the electrotonic distance between a given synapse and the root point, we summed the electrotonic distances of all edges between the root point and that synapse. For example, suppose the length constants of the edges in the path between a particular synapse and the root point are given by $\lambda_1, \lambda_2, \ldots, \lambda_n$ and the lengths (or weights) of these edges are $w_1, w_2, \ldots, w_n$, then the electrotonic distance between this synapse and the rootpoint is given by $w_1\lambda_1 + w_2\lambda_2 + \ldots + w_n\lambda_n$. We assumed that the specific intracellular resistivity $R_i$ and the specific membrane resistivity $R_m$ are constant across the neuron. Hence, we did not need to include these values in our computation of the length constants as they become a constant factor on each electrotonic distance. Since we are comparing the electrotonic distances of synapses on the same neuron, this factor will not affect their relation. The distributions of lengths between synapses and rootpoints were unimodal for all EPG neurons (n=44) included in this analysis. The two EPG neurons that innervate the R1 glomeruli of the PB had bimodal distributions of lengths between synapses and rootpoints (not shown here), and as such were excluded from this analysis. The EPGt neurons were also excluded from this analysis.

## Analysis of synapse locations: 2D histograms, synapse densities, and mean synapse locations

The hemibrain database contains information regarding the spatial location of every synapse, along with the identity of the presynaptic and postsynaptic neuron. In addition, each neuropil has an associated mesh that defines its boundary in three dimensions, which can be used to restrict a neuron's synapses to the subset contained within a given ROI. We used these two sources of information to quantify and visualize the spatial extent of neuronal innervation patterns in various ROIs. Our plots showing synapse locations or distributions are viewed from an ROI-specific perspective, which is in most cases different from the front, top, and side views defined by the x, y, and z axes used by the hemibrain database. For example, to visualize the concentric innervation pattern of EB ring neurons, it helps to view the EB from directly above, looking through its central canal (e.g., *Figure 10—figure supplement 1A*). Similarly, an FB neuron's columnar location is most easily seen by choosing a perspective that places the FB's layer in the image plane (e.g., *Figure 29C*). To define these perspectives, we first performed PCA on the x, y, and z locations of all synapses within the ROI. In most cases, the individual principal components defined the major axes of each structure but in some cases, small rotations were performed to manually adjust PCA-derived axes. Finally, a coordinate transformation was used to convert each synapse location from the hemibrain reference frame to the PCA-derived reference frame. For some ROIs, synapse distributions are shown from several orthogonal perspectives (e.g., *Figure 6D*). The approximate direction of the anterior-posterior, dorsal-ventral, or medial-lateral directions is indicted for each plot. For anterior-radial projections along the EB circumference, the distance of each synapse from the center of mass of the EB ROI mesh was used as an additional coordinate. The outline of each ROI was computed by finding a convex hull that traced the border around the vertex locations of the ROI mesh using the 'ahull' function from the R package 'alphahull.' To visualize projections of synapse distributions, we used two-dimensional histograms of synapse counts (e.g., *Figure 10—figure supplement 1A*) and normalized synapse densities (e.g., *Figure 6D*)

generated with R's ggplot2 package. In some analyses, such as for defining the columnar structure of FB neuron types (e.g., *Figure 29C*), we approximated the spatial location of a neuron's arbor in a given ROI as the average location of all of its synapses.

## EB modularity analysis

We used Pearson's correlation to compute the similarity between the ring neuron inputs to two different EPG neurons. $\lambda_{ab} = (\bar{w}_a \lambda \bar{w}_b)/(\lambda \bar{w}_a \lambda \lambda \bar{w}_b \lambda)$ , where $\bar{w}_a = w_a - \frac{1}{n}\sum_{i=1}^{n} w_{ai}$ is the mean-subtracted vector of inputs onto neuron a, and $\bar{w}_b$ is the mean-subtracted vector of inputs onto neuron b for the same set of inputs. We chose this similarity measure (as opposed to cosine similarity, used elsewhere) because of the high density of connections between ring neurons and EPG neurons. We then normalized the correlation values between 0 and 1, and used this as a measure of adjacency between neurons a and b. $A_{ab} = (\lambda_{ab} + 1)/2$. We ordered the resulting adjacency matrix according to the EB wedges that the EPG neurons innervate. We used these wedges to define clusters c, and then we computed the modularity of this matrix with respect to these clusters. We interpret the elements of the adjacency matrix as a measure of connection weight between the network of EPG neurons. For weighted networks, the modularity Q is given by: $Q = \frac{1}{2m}\sum_{ab} A_{ab} - \frac{k_a k_b}{2m}\delta_{c_a, c_b}$, where $m = \frac{1}{2}\sum_{ab} A_{ab}$ is the total weight of the edges in the network, $k_a = \sum_b A_{ab}$ is the degree of neuron a, and $\delta_{c_a, c_b}$ is a function that equals 1 if neurons a and b are in the same cluster (i.e., neurons a and b innervate the same EB wedge, and $c_a = c_b$), and 0 otherwise. Q can take values between 0 and 1; in our case, a value of 1 indicates perfect correlation within clusters and perfect anticorrelation between clusters. Conversely, a value of 0 indicates that correlations within each cluster are no stronger than would be expected by chance, given the average correlation of each EPG with all other EPG neurons. To measure the statistical significance of modularity values, we computed the distribution of modularity values for shuffled version of the connectivity matrix from ring neurons onto EPG neurons. For each shuffling, we randomly permuted the connections from each ring neuron onto its set of EPG outputs and repeated this independently for each ring neuron. We then recomputed the adjacency matrix and the corresponding modularity. We reported p-values as the fraction of 1000 permutations for which the shuffled connectivity matrix had a higher modularity value than the unshuffled matrix. We repeated this analysis using both the relative weight and the synapse count as a measure of connectivity, and using connections from rings neurons to EPG neurons and vice versa. MATLAB (MathWorks Inc, Natick, MA) was used for this analysis.

## Analyzing the Δ7 connectivity profile

Each Δ7 neuron outputs in two or three glomeruli that are spaced seven glomeruli apart and receives varying input from the EPG neurons in the glomeruli in between. This creates a double-peaked EPG to Δ7 connectivity profile. To find the mean profile shape, the EPG inputs were first grouped by PB glomerulus and the Δ7 neurons were grouped by the glomeruli in which they output. The peaks of the connectivity profiles were then circularly shifted to bring them into alignment. The mean and standard deviation of the aligned profiles was then calculated.

## Propagating fictive EPG activity through the Δ7 neurons

The EPG activity was assumed to have one of two shapes: a von Mises or an impulse profile. In the former case, the 16 wedges of the EPG were each assigned a value from 0 to $2\pi$, and a von Mises function with $\kappa = 2.5$ was assumed. The profile was normalized so that the fictive activity ranged from 0 to 1. For the impulse function, all EPG neurons that arborize within a given PB glomerulus were assumed to have an activity of 1 while all others were assumed to have an activity of 0. When generating the summary statistic, the mean of the von Mises function or the location of the impulse were each shifted to cover all 16 possible permutations (one for each glomerulus).

The fictive EPG activity was then converted into a vector. This vector was first multiplied by the EPG to Δ7 connectivity matrix and then by the Δ7 to X connectivity matrix, where X is a columnar PB-EB or PB-FB neuron. The activity was then averaged across all neurons within each glomerulus. This generated a 16 × Y matrix, where Y is the number of glomeruli that each columnar type X covers in the PB. The 16 comes from the 16 different permutations. The 16 activity profiles were then aligned by circularly permuting them by their order in the permutation (e.g., the first profile was not changed, the second was shifted by one glomerulus, the third by two, etc.). Additional alignment corrections

were made to those neuron types that arborize in all PB glomeruli (the PFGs and PEG neurons) or those that arborize in fewer than 16 glomeruli (the PFL1, PFL3, and PFR_b neurons) as appropriate. The mean of the aligned profiles was then calculated. A cosine was fit to either the right or the left mean PB activity using the 'nls' function in R, and the mean of the residual from the two sides was taken as the summary statistic.

## Determining the offset between Δ7 inputs and outputs

EPG neurons were assigned angles based on the wedge they innervate in the EB. The number of synapses from each EPG neuron to each Δ7 neuron in the PB was then used to weight these angles, and a circular mean of the weighted angles was calculated to determine the average directional tuning of each Δ7 neuron. We then compared this tuning to the Δ7 neuron's average directional output tuning on either the right or the left side of the PB. The output tuning was calculated in a similar manner to the input tuning, with the angle of each glomerulus being taken from the EPG neurons that arborize there, and the weights now being the number of synapses from the Δ7 neuron to the EPG neurons in the given glomerulus.

## Assigning neurons to FB columns

FB columnar neurons were assigned to FB columns using several complementary methods that depended on cell type. First, neurons belonging to an FX type (FS, FR, or FC) were manually assigned to one of nine columns (C1–C9) by viewing population morphological renderings in 3D. Several types, such as FR1 and FC2A, are composed of 18 neurons total with two neurons per column, which made columnar assignments unambiguous. For types with a less clear columnar structure, direct comparisons to strongly columnar types could be used to aid in column assignment. FX types were assigned to columns first since they collectively innervate all FB layers and could therefore be used as a backbone for defining the average position of the nine FB columns. Second, PB-FB-* and vΔ neurons were assigned to FB columns using an automated approach. For each type, we began by finding the FB layer that contained the most synapses (presynaptic and postsynaptic) and then calculated each neuron's average synapse location using all three dimensions and assigned it an FB column corresponding to the closest FX-defined column from that layer. Automated column assignments were manually checked by viewing 3D population morphological renderings and plots of average neuron locations (e.g., *Figure 29*). vΔ types contain individual neurons that innervate both C1 and C9. These neurons were manually identified and assigned to the 'C0' column as a way to separate them from neurons that innervate single columns (C1–C9). Third, hΔ types form a variable number of FB columns that depends on cell type (*Figure 31—figure supplement 2*), ranging from 6 to 12 columns. Many types, such as hΔA and hΔG, have as many neurons as they have columns, and each column shows minimal overlap with neighboring columns, which made columnar assignments unambiguous. Other types show a less clear columnar structure, such as hΔC. These types were manually assigned to a 12-column scheme by finding leftward and rightward projecting pairs that were approximately mirror symmetric.

Every FB columnar neuron in the database contains its FB column assignment as part of the 'instance field,' but each type's total column number needs to be taken into account, since 'C2' for an FX type is not the same as 'C2' for an hΔ type. In addition, it is important to recognize that for many types column assignments are discrete, despite spatial variation in neuron locations that are rather continuous.

## Connectivity-based estimates of FB neuron directional tuning and PB-FB phase shifts

We used an approach similar to that developed by *Lyu et al., 2020* to estimate the average directional tuning of FB columnar neurons (vΔ, hΔ, and FX types) as well as the magnitude of PB-FB phase shifts. Every PB glomerulus was assigned an angle based on the EB-to-PB projection pattern of EPG neurons (*Wolff et al., 2015*; *Figure 16*). Glomeruli R9 and L9 do not receive direct EPG input but were assigned the same angles as R1 and L1, respectively. This preserves the 45° sampling interval between adjacent glomeruli in the left and right PB. However, the angles assigned to R9 and L9 are different from those predicted by their input from EPGt neurons (*Figure 18*), though other neurons belonging to the P6-8P9 and Δ7 types may also impact the directional tuning of these lateral glomeruli

(*Figures 20 and 21*). Overall, these angular assignments indicate that the directional tuning of the left and right PB are shifted by 22.5°, a prediction supported by recent physiological recordings (*Lyu et al., 2020*). Next, PB-FB connectivity was used to estimate the average directional tuning of the postsynaptic FB neurons that belong to the vΔ, hΔ, and FX types. The average directional tuning of each postsynaptic FB neuron was computed by taking a circular mean across the angles it inherits from its PB-FB inputs, weighted by connection strength (i.e., number of synapses). This average was computed for each PB-FB input type separately, so each FB neuron could have several directional tuning estimates, one for each presynaptic type (*Figure 34D*). Similarly, to compute the magnitude of the PB-FB phase shift, we calculate the angular difference between the average phase inherited from the left PB population compared to the right PB population (*Figure 34E*). Only neurons that receive input from both and left and right PB populations were used for these estimates. In addition, only neurons belonging to significant type-to-type connections that involved at least 80% of the neurons in the postsynaptic population were used. Finally, we excluded all connections where the presynaptic PB-FB type contacts postsynaptic hΔ neurons on both their axonal and dendritic compartments, which complicates phase estimates. Together, these criteria prevented accurate estimates for PFR_b, PFNp_a, and PFNp_d types.

## FB motif analysis

*Figure 37* employs PCA on column-to-column FB connectivity matrices, establishing the existence of three connectivity motifs. This analysis starts with column-to-column connectivity matrices, three examples of which are shown in the bottom panels of *Figure 37B*. These matrices were constructed by averaging relative weights across presynaptic and postsynaptic neuron pairs, grouped by their columnar locations. Because of this, the analysis does not take into account PB-FB offsets, only how neurons in each FB column connect to one another. Next, since FB neurons have type-specific column numbers, all column-to-column connectivity matrices were coerced to a nine-column scheme by rounding to the nearest column. For example, hΔB has a total of 12 columns, and neurons in C11 (in the 12-column scheme) would get mapped to C8 (in the 9-column scheme), since 91112=8.25, which rounds to C8 (in the 9-column scheme). To these 9 × 9 column-to-column connectivity matrices were added a C0 column to include those vΔ neurons that innervate C1 and C9. Together, this produced 903 column-to-column connectivity matrices that all had the same dimensions: 10 × 10. Next, every column-to-column connectivity matrix was transformed into a vector by concatenating rows, and the resulting vectors were grouped into a new matrix whose dimensions were 903 × 100. We refer to this grouped matrix as the 'connectivity feature matrix' since each row in the matrix contains a vectorized column-to-column connectivity matrix between two FB neuron types. Because strong connections will account for more variance than weak connections, even though their column-to-column structure could be the same, we performed PCA on the binarized connectivity feature matrix. As discussed in the 'Results' section (*Figure 37*), the first two PCs accounted for much more variance than any of the subsequent PCs. To visualize the column-to-column connectivity space, each row in the connectivity feature matrix was projected onto the first two PCs to generate the scatter plot shown in *Figure 37C*.

## FB columnar steps from inputs to outputs

To count the steps between FB columnar inputs and outputs (shown in *Figure 33B*), the connectivity matrix shown in *Figure 33B* was converted to a network. The shortest path between the input node and the output node was then used to determine the number of steps that connected inputs to outputs.

## Identification of R23E10 and PPL1 dopaminergic dorsal FB tangential neurons

To assign neurons in a GAL4 driver line to their EM-defined neuron types, we warped raw confocal stacks to a standard reference brain, which allowed for a direct comparison between light- and EM-level morphologies using VVDviewer. For neurons contained in R23E10 (*Figure 48*) and PPL1 dopaminergic types (*Figure 49*), we began by identifying potential EM candidates based on broad agreement in overall morphology (*Figure 48—figure supplements 5–7*). Two anatomical features of R23E10 neurons – the lateral location of their soma and a fiber track that enters the FB slightly medial to the lateral border – unambiguously identified 14 candidate tangential neuron types with processes

in layers 6 and 7 whose general morphology matched that of the R23E10 pattern (*Figure 48—figure supplement 2*). Next, directly comparing the light-level morphology of individual R23E10 neurons, generated by MCFO stochastic labeling (*Nern et al., 2015*), to the 14 candidate EM neuron types allowed us to exclude 5 of the 14 candidates based on the presence of arbors that lie well outside the R23E10 pattern (*Figure 48—figure supplement 4*). Of the remaining nine candidates, seven neuron types had one or more high-quality matches between individual R23E10 neurons and corresponding EM morphologies: FB6A, FB6C_a, FB6C_b, FB6E, FB6G, FB6I, and FB6Z (*Figure 48—figure supplement 3*). The remaining two candidates – FB7A and FB7K – are also likely to be in the R23E10. Not only does R23E10 contain processes in layer 7, but we were able to identify high-quality matches with a subset of FB7A neurons and a moderate-quality match to FB7K. As presently defined, the FB7A neuron type contains three neurons per hemisphere. Two of these neurons send processes to the lateral portion of the SMP/SIP/SLP – a feature not observed in R23E10 – while the remaining neuron showed a high-quality match to several individual R23E10 neurons. Therefore, we include all FB7A neurons while recognizing that future work may further refine this neuron type and its relation to the R23E10 line.

Similarly, several lines of evidence support the identification of FB5H, FB6H, and FB7B as the three PPL1 dopaminergic types in the dorsal FB (*Figure 48—figure supplement 2*). First, a stable split GAL4 that uses a 10 kb segment of the TH genomic region as one of its hemidrivers (*Aso et al., 2014a*) labels three neurons per hemisphere whose cell bodies express tyrosine hydroxylase (TH). Matching morphologies of individual neurons from this split to EM morphologies yielded high-quality matches to FB5H, FB6H, and FB7B (*Figure 49*). Second, a driver line that specifically targets DANs, TH-GAL4, contains individual neurons whose morphology matches that of FB6H and FB7B (Fly Circuit TH collection, *Chiang et al., 2011*). Third, although the general morphology of the FB5H, FB6H, and FB7B neuron types is similar to that of R23E10 neurons, all three neuron types have arbors that lie partly outside the R23E10 pattern, demonstrating that these DAN types are not present in R23E10. These PPL1 DANs are likely the primary wake-promoting DANs of the dFB, as previously indicated by experimental studies (*Liu et al., 2012*; *Ueno et al., 2012*). Although additional DANs in the PPM3 cluster – FB2A, FB4L, FB4M – innervate more ventral FB layers, there is as yet no data to indicate that they regulate sleep. However, it remains possible that unidentified DANs in the dFB, perhaps belonging to the PPM3 cluster, may also be involved.

## ROI groups

For the analysis in the Output section, all ROIs on the right side of the brain (with the exception of the CX) were grouped as a super-ROI. All statistics were then recomputed within that super-ROI. This approach avoids ROI-specific artifacts that result from large neurons that innervate small, sometimes ill-defined regions with processes that are part of a much larger arbor that spans multiple ROIs.

## Similarity plots/clustering

To compare the similarity of the we computed their cosine similarity: (waλwb)/(λλwaλλλλwbλλ), where wa is the vector of all input weights to a neuron a and wb is the vector of all input weights to a neuron b for the same set of inputs. We used the same method to compare outputs, though, in that case, wa and wb contained all output weights. We chose this metric as it is suitable for very sparse vectors while still conveying information about the proportions in which a neuron contacts its targets. Similarity matrices were ordered (and in some cases clustered) by using hierarchical clustering based on the complete linkage algorithm.

## Lateralizing neuron types

For some analyses, we subdivided types into left and right populations (respectively post-fixed with _L or _R). Right or left lateralization was determined by the position of the cell bodies, as taken from the 'name' field in neuprint. The only exception to this rule was made for PFL1, PFL3, PFR_a, and PFR_b. For these types, their assigned side corresponds to the side opposite to the hemisphere that the neuron innervates outside of the CX (the LAL or the round body). In a few instances, this side assignment differs from the cell body side.

## Pathways

Pathways between neuron types were determined by walking downstream (or upstream) of a set of types of interest. All significant downstream (upstream) types were determined, then all significant downstream (upstream) types of those types were found, and so on, for a defined number of steps. For the analysis in the Output section, for example, five steps were considered. Further, for this output analysis, type-to-type significance was determined using the criterion described above, which is a t-test on the vector of the type-to-type relative weights for connections to postsynaptic types containing multiple neuron instances, or a type-to-type connection whose relative weight exceeds 0.01 for connnections to postsynaptic types (or half-type as they are lateralized) constituted of a single instance (see section 'Quantification of connectivity between neuron types'). Moreover, since the output pathways contain a lot of types constituted by a single very large neuron for which relative input weights can get very small, we also kept all type-to-type connections containing more than 50 synapses. Finally, to eliminate spurious connections stemming from small neuronal processes crossing ROI boundaries, we did not consider any connection to types containing less than 20 synapses in the super-ROI used (see above, ROI groups).

For sleep pathway analysis, significance was defined as any type-to-type connection whose relative weight exceeded 0.01. We adopted this alternative criterion because some FB tangential types are composed of only two neurons that can have strong but variable connections with neurons of another type, and these connections would occasionally be filtered out by our first criterion. This procedure leads to a connectivity graph, which we processed further as described below.

In the output analysis, we extended the connectivity graph by simulating contralateral connections at every step. These simulations mitigate the incompleteness of the EM volume on the left side. For example, if A_R contacts B_R with weight w in a lateralized region (in this case, the full right side of the brain excluding the CX), we added to the graph an equivalent connection from A_L to B_L on the other side. This was particularly useful when neurons crossed over the midline, but did not propagate if the symmetric neuron was not identified in the dataset.

## Pathway weight

Given two types A and B in a connectivity graph, we defined the *pathway weight* from A to B as the summed weight over all pathways, where the weights of individual pathways were obtained by multiplying relative weights along the steps of a given pathway, that is, $W_{pathway} = \sum_{pPathways} \prod_{sSteps} w_{ps}$ . $w_{ps}$ is the weight between partners $p$ at step $s$ (*Appendix 1—figure 3*). This is equivalent to multiplying the adjacency matrix $A_g$ with itself. In practice, because of recurrence, there are infinitely many pathways, and we therefore cannot loop over all of them. For the sleep pathway analysis, we used pathways that are no longer than the number of steps used to build the graph in the first place. For the output pathway analysis, given the high recurrence in the circuit, we multiplied $A_g$ until its norm converged to 0 (in practice when it becomes smaller than 10-8). This multiplication always converges because the metric used (the relative weight) is smaller than 1.

To determine the relative contribution a neuron makes to a set of pathways, we ran the same computation but instead of using $w_{ps}$ , we used the weight relative normalized by the sum of weight relative contributed by the presynaptic type. This yielded a metric that sums to 1 if all the pathways emanating from a given type are summed. The CX-to-CX fraction of individual synapses in *Figure 56C* is the relative contribution made to CX neurons by the neuron targeted by the synapse considered.

When considering pathways from the CX, we ended pathways as soon as they looped back to the CX. When considering pathways to a set of known types, we ended pathways as soon as a known neuron was reached. This was done by removing the corresponding line (column) in the resulting matrix to be recursively multiplied by the adjacency matrix.

## Immunohistochemistry

To determine whether neurons in the SS56699 split GAL4 line expressed TH, immunohistochemical processes was performed as described in *Aso et al., 2014b*. Briefly, 10 GFP-expressing brains were fixed (2% paraformaldehyde in Schneider's medium), permeabilized in PBT (0.5% Triton X-100 in PBS), and blocked (5% normal goat serum for 90 min). Subsequently, brains were incubated in primary antibodies (diluted in 5% serum in PBT at 4 °C for 2–4 days). Chicken anti-GFP (Abcam ab13970; 1:1000); anti-TH mouse monoclonal (Sigma MAB318; 1:200); anti-TH rabbit polyclonal (Sigma AB152; 1:200).

After several washes (PBT for 30 min), brains were then incubated in secondary antibodies (diluted in 5% serum in PBT at 4 °C for 2–4 days). Alexa 488 anti-chicken IgY (Invitrogen A11039; 1:400); Atto 647N anti-mouse IgG (Rockland 610-156-121; 1:400); Alexa 568 anti-rabbit IgG (Invitrogen A11036; 1:400). Finally, brains were washed thoroughly (PBT four times for 30 min or longer) and mounted on glass slides for confocal imaging.

## Acknowledgements

We thank Stephen Plaza, Lou Scheffer, and the Janelia FlyEM team for helpful advice on the hemibrain connectome, neuPrint database support, and the development and support of a clustering algorithm (SP) to determine neuron types based on connectivity patterns; Shan Xu and Harald Hess for patiently accommodating requests to expand the extent of the imaged hemibrain volume to include most accessory regions of the CX; members of Janelia's Connectome Annotation Team (CAT) for their CX-targeted proofreading efforts; Hideo Otsuna for providing tools for neuron type identification through comparing EM and LM images in VVDViewer; Gudrun Ihrke, Christina Chrstoforou, and the Project Technical Resources and FlyLight team for help in imaging selected GAL4 lines for neuron-type identification; Philip Hubbard for producing videos showing 3D renderings of neuronal morphologies and synaptic connections; Yoshi Aso for help in identifying the dorsal FB PPL1 dopamine neuron types and for providing the immunohistochemical results shown in Figure 49; Michael Reiser for useful discussions regarding optic lobe inputs to the AOTU; Aljoscha Nern for help in identifying hemibrain neurons with optic lobe projections; Greg Jefferis and Alexander Bates for Natverse support; Masayoshi Ito for helping to identify left/right neuron pairs in the LAL; Nicole Neubarth for help with Cypher queries; Pratyush Kandimalla for helpful discussions about TuBu and ring neurons, and about the manuscript more generally; Kazunori Shinomiya for generating the hemibrain rendering in Figure 1. We thank Sandro Romani, Yoshi Aso, Dennis Goldschmidt, Sung Soo Kim, Eugenia Chiappe, Krystyna Keleman, Shivam Chitnis, Sasha Rayshubskiy, Larry Abbott, Gaby Maimon, Cheng Lyu, Peter Mussells Pires, Ben Hardcastle, Sha Liu, Jens Goldammer, Shigehiro Namiki, and eLife reviewers, Stanley Heinze and Jason Pipkin for helpful feedback on the manuscript. This work was supported by the Howard Hughes Medical Institute.

## Additional information

### Funding

| Funder | Grant reference number | Author |
|---|---|---|
| Howard Hughes Medical Institute | | Brad K Hulse<br>Hannah Haberkern<br>Romain Franconville<br>Daniel Turner-Evans<br>Shin-ya Takemura<br>Tanya Wolff<br>Marcella Noorman<br>Marisa Dreher<br>Chuntao Dan<br>Ruchi Parekh<br>Ann M Hermundstad<br>Gerald M Rubin<br>Vivek Jayaraman |

The funders had no role in study design, data collection and interpretation, or the decision to submit the work for publication.

### Author contributions

Brad K Hulse, Hannah Haberkern, Daniel Turner-Evans, Conceptualization, Data curation, Formal analysis, Investigation, Methodology, Software, Validation, Visualization, Writing – original draft, Writing – review and editing; Romain Franconville, Conceptualization, Data curation, Formal analysis, Investigation, Methodology, Resources, Software, Validation, Visualization, Writing – original draft, Writing – review and editing; Shin-ya Takemura, Contributed to Figure 56-figure supplement 1, Methods Figure

1 and 2, and Table 2 and 3. Responsible for the accuracy of reconstructing neurons and connectomes., Data curation, Formal analysis, Investigation, Methodology, Validation, Visualization, Writing – review and editing; Tanya Wolff, Contributed to Table 2 and Table 3. Primarily responsible for providing each neuron type with a 'long name' that takes into account neuronal morphology., Data curation, Formal analysis, Investigation, Methodology, Validation, Visualization, Writing – review and editing; Marcella Noorman, Formal analysis, Primary responsibility for Figure 12 and supplements., Software, Validation, Visualization, Writing – review and editing; Marisa Dreher, Data curation, Validation, Visualization; Chuntao Dan, Contributed to Figures 46-47., Data curation, Formal analysis, Investigation, Validation, Visualization, Writing – review and editing; Ruchi Parekh, Data curation, Supervised proofreading., Supervision; Ann M Hermundstad, Formal analysis, Funding acquisition, Investigation, Methodology, Primarily responsible for Figure 11-figure supplement 1 and associated analyses. Contributed to analyses underlying Figure 12 and supplements. Contributed to Figures 66, 67, 73., Software, Supervision, Validation, Visualization, Writing – review and editing; Gerald M Rubin, Conceptualization, Contributed to Figures 46-53. Primarily responsible for Figure 48A,B and generating the SS56699 line associated with Figure 49A,B., Formal analysis, Funding acquisition, Investigation, Methodology, Project administration, Supervision, Validation, Visualization, Writing – original draft, Writing – review and editing; Vivek Jayaraman, Conceptualization, Formal analysis, Funding acquisition, Investigation, Methodology, Primary responsibility for original draft. Contributed to Figures 2-4, Figure 11-figure supplement 1, Figure 12 and supplements, Figure 73-figure supplement 1, Figure 75., Project administration, Supervision, Validation, Visualization, Writing – original draft, Writing – review and editing

### Author ORCIDs
Brad K Hulse ⓘD http://orcid.org/0000-0002-7117-7036
Hannah Haberkern ⓘD http://orcid.org/0000-0002-6135-131X
Romain Franconville ⓘD http://orcid.org/0000-0002-4440-7297
Daniel Turner-Evans ⓘD http://orcid.org/0000-0002-8020-0170
Shin-ya Takemura ⓘD http://orcid.org/0000-0003-2400-6426
Tanya Wolff ⓘD http://orcid.org/0000-0002-8681-1749
Marisa Dreher ⓘD http://orcid.org/0000-0002-0041-9229
Chuntao Dan ⓘD http://orcid.org/0000-0002-8951-4248
Ruchi Parekh ⓘD http://orcid.org/0000-0002-8060-2807
Ann M Hermundstad ⓘD http://orcid.org/0000-0002-0377-0516
Gerald M Rubin ⓘD http://orcid.org/0000-0001-8762-8703
Vivek Jayaraman ⓘD http://orcid.org/0000-0003-3680-7378

### Decision letter and Author response
Decision letter https://doi.org/10.7554/eLife.66039.sa1
Author response https://doi.org/10.7554/eLife.66039.sa2

## Additional files

### Supplementary files
• Transparent reporting form

### Data availability
All data are freely available at https://neuprint.janelia.org/.

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

## Appendix 1

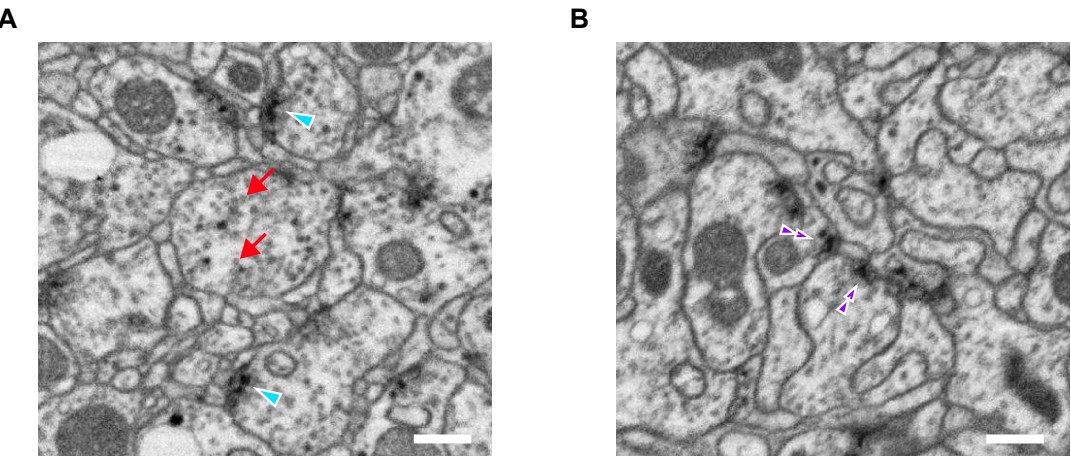

**Appendix 1—figure 1.** Regular and convergent synapses in the CX EM micrographs from the CX. Scale bars: 200 nm. Typical polyadic synapses (in FB, arrowheads), and synaptic vesicles (red arrows). Convergent synapses found in EB (double arrowheads).

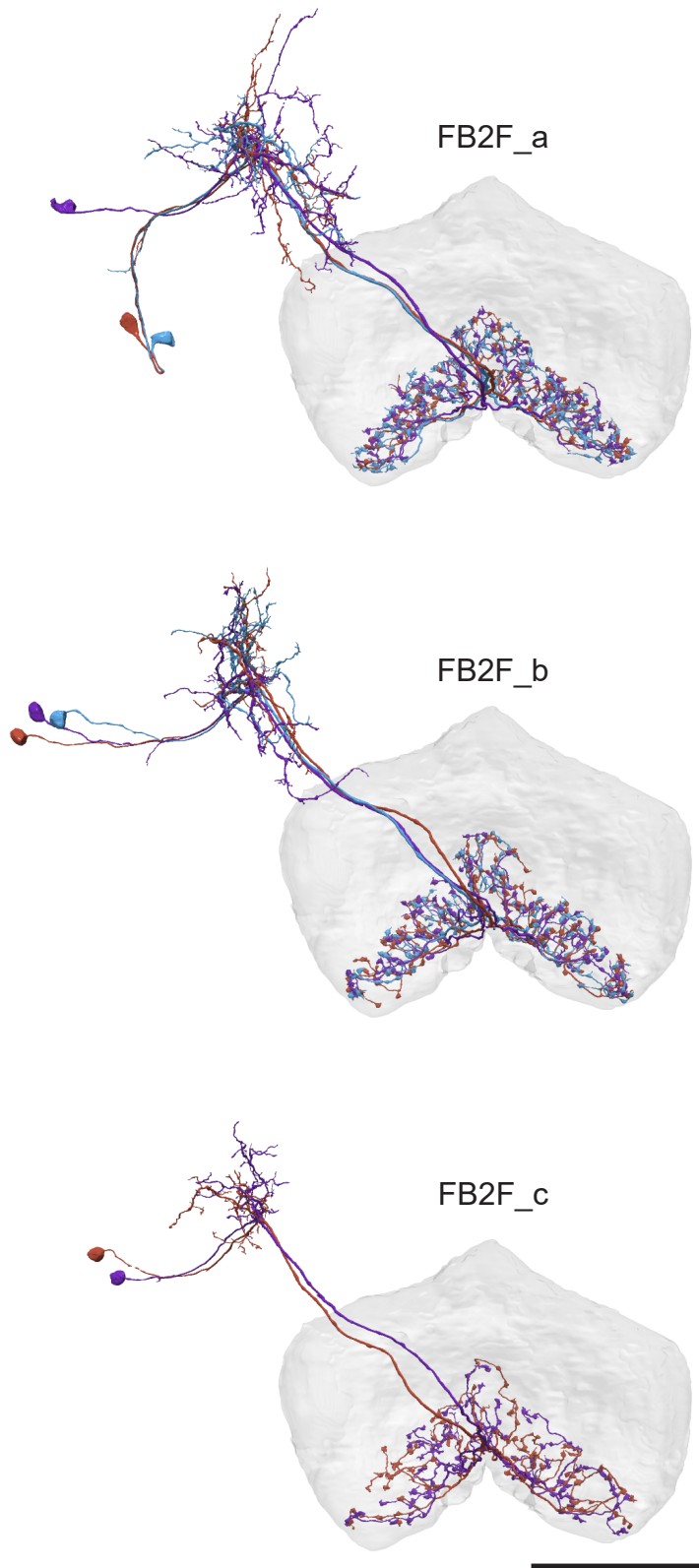

FB2F_a

FB2F_b

FB2F_c

**Appendix 1—figure 2.** An example of connectivity subtypes within a single morphology type. All these neurons were classified as type 'FB2F' but subdivided into three connectivity types. Scale bar: 50 μm.

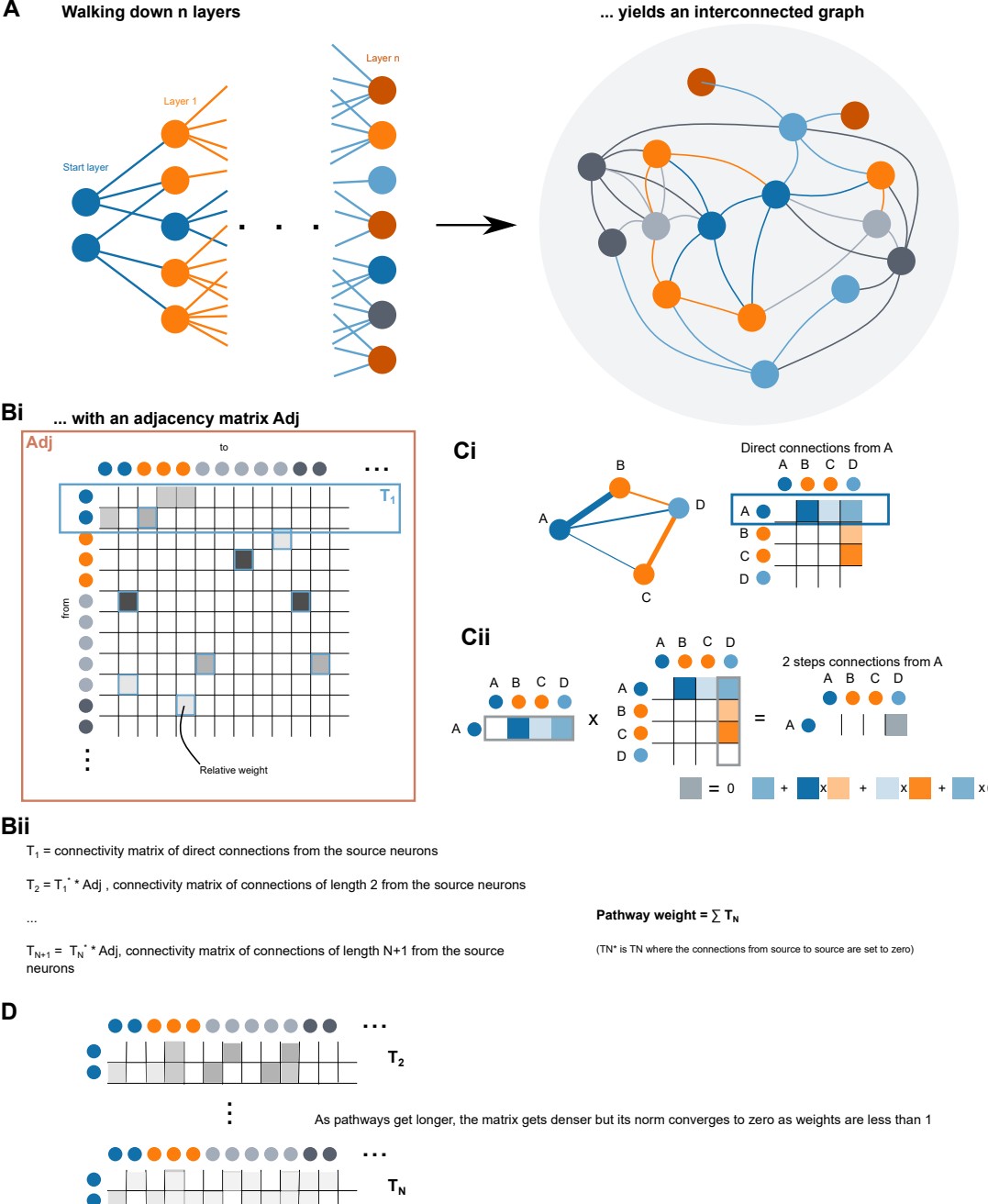

**Appendix 1—figure 3.** Graphical methods for pathway tracing and computation of pathway weights. (**A**) By walking n layers (5 in the case of the outputs network) downstream from a starting layer (here, potential CX output neurons), one obtains a complex interconnected graph. (**B**) Computing pathway weights: **Bi** The graph obtained in A yields an adjacency matrix Adj of relative weights. $T_1$ is the connectivity matrix of direct connections from the source neurons. **Bii** Formulas used to compute the pathway weight. The full pathway weight can be obtained by summing the powers of the adjacency matrix. (**C**)Toy example for a network with four neurons. **Ci** Network graph and associated adjacency matrix. The first line of the matrix is the output connectivity vector of neuron A. **Cii** Multiplying the first line of the matrix by the full adjacency matrix yields the two step connectivity vector from neuron A (this would be $T_2$ in Bii). In this case only A to D is non zero. The 2-steps weight from A to D is obtained by multiplying weights along paths and summing across path as shown in the schematic formula below. (**D**) Since the metrics used are between zero and one, the norm of the connectivity matrix of connections of length n converges to zero as n grows. Intuitively, when considering long paths, connectivity gets very weak and diffuse. As a consequence, the pathway weights matrix, which is the sum of $T_N$ converges to a stable value.

