## [Decision Letter]

**Acceptance summary:**

It is difficult to overestimate the importance of this paper. The full connectome of the *Drosophila* central complex is both the beginning and the end of an era. It provides the first comprehensive dataset of arguably the most enigmatic brain region in the insect brain. This endeavor has generated ground truth data for years of functional work on the neural circuits the connectome outlines and constitutes an unparalleled foundation for exploring the structure function relations in nervous systems in general. While significantly going beyond models of central-complex function that existed previously, the authors have to be much credited for incorporating huge amounts of existing knowledge and data into their interpretations, not only work from *Drosophila*, but also from many other insects. This effort makes this paper not only an invaluable resource on the connectome of the *Drosophila* central complex, but also a most comprehensive review on the current state of the art in central-complex research. This unifying approach of the paper clearly marks a reset of central-complex research, essentially providing a starting point of hundreds of new lines of enquiry, probably for decades to come.

**Decision letter after peer review:**

Thank you for submitting your article "A connectome of *Drosophila*'s central complex reveals network motifs for flexible navigation and context-dependent action" for consideration by *eLife*. Your article has been reviewed by 2 peer reviewers, and the evaluation has been overseen by Ronald Calabrese as the Senior and Reviewing Editor. The following individuals involved in review of your submission have agreed to reveal their identity: Stanley Heinze (Reviewer #1); Jason Pipkin (Reviewer #2).

Essential revisions:

The reviewers were impressed with the quality of the data and the presentation and have only a few substantive concerns, which should be addressed as detailed in "Recommendations for the authors."

1) Address the concern of Reviewer #1 about how synaptic strength is reflected in the connectome and the effect of experience/plasticity on the connectome and synaptic strength. Please include information on the sex and life experience of the fly from which the hemibrain was taken.

2) Address the concern of review #1 about additional data on neurite diameter and Neurite position for neurons originating from the w,x,y,z neuroblasts.

3) It would be helpful to reprise Table 1 from Scheffer et al. 2020 with names of brain regions (perhaps limited just to the CX regions).

4) Reviewer #2 has feedback on neuPrint generated figure that would make the manuscript more accessible and useful as a tool.

5) The paper is very well written and beautifully illustrated but is such a massive undertaking, it is to be expected that some minor details remain to be fixed. Please address all the minor concerns of the reviewers.

*Reviewer #1 (Recommendations for the authors):*

1) When covering the sensory input of the CX, especially when discussing how information from the anterior optic pathway is incorporated into the ellipsoid body (and hence the head direction circuit), there is much emphasis on hierarchies established by differences in synaptic strengths. Yet, at the same time it is clear that these exact synapses are subject to considerable plasticity, in particular synaptic depression based on visual input (needed for shaping visual scenes into a head direction signal). One wonders then what the particular fly used for obtaining the connectome has actually experienced before it was dissected. In particular, the fly has likely not been exposed to any polarized light signals, or other naturalistic features present in the outside world. The question arises whether the synaptic modulations shaping the ER-EPG connections have taken place in only a subset of pathways and hence the visible hierarchies and synaptic strengths are a mixture of default wiring (not shaped by experience) and wiring that has been sculpted by experience, giving a potentially distorted picture of the connection strengths. Information about the particular fly's sensory history should thus be provided and the potential implications, in particular to sensory input mapping, should be discussed.

2) While many dimensions of the data have been superbly analyzed in the paper, I am missing data on two aspects. Firstly, neurons of the CX are quite variable in their fiber diameter, and much data from other species is biased towards neurons with large main neurites. To aid interspecies comparisons, I wonder if diameter data can be provided for the cell types presented in the paper. For columnar cells, diameters could be measured in the bundle connecting the PB and CB, while for tangential cells, the main midline crossing neurite could be informative (as for hDelta cells). Given the large-cell bias in other insects, if large diameter cells from *Drosophila* are not found in other insects, they are more likely to not exist there, than if neurons with very small fiber diameters are not yet reported. Second: For neurons originating from the w,x,y,z neuroblasts, presenting the relative location of their main neurite within the w,x,y,z bundles would be informative for interspecies comparison as well, as the sequence of cells in the bundles from anterior to posterior roughly corresponds to the birth order of the neurons (for columnar, hDelta, and vDelta cells). I thus suggest to include one more figure illustrating the location of these cells within their bundle, and combine this with data about the diameter of the main neurites.

*Reviewer #2 (Recommendations for the authors):*

I have no substantive scientific criticisms; the work is beyond thorough, and the analyses are well-founded. The biggest challenge for this paper is its length. There's very interesting analysis throughout, and particularly in the Discussion, alongside a whole lot of careful mapping that can be a lot drier (but of course, explaining the map properly is the whole point).

Here are my suggestions to the authors:

– It would be helpful to reprise Table 1 from Scheffer et al. 2020 with names of brain regions (perhaps limited just to the CX regions) for those readers that are not fly aficionados already. Such a table could be referenced near the top of the paper.

– The figures are uniformly beautiful. In fact I found myself wanting to interact with the 3D reconstructions (to zoom in, flip them around, and so forth). I understand that these were not generated solely in neuPrint, but is it possible to perhaps provide a hyperlink in the figure legend to the cells involved in neuPrint? That way people could at least look at the skeletons without searching for the cell name themselves. If it's possible to link to multiple cells at once and region highlights as well so that you basically recreate the entire figure (at least with skeletons) that would be excellent, and I think would also serve to increase familiarity with using neuPrint for the grad students and so on in fly labs everywhere that are going to be referring to this paper for years to come.

– There are likely a few minor typos here and there; I did not exhaustively search for them. The only thing I spotted that seemed remotely confusing was in the legend for Figure 66, panels Di and Dii are indicated by Ci and Cii.

To both the editors and the authors:

– I think it would be really useful if the online article featured an alt-text feature such that when your cursor hovers over an abbreviated neuron's name or brain region in the text, you saw (after a little delay) some information pop up in a small box that spells out what the brain region is or what the neuron's full name is. This could be as rich or as simple as you think fitting.

As you can see, my only concerns are on how to improve the practical usefulness of the paper – perhaps these suggestions are impossible or too time-consuming to implement! But thinking down the road, this is the kind of paper like John White's 1986 *C. elegans* connectome that labs are going to keep returning to. And because it will live online, primarily, and because the dataset itself is evolving online, I think anything that can be done to make it easier to use would be worthwhile.

---

## [Author Response]

Essential revisions:1) Address the concern of Reviewer #1 about how synaptic strength is reflected in the connectome and the effect of experience/plasticity on the connectome and synaptic strength.

We discuss our use of synapse counts to assess synaptic strength in the Results section that examines ring-neuron to EPG neuron connectivity patterns. We have added further mention of this issue —and the effect of experience/plasticity that cannot be accounted for without comparative connectomics— in Discussion (see section titled ‘Connectivity and neural processing beyond the typical synapse’).

Please include information on the sex and life experience of the fly from which the hemibrain was taken.

We have added this information to the beginning of the Materials and Methods section.

2) Address the concern of review #1 about additional data on neurite diameter and Neurite position for neurons originating from the w,x,y,z neuroblasts.

We have added this additional data to the manuscript (see Figure 1 —figure supplements 2 and 3).

3) It would be helpful to reprise Table 1 from Scheffer et al. 2020 with names of brain regions (perhaps limited just to the CX regions).

We have reprised this table with only the CX regions as suggested.

4) Reviewer #2 has feedback on neuPrint generated figure that would make the manuscript more accessible and useful as a tool.

We have added links to neuPrint populations of the major neuron types referred to in the main figures of the manuscript.

Reviewer #1 (Recommendations for the authors):1) When covering the sensory input of the CX, especially when discussing how information from the anterior optic pathway is incorporated into the ellipsoid body (and hence the head direction circuit), there is much emphasis on hierarchies established by differences in synaptic strengths. Yet, at the same time it is clear that these exact synapses are subject to considerable plasticity, in particular synaptic depression based on visual input (needed for shaping visual scenes into a head direction signal). One wonders then what the particular fly used for obtaining the connectome has actually experienced before it was dissected. In particular, the fly has likely not been exposed to any polarized light signals, or other naturalistic features present in the outside world. The question arises whether the synaptic modulations shaping the ER-EPG connections have taken place in only a subset of pathways and hence the visible hierarchies and synaptic strengths are a mixture of default wiring (not shaped by experience) and wiring that has been sculpted by experience, giving a potentially distorted picture of the connection strengths. Information about the particular fly's sensory history should thus be provided and the potential implications, in particular to sensory input mapping, should be discussed.

We have added text to Results, Discussion and Methods to address this important issue.

2) While many dimensions of the data have been superbly analyzed in the paper, I am missing data on two aspects. Firstly, neurons of the CX are quite variable in their fiber diameter, and much data from other species is biased towards neurons with large main neurites. To aid interspecies comparisons, I wonder if diameter data can be provided for the cell types presented in the paper. For columnar cells, diameters could be measured in the bundle connecting the PB and CB, while for tangential cells, the main midline crossing neurite could be informative (as for hDelta cells). Given the large-cell bias in other insects, if large diameter cells from *Drosophila* are not found in other insects, they are more likely to not exist there, than if neurons with very small fiber diameters are not yet reported. Second: For neurons originating from the w,x,y,z neuroblasts, presenting the relative location of their main neurite within the w,x,y,z bundles would be informative for interspecies comparison as well, as the sequence of cells in the bundles from anterior to posterior roughly corresponds to the birth order of the neurons (for columnar, hDelta, and vDelta cells). I thus suggest to include one more figure illustrating the location of these cells within their bundle, and combine this with data about the diameter of the main neurites.

We thank the reviewer for pointing out how we could make our manuscript more useful to those working in other insects. We have added Figure 1—figure supplements 2 and 3 to provide the missing information (anatomy + tract diameters plots).

Reviewer #2 (Recommendations for the authors):I have no substantive scientific criticisms; the work is beyond thorough, and the analyses are well-founded. The biggest challenge for this paper is its length. There's very interesting analysis throughout, and particularly in the Discussion, alongside a whole lot of careful mapping that can be a lot drier (but of course, explaining the map properly is the whole point).Here are my suggestions to the authors:– It would be helpful to reprise Table 1 from Scheffer et al. 2020 with names of brain regions (perhaps limited just to the CX regions) for those readers that are not fly aficionados already. Such a table could be referenced near the top of the paper.

Figure 1 provides something very similar and we hope that orients readers to the CX and its regions.

– The figures are uniformly beautiful. In fact I found myself wanting to interact with the 3D reconstructions (to zoom in, flip them around, and so forth). I understand that these were not generated solely in neuPrint, but is it possible to perhaps provide a hyperlink in the figure legend to the cells involved in neuPrint? That way people could at least look at the skeletons without searching for the cell name themselves. If it's possible to link to multiple cells at once and region highlights as well so that you basically recreate the entire figure (at least with skeletons) that would be excellent, and I think would also serve to increase familiarity with using neuPrint for the grad students and so on in fly labs everywhere that are going to be referring to this paper for years to come.

We have added hyperlinks to neuprint for most neuron types referred to in the study.

– There are likely a few typos here and there; I did not exhaustively search for them. The only thing I spotted that seemed remotely confusing was in the legend for Figure 66, panels Di and Dii are indicated by Ci and Cii.

We have fixed this, and hopefully other similar issues.

To both the editors and the authors:– I think it would be really useful if the online article featured an alt-text feature such that when your cursor hovers over an abbreviated neuron's name or brain region in the text, you saw (after a little delay) some information pop up in a small box that spells out what the brain region is or what the neuron's full name is. This could be as rich or as simple as you think fitting.

We understand what the reviewer is asking for, but we could not find an easy way to achieve this.

As you can see, my only concerns are on how to improve the practical usefulness of the paper – perhaps these suggestions are impossible or too time-consuming to implement! But thinking down the road, this is the kind of paper like John White's 1986 *C. elegans* connectome that labs are going to keep returning to. And because it will live online, primarily, and because the dataset itself is evolving online, I think anything that can be done to make it easier to use would be worthwhile.

We appreciate the reviewer’s comments in this regard, and hope they find this version easier to digest.